# Transformers as Statisticians: Provable In-Context Learning with In-Context Algorithm Selection

**Yu Bai**[*]
Salesforce Research
yu.bai@salesforce.com

**Fan Chen**[*]
Massachusetts Institute of Technology
fanchen@mit.edu

**Huan Wang**
Salesforce Research
huan.wang@salesforce.com

**Caiming Xiong**
Salesforce Research
cxiong@salesforce.com

**Song Mei**[*]
UC Berkeley
songmei@berkeley.edu

## Abstract

Neural sequence models based on the transformer architecture have demonstrated remarkable *in-context learning* (ICL) abilities, where they can perform new tasks when prompted with training and test examples, without any parameter update to the model. This work first provides a comprehensive statistical theory for transformers to perform ICL. Concretely, we show that transformers can implement a broad class of standard machine learning algorithms in context, such as least squares, ridge regression, Lasso, learning generalized linear models, and gradient descent on two-layer neural networks, with near-optimal predictive power on various in-context data distributions. Using an efficient implementation of in-context gradient descent as the underlying mechanism, our transformer constructions admit mild size bounds, and can be learned with polynomially many pretraining sequences.

Building on these "base" ICL algorithms, intriguingly, we show that transformers can implement more complex ICL procedures involving *in-context algorithm selection*, akin to what a statistician can do in real life—A *single* transformer can adaptively select different base ICL algorithms—or even perform qualitatively different tasks—on different input sequences, without any explicit prompting of the right algorithm or task. We both establish this in theory by explicit constructions, and also observe this phenomenon experimentally. In theory, we construct two general mechanisms for algorithm selection with concrete examples: pre-ICL testing, and post-ICL validation. As an example, we use the post-ICL validation mechanism to construct a transformer that can perform nearly Bayes-optimal ICL on a challenging task—noisy linear models with mixed noise levels. Experimentally, we demonstrate the strong in-context algorithm selection capabilities of standard transformer architectures.

## 1 Introduction

Large neural sequence models have demonstrated remarkable *in-context learning* (ICL) capabilities [12], where models can make accurate predictions on new tasks when prompted with training examples from the same task, in a zero-shot fashion without any parameter update to the model. A prevalent example is large language models based on the transformer architecture [84], which can perform a diverse range of tasks in context when trained on enormous text [12, 90]. Recent models

---

[*]Equal technical and directional contributions.

Code is available at https://github.com/allenbai01/transformers-as-statisticians.

37th Conference on Neural Information Processing Systems (NeurIPS 2023).

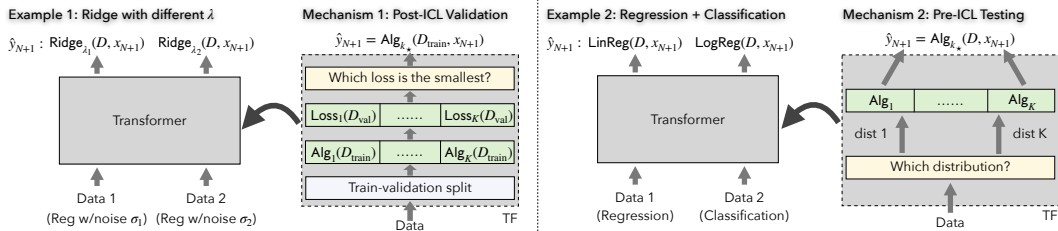

Figure 1: **Illustration of in-context algorithm selection, and two mechanisms constructed in our theory.** *Left, middle-left*: A single transformer can perform ridge regression with different $\lambda$'s on input sequences with different observation noise; we prove this by the **post-ICL validation** mechanism (Section 4.1). *Middle-right, right*: A single transformer can perform linear regression on regression data and logistic regression on classification data; we prove this via the **pre-ICL testing** mechanism (Section 4.2).

in this paradigm such as GPT-4 achieve surprisingly impressive ICL performance that makes them akin to a general-purpose agent in many aspects [65, 14]. Such strong capabilities call for better understandings, which a recent line of work tackles from various aspects [49, 94, 28, 72, 15, 57, 64].

Recent pioneering work of Garg et al. [31] proposes an interpretable and theoretically amenable setting for understanding ICL in transformers. They perform ICL experiments where input tokens are real-valued (input, label) pairs generated from standard statistical models such as linear models (and the sparse version), neural networks, and decision trees. Garg et al. [31] find that transformers can learn to perform ICL with prediction power (and fitted functions) matching standard machine learning algorithms for these settings, such as least squares for linear models, and Lasso for sparse linear models. Subsequent work further studies the internal mechanisms [2, 86, 18], expressive power [2, 32], and generalization [47] of transformers in this setting. However, these works only showcase simple mechanisms such as regularized regression [31, 2, 47] or gradient descent [2, 86, 18], which are arguably only a small subset of what transformers are capable of in practice; or expressing universal function classes not specific to ICL [89, 32]. This motivates the following question:

*How do transformers learn in context beyond implementing simple algorithms?*

This paper makes steps on this question by making two main contributions: (1) We **unveil a general mechanism—*in-context algorithm selection***—by which a *single* transformer can adaptively *select different "base" ICL algorithms* to use on *different ICL instances*, without any explicit prompting of the right algorithm to use in the input sequence. For example, a transformer may choose to perform ridge regression with regularization $\lambda_1$ on ICL instance 1, and $\lambda_2$ on ICL instance 2 (Figure 2); or perform regression on ICL instance 1 and classification on ICL instance 2 (Figure 5). This adaptivity allows transformers to achieve much stronger ICL performance than the base ICL algorithms. We both prove this in theory, and demonstrate this phenomenon empirically on standard transformer architectures. (2) Along the way, equally importantly, we present a comprehensive theory for ICL in transformers by establishing end-to-end quantitative guarantees for the **expressive power, in-context prediction performance, and sample complexity of pretraining**. These results add upon the recent line of work on the statistical learning theory of transformers [97, 89, 27, 39], and lay out a foundation for the intriguing special case where the *learning targets are themselves ICL algorithms*.

A detailed summary of our contributions is as follows.

- We prove that transformers can implement a broad class of standard machine learning algorithms in context, such as least squares, ridge regression, Lasso, convex risk minimization for learning generalized linear models (such as logistic regression), and gradient descent for two-layer neural networks (Section 3). Our constructions admit mild bounds on the number of layers, heads, and weight norms, and achieve near-optimal prediction power on many in-context data distributions.

- Technically, the above transformer constructions build on a new efficient implementation of in-context gradient descent (Appendix D), which could be broadly applicable. For a broad class of smooth convex empirical risks over the in-context training data, we construct an $(L+1)$-layer transformer that approximates $L$ steps of gradient descent. Notably, the approximation error accumulates only *linearly* in $L$, utilizing a stability-like property of smooth convex optimization.

- We prove that transformers can perform in-context algorithm selection (Section 4). We construct two algorithm selection mechanisms: Post-ICL validation (Section 4.1), and Pre-ICL testing

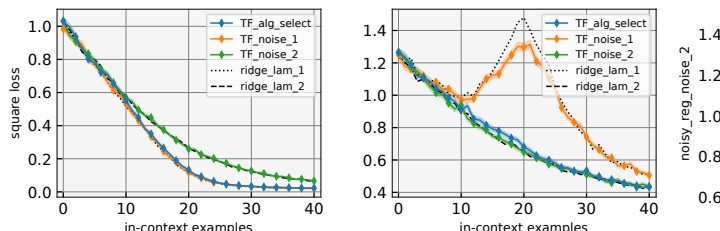

(a) Noisy linear reg with noise $\sigma_1$   (b) Noisy linear reg with noise $\sigma_2$   (c) Task 1 vs. task 2 at token 20

Figure 2: In-context algorithm selection on two separate noisy linear regression tasks with noise $(\sigma_1, \sigma_2) = (0.1, 0.5)$. *(a,b)* A **single transformer** `TF_alg_select` **simultaneously approaches the performance of the two individual Bayes predictors** `ridge_lam_1` on task 1 and `ridge_lam_2` on task 2. *(c)* At token 20 (using example $\{0, \dots, 19\}$ for training), `TF_alg_select` approaches the Bayes error on two tasks simultaneously, and **outperforms ridge regression with any fixed** $\lambda$. *(a,b,c)* Note that transformers pretrained on a single task (`TF_noise_1`, `TF_noise_2`) perform near-optimally on that task but suboptimally on the other task. More details about the setup and training method can be found in Appendix M.2.

(Section 4.2). For both mechanisms, we provide general constructions as well as concrete examples. Figure 1 provides a pictorial illustration of the two mechanisms.

- As a concrete application, using the post-ICL validation mechanism, we construct a transformer that can perform nearly Bayes-optimal ICL on noisy linear models with *mixed* noise levels (Section 4.1.1), a more complex task than those considered in existing work.

- We provide the first line of results for *pretraining* transformers to perform the various ICL tasks above, from polynomially many training sequences (Section 5 & Appendix K).

- Experimentally, we find that learned transformers indeed exhibit strong in-context algorithm selection capabilities in the settings considered in our theory (Section 6). For example, Figure 2 shows that a *single* transformer can approach the individual Bayes risks (the optimal risk among all possible algorithms) simultaneously on two noisy linear models with different noise levels.

**Transformers as statisticians**   We humbly remark that the typical toolkit of a statistician contains much more beyond those covered in this work, including and not limited to inference, uncertainty quantification, and theoretical analysis. This work merely aims to show the algorithm selection capability of transformers, akin to what a statistician *can* do.

**Related work**   Our work is intimately related to the lines of work on in-context learning, theoretical understandings of transformers, as well as other formulations for learning-to-learn such as meta-learning. Due to limited space, we discuss these related work in Appendix A.

## 2   Preliminaries

We consider a sequence of $N$ input vectors $\{\mathbf{h}_i\}_{i=1}^N \subset \mathbb{R}^D$, written compactly as an input matrix $\mathbf{H} = [\mathbf{h}_1, \dots, \mathbf{h}_N] \in \mathbb{R}^{D \times N}$, where each $\mathbf{h}_i$ is a column of $\mathbf{H}$ (also a *token*). Throughout this paper, we let $\sigma(t) := \mathrm{ReLU}(t) = \max\{t, 0\}$ denote the standard relu activation.

### 2.1   Transformers

We consider transformer architectures that process any input sequence $\mathbf{H} \in \mathbb{R}^{D \times N}$ by applying (encoder-mode[2]) attention layers and MLP layers formally defined as follows.

**Definition 1** (Attention layer). *A (self-)attention layer with $M$ heads is denoted as $\mathrm{Attn}_{\boldsymbol{\theta}}(\cdot)$ with parameters $\boldsymbol{\theta} = \{(\mathbf{V}_m, \mathbf{Q}_m, \mathbf{K}_m)\}_{m \in [M]} \subset \mathbb{R}^{D \times D}$. On any input sequence $\mathbf{H} \in \mathbb{R}^{D \times N}$,*

$$\widetilde{\mathbf{H}} = \mathrm{Attn}_{\boldsymbol{\theta}}(\mathbf{H}) := \mathbf{H} + \frac{1}{N} \sum_{m=1}^M (\mathbf{V}_m \mathbf{H}) \times \sigma\big((\mathbf{Q}_m \mathbf{H})^\top (\mathbf{K}_m \mathbf{H})\big) \in \mathbb{R}^{D \times N}, \tag{1}$$

*where $\sigma : \mathbb{R} \to \mathbb{R}$ is the ReLU function. In vector form,*

$$\widetilde{\mathbf{h}}_i = [\mathrm{Attn}_{\boldsymbol{\theta}}(\mathbf{H})]_i = \mathbf{h}_i + \sum_{m=1}^M \frac{1}{N} \sum_{j=1}^N \sigma(\langle \mathbf{Q}_m \mathbf{h}_i, \mathbf{K}_m \mathbf{h}_j \rangle) \cdot \mathbf{V}_m \mathbf{h}_j.$$

---

[2]Many of our results can be generalized to decoder-based architectures; see Appendix C for a discussion.

Above, (1) uses a normalized ReLU[3] activation $t \mapsto \sigma(t)/N$ in place of the standard softmax activation; we remark this activation is also found to work well empirically in recent studies [78, 93].

**Definition 2** (MLP layer). *A (token-wise) MLP layer with hidden dimension $D'$ is denoted as* $\mathrm{MLP}_{\boldsymbol{\theta}}(\cdot)$ *with parameters* $\boldsymbol{\theta} = (\mathbf{W}_1, \mathbf{W}_2) \in \mathbb{R}^{D' \times D} \times \mathbb{R}^{D \times D'}$. *On any input sequence* $\mathbf{H} \in \mathbb{R}^{D \times N}$,

$$\widetilde{\mathbf{H}} = \mathrm{MLP}_{\boldsymbol{\theta}}(\mathbf{H}) := \mathbf{H} + \mathbf{W}_2 \sigma(\mathbf{W}_1 \mathbf{H}),$$

*where* $\sigma : \mathbb{R} \to \mathbb{R}$ *is the ReLU function. In vector form, we have* $\widetilde{\mathbf{h}}_i = \mathbf{h}_i + \mathbf{W}_2 \sigma(\mathbf{W}_1 \mathbf{h}_i)$.

We consider a transformer architecture with $L \geq 1$ transformer layers, each consisting of a self-attention layer followed by an MLP layer.

**Definition 3** (Transformer). *An $L$-layer transformer, denoted as* $\mathrm{TF}_{\boldsymbol{\theta}}(\cdot)$, *is a composition of $L$ self-attention layers each followed by an MLP layer:* $\mathbf{H}^{(L)} = \mathrm{TF}_{\boldsymbol{\theta}}(\mathbf{H}^{(0)})$, *where* $\mathbf{H}^{(0)} \in \mathbb{R}^{D \times N}$ *is the input sequence, and*

$$\mathbf{H}^{(\ell)} = \mathrm{MLP}_{\boldsymbol{\theta}_{\mathrm{mlp}}^{(\ell)}}\Big(\mathrm{Attn}_{\boldsymbol{\theta}_{\mathrm{attn}}^{(\ell)}}\big(\mathbf{H}^{(\ell-1)}\big)\Big), \quad \ell \in \{1, \dots, L\}.$$

*Above, the parameter* $\boldsymbol{\theta} = (\boldsymbol{\theta}_{\mathrm{attn}}^{(1:L)}, \boldsymbol{\theta}_{\mathrm{mlp}}^{(1:L)})$ *consists of the attention layers* $\boldsymbol{\theta}_{\mathrm{attn}}^{(\ell)} = \{(\mathbf{V}_m^{(\ell)}, \mathbf{Q}_m^{(\ell)}, \mathbf{K}_m^{(\ell)})\}_{m \in [M^{(\ell)}]} \subset \mathbb{R}^{D \times D}$ *and the MLP layers* $\boldsymbol{\theta}_{\mathrm{mlp}}^{(\ell)} = (\mathbf{W}_1^{(\ell)}, \mathbf{W}_2^{(\ell)}) \in \mathbb{R}^{D^{(\ell)} \times D} \times \mathbb{R}^{D \times D^{(\ell)}}$. *We will frequently consider* "attention-only" *transformers with* $\mathbf{W}_1^{(\ell)}, \mathbf{W}_2^{(\ell)} = \mathbf{0}$, *which we denote as* $\mathrm{TF}_{\boldsymbol{\theta}}^0(\cdot)$ *for shorthand, with* $\boldsymbol{\theta} = \boldsymbol{\theta}^{(1:L)} := \boldsymbol{\theta}_{\mathrm{attn}}^{(1:L)}$.

We additionally define the following norm of a transformer $\mathrm{TF}_{\boldsymbol{\theta}}$:

$$\|\|\boldsymbol{\theta}\|\| := \max_{\ell \in [L]} \left\{ \max_{m \in [M]} \left\{ \|\mathbf{Q}_m^{(\ell)}\|_{\mathrm{op}}, \|\mathbf{K}_m^{(\ell)}\|_{\mathrm{op}} \right\} + \sum_{m=1}^{M} \|\mathbf{V}_m^{(\ell)}\|_{\mathrm{op}} + \|\mathbf{W}_1^{(\ell)}\|_{\mathrm{op}} + \|\mathbf{W}_2^{(\ell)}\|_{\mathrm{op}} \right\}. \quad (2)$$

In (2), the choices of the operator norm and max/sums are for convenience only and not essential, as our results (e.g. for pretraining) depend only logarithmically on $\|\|\boldsymbol{\theta}\|\|$.

## 2.2 In-context learning

In an in-context learning (ICL) instance, the model is given a dataset $\mathcal{D} = \{(\mathbf{x}_i, y_i)\}_{i \in [N]} \overset{\mathrm{iid}}{\sim} \mathsf{P}$ and a new test input $\mathbf{x}_{N+1} \sim \mathsf{P}_{\mathbf{x}}$ for some data distribution $\mathsf{P}$, where $\{\mathbf{x}_i\}_{i \in [N]} \subseteq \mathbb{R}^d$ are the input vectors, $\{y_i\}_{i \in [N]} \subseteq \mathbb{R}$ are the corresponding labels (e.g. real-valued for regression, or $\{0, 1\}$-valued for binary classification), and $\mathbf{x}_{N+1}$ is the test input on which the model is required to make a prediction. Different from standard supervised learning, in ICL, each instance $(\mathcal{D}, \mathbf{x}_{N+1})$ is in general drawn from a different distribution $\mathsf{P}_j$, such as a linear model with a new ground truth coefficient $\mathbf{w}_{\star,j} \in \mathbb{R}^d$. Our goal is to construct *fixed* transformer to perform ICL on a large set of $\mathsf{P}_j$'s.

We consider using transformers to perform ICL, in which we encode $(\mathcal{D}, \mathbf{x}_{N+1})$ into an input sequence $\mathbf{H} \in \mathbb{R}^{D \times (N+1)}$. In our theory, we use the following format, where the first two rows contain $(\mathcal{D}, \mathbf{x}_{N+1})$ (zero at the location for $y_{N+1}$), and the third row contains fixed vectors $\{\mathbf{p}_i\}_{i \in [N+1]}$ with ones, zeros, and indicator for being the train token (similar to a positional encoding vector):

$$\mathbf{H} = \begin{bmatrix} \mathbf{x}_1 & \mathbf{x}_2 & \dots & \mathbf{x}_N & \mathbf{x}_{N+1} \\ y_1 & y_2 & \dots & y_N & 0 \\ \mathbf{p}_1 & \mathbf{p}_2 & \dots & \mathbf{p}_N & \mathbf{p}_{N+1} \end{bmatrix} \in \mathbb{R}^{D \times (N+1)}, \quad \mathbf{p}_i := \begin{bmatrix} \mathbf{0}_{D-(d+3)} \\ 1 \\ \mathbb{1}\{i < N+1\} \end{bmatrix} \in \mathbb{R}^{D-(d+1)}. \quad (3)$$

We will choose $D = \Theta(d)$, so that the hidden dimension of $\mathbf{H}$ is at most a constant multiple of $d$. We then feed $\mathbf{H}$ into a transformer to obtain the output $\widetilde{\mathbf{H}} = \mathrm{TF}_{\boldsymbol{\theta}}(\mathbf{H}) \in \mathbb{R}^{D \times (N+1)}$ with the same shape, and *read out* the prediction $\widehat{y}_{N+1}$ from the $(d+1, N+1)$-th entry of $\widetilde{\mathbf{H}} = [\widetilde{\mathbf{h}}_i]_{i \in [N+1]}$ (the entry corresponding to the missing test label): $\widehat{y}_{N+1} = \mathrm{read}_y(\widetilde{\mathbf{H}}) := (\widetilde{\mathbf{h}}_{N+1})_{d+1}$. The goal is to predict

---

[3]For each query index $i$, the attention weights $\{\sigma(\langle \mathbf{Q}_m \mathbf{h}_i, \mathbf{K}_m \mathbf{h}_j \rangle)/N\}_{j \in [N]}$ is also a set of non-negative weights that sum to $O(1)$ (similar as a softmax probability distribution) in typical scenarios. Also, our approximation results can potentially be generalized to softmax attention e.g. using the technique of [32].

$\widehat{y}_{N+1}$ that is close to $y_{N+1} \sim \mathsf{P}_{y|\mathbf{x}_{N+1}}$ measured by proper losses. We emphasize that we consider predicting only at the last token $\mathbf{x}_{N+1}$, which is without much loss of generality.[4]

**Miscellaneous setups** We assume bounded features and labels throughout the paper (unless otherwise specified, e.g. when $\mathbf{x}_i$ is Gaussian): $\|\mathbf{x}_i\|_2 \leq B_x$ and $|y_i| \leq B_y$ with probability one. We use the standard notation $\mathbf{X} = [\mathbf{x}_1^\top; \ldots; \mathbf{x}_N^\top] \in \mathbb{R}^{N \times d}$ and $\mathbf{y} = [y_1; \ldots; y_N] \in \mathbb{R}^N$ to denote the matrix of inputs and vector of labels, respectively. To prevent the transformer from blowing up on tail events, in all our results concerning (statistical) in-context prediction powers, we consider a clipped prediction $\widehat{y}_{N+1} = \widetilde{\mathrm{read}}_{\mathsf{y}}(\widetilde{\mathbf{H}}) := \mathrm{clip}_R((\widetilde{\mathbf{h}}_{N+1})_{d+1})$, where $\mathrm{clip}_R(t) := \mathrm{Proj}_{[-R,R]}(t)$ is the standard clipping operator with (a suitably large) radius $R \geq 0$ that varies in different problems.

# 3 Basic in-context learning algorithms

We begin by constructing transformers that approximately implement a variety of standard machine learning algorithms in context, with mild size bounds and near-optimal prediction power on many standard in-context data distributions.

## 3.1 In-context ridge regression and least squares

Consider the standard ridge regression estimator over the in-context training examples $\mathcal{D}$ with regularization $\lambda \geq 0$ (reducing to least squares at $\lambda = 0$ and $N \geq d$):

$$\mathbf{w}_{\mathrm{ridge}}^\lambda := \arg\min_{\mathbf{w} \in \mathbb{R}^d} \frac{1}{2N} \sum_{i=1}^N \left(\langle \mathbf{w}, \mathbf{x}_i \rangle - y_i\right)^2 + \frac{\lambda}{2} \|\mathbf{w}\|_2^2. \tag{ICRidge}$$

We show that transformers can approximately implement (ICRidge) (proof in Appendix F.1).

**Theorem 4** (Implementing in-context ridge regression). *For any $\lambda \geq 0$, $0 \leq \alpha \leq \beta$ with $\kappa := \frac{\beta+\lambda}{\alpha+\lambda}$, $B_w > 0$, and $\varepsilon < B_x B_w / 2$, there exists an $L$-layer attention-only transformer $\mathrm{TF}_{\boldsymbol{\theta}}^0$ with*

$$L = \lceil 2\kappa \log(B_x B_w / (2\varepsilon)) \rceil + 1, \quad \max_{\ell \in [L]} M^{(\ell)} \leq 3, \quad \|\boldsymbol{\theta}\| \leq 4R + 8(\beta + \lambda)^{-1}. \tag{4}$$

*(with $R := \max\{B_x B_w, B_y, 1\}$) such that the following holds. On any input data $(\mathcal{D}, \mathbf{x}_{N+1})$ such that the problem (ICRidge) is well-conditioned and has a bounded solution:*

$$\alpha \leq \lambda_{\min}(\mathbf{X}^\top \mathbf{X}/N) \leq \lambda_{\max}(\mathbf{X}^\top \mathbf{X}/N) \leq \beta, \quad \|\mathbf{w}_{\mathrm{ridge}}^\lambda\|_2 \leq B_w/2, \tag{5}$$

*$\mathrm{TF}_{\boldsymbol{\theta}}^0$ approximately implements (ICRidge): The prediction $\widehat{y}_{N+1} = \mathrm{read}_{\mathsf{y}}(\mathrm{TF}_{\boldsymbol{\theta}}^0(\mathbf{H}))$ satisfies*

$$\left| \widehat{y}_{N+1} - \langle \mathbf{w}_{\mathrm{ridge}}^\lambda, \mathbf{x}_{N+1} \rangle \right| \leq \varepsilon. \tag{6}$$

Theorem 4 presents the first quantitative construction for end-to-end in-context ridge regression up to arbitrary precision, and improves upon Akyürek et al. [2] whose construction does not give (or directly imply) an explicit error bound like (6). Further, the bounds on the number of layers and heads in (4) are mild (constant heads and logarithmically many layers).

**Near-optimal in-context prediction power for linear problems** Combining Theorem 4 with standard analyses of linear regression yields the following corollaries (proofs in Appendix F.3 & F.4).

**Corollary 5** (Near-optimal linear regression with transformers by approximating least squares). *For any $N \geq \widetilde{\mathcal{O}}(d)$, there exists an $\mathcal{O}(\kappa \log(\kappa N/\sigma))$-layer transformer $\boldsymbol{\theta}$, such that on any $\mathsf{P}$ satisfying standard statistical assumptions for least squares (Assumption A), its ICL prediction $\widehat{y}_{N+1}$ achieves*

$$\mathbb{E}_{(\mathcal{D}, \mathbf{x}_{N+1}, y_{N+1}) \sim \mathsf{P}}[(\widehat{y}_{N+1} - y_{N+1})^2] \leq \inf_{\mathbf{w}} \mathbb{E}_{(\mathbf{x},y) \sim \mathsf{P}}[(y - \langle \mathbf{w}, \mathbf{x} \rangle)^2] + \widetilde{\mathcal{O}}(d\sigma^2/N).$$

Assumption A requires only generic tail properties such as sub-Gaussianity, and *not* realizability (i.e., $\mathsf{P}$ follows a true linear model); $\kappa, \sigma$ above denote the covariance condition number and the

---

[4]Our constructions may be generalized to predicting at every token, by using a decoder architecture and potentially different input formats correspondingly (cf. Appendix C). Our theory focuses on predicting at the last token only, which simplifies the setting. Our experiments test both settings.

noise level therein. The $\widetilde{\mathcal{O}}(d\sigma^2/N)$ excess risk is known to be rate-optimal for linear regression [38], and Corollary 5 achieves this in context with a transformer with only logarithmically many layers.

Next, consider Bayesian linear models where each in-context data distribution $\mathsf{P} = \mathsf{P}_{\mathbf{w}_\star}^{\mathsf{lin}}$ is drawn from a Gaussian prior $\pi: \mathbf{w}_\star \sim \mathsf{N}(0, \mathbf{I}_d/d)$, and $(\mathbf{x}, y) \sim \mathsf{P}_{\mathbf{w}_\star}^{\mathsf{lin}}$ is sampled as $\mathbf{x} \sim \mathsf{N}(\mathbf{0}, \mathbf{I}_d)$, $y = \langle \mathbf{w}_\star, \mathbf{x} \rangle + \mathsf{N}(0, \sigma^2)$. It is a standard result that the Bayes estimator of $y_{N+1}$ given $(\mathcal{D}, \mathbf{x}_{N+1})$ is given by ridge regression (ICRidge): $\widehat{y}_{N+1}^{\mathsf{Bayes}} := \langle \mathbf{w}_{\mathrm{ridge}}^\lambda, \mathbf{x}_{N+1} \rangle$ with $\lambda = d\sigma^2/N$. We show that transformers achieve nearly-Bayes risk for this problem, and we use

$$\mathsf{BayesRisk}_\pi := \mathbb{E}_{\mathbf{w}_\star \sim \pi, (\mathcal{D}, \mathbf{x}_{N+1}, y_{N+1}) \sim \mathsf{P}_{\mathbf{w}_\star}^{\mathsf{lin}}} \left[ \tfrac{1}{2} \left( \widehat{y}_{N+1}^{\mathsf{Bayes}} - y_{N+1} \right)^2 \right]$$

to denote the Bayes risk of this problem under prior $\pi$.

**Corollary 6** (Nearly-Bayes linear regression with transformers by approximating ridge regression). *Under the Bayesian linear model above with $N \geq \max\{d/10, \mathcal{O}(\log(1/\varepsilon))\}$, there exists a $L = \mathcal{O}(\log(1/\varepsilon))$-layer transformer such that $\mathbb{E}_{\mathbf{w}_\star, (\mathcal{D}, \mathbf{x}_{N+1}, y_{N+1})} \left[ \tfrac{1}{2} (\widehat{y}_{N+1} - y_{N+1})^2 \right] \leq \mathsf{BayesRisk}_\pi + \varepsilon$.*

**Generalized linear models** In Appendix G, we extend the above results to generalized linear models [53] and show that transformers can approximate the corresponding convex risk minimization algorithm in context (which includes logistic regression for linear classification as an important special case), and achieve near-optimal excess risk under standard statistical assumptions.

## 3.2 In-context Lasso

Consider the standard Lasso estimator [82] which minimizes an $\ell_1$-regularized linear regression loss $\widehat{L}_{\mathrm{lasso}}$ over the in-context training examples $\mathcal{D}$:

$$\mathbf{w}_{\mathrm{lasso}} := \arg\min_{\mathbf{w} \in \mathbb{R}^d} \widehat{L}_{\mathrm{lasso}}(\mathbf{w}) = \tfrac{1}{2N} \sum_{i=1}^N \left( \langle \mathbf{w}, \mathbf{x}_i \rangle - y_i \right)^2 + \lambda_N \|\mathbf{w}\|_1. \qquad \text{(ICLasso)}$$

We show that transformers can also approximate in-context Lasso with a mild number of layers, and can perform sparse linear regression in standard sparse linear models (proofs in Appendix H).

**Theorem 7** (Implementing in-context Lasso). *For any $\lambda_N \geq 0$, $\beta > 0$, $B_w > 0$, and $\varepsilon > 0$, there exists a $L$-layer transformer $\mathrm{TF}_{\boldsymbol{\theta}}$ with*

$$L = \lceil \beta B_w^2/\varepsilon \rceil + 1, \quad \max_{\ell \in [L]} M^{(\ell)} \leq 2, \quad \max_{\ell \in [L]} D^{(\ell)} \leq 2d, \quad \|\boldsymbol{\theta}\| \leq \mathcal{O}\left(R + (1 + \lambda_N)\beta^{-1}\right)$$

*(where $R := \max\{B_x B_w, B_y, 1\}$) such that the following holds. On any input data $(\mathcal{D}, \mathbf{x}_{N+1})$ such that $\lambda_{\max}(\mathbf{X}^\top \mathbf{X}/N) \leq \beta$ and $\|\mathbf{w}_{\mathrm{lasso}}\|_2 \leq B_w/2$, $\mathrm{TF}_{\boldsymbol{\theta}}(\mathbf{H}^{(0)})$ approximately implements (ICLasso), in that it outputs $\widehat{y}_{N+1} = \langle \mathbf{x}_{N+1}, \widehat{\mathbf{w}} \rangle$ with $\widehat{L}_{\mathrm{lasso}}(\widehat{\mathbf{w}}) - \widehat{L}_{\mathrm{lasso}}(\mathbf{w}_{\mathrm{lasso}}) \leq \varepsilon$.*

**Theorem 8** (Near-optimal sparse linear regression with transformers by approximating Lasso). *For any $d, N \geq 1, \delta > 0, B_w^\star, \sigma > 0$, there exists a $\widetilde{\mathcal{O}}((B_w^\star)^2/\sigma^2 \times (1 + (d/N)))$-layer transformer $\boldsymbol{\theta}$ such that the following holds: For any $s$ and $N \geq \mathcal{O}(s \log(d/\delta))$, suppose that $\mathsf{P}$ is an $s$-sparse linear model: $\mathbf{x}_i \sim \mathsf{N}(0, \mathbf{I}_d)$, $y_i = \langle \mathbf{w}_\star, \mathbf{x}_i \rangle + \mathsf{N}(0, \sigma^2)$ for any $\|\mathbf{w}_\star\|_2 \leq B_w^\star$ and $\|\mathbf{w}_\star\|_0 \leq s$, then with probability at least $1 - \delta$ (over the randomness of $\mathcal{D}$), the transformer output $\widehat{y}_{N+1}$ achieves*

$$\mathbb{E}_{(\mathbf{x}_{N+1}, y_{N+1}) \sim \mathsf{P}} \left[ (\widehat{y}_{N+1} - y_{N+1})^2 \right] \leq \sigma^2 [1 + \mathcal{O}(s \log(d/\delta)/N)].$$

The $\widetilde{\mathcal{O}}(s \log d/N)$ excess risk obtained in Theorem 8 is optimal up to log factors [62, 87]. We remark that Theorem 8 is not a direct corollary of Theorem 7; Rather, the bound on the number of layers in Theorem 8 requires a sharper convergence analysis of the (ICLasso) problem under sparse linear models (Appendix H.2), similar to [1].

## 3.3 Proof technique: In-context gradient descent

The constructions in Section 3.1 and 3.2 is built on the following result for approximating in-context (proximal) gradient descent on (regularized) convex losses.

**Theorem 9** (ICGD; Informal version of Theorem D.1 & D.2). *For a broad class of convex losses of form $\mathbf{w} \mapsto \tfrac{1}{N} \sum_{i=1}^N \ell(\mathbf{w}^\top \mathbf{x}_i, y_i) + R(\mathbf{w})$, there exists an $L$-layer transformer that takes in any $(\mathcal{D}, \mathbf{w}^0)$ and outputs $\widehat{\mathbf{w}}^L$ such that $\|\widehat{\mathbf{w}}^L - \mathbf{w}_{\{\mathrm{GD}, \mathrm{PGD}\}}^L\|_2 \leq \mathcal{O}(L\varepsilon)$, by composing $L$ identical layers each $\mathcal{O}(\varepsilon)$-approximating a single step of GD (so that $\mathcal{O}(L\varepsilon)$ is a linear error accumulation).*

Theorem 9 is established in two main steps:

- Approximating one-step of ICGD using one attention layer (Proposition E.1), which substantially generalizes that of von Oswald et al. [86] (which only does GD on square losses with a *linear* self-attention), and is simpler than the ones in Akyürek et al. [2] and Giannou et al. [32].
- Stacking $L$ of the above layer to approximate $L$ steps of ICGD. Done naively, the error accumulation of this stacking operation is exponential in $L$ in the worst case. We utilize the stability of *convex* gradient descent (Lemma D.1) to obtain the *linear* in $L$ error accumulation in Theorem 9.

In Appendix D.3, we also give results for *non-convex* GD on two-layer neural nets, though with a worse (exponential in $L$) error accumulation as expected.

## 4 In-context algorithm selection

We now show that transformers can perform various kinds of *in-context algorithm selection*, which allows them to implement more complex ICL procedures by adaptively selecting different "base" algorithms on different input sequences. We construct two general mechanisms: *Post-ICL validation*, and *Pre-ICL testing*; See Figure 1 for a pictorial illustration.

### 4.1 Post-ICL validation mechanism

In our first mechanism, post-ICL validation, the transformer begins by implementing a *train-validation split* $\mathcal{D} = (\mathcal{D}_{\mathsf{train}}, \mathcal{D}_{\mathsf{val}})$, and running $K$ *base* ICL algorithms on $\mathcal{D}_{\mathsf{train}}$. Let $\{f_k\}_{k \in [K]} \subset (\mathbb{R}^d \to \mathbb{R})$ denote the $K$ learned predictors, and

$$\widehat{L}_{\mathsf{val}}(f) := \tfrac{1}{|\mathcal{D}_{\mathsf{val}}|} \sum_{(\mathbf{x}_i, y_i) \in \mathcal{D}_{\mathsf{val}}} \ell(f(\mathbf{x}_i), y_i) \tag{7}$$

denote the validation loss of any predictor $f$.

We show that (proof in Appendix I.1) a 3-layer transformer can output a predictor $\widehat{f}$ that achieves nearly the smallest validation loss, and thus nearly optimal expected loss if $\widehat{L}_{\mathsf{val}}$ concentrates around the expected loss $L$. Below, the input sequence $\mathbf{H}$ uses a generalized positional encoding $\mathbf{p}_i := [\mathbf{0}_{D-(d+3)}; 1; t_i]$ in (3), where $t_i := 1$ for $i \in \mathcal{D}_{\mathsf{train}}$, $t_i := -1$ for $i \in \mathcal{D}_{\mathsf{val}}$, and $t_{N+1} := 0$.

**Proposition 10** (In-context algorithm selection via train-validation split). *Suppose that $\ell(\cdot, \cdot)$ in (7) is approximable by sum of relus (Definition D.1, which includes all $C^3$-smooth bivariate functions). Then there exists a 3-layer transformer $\mathrm{TF}_{\boldsymbol{\theta}}$ that maps (defining $y_i' = y_i 1\{i < N+1\}$)*

$$\mathbf{h}_i = [\mathbf{x}_i; y_i'; *; f_1(\mathbf{x}_i); \cdots; f_K(\mathbf{x}_i); \mathbf{0}_{K+1}; 1; t_i] \quad \rightarrow \quad \mathbf{h}_i' = [\mathbf{x}_i; y_i'; *; \widehat{f}(\mathbf{x}_i); 1; t_i], \; i \in [N+1],$$

*where the predictor $\widehat{f} : \mathbb{R}^d \to \mathbb{R}$ is a convex combination of $\{f_k : \widehat{L}_{\mathsf{val}}(f_k) \leq \min_{k_\star \in [K]} \widehat{L}_{\mathsf{val}}(f_{k_\star}) + \gamma\}$. As a corollary, for any convex risk $L : (\mathbb{R}^d \to \mathbb{R}) \to \mathbb{R}$, $\widehat{f}$ satisfies*

$$L(\widehat{f}) \leq \min_{k_\star \in [K]} L(f_{k_\star}) + \max_{k \in [K]} \left| \widehat{L}_{\mathsf{val}}(f_k) - L(f_k) \right| + \gamma.$$

**Ridge regression with in-context regularization selection** As an example, we use Proposition 10 to construct a transformer to perform in-context ridge regression with regularization selection according to the *unregularized* validation loss $\widehat{L}_{\mathsf{val}}(\mathbf{w}) := \frac{1}{2|\mathcal{D}_{\mathsf{val}}|} \sum_{(x_i, y_i) \in \mathcal{D}_{\mathsf{val}}} (\langle \mathbf{w}, \mathbf{x}_i \rangle - y_i)^2$ (proof in Appendix I.2). Let $\lambda_1, \ldots, \lambda_K \geq 0$ be $K$ fixed regularization strengths.

**Theorem 11** (Ridge regression with in-context regularization selection). *There exists a transformer with $\mathcal{O}(\log(1/\varepsilon))$ layers and $\mathcal{O}(K)$ heads such that the following holds: On any $(\mathcal{D}, \mathbf{x}_{N+1})$ well-conditioned (cf. (5)) for all $\{\lambda_k\}_{k \in [K]}$, it outputs $\widehat{y}_{N+1} = \langle \widehat{\mathbf{w}}, \mathbf{x}_{N+1} \rangle$, where*

$$\mathrm{dist}\left( \widehat{\mathbf{w}}, \mathrm{conv}\{\widehat{\mathbf{w}}_{\mathsf{ridge,train}}^{\lambda_k} : \widehat{L}_{\mathsf{val}}(\widehat{\mathbf{w}}_{\mathsf{ridge,train}}^{\lambda_k}) \leq \min_{k_\star \in [K]} \widehat{L}_{\mathsf{val}}(\widehat{\mathbf{w}}_{\mathsf{ridge,train}}^{\lambda_{k_\star}}) + \gamma\} \right) \leq \varepsilon.$$

*Above, $\widehat{\mathbf{w}}_{\mathsf{ridge,train}}^{\lambda}$ denotes the solution to (ICRidge) on the training split $\mathcal{D}_{\mathsf{train}}$.*

#### 4.1.1 Nearly Bayes-optimal ICL on noisy linear models with mixed noise levels

We build on Theorem 11 to show that transformers can perform nearly Bayes-optimal ICL when data come from noisy linear models with a *mixture of $K$ different noise levels* $\sigma_1, \ldots, \sigma_K > 0$.

Concretely, consider the following data generating model, where we first sample $\mathsf{P} = \mathsf{P}_{\mathbf{w}_\star, \sigma_k} \sim \pi$ from $k \sim \Lambda \in \Delta([K])$, $\mathbf{w}_\star \sim \mathsf{N}(\mathbf{0}, \mathbf{I}_d/d)$, and then sample data $\{(\mathbf{x}_i, y_i)\}_{i \in [N+1]} \stackrel{\text{iid}}{\sim} \mathsf{P}_{\mathbf{w}_\star, \sigma_k}$ as

$$\mathsf{P}_{\mathbf{w}_\star, \sigma_k} : \mathbf{x}_i \sim \mathsf{N}(\mathbf{0}, \mathbf{I}_d), \quad y_i = \langle \mathbf{x}_i, \mathbf{w}_\star \rangle + \mathsf{N}(0, \sigma_k^2).$$

For any fixed $(N, d)$, consider the Bayes risk for predicting $y_{N+1}$ under this model:

$$\mathsf{BayesRisk}_\pi := \inf_{\mathcal{A}} \mathbb{E}_\pi \left[ \tfrac{1}{2} (\mathcal{A}(\mathcal{D})(\mathbf{x}_{N+1}) - y_{N+1})^2 \right].$$

By standard Bayesian calculations, the above Bayes risk is attained when $\mathcal{A}$ is a certain *mixture of $K$ ridge regressions* with regularization $\lambda_k = d\sigma_k^2/N$; however, the mixing weights depend on $\mathcal{D}$ in a highly non-trivial fashion (see Appendix J.2 for a derivation). By using the post-ICL validation mechanism in Theorem 11, we construct a transformer that achieves nearly the Bayes risk.

**Theorem 12** (Nearly Bayes-optimal ICL; Informal version of Theorem J.1). *For sufficiently large $N, d$, there exists a transformer with $\mathcal{O}(\log N)$ layers and $\mathcal{O}(K)$ heads such that on the above model, it outputs a prediction $\widehat{y}_{N+1}$ that is nearly Bayes-optimal:*

$$\mathbb{E}_\pi \left[ \tfrac{1}{2} (y_{N+1} - \widehat{y}_{N+1})^2 \right] \leq \mathsf{BayesRisk}_\pi + \mathcal{O}\left( (\log K/N)^{1/3} \right). \tag{8}$$

In particular, Theorem 12 applies in the *proportional setting* where $N, d$ are large and $N/d = \Theta(1)$ [22], in which case $\mathsf{BayesRisk}_\pi = \Theta(1)$, and thus the transformer achieves vanishing excess risk relative to the Bayes risk at large $N$.

This substantially strengthens the results of Akyürek et al. [2], who empirically find that transformers can achieve nearly Bayes risk under any *fixed* noise level. By contrast, Theorem 12 shows that a *single* transformer can achieve nearly Bayes risk even under a mixture of $K$ noise levels, with quantitative guarantees. Also, our proof in fact gives a stronger guarantee: The transformer approaches the *individual Bayes risks on all $K$ noise levels simultaneously* (in addition to the overall Bayes risk for $k \sim \Lambda$ as in Theorem 12). We demonstrate this empirically in Section 6 (cf. Figure 3b & 2).

**Exact Bayes predictor vs. Post-ICL validation mechanism**   As $\mathsf{BayesRisk}_\pi$ is the theoretical lower bound for the risk of any possible ICL algorithm, Theorem 12 implies that our transformer performs similarly as the exact Bayes estimator[5]. Notice that our construction builds on the (generic) post-ICL validation mechanism, rather than a direct attempt of approximating the exact Bayes predictor, whose structure may vary significantly case-by-case. This highlights post-ICL validation as a promising mechanism for approximating the Bayes predictor on broader classes of problems beyond noisy linear models, which we leave as future work.

**Generalized linear models with adaptive link function selection**   As another example of the post-ICL validation mechanism, we construct a transformer that can learn a generalized linear model with adaptively chosen link function for the particular ICL instance; see Theorem J.2.

### 4.2 Pre-ICL testing mechanism

In our second mechanism, pre-ICL testing, the transformer runs a *distribution testing* procedure on the input sequence to determine the right ICL algorithm to use. While the test (and thus the mechanism itself) could in principle be general, we focus on cases where the test amounts to computing some simple summary statistics of the input sequence.

To showcase pre-ICL testing, we consider the toy problem of selecting between in-context regression and in-context classification, by running the following *binary type check* on the input labels $\{y_i\}_{i \in [N]}$.

$$\Psi^{\mathsf{binary}}(\mathcal{D}) = \frac{1}{N} \sum_{i=1}^N \psi(y_i), \quad \psi(y) := \begin{cases} 1, & y \in \{0, 1\}, \\ 0, & y \notin [-\varepsilon, \varepsilon] \cup [1 - \varepsilon, 1 + \varepsilon], \\ \text{linear interpolation}, & \text{otherwise}. \end{cases}$$

---

[5] By the Bayes risk decomposition for square loss, (8) implies that $\mathbb{E}[(\widehat{y}_{N+1} - \widehat{y}_{N+1}^{\mathsf{Bayes}})^2] \leq \mathcal{O}((\log K/N)^{1/3})$.

**Lemma 13.** *There exists a single attention layer with 6 heads that implements $\Psi^{\text{binary}}$ exactly.*

Using this test, we construct a transformer that performs logistic regression when labels are binary, and linear regression with high probability if the label admits a continuous distribution.

**Proposition 14** (Adaptive regression or classification; Informal version of Proposition I.4). *There exists a transformer with $\mathcal{O}(\log(1/\varepsilon))$ layers such that the following holds: On any $\mathcal{D}$ such that $y_i \in \{0, 1\}$, it outputs $\widehat{y}_{N+1}$ that $\varepsilon$-approximates the prediction of in-context logistic regression.*

*By contrast, for any distribution $\mathsf{P}$ whose marginal distribution of $y$ is not concentrated around $\{0, 1\}$, with high probability (over $\mathcal{D}$), $\widehat{y}_{N+1}$ $\varepsilon$-approximates the prediction of in-context least squares.*

The proofs can be found in Appendix I.3. We additionally show that transformers can implement more complex tests such as a *linear correlation test*, which can be useful in certain scenarios such as "confident linear regression" (predict only when the signal-to-noise ratio is high); see Appendix I.4.

## 5  Analysis of pretraining

Building on the expressivity results in Section 3 & 4, we provide the first line of polynomial sample complexity results for *pretraining* transformers to perform ICL (including with in-context algorithm selection). We begin by providing a generic generalization guarantee for pretraining transformers.

Consider the pretraining ERM problem (TF-ERM), which minimizes the pretraining risk $\widehat{L}_{\text{icl}}(\cdot)$ over $n$ pretraining sequences. Let $L_{\text{icl}}(\cdot)$ denote the corresponding population risk.

**Theorem 15** (Generalization of transformers; Informal version of Theorem K.1). *The solution $\widehat{\boldsymbol{\theta}}$ to (TF-ERM) over transformers with $L$ layers, $M$ heads per layer, and hidden dimension $D'$ satisfies*

$$L_{\text{icl}}(\widehat{\boldsymbol{\theta}}) \leq \inf_{\boldsymbol{\theta}} L_{\text{icl}}(\boldsymbol{\theta}) + \widetilde{\mathcal{O}}\left(\sqrt{\frac{L^2(MD^2 + DD')}{n}}\right).$$

Theorem 15 builds on standard uniform concentration analysis via chaining (Proposition B.4). Combining Theorem 15 with the in-context linear regression construction in Theorem 4 gives the following end-to-end result on the excess in-context prediction risk of trained transformers.

**Theorem 16** (Pretraining transformers for in-context linear regression; Informal version of Theorem K.2). *Under Assumption A and $N \geq \widetilde{\mathcal{O}}(d)$, the solution $\widehat{\boldsymbol{\theta}}$ to (TF-ERM) with $L = \mathcal{O}(\kappa \log(\kappa N/\sigma))$ layers, $M = 3$ heads, $D' = 0$ (attention-only as in Theorem 4) achieves small excess ICL risk over the best linear predictor $\mathbf{w}_{\mathsf{P}}^\star := \mathbb{E}_{\mathsf{P}}[\mathbf{x}\mathbf{x}^\top]^{-1}\mathbb{E}_{\mathsf{P}}[\mathbf{x}y]$ for each $\mathsf{P}$:*

$$L_{\text{icl}}(\widehat{\boldsymbol{\theta}}) - \mathbb{E}_{\mathsf{P}\sim\pi}\mathbb{E}_{(\mathbf{x},y)\sim\mathsf{P}}\left[\frac{1}{2}(y - \langle\mathbf{w}_{\mathsf{P}}^\star, \mathbf{x}\rangle)^2\right] \leq \widetilde{\mathcal{O}}\left(\sqrt{\frac{\kappa^2 d^2}{n}} + \frac{d\sigma^2}{N}\right),$$

See Appendix K.2 for similar results in several additional settings.

## 6  Experiments

We test our theory by studying the ICL and in-context algorithm selection capabilities of transformers, using the encoder-based architecture in our theoretical constructions (Definition 3). Due to limited space, additional experimental details can be found in Appendix M.1. Results with a decoder architecture as in [31, 47] (including the setup of Figure 2) can be found in Appendix M.2.

**Training data distributions and evaluation**  We train a 12-layer transformer, with two modes for the training sequence (instance) distribution $\pi$. In the "base" mode, similar to [31, 2, 86, 47], we sample the training instances from *one* of the following base distributions (tasks), where we first sample $\mathsf{P} = \mathsf{P}_{\mathbf{w}_\star} \sim \pi$ by sampling $\mathbf{w}_\star \sim \mathsf{N}(\mathbf{0}, \mathbf{I}_d/d)$, and then sample $\{(\mathbf{x}_i, y_i)\}_{i\in[N+1]} \overset{\text{iid}}{\sim} \mathsf{P}_{\mathbf{w}_\star}$ as $\mathbf{x}_i \overset{\text{iid}}{\sim} \mathsf{N}(\mathbf{0}, \mathbf{I}_d)$, and $y_i$ from one of the following models studied in Section 3:

1. Linear model: $y_i = \langle\mathbf{w}_\star, \mathbf{x}_i\rangle$;
2. Noisy linear model: $y_i = \langle\mathbf{w}_\star, \mathbf{x}_i\rangle + \sigma z_i$, where $\sigma > 0$ is a fixed noise level, and $z_i \sim \mathsf{N}(0, 1)$.

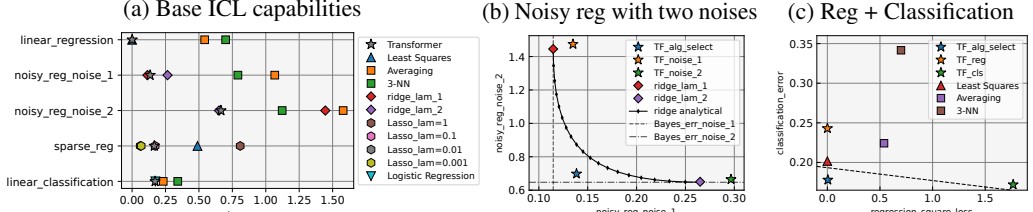

Figure 3: ICL capabilities of the transformer architecture used in our theoretical constructions. *(a)* On five representative base tasks, transformers approximately match the best baseline algorithm for each task, when pretrained on the corresponding task. *(b,c)* A **single transformer** `TF_alg_select` **simultaneously approaches the performance of the strongest baseline algorithm** on two separate tasks: *(b)* noisy linear regression with two different noise levels $\sigma \in \{0.1, 0.5\}$, and *(c)* adaptively selecting between regression and classification.

3. Sparse linear model: $y_i = \langle \mathbf{w}_\star, \mathbf{x}_i \rangle$ with $\|\mathbf{w}_\star\|_0 \leq s$, where $s < d$ is a fixed sparsity level, and in this case we sample $\mathbf{w}_\star$ from a special prior supported on $s$-sparse vectors;

4. Linear classification model: $y_i = \mathrm{sign}(\langle \mathbf{w}_\star, \mathbf{x}_i \rangle)$.

These base tasks have been empirically investigated by Garg et al. [31], though we remark that our architecture (used in our theory) differs from theirs in several aspects, such as encoder-based architecture instead of decoder-based, and ReLU activation instead of softmax. All experiments use $d = 20$. We choose $\sigma \in \{\sigma_1, \sigma_2\} = \{0.1, 0.5\}$ and $N = 20$ for noisy linear regression, $s = 3$ and $N = 10$ for sparse linear regression, and $N = 40$ for linear regression and linear classification.

In the "mixture" mode, $\pi$ is the uniform *mixture of two or more base distributions*. We consider two representative mixture modes studied in Section 4:

- Linear model + linear classification model;
- Noisy linear model with four noise levels $\sigma \in \{0.1, 0.25, 0.5, 1\}$.

Transformers trained with the mixture mode will be evaluated on *multiple* base distributions simultaneously. When the base distributions are sufficiently diverse, a transformer performing well on all of them will *likely* be performing some level of in-context algorithm selection. We evaluate transformers against standard machine learning algorithms in context (for each task respectively) as baselines.

**Results** Figure 3a shows the ICL performance of transformers on five base tasks, within each the transformer is trained on the same task. Transformers match the best baseline algorithm in four out of the five cases, except for the sparse regression task where the Transformer still outperforms least squares and matches Lasso with some choices of $\lambda$ (thus utilizing sparsity to some extent). This demonstrates the strong ICL capability of the transformer architecture considered in our theory.

Figure 3b & 3c examine the in-context algorithm selection capability of transformers, on noisy linear regression with two different noise levels (Figure 3b), and regression + classification (Figure 3c). In both figures, the transformer trained in the mixture mode (`TF_alg_select`) approaches the best baseline algorithm on both tasks simultaneously. By contrast, transformers trained in the base mode for one of the tasks perform well on that task but behave suboptimally on the other task as expected. The existence of `TF_alg_select` showcases a single transformer that performs well on multiple tasks simultaneously (and thus has to perform in-context algorithm selection to some extent), supporting our theoretical results in Section 4.

## 7 Conclusion

This work shows that transformers can perform complex in-context learning procedures with strong in-context algorithm selection capabilties, by both explicit theoretical constructions and experiments. We believe our work opens up many exciting directions, such as (1) more mechanisms for in-context algorithm selection; (2) Bayes-optimal ICL on other problems by either the post-ICL validation mechanism or new approaches; (3) understanding the internal workings of transformers performing in-context algorithm selection; (4) other mechanisms for implementing complex ICL procedures beyond in-context algorithm selection; (5) further statistical analyses, e.g. of pretraining. Besides, this work focuses on the transformer architecture; alternative sequence-to-sequence architectures (such as RNNs) are beyond our scope but would be interesting directions for future work.

## Acknowledgment

The authors would like to thank Tengyu Ma and Jason D. Lee for the many insightful discussions. S. Mei is supported in part by NSF DMS-2210827 and NSF CCF-2315725.

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

## A Related work

**In-context learning** The in-context learning (ICL) capability of large language models (LLMs) has gained significant attention since demonstrated on GPT-3 Brown et al. [12]. A number of subsequent empirical studies have contributed to a better understanding of the capabilities and limitations of ICL in LLM systems, which include but are not limited to [49, 55, 56, 50, 100, 74, 72, 28, 45, 91]. For an overview of ICL, see the survey by Dong et al. [24] which highlights some key findings and advancements in this direction.

A line of recent work investigates why and how LLMs perform ICL [94, 31, 86, 2, 18, 32, 47, 70]. In particular, Xie et al. [94] propose a Bayesian inference framework explaining how ICL works despite formatting differences between training and inference distributions. Garg et al. [31] show empirically that transformers could be trained from scratch to perform ICL of linear models, sparse linear models, two-layer neural networks, and decision trees. Li et al. [47] analyze the generalization error of trained ICL transformers from a stability viewpoint. They also experimentally show that transformers could perform "in-context model selection" (conceptually similar to in-context algorithm selection considered in this work) in specific tasks and presented related theoretical hypotheses. However, they do not provide concrete mechanisms or constructions for in-context model selection. A recent work [99] shows that pretrained transformers can perform Bayesian inference in latent variable models, which may also be interpreted as a mechanism for ICL. Our experimental findings extend these results by unveiling and demonstrating the in-context algorithm selection capabilities of transformers.

Closely related to our theoretical results are [86, 2, 18, 32], which show (among many things) that transformers can perform ICL by simulating gradient descent. However, these results do not provide quantitative error bounds for simulating multi-step gradient descent, and only handle linear regression models or their simple variants. Among these works, Akyürek et al. [2] showed that transformers can implement learning algorithms for linear models based on gradient descent and closed-form ridge regression; it also presented preliminary evidence that learned transformers perform ICL similar to Bayes-optimal ridge regression. Our work builds upon and substantially extends this line of work by (1) providing a more efficient construction for in-context gradient descent; (2) providing an end-to-end theory with additional results for pretraining and statistical power; (3) analyzing a broader spectrum of ICL algorithms, including least squares, ridge regression, Lasso, convex risk minimization for generalized linear models, and gradient descent on two-layer neural networks; and (4) constructing more complex ICL procedures using in-context algorithm selection.

When in-context data are generated from a prior, the Bayes risk is a theoretical lower bound for the risk of any possible ICL algorithm, including transformers. Xie et al. [94], Akyürek et al. [2] observe

that learned transformers behave closely to the Bayes predictor on a variety of tasks such as hidden Markov models [94] and noisy linear regression with a fixed noise level [2, 47]. Using the in-context algorithm selection mechanism (more precisely the post-ICL validation mechanism), we show that transformers can perform nearly-Bayes optimal ICL in noisy linear models with mixed noise levels (a strictly more challenging task than considered in [2, 47]), with both concrete theoretical guarantees (Section 4.1.1) and empirical evidence (Figure 2 & 3b). Complementary to these works, a line of work on "prior-data fitted networks" [59, 60, 35] also empirically demonstrates the Bayesian optimality of transformers in various settings. Our expressivity results support these empirical findings and are applicable beyond the Bayesian setting, e.g. for providing frequentist in-context prediction guarantees for transformers.

**Transformers and its theory**    The transformer architecture, introduced by [84], has revolutionized natural language processing and been adopted in most of the recently developed large language models such as BERT and GPT [68, 21, 12]. Broadly, transformers have demonstrated remarkable performance in many other fields of artificial intelligence such as computer vision, speech, graph processing, reinforcement learning, and biological applications [23, 25, 51, 69, 96, 16, 41, 73, 65, 14]. Towards a better theoretical understanding, recent work has studied the capabilities [97, 67, 37, 95, 11, 98, 48], limitations [33, 10], and internal workings [28, 79, 92, 27, 64] of transformers.

We remark that the transformer architecture used in our theoretical constructions differs from the standard one by replacing the softmax activation (in the attention layers) with a (normalized) ReLU function. Transformers with ReLU activations is experimentally studied in the recent work of Shen et al. [78], who find that they perform as well as the standard softmax activation in many NLP tasks.

**Meta-learning**    Training models (such as transformers) to perform ICL can be viewed as an approach for the broader problem of learning-to-learn or meta-learning [77, 61, 81]. A number of other approaches has been studied extensively for this problem, including (and not limited to) training a meta-learner on how to update the parameters of a downstream learner [9, 46], learning parameter initializations that quickly adapt to downstream tasks [29, 71], learning latent embeddings that allow for effective similarity search [80]. Most relevant to the ICL setting are approaches that directly take as input examples from a downstream task and a query input and produce the corresponding output [34, 58, 75, 44]. For a comprehensive overview, see the survey [36].

Theoretical aspects of meta-learning have received significant recent interest [7, 52, 26, 83, 19, 30, 43, 40, 88, 20, 5, 76, 17, 101]. In particular, [52, 26, 83] analyzed the benefit of multi-task learning through a representation learning perspective, and [88, 20, 5, 76, 101] studied the statistical properties of learning the parameter initialization for downstream tasks.

**Techniques**    We build on various existing techniques from the statistics and learning theory literature to establish our approximation and generalization guarantees for transformers. For the approximation component, we rely on a technical result of Bach [4] on the approximation power of ReLU networks. We use this result to show that transformers can approximate gradient descent (GD) on a broad range of loss functions, substantially extending the results of [86, 2, 18] who primarily consider the square loss. The recent work of Giannou et al. [32] also approximates GD with general loss functions by transformers, though using a different technique of forcing the softmax activations to act as sigmoids. Our analyses of Lasso and generalized linear models build on [87, 62, 1, 54]. Our generalization bound for transformers (used in our pretraining results) build on a chaining argument [87].

# B    Technical tools

**Additional notation for proofs**    We say a random variable $X$ is $\sigma^2$-sub-Gaussian (or $\mathrm{SG}(\sigma)$ interchangeably) if $\mathbb{E}[\exp(X^2/\sigma^2)] \leq 2$. A random vector $\mathbf{x} \in \mathbb{R}^d$ is $\sigma^2$-sub-Gaussian if $\langle \mathbf{v}, \mathbf{x} \rangle$ is $\sigma^2$-sub-Gaussian for all $\|\mathbf{v}\|_2 = 1$. A random variable $X$ is $K$-sub-Exponential (or $\mathrm{SE}(K)$ interchangeably) if $\mathbb{E}[\exp(|X|/K)] \leq 2$.

## B.1    Concentration inequalities

**Lemma B.1.** *Let $\boldsymbol{\beta} \sim \mathsf{N}(\mathbf{0}, \mathbf{I}_d/d)$. Then we have*

$$\mathbb{P}\Big(\|\boldsymbol{\beta}\|_2^2 \geq (1+\delta)^2\Big) \leq e^{-d\delta^2/2}.$$

**Lemma B.2** (Theorem 6.1 of [87]). *Let $X = [X_{ij}] \in \mathbb{R}^{n \times d}$ be a Gaussian random matrix with $X_{ij} \sim \mathsf{N}(0,1)$. Let $\sigma_{\min}(X)$ and $\sigma_{\min}(X)$ be the minimum and maximum singular value of $X$, respectively. Then we have*

$$\mathbb{P}\Big(\sigma_{\max}(X)/\sqrt{n} \geq 1 + \sqrt{d/n} + \delta\Big) \leq e^{-n\delta^2/2},$$

$$\mathbb{P}\Big(\sigma_{\min}(X)/\sqrt{n} \leq 1 - \sqrt{d/n} - \delta\Big) \leq e^{-n\delta^2/2}.$$

The following lemma is a standard result of covariance concentration, see e.g. [85, Theorem 4.6.1].

**Lemma B.3.** *Suppose that $\mathbf{x}_1, \cdots, \mathbf{x}_N$ are independent $d$-dimensional $K$-sub-Gaussian random vectors. Then as long as $N \geq C_0 d$, with probability at least $1 - \exp(-N/C_0)$ we have*

$$\left\| \frac{1}{N} \sum_{i=1}^{N} \mathbf{x}_i \mathbf{x}_i^\top \right\|_{\mathrm{op}} \leq 8K^2,$$

*where $C_0$ is a universal constant.*

**Lemma B.4.** *For random matrix $\mathbf{X} = [x_{ij}] \in \mathbb{R}^{N \times d}$ with $x_{ij} \overset{\mathrm{iid}}{\sim} \mathsf{N}(0,1)$ and $\boldsymbol{\varepsilon} = [\varepsilon_i] \in \mathbb{R}^N$ with $\varepsilon_i \overset{\mathrm{iid}}{\sim} \mathsf{N}(0, \sigma^2)$, it holds that*

$$\mathbb{P}\Big( \left\| \mathbf{X}^\top \boldsymbol{\varepsilon} \right\|_\infty \geq \sqrt{8N\sigma^2 \log(2d/\delta)} \Big) \leq \delta + \exp(-N/2).$$

*Proof.* We consider $\mathbf{u}_j := [x_{ij}]_i \in \mathbb{R}^N$, then $\left\| \mathbf{X}^\top \boldsymbol{\varepsilon} \right\|_\infty = \max_{i \in [d]} |\langle \mathbf{u}_j, \boldsymbol{\varepsilon} \rangle|$. Notice that the random variables $\langle \mathbf{u}_1, \boldsymbol{\varepsilon} \rangle, \cdots, \langle \mathbf{u}_d, \boldsymbol{\varepsilon} \rangle$ are independent $\mathsf{N}(0, \|\boldsymbol{\varepsilon}\|_2^2)$, and hence

$$\mathbb{P}\left( \max_{i \in [d]} |\langle \mathbf{u}_j, \boldsymbol{\varepsilon} \rangle| \geq t \,\Big|\, \boldsymbol{\varepsilon} \right) \leq 2d \exp\left( -\frac{t^2}{2 \|\boldsymbol{\varepsilon}\|_2^2} \right).$$

Further, by Lemma B.1, $\mathbb{P}(\|\boldsymbol{\varepsilon}\|_2 \geq 2\sigma\sqrt{N}) \leq \exp(-N/2)$. Taking $t = \sqrt{8N\sigma^2 \log(2d/\delta)}$ completes the proof. $\square$

## B.2 Approximation theory

For any signed measure $\mu$ over a space $\mathcal{W}$, let $\mathrm{TV}(\mu) := \int_{\mathcal{W}} |d\mu(\mathbf{w})| \in [0, \infty]$ denote its total measure. Recall $\sigma(\cdot) = \mathrm{ReLU}(\cdot)$ is the standard relu activation, and $\mathsf{B}_\infty^k(R) = [-R, R]^k$ denotes the standard $\ell_\infty$ ball in $\mathbb{R}^k$ with radius $R > 0$.

**Definition B.1** (Sufficiently smooth $k$-variable function). *We say a function $g : \mathbb{R}^k \to \mathbb{R}$ is $(R, C_\ell)$-smooth, if for $s = \lceil (k-1)/2 \rceil + 2$, $g$ is a $C^s$ function on $\mathsf{B}_\infty^k(R)$, and*

$$\sup_{\mathbf{z} \in \mathsf{B}_\infty^k(R)} \left\| \nabla^i g(\mathbf{z}) \right\|_\infty = \sup_{\mathbf{z} \in \mathsf{B}_\infty^k(R)} \max_{j_1, \ldots, j_i \in [k]} |\partial_{x_{j_1} \ldots x_{j_i}} g(\mathbf{x})| \leq L_i$$

*for all $i \in \{0, 1, \ldots, s\}$, with $\max_{0 \leq i \leq s} L_i R^i \leq C_\ell$.*

The following result for expressing smooth functions as a random feature model with relu activation is adapted from Bach [4, Proposition 5].

**Lemma B.5** (Expressing sufficiently smooth functions by relu random features). *Suppose function $g : \mathcal{R}^k \to \mathbb{R}$ is $(R, C_\ell)$ smooth. Then there exists a signed measure $\mu$ over $\mathcal{W} = \{\mathbf{w} \in \mathbb{R}^{k+1} : \|\mathbf{w}\|_1 = 1\}$ such that*

$$g(\mathbf{x}) = \int_{\mathcal{W}} \frac{1}{R} \sigma(\mathbf{w}^\top [\mathbf{x}; R]) d\mu(\mathbf{w}), \qquad \forall \mathbf{x} \in \mathcal{X}$$

*and $\mathrm{TV}(\mu) \leq C(k) C_\ell$, where $C(k) < \infty$ is a constant that only depends on $k$.*

**Lemma B.6** (Uniform finite-neuron approximation). *Let $\mathcal{X}$ be a space equipped with a distance function $d_{\mathcal{X}}(\cdot, \cdot) : \mathcal{X} \times \mathcal{X} \to \mathbb{R}_{\geq 0}$. Suppose function $g : \mathcal{X} \to \mathbb{R}$ is given by*

$$g(\mathbf{x}) = \int_{\mathcal{W}} \phi(\mathbf{x}; \mathbf{w}) d\mu(\mathbf{w}),$$

where $\phi(\cdot;\cdot) : \mathcal{X} \times \mathcal{W} \to [-B, B]$ is $L$-Lipschitz (in $d_{\mathcal{X}}$) in the first argument, and $\mu$ is a signed measure over $\mathcal{W}$ with finite total measure $A = \mathrm{TV}(\mu) < \infty$. Then for any $\varepsilon > 0$, there exists $\alpha_1, \cdots, \alpha_K \in \{\pm 1\}$, $\mathbf{w}_1, \cdots, \mathbf{w}_K \in \mathcal{W}$ with $K = \mathcal{O}(A^2 B^2 \log \mathcal{N}(\mathcal{X}, d_{\mathcal{X}}, \frac{\varepsilon}{3AL})/\varepsilon^2)$, such that

$$\sup_{\mathbf{x} \in \mathcal{X}} \left| g(\mathbf{x}) - \frac{A}{K} \sum_{i=1}^{K} \alpha_i \phi(\mathbf{x}; \mathbf{w}_i) \right| \leq \varepsilon,$$

where $\mathcal{N}(\mathcal{X}, d_{\mathcal{X}}, \frac{\varepsilon}{3AL})$ denotes the $(\frac{\varepsilon}{3AL})$-covering number of $\mathcal{X}$ in $d_{\mathcal{X}}$.

*Proof.* Let $\alpha(\mathbf{w}) := \mathrm{sign}(d\mu(\mathbf{w})) \in \{\pm 1\}$ denote the sign of the density $d\mu(\mathbf{w})$. We have

$$g(\mathbf{x}) = A \int_{\mathcal{W}} \alpha(\mathbf{w}) \phi(\mathbf{x}; \mathbf{w}) \times \frac{|d\mu(\mathbf{w})|}{A}. \tag{9}$$

Note that $|d\mu(\mathbf{w})|/A$ is the density of a probability distribution over $\mathcal{W}$. Thus for any $\mathbf{x} \in \mathcal{X}$, as long as $K \geq \mathcal{O}(A^2 B^2 \log(1/\delta)/\varepsilon^2)$, we can sample $\mathbf{w}_1, \ldots, \mathbf{w}_K \overset{\mathrm{iid}}{\sim} |d\mu(\cdot)|/A$, and obtain by Hoeffding's inequality that with probability at least $1 - \delta$,

$$\left| g(\mathbf{x}) - \frac{A}{K} \sum_{i=1}^{K} \alpha(\mathbf{w}_i) \phi(\mathbf{x}; \mathbf{w}_i) \right| \leq \varepsilon.$$

Let $\mathcal{N}(\frac{\varepsilon}{3AL}) := \mathcal{N}(\mathcal{X}, d_{\mathcal{X}}, \frac{\varepsilon}{3AL})$ for shorthand. By union bound, as long as $K \geq \mathcal{O}(A^2 B^2 \log(\mathcal{N}(\frac{\varepsilon}{3AL})/\delta)/\varepsilon^2)$, we have with probability at least $1 - \delta$ that for every $\widehat{\mathbf{x}}$ in the covering set corresponding to $\mathcal{N}(\frac{\varepsilon}{3AL})$,

$$\left| g(\widehat{\mathbf{x}}) - \frac{A}{K} \sum_{i=1}^{K} \alpha(\mathbf{w}_i) \phi(\widehat{\mathbf{x}}; \mathbf{w}_i) \right| \leq \varepsilon/3.$$

Taking $\delta = 1/2$ (for which $K = \mathcal{O}(A^2 B^2 \log \mathcal{N}(\frac{\varepsilon}{3AL})/\varepsilon^2)$), by the probabilistic method, there exists a deterministic set $\{\mathbf{w}_i\}_{i \in [K]} \subset \mathcal{W}$ and $\{\alpha_i := \alpha(\mathbf{w}_i)\}_{i \in [K]} \in \{\pm 1\}$ such that the above holds.

Next, note that both $g$ (by (9)) and the function $\mathbf{x} \mapsto \frac{A}{K} \sum_{i=1}^{K} \alpha(\mathbf{w}_i) \phi(\mathbf{x}; \mathbf{w}_i)$ are $(AL)$-Lipschitz. Therefore, for any $\mathbf{x} \in \mathcal{X}$, taking $\widehat{\mathbf{x}}$ to be the point in the covereing set with $d_{\mathcal{X}}(\mathbf{x}, \widehat{\mathbf{x}}) \leq \frac{\varepsilon}{3AL}$, we have

$$\left| g(\mathbf{x}) - \frac{A}{K} \sum_{i=1}^{K} \alpha(\mathbf{w}_i) \phi(\mathbf{x}; \mathbf{w}_i) \right|$$

$$\leq |g(\mathbf{x}) - g(\widehat{\mathbf{x}})| + \left| g(\widehat{\mathbf{x}}) - \frac{A}{K} \sum_{i=1}^{K} \alpha(\mathbf{w}_i) \phi(\widehat{\mathbf{x}}; \mathbf{w}_i) \right| + \left| \frac{A}{K} \sum_{i=1}^{K} \alpha(\mathbf{w}_i) \phi(\widehat{\mathbf{x}}; \mathbf{w}_i) - \frac{A}{K} \sum_{i=1}^{K} \alpha(\mathbf{w}_i) \phi(\mathbf{x}; \mathbf{w}_i) \right|$$

$$\leq AL \cdot \frac{\varepsilon}{3AL} + \frac{\varepsilon}{3} + AL \cdot \frac{\varepsilon}{3AL} = \varepsilon.$$

This proves the lemma. $\qquad \square$

**Proposition B.1** (Approximating smooth $k$-variable functions). *For any $\varepsilon_{\mathrm{approx}} > 0$, $R \geq 1$, $C_\ell > 0$, we have the following: Any $(R, C_\ell)$-smooth function (Definition B.1) $g : \mathbb{R}^k \to \mathbb{R}$ is $(\varepsilon_{\mathrm{approx}}, R, M, C)$-approximable by sum of relus (Definition D.1) with $M \leq C(k) C_\ell^2 \log(1 + C_\ell/\varepsilon_{\mathrm{approx}})/\varepsilon_{\mathrm{approx}}^2)$ and $C \leq C(k) C_\ell$, where $C(k) > 0$ is a constant that depends only on $k$. In other words, there exists*

$$f(\mathbf{z}) = \sum_{m=1}^{M} c_m \sigma(\mathbf{a}_m^\top [\mathbf{z}; 1]) \quad \text{with} \quad \sum_{m=1}^{M} |c_m| \leq C, \qquad \max_{m \in [M]} \|\mathbf{a}_m\|_1 \leq 1,$$

*such that $\sup_{\mathbf{z} \in [-R, R]^k} |f(\mathbf{z}) - g(\mathbf{z})| \leq \varepsilon_{\mathrm{approx}}$.*

*Proof.* As function $g : \mathsf{B}_\infty^k(R) \to \mathbb{R}$ is $(R, C_\ell)$-smooth, we can apply Lemma B.5 to obtain that there exists a signed measure $\mu$ over $\mathcal{W} := \{\mathbf{w} \in \mathbb{R}^{k+1} : \|\mathbf{w}\|_1 \leq 1\}$ such that

$$g(\mathbf{z}) = \int_\mathcal{W} \frac{1}{R}\sigma(\mathbf{w}^\top[\mathbf{z}; R])d\mu(\mathbf{w}), \qquad \forall \mathbf{z} \in [-R, R]^k,$$

and $A = \mathrm{TV}(\mu) \leq C(k)C_\ell$ where $C(k) > 0$ denotes a constant depending only on $k$.

We now apply Lemma B.6 to approximate the above random feature by finitely many neurons. Let $\mathbf{x} := [\mathbf{z}; R] \in \mathcal{X} := [-R, R]^k \times \{R\}$. Then, the function $\phi(\mathbf{x}; \mathbf{w}) := \frac{1}{R}\sigma(\mathbf{w}^\top \mathbf{x}) = \sigma(\frac{1}{R}\mathbf{w}^\top[\mathbf{z}; R])$ is bounded by $B = 1$ and $(1/R)$-Lipschitz in $\mathbf{x}$ (in the standard $\ell_\infty$-distance). Further, we have $\log \mathcal{N}(\mathcal{X}, \|\cdot - \cdot\|_\infty, \frac{\varepsilon_{\mathrm{approx}}}{3A/R}) \leq \mathcal{O}(k\log(1 + A/\varepsilon_{\mathrm{approx}}))$. We can thus apply Lemma B.6 to obtain that, for

$$M = \mathcal{O}\big(kA^2 \log(1 + A/\varepsilon_{\mathrm{approx}})/\varepsilon_{\mathrm{approx}}^2\big) = C(k)C_\ell^2 \log(1 + C_\ell/\varepsilon_{\mathrm{approx}})/\varepsilon_{\mathrm{approx}}^2,$$

there exists $\boldsymbol{\alpha} = \{\alpha_m\}_{m \in [M]} \subset \{\pm 1\}$ and $\mathbf{W} = \{\mathbf{w}_m\}_{m \in [M]} \subset \mathcal{W} = \{\mathbf{w} \in \mathbb{R}^{k+1} : lone\mathbf{w} = 1\}$ such that

$$\sup_{\mathbf{z} \in [-R,R]^2} |g(\mathbf{z}) - f_{\boldsymbol{\alpha}, \mathbf{W}}(\mathbf{z})| \leq \varepsilon_{\mathrm{approx}},$$

where (recalling $\mathbf{z} = [s; t]$)

$$f_{\boldsymbol{\alpha}, \mathbf{W}}(\mathbf{z}) = \frac{A}{M}\sum_{m=1}^M \alpha_m \sigma\left(\frac{1}{R}\mathbf{w}_m^\top[\mathbf{z}; R]\right) = \sum_{m=1}^M \underbrace{\frac{A\alpha_m}{M}}_{c_m} \sigma\left(\underbrace{\left[\frac{1}{R}\mathbf{w}_{m,1:k}; w_{m,k+1}\right]^\top}_{\mathbf{a}_m^\top}[\mathbf{z}; 1]\right).$$

Note that we have $\sum_{m=1}^M |c_m| = A \leq C(k)C_\ell$, and $\|\mathbf{a}_m\|_1 \leq \|\mathbf{w}_m\|_1 = 1$. This is the desired result. $\qquad\square$

## B.3 Optimization

The following convergence result for minimizing a smooth and strongly convex function is standard from the convex optimization literature, see e.g. Bubeck [13, Theorem 3.10].

**Proposition B.2** (Gradient descent for smooth and strongly convex functions). *Suppose $L : \mathbb{R}^d \to \mathbb{R}$ is $\alpha$-strongly convex and $\beta$-smooth for some $0 < \alpha \leq \beta$. Then, the gradient descent iterates $\mathbf{w}_{\mathrm{GD}}^{t+1} := \mathbf{w}_{\mathrm{GD}}^t - \eta\nabla L(\mathbf{w}_{\mathrm{GD}}^t)$ with learning rate $\eta = 1/\beta$ and initialization $\mathbf{w}_{\mathrm{GD}}^0 \in \mathbb{R}^d$ satisfies for any $t \geq 1$,*

$$\left\|\mathbf{w}_{\mathrm{GD}}^t - \mathbf{w}^\star\right\|_2^2 \leq \exp\left(-t/\kappa\right) \cdot \left\|\mathbf{w}_{\mathrm{GD}}^0 - \mathbf{w}^\star\right\|_2^2,$$

$$L(\mathbf{w}_{\mathrm{GD}}^t) - L(\mathbf{w}^\star) \leq \frac{\beta}{2}\exp\left(-t/\kappa\right) \cdot \left\|\mathbf{w}_{\mathrm{GD}}^0 - \mathbf{w}^\star\right\|_2^2,$$

*where $\kappa := \beta/\alpha$ is the condition number of $L$, and $\mathbf{w}^\star := \arg\min_{\mathbf{w} \in \mathbb{R}^d} L(\mathbf{w})$ is the minimizer of $L$.*

The following convergence result of proximal gradient descent (PGD) on convex composite minimization problem is also standard, see e.g. [8].

**Proposition B.3** (Proximal gradient descent for convex function). *Suppose $L = f + h$, $f : \mathbb{R}^d \to \mathbb{R}$ is convex and $\beta$-smooth for some $\beta > 0$, $h : \mathbb{R}^d \to \mathbb{R}$ is a simple convex function. Then, the proximal gradient descent iterates $\mathbf{w}_{\mathrm{PGD}}^{t+1} := \mathbf{prox}_{\eta h}(\mathbf{w}_{\mathrm{PGD}}^t - \eta\nabla f(\mathbf{w}_{\mathrm{PGD}}^t))$ with learning rate $\eta = 1/\beta$ and initialization $\mathbf{w}_{\mathrm{GD}}^0 \in \mathbb{R}^d$ satisfies the following for any $t \geq 1$:*

1. *$\{L(\mathbf{w}_{\mathrm{PGD}}^t)\}$ is a decreasing sequence.*

2. *For any minimizer $\mathbf{w}^\star \in \arg\min_{\mathbf{w} \in \mathbb{R}^d} L(\mathbf{w})$,*

$$L(\mathbf{w}_{\mathrm{GD}}^{t+1}) - L(\mathbf{w}^\star) \leq \frac{\beta}{2}\left(\left\|\mathbf{w}_{\mathrm{PGD}}^t - \mathbf{w}^\star\right\|_2^2 - \left\|\mathbf{w}_{\mathrm{PGD}}^{t+1} - \mathbf{w}^\star\right\|_2^2\right),$$

*and hence $\left\{\left\|\mathbf{w}_{\mathrm{PGD}}^t - \mathbf{w}^\star\right\|_2^2\right\}$ is also a decreasing sequence.*

3. *For $k \geq 1, t \geq 0$, it holds that*

$$L(\mathbf{w}_{\mathrm{GD}}^{t+k}) - L(\mathbf{w}^\star) \leq \frac{\beta}{2k}\left\|\mathbf{w}_{\mathrm{PGD}}^t - \mathbf{w}^\star\right\|_2^2.$$

## B.4 Uniform convergence

The following result is shown in [87, Section 5.6].

**Theorem B.1.** *Suppose that $\psi : [0, +\infty) \to [0, +\infty)$ is a convex, non-decreasing function that satisfies $\psi(x + y) \geq \psi(x)\psi(y)$. For any random variable $X$, we consider the Orlicz norm induced by $\psi$: $\|X\|_\psi := \inf \{K > 0 : \mathbb{E}\psi(|X|/K)\} \leq 1$.*

*Suppose that $\{X_\theta\}_\theta$ is a zero-mean random process indexed by $\theta \in \Theta$ such that $\|X_\theta - X_{\theta'}\|_\psi \leq \rho(\theta, \theta')$ for some metric $\rho$ on the space $\Theta$. Then it holds that*

$$\mathbb{P}\left(\sup_{\theta, \theta' \in \Theta} |X_\theta - X_{\theta'}| \leq 8(J + t)\right) \leq \frac{1}{\psi(t/D)} \quad \forall t \geq 0,$$

*where $D$ is the diameter of the metric space $(\Theta, \rho)$, and the generalized Dudley entropy integral $J$ is given by*

$$J := \int_0^D \psi^{-1}(N(\delta; \Theta, \rho))d\delta,$$

*where $N(\delta; \Theta, \rho)$ is the $\delta$-covering number of $(\Theta, \rho)$.*

As a corollary of Theorem B.1, we have the following result.

**Proposition B.4** (Uniform concentration bound by chaining)**.** *Suppose that $\{X_\theta\}_{\theta \in \Theta}$ is a zero-mean random process given by*

$$X_\theta := \frac{1}{N} \sum_{i=1}^N f(z_i; \theta) - \mathbb{E}_z[f(z; \theta)],$$

*where $z_1, \cdots, z_N$ are i.i.d samples from a distribution $\mathbb{P}_z$ such that the following assumption holds:*

(a) *The index set $\Theta$ is equipped with a distance $\rho$ and diameter $D$. Further, assume that for some constant $A$, for any ball $\Theta'$ of radius $r$ in $\Theta$, the covering number admits upper bound $\log N(\delta; \Theta', \rho) \leq d \log(2Ar/\delta)$ for all $0 < \delta \leq 2r$.*

(b) *For any fixed $\theta \in \Theta$ and $z$ sampled from $\mathbb{P}_z$, the random variable $f(z; \theta)$ is a $\mathrm{SG}(B^0)$-sub-Gaussian random variable.*

(c) *For any $\theta, \theta' \in \Theta$ and $z$ sampled from $\mathbb{P}_z$, the random variable $f(z; \theta) - f(z; \theta')$ is a $\mathrm{SG}(B^1 \rho(\theta, \theta'))$-sub-Gaussian random variable.*

*Then with probability at least $1 - \delta$, it holds that*

$$\sup_{\theta \in \Theta} |X_\theta| \leq CB^0 \sqrt{\frac{d \log(2A\kappa) + \log(1/\delta)}{N}},$$

*where $C$ is a universal constant, and we denote $\kappa = 1 + B^1 D/B^0$.*

*Furthermore, if we replace the $\mathrm{SG}$ in assumption (b) and (c) by $\mathrm{SE}$, then with probability at least $1 - \delta$, it holds that*

$$\sup_{\theta \in \Theta} |X_\theta| \leq CB^0 \left[\sqrt{\frac{d \log(2A\kappa) + \log(1/\delta)}{N}} + \frac{d \log(2A\kappa) + \log(1/\delta)}{N}\right].$$

*Proof.* Fix a $D_0 \in (0, D]$ to be specified later. We pick a $(D_0/2)$-covering $\Theta_0$ of $\Theta$ so that $\log |\Theta_0| \leq d \log(2AD/D_0)$. Then, by the standard uniform covering of independent sub-Gaussian random variables, we have with probability at least $1 - \delta/2$,

$$\sup_{\theta \in \Theta_0} |X_\theta| \leq CB^0 \sqrt{\frac{d \log(2AD/D_0) + \log(2/\delta)}{N}}.$$

Assume that $\Theta_0 = \{\theta_1, \cdots, \theta_n\}$. For each $j \in [n]$, we consider $\Theta_j$ is the ball centered at $\theta_j$ of radius $D_0$ in $(\Theta, \rho)$. Then $\theta \in \Theta_j$ has diameter $D_0$ and admits covering number bound $\log \mathcal{N}(\Theta_j, \delta) \leq d \log(AD_0/\delta)$. Hence, we can apply Theorem B.1 with the process $\{X_\theta\}_{\theta \in \Theta_j}$, then

$$\psi = \psi_2, \qquad \|X_\theta - X_{\theta'}\|_\psi \leq \frac{B^1}{\sqrt{N}} \rho(\theta, \theta'),$$

and a simple calculation yields

$$\mathbb{P}\left(\sup_{\theta, \theta' \in \Theta_j} |X_\theta - X_{\theta'}| \leq C' B^1 D_0 \left(\sqrt{\frac{d \log(2A)}{N}} + t\right)\right) \leq 2 \exp(-Nt^2) \ \forall t \geq 0.$$

Therefore, we can let $t \leq \sqrt{\log(2n/\delta)/N}$ in the above inequality and taking the union bound over $j \in [n]$, and hence with probability at least $1 - \delta/2$, it holds that for all $j \in [n]$,

$$\sup_{\theta, \theta' \in \Theta_j} |X_\theta - X_{\theta'}| \leq C' B^1 D_0 \sqrt{\frac{2d \log(2AD/D_0) + \log(4/\delta)}{N}}.$$

Notice that for each $\theta \in \Theta$, there exists $j \in [n]$ such that $\theta \in \Theta_j$, and hence

$$|X_\theta| \leq |X_{\theta_j}| + |X_\theta - X_{\theta_j}|.$$

Thus, with probability at least $1 - \delta$, it holds

$$\sup_{\theta \in \Theta} |X_\theta| \leq \sup_{\theta \in \Theta_0} |X_\theta| + \sup_j \sup_{\theta \in \Theta_j} |X_\theta - X_{\theta_j}| \leq C''(B_0 + B^1 D_0) \sqrt{\frac{d \log(2AD/D_0) + \log(2/\delta)}{N}}.$$

Taking $D_0 = D/\kappa$ completes the proof of SG case.

We next consider the SE case. The idea is the same as the SG case, but in this case we need to consider the following Orlicz-norm:

$$\psi_N(t) := \exp\left(\frac{Nt^2}{t+1}\right) - 1.$$

Then Bernstein's inequality of SE random variables yields

$$\|X_\theta - X_{\theta'}\|_{\psi_N} \leq C_0 B^1 \rho(\theta, \theta')$$

for some universal constant $C_0$. Therefore, we can repeat the argument above to deduce that with probability at least $1 - \delta$, it holds

$$\sup_{\theta \in \Theta} |X_\theta| \leq C''(B_0 + B^1 D_0) \left[\sqrt{\frac{d \log(2AD/D_0) + \log(2/\delta)}{N}} + \frac{d \log(2AD/D_0) + \log(2/\delta)}{N}\right].$$

Taking $D_0 = D/\kappa$ completes the proof. $\qquad \square$

### B.5 Useful properties of transformers

The following result can be obtained immediately by "joining" the attention heads and MLP layers of two single-layer transformers.

**Proposition B.5** (Joining parallel single-layer transformers)**.** *Suppose that $P_1 : \mathbb{R}^{(D_0+D_1) \times N} \to \mathbb{R}^{D_1 \times N}, P_2 : \mathbb{R}^{(D_0+D_2) \times N} \to \mathbb{R}^{D_2 \times N}$ are two sequence-to-sequence functions that are implemented by single-layer transformers, i.e. there exists $\boldsymbol{\theta}_1, \boldsymbol{\theta}_2$ such that*

$$\mathrm{TF}_{\boldsymbol{\theta}_1} : \mathbf{H}_1 = \begin{bmatrix} \mathbf{h}_i^{(0)} \\ \mathbf{h}_i^{(1)} \end{bmatrix}_{1 \leq i \leq N} \in \mathbb{R}^{(D_0+D_1) \times N} \mapsto \begin{bmatrix} \mathbf{H}^{(0)} \\ P_1(\mathbf{H}_1) \end{bmatrix},$$

$$\mathrm{TF}_{\boldsymbol{\theta}_2} : \mathbf{H}_2 = \begin{bmatrix} \mathbf{h}_i^{(0)} \\ \mathbf{h}_i^{(2)} \end{bmatrix}_{1 \leq i \leq N} \in \mathbb{R}^{(D_0+D_2) \times N} \mapsto \begin{bmatrix} \mathbf{H}^{(0)} \\ P_2(\mathbf{H}_2) \end{bmatrix}.$$

Then, there exists $\boldsymbol{\theta}$ such that for $\mathbf{H}'$ that takes form $\mathbf{h}_i' = [\mathbf{h}_i^{(0)}; \mathbf{h}_i^{(1)}; \mathbf{h}_i^{(2)}]$, with $\mathbf{h}_i^{(0)} \in \mathbb{R}^{D_0}, \mathbf{h}_i^{(1)} \in \mathbb{R}^{D_1}, \mathbf{h}_i^{(2)} \in \mathbb{R}^{D_2}$, we have

$$\mathrm{TF}_{\boldsymbol{\theta}} : \mathbf{H}' = \begin{bmatrix} \mathbf{h}_i^{(0)} \\ \mathbf{h}_i^{(1)} \\ \mathbf{h}_i^{(2)} \end{bmatrix}_{1 \le i \le N} \in \mathbb{R}^{(D_0 + D_1 + D_2) \times N} \mapsto \begin{bmatrix} \mathbf{H}^{(0)} \\ P_1(\mathbf{H}_1) \\ P_2(\mathbf{H}_2) \end{bmatrix}.$$

Further, $\boldsymbol{\theta}$ has at most $M \le M_1 + M_2$ heads, $D' \le D_1' + D_2'$ hidden dimension in its MLP layer, and norm bound $\|\boldsymbol{\theta}\| \le \|\boldsymbol{\theta}_1\| + \|\boldsymbol{\theta}_2\|$.

**Proposition B.6** (Joining parallel multi-layer transformers). *Suppose that $P_1 : \mathbb{R}^{(D_0 + D_1) \times N} \to \mathbb{R}^{D_1 \times N}, P_2 : \mathbb{R}^{(D_0 + D_2) \times N} \to \mathbb{R}^{D_2 \times N}$ are two sequence-to-sequence functions that are implemented by multi-layer transformers, i.e. there exists $\boldsymbol{\theta}_1, \boldsymbol{\theta}_2$ such that*

$$\mathrm{TF}_{\boldsymbol{\theta}_1} : \mathbf{H}_1 = \begin{bmatrix} \mathbf{h}_i^{(0)} \\ \mathbf{h}_i^{(1)} \end{bmatrix}_{1 \le i \le N} \in \mathbb{R}^{(D_0 + D_1) \times N} \mapsto \begin{bmatrix} \mathbf{H}^{(0)} \\ P_1(\mathbf{H}_1) \end{bmatrix},$$

$$\mathrm{TF}_{\boldsymbol{\theta}_2} : \mathbf{H}_2 = \begin{bmatrix} \mathbf{h}_i^{(0)} \\ \mathbf{h}_i^{(2)} \end{bmatrix}_{1 \le i \le N} \in \mathbb{R}^{(D_0 + D_2) \times N} \mapsto \begin{bmatrix} \mathbf{H}^{(0)} \\ P_2(\mathbf{H}_2) \end{bmatrix}.$$

*Then, there exists $\boldsymbol{\theta}$ such that for $\mathbf{H}'$ that takes form $\mathbf{h}_i' = [\mathbf{h}_i^{(0)}; \mathbf{h}_i^{(1)}; \mathbf{h}_i^{(2)}]$, with $\mathbf{h}_i^{(0)} \in \mathbb{R}^{D_0}, \mathbf{h}_i^{(1)} \in \mathbb{R}^{D_1}, \mathbf{h}_i^{(2)} \in \mathbb{R}^{D_2}$, we have*

$$\mathrm{TF}_{\boldsymbol{\theta}} : \mathbf{H}' = \begin{bmatrix} \mathbf{h}_i^{(0)} \\ \mathbf{h}_i^{(1)} \\ \mathbf{h}_i^{(2)} \end{bmatrix}_{1 \le i \le N} \in \mathbb{R}^{(D_0 + D_1 + D_2) \times N} \mapsto \begin{bmatrix} \mathbf{H}^{(0)} \\ P_1(\mathbf{H}_1) \\ P_2(\mathbf{H}_2) \end{bmatrix}.$$

*Further, $\boldsymbol{\theta}$ has at most $L \le \max\{L_1, L_2\}$ layers, $\max_{\ell \in [L]} M^{(\ell)} \le \max_{\ell \in [L]} \left( M_1^{(\ell)} + M_2^{(\ell)} \right)$ heads, $\max_{\ell \in [L]} D^{(\ell)} \le \max_{\ell \in [L]} \left( D_1^{(\ell)} + D_2^{(\ell)} \right)$ hidden dimension in its MLP layer (understanding the size of the empty layers as 0), and norm bound $\|\boldsymbol{\theta}\| \le \|\boldsymbol{\theta}_1\| + \|\boldsymbol{\theta}_2\|$.*

*Proof.* When $L_1 = L_2$ ($\boldsymbol{\theta}_1$ and $\boldsymbol{\theta}_2$ have the same number of layers), the result follows directly by applying Proposition B.5 repeatedly for all $L_1$ layers and the definition of the norm (2).

If (without loss of generality) $L_1 < L_2$, we can augment $\boldsymbol{\theta}_1$ to $L_2$ layers by adding $(L_2 - L_1)$ layers with zero attention heads, and zero MLP hidden dimension (note that this does not change $M_1, D_1'$, and $\|\boldsymbol{\theta}_1\|$). Due to the residual structure, the transformer maintains the output $P_1(\mathbf{H}_1)$ throughout layer $L_1 + 1, \ldots, L_2$, and it reduces to the case $L_1 = L_2$. $\square$

## C  Extension to decoder-based architecture

Here we briefly discuss how our theoretical results can be adapted to decoder-based architectures (henceforth decoder TFs). Adopting the setting as in Section 2, we consider a sequence of $N$ input vectors $\{\mathbf{h}_i\}_{i=1}^N \subset \mathbb{R}^D$, written compactly as an input matrix $\mathbf{H} = [\mathbf{h}_1, \ldots, \mathbf{h}_N] \in \mathbb{R}^{D \times N}$. Recall that $\sigma(t) := \mathrm{ReLU}(t) = \max\{t, 0\}$ denotes the standard relu activation.

### C.1  Decoder-based transformers

Decoder TFs are the same as encoder TFs, except that the attention layers are replaced by masked attention layers with a specific decoder-based (causal) attention mask.

**Definition C.1** (Masked attention layer). *A masked attention layer with $M$ heads is denoted as $\mathrm{MAttn}_{\boldsymbol{\theta}}(\cdot)$ with parameters $\boldsymbol{\theta} = \{(\mathbf{V}_m, \mathbf{Q}_m, \mathbf{K}_m)\}_{m \in [M]} \subset \mathbb{R}^{D \times D}$. On any input sequence $\mathbf{H} \in \mathbb{R}^{D \times N'}$ with $N' \le N$,*

$$\widetilde{\mathbf{H}} = \mathrm{MAttn}_{\boldsymbol{\theta}}(\mathbf{H}) := \mathbf{H} + \sum_{m=1}^M (\mathbf{V}_m \mathbf{H}) \times \left( (\mathrm{MSK}_{1:N', 1:N'}) \circ \sigma\left( (\mathbf{Q}_m \mathbf{H})^\top (\mathbf{K}_m \mathbf{H}) \right) \right) \in \mathbb{R}^{D \times N'},$$

(10)

*where $\circ$ denotes the entry-wise (Hadamard) product of two matrices, and $\mathrm{MSK} \in \mathbb{R}^{N \times N}$ is the mask matrix given by*

$$\mathrm{MSK} = \begin{bmatrix} 1 & 1/2 & 1/3 & \cdots & 1/N \\ 0 & 1/2 & 1/3 & \cdots & 1/N \\ 0 & 0 & 1/3 & \cdots & 1/N \\ \cdots & \cdots & \cdots & \cdots & \cdots \\ 0 & 0 & 0 & \cdots & 1/N \end{bmatrix}.$$

*In vector form, we have*

$$\widetilde{\mathbf{h}}_i = [\mathrm{Attn}_{\boldsymbol{\theta}}(\mathbf{H})]_i = \mathbf{h}_i + \sum_{m=1}^M \tfrac{1}{i} \sum_{j=1}^i \sigma(\langle \mathbf{Q}_m \mathbf{h}_i, \mathbf{K}_m \mathbf{h}_j \rangle) \cdot \mathbf{V}_m \mathbf{h}_j.$$

Notice that standard masked attention definitions use the pre-activation additive masks (with mask value $-\infty$) [84]. The post-activation multiplicative masks we use is equivalent to the pre-activation additive masks, and the modified presentation is for notational convenience. We also use a normalized ReLU activation $t \mapsto \sigma(t)/i$ in place of the standard softmax activation to be consistent with Definition 1. Note that the normalization $1/i$ is to ensure that the attention weights $\{\sigma(\langle \mathbf{Q}_m \mathbf{h}_i, \mathbf{K}_m \mathbf{h}_j \rangle)/i\}_{j \in [i]}$ is a set of non-negative weights that sum to $O(1)$. The motivation of masked attention layer is to ensure that, when processing a sequence of tokens, the computations at any token do not see any later token.

We next define the decoder-based transformers with $L \geq 1$ transformer layers, each consisting of a masked attention layer (c.f. Definition C.1) followed by an MLP layer (c.f. Definition 2). This definition is similar to the definition of encoder-based transformers (c.f., Definition 3), except that we replace the attention layers by masked attention layers.

**Definition C.2** (Decoder-based Transformer). *An $L$-layer decoder-based transformer, denoted as $\mathrm{DTF}_{\boldsymbol{\theta}}(\cdot)$, is a composition of $L$ self-attention layers each followed by an MLP layer: $\mathbf{H}^{(L)} = \mathrm{DTF}_{\boldsymbol{\theta}}(\mathbf{H}^{(0)})$, where $\mathbf{H}^{(0)} \in \mathbb{R}^{D \times N}$ is the input sequence, and*

$$\mathbf{H}^{(\ell)} = \mathrm{MLP}_{\boldsymbol{\theta}_{\mathtt{mlp}}^{(\ell)}}\Big(\mathrm{MAttn}_{\boldsymbol{\theta}_{\mathtt{mattn}}^{(\ell)}}\big(\mathbf{H}^{(\ell-1)}\big)\Big), \quad \ell \in \{1, \ldots, L\}.$$

*Above, the parameter $\boldsymbol{\theta} = (\boldsymbol{\theta}_{\mathtt{mattn}}^{(1:L)}, \boldsymbol{\theta}_{\mathtt{mlp}}^{(1:L)})$ is the parameter consisting of the attention layers $\boldsymbol{\theta}_{\mathtt{mattn}}^{(\ell)} = \{(\mathbf{V}_m^{(\ell)}, \mathbf{Q}_m^{(\ell)}, \mathbf{K}_m^{(\ell)})\}_{m \in [M^{(\ell)}]} \subset \mathbb{R}^{D \times D}$ and the MLP layers $\boldsymbol{\theta}_{\mathtt{mlp}}^{(\ell)} = (\mathbf{W}_1^{(\ell)}, \mathbf{W}_2^{(\ell)}) \in \mathbb{R}^{D^{(\ell)} \times D} \times \mathbb{R}^{D \times D^{(\ell)}}$. We will frequently consider "attention-only" decoder-based transformers with $\mathbf{W}_1^{(\ell)}, \mathbf{W}_2^{(\ell)} = \mathbf{0}$, which we denote as $\mathrm{DTF}_{\boldsymbol{\theta}}^0(\cdot)$ for shorthand, with $\boldsymbol{\theta} = \boldsymbol{\theta}^{(1:L)} := \boldsymbol{\theta}_{\mathtt{mattn}}^{(1:L)}$.*

We also use (2) to define the norm of $\mathrm{DTF}_{\boldsymbol{\theta}}$.

## C.2 In-context learning with decoder-based transformers

We consider using decoder-based TFs to perform ICL. We encode $(\mathcal{D}, \mathbf{x}_{N+1})$, which follows the generating rule as described in Section 2.2, into an input sequence $\mathbf{H} \in \mathbb{R}^{D \times (2N+1)}$. In our theory, we use the following format, where the first two rows contain $(\mathcal{D}, \mathbf{x}_{N+1})$ which alternating between $[\mathbf{x}_i; 0] \in \mathbb{R}^{d+1}$ and $[\mathbf{0}_{d \times 1}; y_i] \in \mathbb{R}^{d+1}$ (the same setup as adopted in [31, 2]); The third row contains fixed vectors $\{\mathbf{p}_i\}_{i \in [N+1]}$ with ones, zeros, the example index, and indicator for being the covariate token (similar to a positional encoding vector):

$$\mathbf{H} = \begin{bmatrix} \mathbf{x}_1 & \mathbf{0} & \ldots & \mathbf{x}_N & \mathbf{0} & \mathbf{x}_{N+1} \\ 0 & y_1 & \ldots & 0 & y_N & 0 \\ \mathbf{p}_1 & \mathbf{p}_2 & \ldots & \mathbf{p}_{2N-1} & \mathbf{p}_{2N} & \mathbf{p}_{2N+1} \end{bmatrix}, \quad \mathbf{p}_i := \begin{bmatrix} \mathbf{0}_{D-(d+4)} \\ \lceil i/2 \rceil \\ 1 \\ \mathrm{mod}(i+1, 2) \end{bmatrix} \in \mathbb{R}^{D-(d+1)}. \quad (11)$$

(11) is different from out input format (3) for encoder-based TFs. The main difference is that $(\mathbf{x}_i, y_i)$ are in different tokens in (11), whereas $(\mathbf{x}_i, y_i)$ are in the same token in (3). The reason for the former (i.e., different tokens in decoder) is that we want to avoid every $[\mathbf{x}_i; 0]$ token seeing the information of $y_i$, since we will evaluate the loss at every token. The reason for the latter (i.e., the same token in encoder) is for presentation convenience: since we only evaluate the loss at the last token, it is not necessary to alternate between $[\mathbf{x}_i; 0]$ and $[\mathbf{0}; y_i]$ to avoid information leakage.

We then feed $\mathbf{H}$ into a decoder TF to obtain the output $\widetilde{\mathbf{H}} = \mathrm{DTF}_{\boldsymbol{\theta}}(\mathbf{H}) \in \mathbb{R}^{D \times (2N+1)}$ with the same shape, and *read out* the prediction $\widehat{y}_{N+1}$ from the $(d+1, 2N+1)$-th entry of $\widetilde{\mathbf{H}} = [\widetilde{\mathbf{h}}_i]_{i \in [2N+1]}$ (the entry corresponding to the last missing test label): $\widehat{y}_{N+1} = \mathrm{read}_y(\widetilde{\mathbf{H}}) := (\widetilde{\mathbf{h}}_{2N+1})_{d+1}$. The goal is to predict $\widehat{y}_{N+1}$ that is close to $y_{N+1} \sim \mathsf{P}_{y|\mathbf{x}_{N+1}}$ measured by proper losses.

The benefit of using the decoder architecture is that, during the pre-training phase, one can construct the training loss function by using all the predictions $\{\widehat{y}_j\}_{j \in [N+1]}$, where $\widehat{y}_j$ gives the $(d+1, 2j-1)$-th entry of $\widetilde{\mathbf{H}} = [\widetilde{\mathbf{h}}_i]_{i \in [2N+1]}$ for each $j \in [N+1]$ (the entry corresponding to the missing test label of the $2j-1$'th token): $\widehat{y}_j = \mathrm{read}_{y,j}(\widetilde{\mathbf{H}}) := (\widetilde{\mathbf{h}}_{2j-1})_{d+1}$. Given a loss function $\ell : \mathbb{R} \times \mathbb{R} \to \mathbb{R}$ associated to a single response, the training loss associated to the whole input sequence can be defined by $\ell(\mathbf{H}) = \sum_{j=1}^{N+1} \ell(y_j, \widehat{y}_j)$. This potentially enables less training sequences in the pre-training stage, and some generalization bound analysis justifying this benefit was provided in [47].

## C.3 Results

We discuss how our theoretical results upon encoder TFs can be converted to those of the decoder TFs. Taking the implementation of (ICGD) (a key mechanism that enables most basic ICL algorithms such as ridge regression; cf. Appendix D.1) as an example, this conversion is enabled by the following facts: (a) the input format (11) of decoders can be converted to the input format (3) of encoders by a 2-layer decoder TF; (b) the encoder TF that implements (ICGD) with input format (3), by a slight parameter modification, can be converted to a decoder TF that implements the (ICGD) algorithm with a converted input format.

**Input format conversion** Despite the difference between the input format (11) and (3), we show that there exists a 2-layer decoder TF that can convert the input format (11) to format (3). The proof can be found in Appendix C.4.

**Proposition C.1** (Input format conversion)**.** *There exists a 2-layer decoder TF* DTF *with* 3 *heads per layer, hidden dimension* 2 *and* $\|\boldsymbol{\theta}\| \leq 12$ *such that upon taking input* $\mathbf{H}$ *of format* (11)*, it outputs* $\widetilde{\mathbf{H}} = \mathrm{DTF}(\mathbf{H})$ *with*

$$\widetilde{\mathbf{H}} = \begin{bmatrix} \mathbf{x}_1 & \mathbf{x}_1 & \dots & \mathbf{x}_N & \mathbf{x}_N & \mathbf{x}_{N+1} \\ 0 & y_1 & \dots & 0 & y_N & 0 \\ \mathbf{p}_1 & \mathbf{p}_2 & \dots & \mathbf{p}_{2N-1} & \mathbf{p}_{2N} & \mathbf{p}_{2N+1} \end{bmatrix}. \tag{12}$$

*In particular, format* (12) *contains format* (3) *as a submatrix, by restricting to the* $\{1, 2, \dots, D-1, D-2, D\}$ *rows and* $\{2, 4, \dots, 2N-2, 2N, 2N+1\}$ *columns.*

**Generalization TF constructions to decoder architecture** The construction in Theorem D.1 can be generalized to using the input format (12) along with a decoder TF, by using the scratch pad within the last token to record the gradient descent iterates. Further, if we slightly change the normalization in MSK from $1/i$ to $1/((i-1) \vee 1)$, then the same construction performs (ICGD) (with training examples $\{1, \dots, j\}$) at every token $i = 2j+1$ (corresponding to predicting at $\mathbf{x}_{j+1}$). Building on this extension, all our constructions in Section 3 and Section 4.2 can be generalized to decoder TFs.

## C.4 Proof of Proposition C.1

For the simplicity of presentation, we write $c_i = \lceil i/2 \rceil$, $t_i = \mathrm{mod}(i+1, 2)$, $\mathbf{u}_i = \mathbf{h}_i[1:d] \in \mathbb{R}^{d+1}$ be the vector of first $d$ entries of $\mathbf{h}_i$ [6], and let $v_i = \mathbf{h}_i[d+1]$ be the $(d+1)$-th entry of $\mathbf{h}_i$. With such notations, the input sequence $\mathbf{H} = [\mathbf{h}_i]_i$ can be compactly written as

$$\mathbf{h}_i = [\mathbf{u}_i; v_i; \mathbf{0}_{D-d-4}; c_i; 1; t_i].$$

In the following, we construct the desired $\boldsymbol{\theta} = (\boldsymbol{\theta}^{(1)}, \boldsymbol{\theta}^{(2)})$ as follows.

**Step 1:** construction of $\boldsymbol{\theta}^{(1)} = (\boldsymbol{\theta}_{\mathtt{mattn}}^{(1)}, \boldsymbol{\theta}_{\mathtt{mlp}}^{(1)})$, so that $\mathrm{MLP}_{\boldsymbol{\theta}_{\mathtt{mlp}}^{(1)}} \circ \mathrm{MAttn}_{\boldsymbol{\theta}_{\mathtt{mattn}}^{(1)}}$ maps

$$\mathbf{h}_i \xrightarrow{\mathrm{MAttn}_{\boldsymbol{\theta}_{\mathtt{mattn}}^{(1)}}} \mathbf{h}'_i = [\mathbf{u}_i; v_i; \mathbf{0}_{D-d-6}; t_i(c_i^2 + 0.5); t_i c_i; c_i; 1; t_i]$$

---

[6]In other words, when $2 \nmid i$, $\mathbf{u}_i = \mathbf{x}_{(i-1)/2}$; when $2 \mid i$, $\mathbf{u}_i = \mathbf{0}_d$.

$$\xrightarrow{\text{MLP}_{\theta_{\mathtt{mlp}}^{(1)}}} \quad \mathbf{h}_i^{(1)} = [\mathbf{u}_i; v_i; \mathbf{0}_{D-d-6}; t_i c_i^2; t_i c_i; c_i; 1; t_i].$$

For $m \in \{0,1\}$, we define matrices $\mathbf{Q}_m^{(1)}, \mathbf{K}_m^{(1)}, \mathbf{V}_m^{(1)} \in \mathbb{R}^{D \times D}$ such that

$$\mathbf{Q}_0^{(1)}\mathbf{h}_i = \mathbf{Q}_1^{(1)}\mathbf{h}_i = \begin{bmatrix} t_i \\ \mathbf{0} \end{bmatrix}, \qquad \mathbf{K}_0^{(1)}\mathbf{h}_j = \mathbf{K}_1^{(1)}\mathbf{h}_j = \begin{bmatrix} c_j \\ \mathbf{0} \end{bmatrix}, \qquad \mathbf{V}_0^{(1)}\mathbf{h}_j = \begin{bmatrix} \mathbf{0}_{D-4} \\ 3c_j \\ \mathbf{0}_3 \end{bmatrix}, \qquad \mathbf{V}_1^{(1)}\mathbf{h}_j = \begin{bmatrix} \mathbf{0}_{D-3} \\ 2 \\ \mathbf{0}_2 \end{bmatrix},$$

for all $i, j$. By the structure of $\mathbf{h}_i$, these matrices indeed exist, and further it is straightforward to check that they have norm bounds

$$\max_m \left\| \mathbf{Q}_m^{(1)} \right\|_{\mathrm{op}} \le 1, \quad \max_m \left\| \mathbf{K}_m^{(1)} \right\|_{\mathrm{op}} \le 1, \quad \sum_m \left\| \mathbf{V}_m^{(1)} \right\|_{\mathrm{op}} \le 5.$$

Now, for every $i$,

$$\frac{1}{i} \sum_{j=1}^{i} \sum_{m \in \{0,1\}} \sigma\Big(\big\langle \mathbf{Q}_m^{(1)}\mathbf{h}_i, \mathbf{K}_m^{(1)}\mathbf{h}_j \big\rangle\Big)\mathbf{V}_m^{(1)}\mathbf{h}_j = \frac{1}{i}\sum_{j=1}^{i} t_i \cdot [\mathbf{0}_{D-4}; 3c_j^2; 2c_j; 0; 0].$$

Notice that $t_i \ne 0$ only when $2 \mid i$, we then compute for $i = 2k$ that

$$\sum_{j=1}^{i} 3c_j^2 = 3 \cdot \frac{k(k-1)(2k-1)}{3} + 3k^2 = 2k^3 + k, \qquad \sum_{j=1}^{i} 2c_j = 2 \cdot k(k-1) + 2k = 2k^2.$$

Therefore, the $\theta_{\mathtt{mattn}}^{(1)} = \{(\mathbf{Q}_m^{(1)}, \mathbf{K}_m^{(1)}, \mathbf{V}_m^{(1)} \in \mathbb{R}^{D \times D})\}_{m \in \{0,1\}}$ we construct above is indeed the desired attention layer. The existence of the desired $\theta_{\mathtt{mlp}}^{(1)}$ is clear, and $\theta_{\mathtt{mlp}}^{(1)} = (\mathbf{W}_1^{(1)}, \mathbf{W}_2^{(1)})$ can further be chosen so that $\|\mathbf{W}_1^{(1)}\|_{\mathrm{op}} \le 1, \|\mathbf{W}_2^{(1)}\|_{\mathrm{op}} \le 1$.

**Step 2:** construction of $\boldsymbol{\theta}^{(2)}$. For every $m \in \{-1, 0, 1\}$, we define matrices $\mathbf{Q}_m^{(2)}, \mathbf{K}_m^{(2)}, \mathbf{V}_m^{(2)} \in \mathbb{R}^{D \times D}$ such that

$$\mathbf{Q}_0^{(2)}\mathbf{h}_i^{(1)} = \mathbf{Q}_1^{(2)}\mathbf{h}_i^{(1)} = \mathbf{Q}_{-1}^{(2)}\mathbf{h}_i^{(1)} = \begin{bmatrix} t_i c_i^2 \\ t_i c_i \\ \mathbf{0} \end{bmatrix},$$

$$\mathbf{K}_0^{(2)}\mathbf{h}_j^{(1)} = \begin{bmatrix} 1 \\ -c_j \\ \mathbf{0} \end{bmatrix}, \qquad \mathbf{K}_1^{(2)}\mathbf{h}_j^{(1)} = \begin{bmatrix} 1 \\ -(c_j + 1) \\ \mathbf{0} \end{bmatrix}, \qquad \mathbf{K}_1^{(2)}\mathbf{h}_j^{(1)} = \begin{bmatrix} 1 \\ -(c_j - 1) \\ \mathbf{0} \end{bmatrix},$$

$$\mathbf{V}_0^{(2)}\mathbf{h}_j^{(1)} = \begin{bmatrix} -4\mathbf{u}_j \\ \mathbf{0}_{D-d} \end{bmatrix}, \qquad \mathbf{V}_1^{(2)}\mathbf{h}_j^{(1)} = \mathbf{V}_{-1}^{(2)}\mathbf{h}_j^{(1)} = \begin{bmatrix} 2\mathbf{u}_j \\ \mathbf{0}_{D-d} \end{bmatrix},$$

for all $i, j$. By the structure of $\mathbf{h}_i^{(1)}$, these matrices indeed exist, and further it is straightforward to check that they have norm bounds

$$\max_m \left\| \mathbf{Q}_m^{(2)} \right\|_{\mathrm{op}} \le 1, \quad \max_m \left\| \mathbf{K}_m^{(2)} \right\|_{\mathrm{op}} \le 2, \quad \sum_m \left\| \mathbf{V}_m^{(2)} \right\|_{\mathrm{op}} \le 8.$$

Now, for every $i, j$, we have

$$\sum_{m \in \{-1,0,1\}} \sigma\Big(\big\langle \mathbf{Q}_m^{(2)}\mathbf{h}_i^{(1)}, \mathbf{K}_m^{(2)}\mathbf{h}_j^{(1)} \big\rangle\Big)\mathbf{V}_m^{(2)}\mathbf{h}_j^{(1)}$$

$$= \big\{ -2\sigma\big(t_i c_i^2 - t_i c_i c_j\big) + \sigma\big(t_i c_i^2 - t_i c_i (c_j + 1)\big) + \sigma\big(t_i c_i^2 - t_i c_i (c_j - 1)\big) \big\} \cdot 2[\mathbf{u}_j; \mathbf{0}_{D-d}]$$

$$= \big\{ -2\sigma(c_i - c_j) + \sigma((c_i - c_j) - 1) + \sigma((c_i - c_j) + 1) \big\} \cdot 2c_i t_i [\mathbf{u}_j; \mathbf{0}_{D-d}]$$

$$= \mathbb{I}(c_i = c_j) \cdot 2c_i t_i [\mathbf{u}_j; \mathbf{0}_{D-d}],$$

where the last equality follows from the fact that

$$-2\sigma(x) + \sigma(x - 1) + \sigma(x + 1) = \begin{cases} 0, & x \ge 1 \text{ or } x \le -1, \\ x + 1, & x \in [-1, 0], \\ 1 - x, & x \in [0, 1]. \end{cases}$$

Therefore,

$$\frac{1}{i}\sum_{j=1}^{i}\sum_{m\in\{-1,0,1\}}\sigma\left(\left\langle \mathbf{Q}_m^{(2)}\mathbf{h}_i^{(1)},\mathbf{K}_m^{(2)}\mathbf{h}_j^{(1)}\right\rangle\right)\mathbf{V}_m^{(2)}\mathbf{h}_j^{(1)} = \frac{1}{i}\sum_{j=1}^{i}2\mathbb{I}(c_i = c_j)c_i t_i[\mathbf{u}_j;\mathbf{0}_{D-d}]$$

$$= \begin{cases} [\mathbf{x}_k;\mathbf{0}_{D-d}], & i = 2k \\ \mathbf{0}_D, & \text{otherwise} \end{cases}.$$

Therefore, the $\boldsymbol{\theta}_{\mathtt{mattn}}^{(2)} = \{(\mathbf{Q}_m^{(2)},\mathbf{K}_m^{(2)},\mathbf{V}_m^{(2)}\in\mathbb{R}^{D\times D})\}_{m\in\{-1,0,1\}}$ we construct above maps

$$\mathbf{h}_i^{(1)} \quad\to\quad \mathbf{h}_i'' = [\mathbf{x}_{\lceil i/2\rceil};v_i;\mathbf{0}_{D-d-6};t_i c_i^2;t_i c_i;c_i;1;t_i].$$

Finally, we only need to take a MLP layer $\boldsymbol{\theta}_{\mathtt{mlp}}^{(2)} = (\mathbf{W}_1^{(2)},\mathbf{W}_2^{(2)})$ with hidden dimension 2 that maps

$$\mathbf{h}_i'' \quad\to\quad \mathbf{h}_i^{(2)} = [\mathbf{x}_{\lceil i/2\rceil};v_i;\mathbf{0}_{D-d-6};0;0;c_i;1;t_i],$$

which clearly exists and can be chosen so that $\|\mathbf{W}_1^{(2)}\|_{\mathrm{op}} \leq 1, \|\mathbf{W}_2^{(2)}\|_{\mathrm{op}} \leq 1$.

Combining the two steps above, we complete the proof of Proposition C.1. $\qquad\square$

## D  Mechanism: In-context gradient descent

Technically, the constructions in Section 3.1-3.2 rely on a new efficient construction for transformers to implement in-context gradient descent and its variants, which we present as follows. We begin by presenting the result for implementing (vanilla) gradient descent on convex empirical risks.

**Compact notation of input**  We will often use shorthand $y_i' \in \mathbb{R}$ defined as $y_i' = y_i$ for $i \in [N]$ and $y_{N+1}' = 0$ to simplify our notation, with which the input sequence $\mathbf{H} \in \mathbb{R}^{D\times(N+1)}$ can be compactly written as $\mathbf{h}_i = [\mathbf{x}_i;y_i';\mathbf{p}_i] = [\mathbf{x}_i;y_i';\mathbf{0}_{D-d-3};1;t_i]$ for $i \in [N+1]$, where $t_i := 1\{i < N+1\}$ is the indicator for the training examples.

### D.1  Gradient descent on convex empirical risk

Let $\ell(\cdot,\cdot) : \mathbb{R}^2 \to \mathbb{R}$ be a loss function. Let $\widehat{L}_N(\mathbf{w}) := \frac{1}{N}\sum_{i=1}^{N}\ell(\mathbf{w}^\top\mathbf{x}_i, y_i)$ denote the empirical risk with loss function $\ell$ on dataset $\{(\mathbf{x}_i, y_i)\}_{i\in[N]}$, and

$$\mathbf{w}_{\mathrm{GD}}^{t+1} := \mathbf{w}_{\mathrm{GD}}^t - \eta\nabla\widehat{L}_N(\mathbf{w}_{\mathrm{GD}}^t) \tag{ICGD}$$

denote the gradient descent trajectory on $\widehat{L}_N$ with initialization $\mathbf{w}_{\mathrm{GD}}^0 \in \mathbb{R}^d$ and learning rate $\eta > 0$.

We require the partial derivative of the loss $\partial_s\ell : (s,t) \mapsto \partial_s\ell(s,t)$ (as a bivariate function) to be approximable by a sum of relus, defined as follows.

**Definition D.1** (Approximability by sum of relus). *A function $g : \mathbb{R}^k \to \mathbb{R}$ is $(\varepsilon_{\mathrm{approx}}, R, M, C)$-approximable by sum of relus, if there exists a "$(M,C)$-sum of relus" function*

$$f_{M,C}(\mathbf{z}) = \sum_{m=1}^{M}c_m\sigma(\mathbf{a}_m^\top[\mathbf{z};1]) \quad\text{with}\quad \sum_{m=1}^{M}|c_m| \leq C,\ \max_{m\in[M]}\|\mathbf{a}_m\|_1 \leq 1,\ \mathbf{a}_m \in \mathbb{R}^{k+1},\ c_m \in \mathbb{R},$$

*such that $\sup_{\mathbf{z}\in[-R,R]^k}|g(\mathbf{z}) - f_{M,C}(\mathbf{z})| \leq \varepsilon_{\mathrm{approx}}$.*

Definition D.1 is known to contain broad class of functions. For example, any mildly smooth $k$-variate function is approximable by a sum of relus for any $(\varepsilon_{\mathrm{approx}}, R)$, with mild bounds on $(M,C)$ (Proposition B.1, building on results of Bach [4]). Also, any function that is a $(M,C)$-sum of relus itself (which includes all piecewise linear functions) is by definition $(0,\infty,M,C)$-approximable by sum of relus.

We show that $L$ steps of (ICGD) can be approximately implemented by an $(L+1)$-layer transformer.

**Theorem D.1** (Convex ICGD). *Fix any $B_w > 0$, $L > 1$, $\eta > 0$, and $\varepsilon \leq B_w/(2L)$. Suppose that*

1. *The loss $\ell(\cdot,\cdot)$ is convex in the first argument;*

2. *$\partial_s\ell$ is $(\varepsilon, R, M, C)$-approximable by sum of relus with $R = \max\{B_x B_w, B_y, 1\}$.*

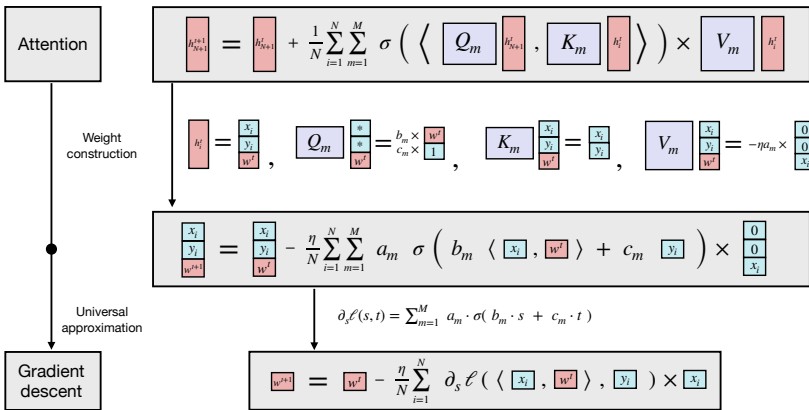

Figure 4: Illustration of our main mechanism for implementing basic ICL algorithms: One attention layer implements a single (ICGD) iterate (Proposition E.1 & Theorem D.1). Top: the attention mechanism as in Definition 1. Bottom: A single (ICGD) iterate. Middle: Linear algebraic illustration of the attention layer for implementing a GD update.

Then, there exists an attention-only transformer $\mathrm{TF}_{\boldsymbol{\theta}}^0$ with $(L+1)$ layers, $\max_{\ell \in [L]} M^{(\ell)} \leq M$ heads within the first $L$ layers, and $M^{(L+1)} = 2$ such that for any input data $(\mathcal{D}, \mathbf{x}_{N+1})$ such that

$$\sup_{\|\mathbf{w}\|_2 \leq B_w} \lambda_{\max}(\nabla^2 \widehat{L}_N(\mathbf{w})) \leq 2/\eta, \qquad \exists \mathbf{w}^\star \in \arg\min_{\mathbf{w} \in \mathbb{R}^d} \widehat{L}_N(\mathbf{w}) \text{ such that } \|\mathbf{w}^\star\|_2 \leq B_w/2,$$

$\mathrm{TF}_{\boldsymbol{\theta}}^0(\mathbf{H}^{(0)})$ approximately implements (ICGD) with initialization $\mathbf{w}_{\mathrm{GD}}^0 = \mathbf{0}$:

1. (Parameter space) For every $\ell \in [L]$, the $\ell$-th layer's output $\mathbf{H}^{(\ell)} = \mathrm{TF}_{\boldsymbol{\theta}(1:\ell)}^0(\mathbf{H}^{(0)})$ approximates $\ell$ steps of (ICGD): We have $\mathbf{h}_i^{(\ell)} = [\mathbf{x}_i; y_i'; \widehat{\mathbf{w}}^\ell; \mathbf{0}_{D-2d-3}; 1; t_i]$ for every $i \in [N+1]$, where

$$\left\|\widehat{\mathbf{w}}^\ell - \mathbf{w}_{\mathrm{GD}}^\ell\right\|_2 \leq \varepsilon \cdot (\ell\eta B_x).$$

Note that the bound scales as $\mathcal{O}(\ell)$, a linear error accumulation.

2. (Prediction space) The final output $\mathbf{H}^{(L+1)} = \mathrm{TF}_{\boldsymbol{\theta}}^0(\mathbf{H}^{(0)})$ approximates the prediction of $L$ steps of (ICGD): We have $\mathbf{h}_{N+1}^{(L+1)} = [\mathbf{x}_{N+1}; \widehat{y}_{N+1}; \widehat{\mathbf{w}}^L; \mathbf{0}_{D-2d-3}; 1; t_i]$, where $\widehat{y}_{N+1} = \langle \widehat{\mathbf{w}}^L, \mathbf{x}_{N+1} \rangle$ so that

$$\left|\widehat{y}_{N+1} - \langle \mathbf{w}_{\mathrm{GD}}^L, \mathbf{x}_{N+1} \rangle\right| \leq \varepsilon \cdot (L\eta B_x^2).$$

Further, the transformer admits norm bound $\|\boldsymbol{\theta}\| \leq 2 + R + 2\eta C$.

The proof can be found in Appendix E.2. Theorem D.1 substantially generalizes that of von Oswald et al. [86] (which only does GD on square losses with a *linear* self-attention), and is simpler than the ones in Akyürek et al. [2] and Giannou et al. [32]. See Figure 4 for a pictorial illustration of the basic component of the construction, which implements a single step of gradient descent using a single attention layer (Proposition E.1).

Technically, we utilize the stability of *convex* gradient descent as in the following lemma (proof in Appendix E.3) to obtain the *linear* error accumulation in Theorem D.1; the error accumulation will become *exponential* in $L$ in the non-convex case in general; see Lemma D.3(b).

**Lemma D.1** (Composition of error for approximating convex GD). *Suppose $f : \mathbb{R}^d \to \mathbb{R}$ is a convex function. Let $\mathbf{w}^\star \in \arg\min_{\mathbf{w} \in \mathbb{R}^d} f(\mathbf{w})$, $R \geq 2\|\mathbf{w}^\star\|_2$, and assume that $\nabla f$ is $L_f$-smooth on $\mathrm{B}_2^d(R)$. Let sequences $\{\widehat{\mathbf{w}}^\ell\}_{\ell \geq 0} \subset \mathbb{R}^d$ and $\{\mathbf{w}_{\mathrm{GD}}^\ell\}_{\ell \geq 0} \subset \mathbb{R}^d$ be given by $\widehat{\mathbf{w}}^0 = \mathbf{w}_{\mathrm{GD}}^0 = \mathbf{0}$,*

$$\begin{cases} \widehat{\mathbf{w}}^{\ell+1} = \widehat{\mathbf{w}}^\ell - \eta \nabla f(\widehat{\mathbf{w}}^\ell) + \boldsymbol{\varepsilon}^\ell, & \|\boldsymbol{\varepsilon}^\ell\|_2 \leq \varepsilon, \\ \mathbf{w}_{\mathrm{GD}}^{\ell+1} = \mathbf{w}_{\mathrm{GD}}^\ell - \eta \nabla f(\mathbf{w}_{\mathrm{GD}}^\ell), \end{cases}$$

*for all $\ell \geq 0$. Then as long as $\eta \leq 2/L_f$, for any $0 \leq L \leq R/(2\varepsilon)$, it holds that $\left\|\widehat{\mathbf{w}}^L - \mathbf{w}_{\mathrm{GD}}^L\right\|_2 \leq L\varepsilon$ and $\|\widehat{\mathbf{w}}^L\|_2 \leq \frac{R}{2} + L\varepsilon \leq R$.*

## D.2 Proximal gradient descent for regularized convex losses

Proximal gradient descent (PGD) is a variant of gradient descent that is suitable for minimizing regularized risks [66], in particular those with a non-smooth regularizer such as the $\ell_1$ norm. In this section, we show that transformers can approximate PGD with similar quantitative guarantees as for GD in Appendix D.1.

Let $\ell(\cdot, \cdot) : \mathbb{R}^2 \to \mathbb{R}$ be a loss function. Let $\widehat{L}_N(\mathbf{w}) := \frac{1}{N} \sum_{i=1}^N \ell(\mathbf{w}^\top \mathbf{x}_i, y_i) + \mathcal{R}(\mathbf{w})$ denote the regularized empirical risk with loss function $\ell$ on dataset $\{(\mathbf{x}_i, y_i)\}_{i \in [N]}$ and regularizer $\mathcal{R}$. To minimize $\widehat{L}_N$, we consider the proximal gradient descent trajectory on $\widehat{L}_N$ with initialization $\mathbf{w}_{\mathrm{GD}}^0 = \mathbf{0} \in \mathbb{R}^d$ and learning rate $\eta > 0$:

$$\mathbf{w}_{\mathrm{PGD}}^{t+1} := \mathbf{prox}_{\eta\mathcal{R}}\Big( \mathbf{w}_{\mathrm{PGD}}^t - \eta \nabla \widehat{L}_N^0(\mathbf{w}_{\mathrm{PGD}}^t) \Big), \tag{ICPGD}$$

where we denote $\widehat{L}_N^0(\mathbf{w}) := \frac{1}{N} \sum_{i=1}^N \ell(\mathbf{w}^\top \mathbf{x}_i, y_i)$.

To approximate (ICPGD) by transformers, in addition to the requirement on the loss $\ell$ as in Theorem D.1, we additionally require the the proximal operator $\mathbf{prox}_{\eta\mathcal{R}}(\cdot)$ to be approximable by an MLP layer (as a vector-valued analog of Definition D.1) defined as follows.

**Definition D.2** (Approximability by MLP). *An operator $P : \mathbb{R}^d \to \mathbb{R}^d$ is $(\varepsilon, R, D, C)$-approximable by MLP, if there exists a there exists a MLP $\boldsymbol{\theta}_{\mathtt{mlp}} = (\mathbf{W}_1, \mathbf{W}_2) \in \mathbb{R}^{D \times d} \times \mathbb{R}^{d \times D}$ with hidden dimension $D$, $\|\mathbf{W}_1\|_{\mathrm{op}} + \|\mathbf{W}_2\|_{\mathrm{op}} \le C'$, such that $\sup_{\|\mathbf{w}\|_2 \le R} \|P(\mathbf{w}) - \mathrm{MLP}_{\boldsymbol{\theta}_{\mathtt{mlp}}}(\mathbf{w})\|_2 \le \varepsilon$.*

The definition above captures the proximal operator $\mathbf{prox}_{\eta\mathcal{R}}$ for a broad class of regularizers, such as the (commonly-used) $L_1$ and $L_2$ regularizer listed in the following proposition, for all of which one can directly check that they can be exactly implemented by an MLP as stated below.

**Proposition D.1** (Proximal operators for commonly-used regularizers). *For regularizer $\mathcal{R}$ in $\{\lambda \|\cdot\|_1, \frac{\lambda}{2} \|\cdot\|_2^2, \mathbb{I}_{\mathsf{B}_\infty(B)}(\cdot)\}$, the operator $\mathbf{prox}_{\eta\mathcal{R}} : \mathbb{R}^d \to \mathbb{R}^d$ is exactly approximable by MLP. More concretely, we have*

1. *For $\mathcal{R} = \lambda \|\cdot\|_1$, $\mathbf{prox}_{\eta\mathcal{R}}$ is $(0, +\infty, 4d, 4 + 2\eta\lambda)$-approximable by MLP.*

2. *For $\mathcal{R} = \frac{\lambda}{2} \|\cdot\|_2^2$, $\mathbf{prox}_{\eta\mathcal{R}}$ is $(0, +\infty, 2d, 2 + 2\eta\lambda)$-approximable by MLP.*

3. *For $\mathcal{R} = \mathbb{I}_{\mathsf{B}_\infty(B)}(\cdot)$, $\mathbf{prox}_{\eta\mathcal{R}} = \mathrm{Proj}_{\mathsf{B}_\infty(B)}$ is $(0, +\infty, 2d, 2 + 2B)$-approximable by MLP.*

**Theorem D.2** (Convex ICPGD). *Fix any $B_w > 0$, $L > 1$, $\eta > 0$, and $\varepsilon + \varepsilon' \le B_w/(2L)$. Suppose that*

1. *The loss $\ell(\cdot, \cdot)$ is convex in the first argument;*

2. *$\partial_s \ell$ is $(\varepsilon, R, M, C)$-approximable by sum of relus with $R = \max\{B_x B_w, B_y, 1\}$.*

3. *$\mathcal{R}$ convex, and the proximal operator $\mathbf{prox}_{\eta\mathcal{R}}(\mathbf{w})$ is $(\eta\varepsilon', R', D', C')$-approximable by MLP with $R' = \sup_{\|\mathbf{w}\|_2 \le B_w} \|\mathbf{w}_\eta^+\|_2 + \eta\varepsilon$.*

*Then there exists a transformer $\mathrm{TF}_{\boldsymbol{\theta}}$ with $(L + 1)$ layers, $\max_{\ell \in [L]} M^{(\ell)} \le M$ heads within the first $L$ layers, $M^{(L+1)} = 2$, and hidden dimension $D'$ such that, for any input data $(\mathcal{D}, \mathbf{x}_{N+1})$ such that*

$$\sup_{\|\mathbf{w}\|_2 \le B_w} \lambda_{\max}(\nabla^2 \widehat{L}_N(\mathbf{w})) \le 2/\eta, \qquad \exists \mathbf{w}^\star \in \arg\min_{\mathbf{w} \in \mathbb{R}^d} \widehat{L}_N(\mathbf{w}) \text{ such that } \|\mathbf{w}^\star\|_2 \le B_w/2,$$

*$\mathrm{TF}_{\boldsymbol{\theta}}(\mathbf{H}^{(0)})$ approximately implements (ICGD):*

1. *(Parameter space) For every $\ell \in [L]$, the $\ell$-th layer's output $\mathbf{H}^{(\ell)} = \mathrm{TF}_{\boldsymbol{\theta}^{(1:\ell)}}(\mathbf{H}^{(0)})$ approximates $\ell$ steps of (ICGD): We have $\mathbf{h}_i^{(\ell)} = [\mathbf{x}_i; y_i'; \widehat{\mathbf{w}}^\ell; \mathbf{0}_{D-2d-3}; 1; t_i]$ for every $i \in [N+1]$, where*

$$\|\widehat{\mathbf{w}}^\ell - \mathbf{w}_{\mathrm{PGD}}^\ell\|_2 \le (\varepsilon + \varepsilon') \cdot (L\eta B_x).$$

2. *(Prediction space) The final output $\mathbf{H}^{(L+1)} = \mathrm{TF}_{\boldsymbol{\theta}}(\mathbf{H}^{(0)})$ approximates the prediction of $L$ steps of (ICGD): We have $\mathbf{h}_{N+1}^{(L+1)} = [\mathbf{x}_{N+1}; \widehat{y}_{N+1}; \widehat{\mathbf{w}}^L; \mathbf{0}_{D-2d-3}; 1; t_i]$, where $\widehat{y}_{N+1} = \langle \widehat{\mathbf{w}}^L, \mathbf{x}_{N+1} \rangle$ so that*

$$\left| \widehat{y}_{N+1} - \langle \mathbf{w}_{\mathrm{PGD}}^L, \mathbf{x}_{N+1} \rangle \right| \le (\varepsilon + \varepsilon') \cdot (2L\eta B_x^2).$$

*Further, the weight matrices have norm bounds* $\|\boldsymbol{\theta}\| \leq 3 + R + 2\eta C + C'$.

The proof of Theorem D.2 is essentially similar to the proof of Theorem D.1, using the following generalized version of Lemma D.1.

**Lemma D.2** (Composition of error for approximating convex PGD). *Suppose* $f : \mathbb{R}^d \to \mathbb{R}$ *is a convex function and* $\mathcal{R}$ *is a convex regularizer. Let* $\mathbf{w}^\star \in \arg\min_{\mathbf{w} \in \mathbb{R}^d} f(\mathbf{w}) + \mathcal{R}(\mathbf{w})$, $R \geq 2\|\mathbf{w}^\star\|_2$, *and assume that* $\nabla f$ *is* $L_f$*-smooth on* $\mathsf{B}_2^d(R)$. *Let sequences* $\{\widehat{\mathbf{w}}^\ell\}_{\ell \geq 0} \subset \mathbb{R}^d$ *and* $\{\mathbf{w}_{\mathrm{GD}}^\ell\}_{\ell \geq 0} \subset \mathbb{R}^d$ *be given by* $\widehat{\mathbf{w}}^0 = \mathbf{w}_{\mathrm{GD}}^0 = \mathbf{0}$,

$$\begin{cases} \widehat{\mathbf{w}}^{\ell+1} = \mathbf{prox}_{\eta\mathcal{R}}\big(\widehat{\mathbf{w}}^\ell - \eta\nabla f(\widehat{\mathbf{w}}^\ell)\big) + \boldsymbol{\varepsilon}^\ell, \qquad \|\boldsymbol{\varepsilon}^\ell\|_2 \leq \varepsilon, \\ \mathbf{w}_{\mathrm{GD}}^{\ell+1} = \mathbf{prox}_{\eta\mathcal{R}}\big(\mathbf{w}_{\mathrm{GD}}^\ell - \eta\nabla f(\mathbf{w}_{\mathrm{GD}}^\ell)\big), \end{cases}$$

*for all* $\ell \geq 0$. *Then as long as* $\eta \leq 2/L_f$, *for any* $0 \leq L \leq R/(2\varepsilon)$, *it holds that* $\big\|\widehat{\mathbf{w}}^L - \mathbf{w}_{\mathrm{GD}}^L\big\|_2 \leq L\varepsilon$ *and* $\|\widehat{\mathbf{w}}^L\|_2 \leq \frac{R}{2} + L\varepsilon \leq R$.

The proof of the above lemma is done by utilizing the non-expansiveness of the PGD operator $\mathbf{w} \mapsto \mathbf{prox}_{\eta\mathcal{R}}(\mathbf{w} - \eta\nabla f(\mathbf{w}))$ and otherwise following the same arguments as for Lemma D.1.

### D.3 Gradient descent on two-layer neural networks

We now move beyond the convex setting by showing that transformers can implement gradient descent on two-layer neural networks in context.

Suppose that the prediction function $\mathrm{pred}(\mathbf{x}; \mathbf{w}) := \sum_{k=1}^K u_k r(\mathbf{v}_k^\top \mathbf{x})$ is given by a two-layer neural network, parameterized by $\mathbf{w} = [\mathbf{v}_k; u_k]_{k \in [K]} \in \mathbb{R}^{K(d+1)}$. Consider the empirical risk minimization problem:

$$\min_{\mathbf{w} \in \mathcal{W}} \widehat{L}_N(\mathbf{w}) := \frac{1}{2N} \sum_{i=1}^N \ell(\mathrm{pred}(\mathbf{x}_i; \mathbf{w}), y_i) = \frac{1}{2N} \sum_{i=1}^N \ell\bigg(\sum_{k=1}^K u_k r(\mathbf{v}_k^\top \mathbf{x}_i), y_i\bigg), \qquad (13)$$

where $\mathcal{W}$ is a bounded domain. For the sake of simplicity, in the following discussion we assume that $\mathrm{Proj}_{\mathcal{W}}$ can be *exactly* implemented by a MLP layer (e.g. $\mathcal{W} = \mathsf{B}_\infty(R_w)$ for some $R_w > 0$).

**Theorem D.3** (Approximate ICGD on two-layer NNs). *Fix any* $B_v, B_u > 0$, $L \geq 1$, $\eta > 0$, *and* $\varepsilon > 0$. *Suppose that*

1. *Both the activation function* $r$ *and the loss function* $\ell$ *is* $C^4$*-smooth;*

2. $\mathcal{W}$ *is a closed domain such that* $\mathcal{W} \subset \{\mathbf{w} = [\mathbf{v}_k; u_k]_{k \in [K]} \in \mathbb{R}^{K(d+1)} : \|\mathbf{v}_k\|_2 \leq B_v, |u_k| \leq B_u\}$, *and* $\mathrm{Proj}_{\mathcal{W}} = \mathrm{MLP}_{\boldsymbol{\theta}_{\mathtt{mlp}}}$ *for some MLP layer* $\boldsymbol{\theta}_{\mathtt{mlp}}$ *with hidden dimension* $D_w$ *and* $\big\|\boldsymbol{\theta}_{\mathtt{mlp}}\big\| \leq C_w$;

*Then there exists a* $(2L)$*-layer transformer* $\mathrm{TF}_{\boldsymbol{\theta}}$ *with*

$$\max_{\ell \in [2L]} M^{(\ell)} \leq \widetilde{\mathcal{O}}\left(\varepsilon^{-2}\right), \qquad \max_{\ell \in [2L]} D^{(\ell)} \leq \widetilde{\mathcal{O}}\left(\varepsilon^{-2}\right) + D_w, \qquad \|\boldsymbol{\theta}\| \leq \mathcal{O}\left(1 + \eta\right) + C_w,$$

*where* $\mathcal{O}(\cdot)$ *hides the constants that depend on* $K$, *the radius parameters* $B_x, B_y, B_u, B_v$ *and the smoothness of* $r$ *and* $\ell$, *such that for* any *input data* $(\mathcal{D}, \mathbf{x}_{N+1})$ *such that input sequence* $\mathbf{H}^{(0)} \in \mathbb{R}^{D \times (N+1)}$ *takes form* (3), $\mathrm{TF}_{\boldsymbol{\theta}}(\mathbf{H}^{(0)})$ *approximately implements in-context gradient descent on risk* (13): *For every* $\ell \in [L]$, *the* $2\ell$*-th layer's output* $\mathbf{h}_i^{(2\ell)} = [\mathbf{x}_i; y_i'; \widehat{\mathbf{w}}^\ell; \mathbf{0}; 1; t_i]$ *for every* $i \in [N+1]$, *and*

$$\widehat{\mathbf{w}}^\ell = \mathrm{Proj}_{\mathcal{W}}\left(\widehat{\mathbf{w}}^{\ell-1} - \eta(\nabla\widehat{L}_N(\widehat{\mathbf{w}}^{\ell-1}) + \boldsymbol{\varepsilon}^{\ell-1})\right), \qquad \widehat{\mathbf{w}}^0 = \mathbf{0}, \qquad (14)$$

*where* $\big\|\boldsymbol{\varepsilon}^{\ell-1}\big\|_2 \leq \varepsilon$ *is an error term.*

As a direct corollary, the transformer constructed above can approximate the true gradient descent trajectory $\{\mathbf{w}_{\mathrm{GD}}^\ell\}_{\ell \geq 0}$ on (16), defined as $\mathbf{w}_{\mathrm{GD}}^0 = \mathbf{0}$ and $\mathbf{w}_{\mathrm{GD}}^{\ell+1} = \mathbf{w}_{\mathrm{GD}}^\ell - \eta\nabla\widehat{L}_N(\mathbf{w}_{\mathrm{GD}}^\ell)$ for all $\ell \geq 0$.

**Corollary D.1** (Approximating multi-step ICGD on two-layer NNs)**.** *For any $L \geq 1$, under the same setting as Theorem D.3, the $(2L)$-layer transformer $\mathrm{TF}_{\boldsymbol{\theta}}$ there approximates the true gradient descent trajectory $\{\mathbf{w}_{\mathrm{GD}}^{\ell}\}_{\ell \geq 0}$: For the intermediate iterates $\{\widehat{\mathbf{w}}^{\ell}\}_{\ell \in [L]}$ considered therein, we have*

$$\left\| \widehat{\mathbf{w}}^{\ell} - \mathbf{w}_{\mathrm{GD}}^{\ell} \right\|_2 \leq L_f^{-1}(1 + \eta L_f)^{\ell} \varepsilon,$$

*where $L_f = \sup_{\mathbf{w} \in \mathcal{W}} \left\| \nabla^2 \widehat{L}_N(\mathbf{w}) \right\|_{\mathrm{op}}$ denotes the smoothness of $\widehat{L}_N$ within $\mathcal{W}$.*

**Remark on error accumulation**   Note that in Corollary D.1, the error accumulates *exponentially* in $\ell$ rather than linearly as in Theorem D.1. This is as expected, since gradient descent on non-convex objectives is inherently unstable at a high level (a slight error added upon each step may result in a drastically different trajectories); technically, this happens as the stability-like property Lemma D.1 no longer holds for the non-convex case.

Corollary D.1 is a simple implication of Theorem D.3 and Part (b) of the following convergence and trajectory closeness result for inexact gradient descent. For any closed convex set $\mathcal{W} \subset \mathbb{R}^d$, any function $f : \mathcal{W} \to \mathbb{R}$, and any initial point $\mathbf{w} \in \mathcal{W}$, let

$$\mathsf{G}_{\mathcal{W},\eta}^{f}(\mathbf{w}) := \frac{\mathbf{w} - \mathrm{Proj}_{\mathcal{W}}(\mathbf{w} - \eta \nabla f(\mathbf{w}))}{\eta}$$

denote the gradient mapping at $\mathbf{w}$ with step size $\eta$, a standard measure of stationarity in constrained optimization [63]. Note that $\mathsf{G}_{\mathcal{W},\eta}^{f}(\mathbf{w}) = \nabla f(\mathbf{w})$ when $\mathbf{w} - \eta \nabla f(\mathbf{w}) \in \mathcal{W}$ (so that the projection does not take effect).

**Lemma D.3** (Convergence and trajectory closeness of inexact GD)**.** *Suppose $f : \mathcal{W} \to \mathbb{R}$, where $\mathcal{W} \subset \mathbb{R}^d$ is a convex closed domain and $\nabla f$ is $L_f$-Lipschitz on $\mathcal{W}$. Let sequence $\{\widehat{\mathbf{w}}^{\ell}\}_{\ell \geq 0} \subset \mathbb{R}^d$ be given by $\widehat{\mathbf{w}}^0 = \mathbf{w}^0$,*

$$\widehat{\mathbf{w}}^{\ell+1} = \mathrm{Proj}_{\mathcal{W}} \left( \widehat{\mathbf{w}}^{\ell} - \eta(\nabla f(\widehat{\mathbf{w}}^{\ell}) + \boldsymbol{\varepsilon}^{\ell}) \right), \qquad \| \boldsymbol{\varepsilon}^{\ell} \|_2 \leq \varepsilon,$$

*for all $\ell \geq 0$. Then the following holds.*

*(a) As long as $\eta \leq 1/L_f$, for all $L \geq 1$,*

$$\min_{\ell \in [L-1]} \left\| \mathsf{G}_{\mathcal{W},\eta}^{f}(\widehat{\mathbf{w}}^{\ell}) \right\|_2^2 \leq \frac{1}{L} \sum_{\ell=0}^{L-1} \left\| \mathsf{G}_{\mathcal{W},\eta}^{f}(\widehat{\mathbf{w}}^{\ell}) \right\|_2^2 \leq \frac{8(f(\mathbf{w}^0) - \inf_{\mathbf{w} \in \mathcal{W}} f(\mathbf{w}))}{\eta L} + 10 \varepsilon^2.$$

*(b) Let the sequences $\{\mathbf{w}_{\mathrm{GD}}^{\ell}\}_{\ell \geq 0} \subset \mathbb{R}^d$ and be given by $\mathbf{w}_{\mathrm{GD}}^0 = \mathbf{w}^0$ and $\mathbf{w}_{\mathrm{GD}}^{\ell+1} = \mathrm{Proj}_{\mathcal{W}}(\mathbf{w}_{\mathrm{GD}}^{\ell} - \eta \nabla f(\mathbf{w}_{\mathrm{GD}}^{\ell}))$. Then it holds that*

$$\left\| \widehat{\mathbf{w}}^{\ell} - \mathbf{w}_{\mathrm{GD}}^{\ell} \right\|_2 \leq L_f^{-1}(1 + \eta L_f)^{\ell} \varepsilon, \qquad \forall \ell \geq 0.$$

# E    Proofs for Section D

## E.1    Approximating a single GD step

**Proposition E.1** (Approximating a single GD step by a single attention layer)**.** *Let $\ell(\cdot, \cdot) : \mathbb{R}^2 \to \mathbb{R}$ be a loss function such that $\partial_1 \ell$ is $(\varepsilon, R, M, C)$-approximable by sum of relus with $R = \max\{B_x B_w, B_y, 1\}$. Let $\widehat{L}_N(\mathbf{w}) := \frac{1}{N} \sum_{i=1}^{N} \ell(\mathbf{w}^{\top} \mathbf{x}_i, y_i)$ denote the empirical risk with loss function $\ell$ on dataset $\{(\mathbf{x}_i, y_i)\}_{i \in [N]}$.*

*Then, for any $\varepsilon > 0$, there exists an attention layer $\boldsymbol{\theta} = \{(\mathbf{Q}_m, \mathbf{K}_m, \mathbf{V}_m)\}_{m \in [M]}$ with $M$ heads such that, for any input sequence that takes form $\mathbf{h}_i = [\mathbf{x}_i; y_i'; \mathbf{w}; \mathbf{0}_{D-2d-3}; 1; t_i]$ with $\|\mathbf{w}\|_2 \leq B_w$, it gives output $\widetilde{\mathbf{h}}_i = [\mathrm{Attn}_{\boldsymbol{\theta}}(\mathbf{H})]_i = [\mathbf{x}_i; y_i'; \widetilde{\mathbf{w}}; \mathbf{0}_{D-2d-3}; 1; t_i]$ for all $i \in [N+1]$, where*

$$\left\| \widetilde{\mathbf{w}} - (\mathbf{w} - \eta \nabla \widehat{L}_N(\mathbf{w})) \right\|_2 \leq \varepsilon \cdot (\eta B_x).$$

*Further, $\|\boldsymbol{\theta}\| \leq 2 + R + 2\eta C$.*

*Proof of Proposition E.1.* As $\partial_s \ell$ is $(\varepsilon, R, M, C)$-approximable by sum of relus, there exists a function $f : [-R, R]^2 \to \mathbb{R}$ of form

$$f(s,t) = \sum_{m=1}^{M} c_m \sigma(a_m s + b_m t + d_m) \quad \text{with} \quad \sum_{m=1}^{M} |c_m| \le C, \ |a_m| + |b_m| + |d_m| \le 1, \ \forall m \in [M],$$

such that $\sup_{(s,t) \in [-R,R]^2} |f(s,t) - \partial_s \ell(s,t)| \le \varepsilon$.

Next, for every $m \in [M]$, we define matrices $\mathbf{Q}_m, \mathbf{K}_m, \mathbf{V}_m \in \mathbb{R}^{D \times D}$ such that

$$\mathbf{Q}_m \mathbf{h}_i = \begin{bmatrix} a_m \mathbf{w} \\ b_m \\ d_m \\ -2 \\ 0 \end{bmatrix}, \quad \mathbf{K}_m \mathbf{h}_j = \begin{bmatrix} \mathbf{x}_j \\ y'_j \\ 1 \\ R(1 - t_j) \\ 0 \end{bmatrix}, \quad \mathbf{V}_m \mathbf{h}_j = -\frac{(N+1)\eta c_m}{N} \cdot \begin{bmatrix} \mathbf{0}_d \\ 0 \\ \mathbf{x}_j \\ \mathbf{0}_{D-2d-1} \end{bmatrix}$$

for all $i, j \in [N+1]$. As the input has structure $\mathbf{h}_i = [\mathbf{x}_i; y'_i; \mathbf{w}; \mathbf{0}_{D-2d-3}; 1; t_i]$, these matrices indeed exist, and further it is straightforward to check that they have norm bounds

$$\max_{m \in [M]} \|\mathbf{Q}_m\|_{\mathrm{op}} \le 3, \quad \max_{m \in [M]} \|\mathbf{K}_m\|_{\mathrm{op}} \le 2 + R, \quad \sum_{m \in [M]} \|\mathbf{V}_m\|_{\mathrm{op}} \le 2\eta C.$$

Consequently, $\|\boldsymbol{\theta}\| \le 2 + R + 2\eta C$.

Now, for every $i, j \in [N+1]$, we have

$$\sigma(\langle \mathbf{Q}_m \mathbf{h}_i, \mathbf{K}_m \mathbf{h}_j \rangle) = \sigma\big(a_m \mathbf{w}^\top \mathbf{x}_j + b_m(1 - t_j)y_j + d_m - 2Rt_j\big)$$
$$= \sigma\big(a_m \mathbf{w}^\top \mathbf{x}_j + b_m y_j + d_m\big)\mathbf{1}\{t_j = 1\},$$

where the last equality follows from the bound

$$\big|a_m \mathbf{w}^\top \mathbf{x}_j + b_m(1 - t_j)y_j + d_m\big| \le |a_m| B_x B_w + R \le 2R, \tag{15}$$

so that the above relu equals $0$ if $t_j \le 0$. Therefore,

$$\sum_{m=1}^{M} \sigma(\langle \mathbf{Q}_m \mathbf{h}_i, \mathbf{K}_m \mathbf{h}_j \rangle)\mathbf{V}_m \mathbf{h}_j$$
$$= \left( \sum_{m=1}^{M} c_m \sigma\big(a_m \mathbf{w}^\top \mathbf{x}_j + b_m y_j + d_m\big) \right) \cdot \frac{-(N+1)\eta}{N}\mathbf{1}\{t_j = 0\}[\mathbf{0}_{d+1}; \mathbf{x}_j; \mathbf{0}_2]$$
$$= f(\mathbf{w}^\top \mathbf{x}_j, y_j) \cdot \frac{-(N+1)\eta}{N}\mathbf{1}\{t_j = 0\}[\mathbf{0}_{d+1}; \mathbf{x}_j; \mathbf{0}_{D-2d-1}].$$

Thus letting the attention layer $\boldsymbol{\theta} = \{(\mathbf{V}_m, \mathbf{Q}_m, \mathbf{K}_m)\}_{m \in [M]}$, we have

$$\widetilde{\mathbf{h}}_i = [\mathrm{Attn}_{\boldsymbol{\theta}}(\mathbf{H})]_i = \mathbf{h}_i + \frac{1}{N+1} \sum_{j=1}^{N+1} \sum_{m=1}^{M} \sigma(\langle \mathbf{Q}_m \mathbf{h}_i, \mathbf{K}_m \mathbf{h}_j \rangle)\mathbf{V}_m \mathbf{h}_j$$

$$= \mathbf{h}_i - \frac{\eta}{N} \sum_{j=1}^{N} f(\mathbf{w}^\top \mathbf{x}_j, y_j)[\mathbf{0}_{d+1}; \mathbf{x}_j; \mathbf{0}_2]$$

$$= [\mathbf{x}_i; y_i; \mathbf{w}; 1; t_i] \underbrace{- \frac{\eta}{N} \sum_{j=1}^{N} \partial_s \ell(\mathbf{w}^\top \mathbf{x}_j, y_j)[\mathbf{0}_{d+1}; \mathbf{x}_j; \mathbf{0}_{D-2d-1}] + [\mathbf{0}_{d+1}; \boldsymbol{\varepsilon}; \mathbf{0}_{D-2d-1}]}_{[\mathbf{0}_{d+1}; -\eta \nabla \widehat{L}_N(\mathbf{w}); \mathbf{0}_{D-2d-1}]}$$

$$= [\mathbf{x}_i; y_i; \mathbf{w}_\eta^+ + \boldsymbol{\varepsilon}; \mathbf{0}_{D-2d-3}; 1; t_i],$$

where the error vector $\boldsymbol{\varepsilon} \in \mathbb{R}^d$ satisfies

$$\|\boldsymbol{\varepsilon}\|_2 = \left\| -\frac{\eta}{N} \sum_{j=1}^{N} \big(f(\mathbf{w}^\top \mathbf{x}_j, y_j) - \partial_s \ell(\mathbf{w}^\top \mathbf{x}_j, y_j)\big)\mathbf{x}_j \right\|_2$$

$$\leq \frac{\eta}{N} \sum_{j=1}^{N} \left| f(\mathbf{w}^\top \mathbf{x}_j, y_j) - \partial_s \ell(\mathbf{w}^\top \mathbf{x}_j, y_j) \right| \cdot \|\mathbf{x}_j\|_2$$

$$\leq \frac{\eta}{N} \cdot N \cdot \varepsilon \cdot B_x = \varepsilon \cdot (\eta B_x).$$

This is the desired result. $\qquad\qquad\square$

### E.2  Proof of Theorem D.1

We first prove part (a), which requires constructing the first $L$ layers of $\boldsymbol{\theta}$. Note that by our precondition $L \leq B_w/(2\varepsilon)$.

By our precondition, the partial derivative of the loss $\partial_s \ell$ is $(\varepsilon, R, M, C)$-approximable by sum of relus. Therefore we can apply Proposition E.1 to obtain that, there exists a single attention layer $\boldsymbol{\theta}^{(1)} = \{(\mathbf{Q}_m, \mathbf{K}_m, \mathbf{V}_m)\}_{m \in [M]}$ with $M$ heads (and norm bounds specified in Proposition E.1), such that for any $\mathbf{w}$ with $\|\mathbf{w}\|_2 \leq B_w$, the attention layer $\mathrm{Attn}_{\boldsymbol{\theta}^{(1)}}$ maps the input $\mathbf{h}_i = [\mathbf{x}_i; y_i'; \mathbf{w}; \mathbf{0}_{D-2d-3}; 1; t_i]$ to output $\mathbf{h}_i' = [\mathbf{x}_i; y_i'; \widehat{\mathbf{w}}; \mathbf{0}_{D-2d-3}; 1; t_i]$ for all $i \in [N+1]$, where

$$\left\| \widehat{\mathbf{w}} - \left( \mathbf{w} - \eta \nabla \widehat{L}_N(\mathbf{w}) \right) \right\|_2 \leq \varepsilon \cdot (\eta B_x) =: \varepsilon'.$$

Consider the $L$-layer transformer $\boldsymbol{\theta}^{1:L} = (\boldsymbol{\theta}^{(1)}, \dots, \boldsymbol{\theta}^{(1)})$ which stacks the same attention layer $\boldsymbol{\theta}^{(1)}$ for $L$ times, and for the given input $\mathbf{h}_i^{(0)} = [\mathbf{x}_i; y_i'; \mathbf{w}^0; \mathbf{0}_{D-2d-3}; 1; t_i]$, its $\ell$-th layer's output $\mathbf{h}_i^{(\ell)} = [\mathbf{x}_i; y_i'; \widehat{\mathbf{w}}^\ell; \mathbf{0}_{D-2d-3}; 1; t_i]$.

We now inductively show that $\|\widehat{\mathbf{w}}^\ell\|_2 \leq B_w$ and $\|\widehat{\mathbf{w}}^\ell - \mathbf{w}_{\mathrm{GD}}^\ell\|_2 \leq \ell\varepsilon$ for all $\ell \in [L]$. The base case of $\ell = 0$ is trivial. Suppose the claim holds for $\ell$. Then for $\ell + 1 \leq L \leq B_w/(2\varepsilon)$, the sequence $\{\widehat{\mathbf{w}}^i\}_{i \leq \ell+1}$ and $\{\mathbf{w}_{\mathrm{GD}}^i\}_{i \leq \ell+1}$ satisfies the precondition of the error composition lemma (Lemma D.1) with error bound $\varepsilon$, from which we obtain $\|\widehat{\mathbf{w}}^{\ell+1}\|_2 \leq B_w$ and

$$\left\| \widehat{\mathbf{w}}^{\ell+1} - \mathbf{w}_{\mathrm{GD}}^{\ell+1} \right\|_2 \leq (\ell+1)\varepsilon'.$$

This finishes the induction, and gives the following approximation guarantee for all $\ell \in [L]$:

$$\left\| \widehat{\mathbf{w}}^\ell - \mathbf{w}_{\mathrm{GD}}^\ell \right\|_2 \leq \ell\varepsilon' \leq \varepsilon \cdot (L\eta B_x),$$

which proves part (a).

We now prove part (b), which requires constructing the last attention layer $\boldsymbol{\theta}^{(L+1)}$. Recall $\mathbf{h}_i^{(L)} = [\mathbf{x}_i; y_i'; \widehat{\mathbf{w}}^L; \mathbf{0}_{D-2d-3}; 1; t_i]$ for all $i \in [N+1]$. We construct a 2-head attention layer $\boldsymbol{\theta}^{(L+1)} = \{(\mathbf{Q}_m^{(L+1)}, \mathbf{K}_m^{(L+1)}, \mathbf{V}_m^{(L+1)})\}_{m=1,2}$ such that for every $i, j \in [N+1]$,

$$\mathbf{Q}_1^{(L+1)} \mathbf{h}_i^{(L)} = [\mathbf{x}_i; \mathbf{0}_{D-d}], \quad \mathbf{K}_1^{(L+1)} \mathbf{h}_j^{(L)} = [\widehat{\mathbf{w}}^L; \mathbf{0}_{D-d}], \quad \mathbf{V}_1^{(L+1)} \mathbf{h}_j^{(L)} = [\mathbf{0}_d; 1; \mathbf{0}_{D-d-1}],$$

$$\mathbf{Q}_2^{(L+1)} \mathbf{h}_i^{(L)} = [\mathbf{x}_i; \mathbf{0}_{D-d}], \quad \mathbf{K}_2^{(L+1)} \mathbf{h}_j^{(L)} = [-\widehat{\mathbf{w}}^L; \mathbf{0}_{D-d}], \quad \mathbf{V}_2^{(L+1)} \mathbf{h}_j^{(L)} = [\mathbf{0}_d; -1; \mathbf{0}_{D-d-1}].$$

Note that the weight matrices have norm bound

$$\max_{i=1,2} \left\| \mathbf{Q}_i^{(L+1)} \right\|_{\mathrm{op}} \leq 1, \quad \max_{i=1,2} \left\| \mathbf{K}_i^{(L+1)} \right\|_{\mathrm{op}} \leq 1, \quad \sum_{i=1}^{2} \left\| \mathbf{V}_i^{(L+1)} \right\|_{\mathrm{op}} \leq 2.$$

Then we have

$$\mathbf{h}_{N+1}^{(L+1)} = \mathbf{h}_{N+1}^{(L)} + \frac{1}{N+1} \sum_{j=1}^{N+1} \sum_{m=1}^{2} \sigma\Big( \big\langle \mathbf{Q}^{(L+1)} \mathbf{h}_{N+1}^{(L)}, \mathbf{K}^{(L+1)} \mathbf{h}_j^{(L)} \big\rangle \Big) \mathbf{V}^{(L+1)} \mathbf{h}_j^{(L)}$$

$$= [\mathbf{x}_i; 0; \widehat{\mathbf{w}}^L; \mathbf{0}_{D-2d-3}; 1; 1] + \big( \sigma(\langle \widehat{\mathbf{w}}^L, \mathbf{x}_{N+1} \rangle) - \sigma(-\langle \widehat{\mathbf{w}}^L, \mathbf{x}_{N+1} \rangle) \big) \cdot [\mathbf{0}_d; 1; \mathbf{0}_{D-d-1}]$$

$$\overset{(i)}{=} [\mathbf{x}_i; 0; \widehat{\mathbf{w}}^L; \mathbf{0}_{D-2d-3}; 1; 1] + [\mathbf{0}_d; \langle \widehat{\mathbf{w}}^L, \mathbf{x}_{N+1} \rangle; \mathbf{0}_{D-d-1}]$$

$$= [\mathbf{x}_i; \underbrace{\langle \widehat{\mathbf{w}}^L, \mathbf{x}_{N+1} \rangle}_{\widehat{y}_{N+1}}; \widehat{\mathbf{w}}^L; \mathbf{0}_{D-2d-3}; 1; 1],$$

Above, (i) uses the identity $t = \sigma(t) - \sigma(-t)$. Further by part (a) we have

$$\left|\widehat{y}_{N+1} - \left\langle \mathbf{w}_{\mathrm{GD}}^L, \mathbf{x}_{N+1} \right\rangle\right| = \left|\left\langle \widehat{\mathbf{w}}^L - \mathbf{w}_{\mathrm{GD}}^L, \mathbf{x}_{N+1} \right\rangle\right| \le \varepsilon \cdot (L\eta B_x^2).$$

This proves part (b), and also finishes the proof Theorem D.1 where the overall $(L+1)$-layer attention-only transformer is given by $\mathrm{TF}_{\boldsymbol{\theta}}^0$ with

$$\boldsymbol{\theta} = (\underbrace{\boldsymbol{\theta}^{(1)}, \ldots, \boldsymbol{\theta}^{(1)}}_{L \text{ times}}, \boldsymbol{\theta}^{(L+1)}).$$

$\square$

## E.3  Proof of Lemma D.1

As $f$ is a convex, $L_f$ smooth function on $\mathsf{B}_2^d(R)$, the mapping $\mathcal{T}_\eta : \mathbf{w} \mapsto \mathbf{w} - \eta\nabla f(\mathbf{w})$ is non-expansive in $\|\cdot\|_2$: Indeed, for any $\mathbf{w}, \mathbf{w}' \in \mathsf{B}_2^d(R)$ we have

$$\begin{aligned}
\left\|\mathcal{T}_\eta(\mathbf{w}) - \mathcal{T}_\eta(\mathbf{w}')\right\|_2 &= \left\|\mathbf{w} - \eta\nabla f(\mathbf{w}) - (\mathbf{w}' - \eta\nabla f(\mathbf{w}'))\right\|_2^2 \\
&= \|\mathbf{w} - \mathbf{w}'\|_2^2 - 2\eta\left\langle \mathbf{w} - \mathbf{w}', \nabla f(\mathbf{w}) - \nabla f(\mathbf{w}')\right\rangle + \eta^2\left\|\nabla f(\mathbf{w}) - \nabla f(\mathbf{w}')\right\|_2^2 \\
&\overset{(i)}{\le} \|\mathbf{w} - \mathbf{w}'\|_2^2 - \left(2\eta/L_f - \eta^2\right)\|\nabla f(\mathbf{w}) - \nabla f(\mathbf{w}')\|_2^2 \overset{(ii)}{\le} \|\mathbf{w} - \mathbf{w}'\|_2^2.
\end{aligned}$$

Above, (i) uses the property $\langle \mathbf{w} - \mathbf{w}', \nabla f(\mathbf{w}) - \nabla f(\mathbf{w}')\rangle \ge \frac{1}{L_f}\|\nabla f(\mathbf{w}) - \nabla f(\mathbf{w}')\|_2^2$ for smooth convex functions [63, Theorem 2.1.5]; (ii) uses the precondition that $\eta \le 2/L_f$.

The lemma then follows directly by induction on $L$. The base case of $L = 0$ follows directly by assumption that $\widehat{\mathbf{w}}^0 = \mathbf{w}_{\mathrm{GD}}^0 \in \mathsf{B}_2^d(R/2)$. Suppose the claim holds for iterate $L$. For iterate $L + 1 \le R/(2\varepsilon)$, we have

$$\begin{aligned}
\left\|\widehat{\mathbf{w}}^{L+1} - \mathbf{w}_{\mathrm{GD}}^{L+1}\right\|_2 &= \left\|\mathcal{T}_\eta(\widehat{\mathbf{w}}^L) + \varepsilon^L - \mathcal{T}_\eta(\mathbf{w}_{\mathrm{GD}}^L)\right\|_2 \\
&\le \left\|\mathcal{T}_\eta(\widehat{\mathbf{w}}^L) - \mathcal{T}_\eta(\mathbf{w}_{\mathrm{GD}}^L)\right\|_2 + \left\|\varepsilon^L\right\|_2 \\
&\overset{(i)}{\le} \left\|\widehat{\mathbf{w}}^L - \mathbf{w}_{\mathrm{GD}}^L\right\|_2 + \varepsilon \overset{(ii)}{\le} (L+1)\varepsilon.
\end{aligned}$$

Above, (i) uses the non-expansiveness, and (ii) uses the inductive hypothesis. Similarly, by our assumption $\mathbf{w}^\star = \mathcal{T}_\eta(\mathbf{w}^\star)$,

$$\left\|\widehat{\mathbf{w}}^{L+1} - \mathbf{w}^\star\right\|_2 = \left\|\mathcal{T}_\eta(\widehat{\mathbf{w}}^L) + \varepsilon^L - \mathcal{T}_\eta(\mathbf{w}^\star)\right\|_2 \le \left\|\widehat{\mathbf{w}}^L - \mathbf{w}^\star\right\|_2 + \left\|\varepsilon^L\right\|_2 \le \frac{R}{2} + (L+1)\varepsilon \le R.$$

This finishes the induction. $\square$

## E.4  Convex ICGD with $\ell_2$ regularization

In the same setting as Theorem D.1, consider the ICGD dynamics over an $\ell_2$-regularized empirical risk:

$$\mathbf{w}_{\mathrm{GD}}^{t+1} := \mathbf{w}_{\mathrm{GD}}^t - \eta\nabla\widehat{L}_N^\lambda(\mathbf{w}_{\mathrm{GD}}^t) \qquad\qquad \text{(ICGD-}\ell_2\text{)}$$

with initialization $\mathbf{w}_{\mathrm{GD}}^0 \in \mathbb{R}^d$ and learning rate $\eta > 0$, where $\widehat{L}_N^\lambda(\mathbf{w}) := \widehat{L}_N(\mathbf{w}) + \frac{\lambda}{2}\|\mathbf{w}\|_2^2$ denotes the $\ell_2$-regularized empirical risk.

**Corollary E.1** (Convex ICGD with $\ell_2$ regularization)**.** *Fix any $B_w > 0$, $L > 1$, $\eta > 0$, and $\varepsilon < B_x B_w$. Suppose the loss $\ell(\cdot, \cdot)$ is convex in the first argument, and $\partial_s\ell$ is $(\varepsilon, R, M, C)$-approximable by sum of relus with $R = \max\{B_x B_w, B_y, 1\}$.*

*Then, there exists an attention-only transformer $\mathrm{TF}_{\boldsymbol{\theta}}^0$ with $(L+1)$ layers, $\max_{\ell \in [L]} M^{(\ell)} \le M + 1$ heads within the first $L$ layers, and $M^{(L+1)} = 2$ such that for any input data $(\mathcal{D}, \mathbf{x}_{N+1})$ with*

$$\sup_{\|\mathbf{w}\|_2 \le B_w} \lambda_{\max}(\nabla^2\widehat{L}_N^\lambda(\mathbf{w})) \le 2\eta^{-1}, \qquad \exists\mathbf{w}^\star \in \underset{\mathbf{w} \in \mathbb{R}^d}{\arg\min}\,\widehat{L}_N^\lambda(\mathbf{w}) \text{ such that } \|\mathbf{w}^\star\|_2 \le B_w/2,$$

*$\mathrm{TF}_{\boldsymbol{\theta}}^0(\mathbf{H}^{(0)})$ approximately implements (ICGD-$\ell_2$):*

1. *(Parameter space) For every $\ell \in [L]$, the $\ell$-th layer's output $\mathbf{H}^{(\ell)} = \mathrm{TF}_{\boldsymbol{\theta}^{(1:\ell)}}(\mathbf{H}^{(0)})$ approximates $\ell$ steps of (ICGD-$\ell_2$): We have $\mathbf{h}_i^{(\ell)} = [\mathbf{x}_i; y_i'; \widehat{\mathbf{w}}^\ell; \mathbf{0}_{D-2d-3}; 1; t_i]$ for every $i \in [N+1]$, where*

$$\left\| \widehat{\mathbf{w}}^\ell - \mathbf{w}_{\mathrm{GD}}^\ell \right\|_2 \leq \varepsilon \cdot (2L\eta B_x).$$

2. *(Prediction space) The final output $\mathbf{H}^{(L+1)} = \mathrm{TF}_{\boldsymbol{\theta}}(\mathbf{H}^{(0)})$ approximates the prediction of $L$ steps of (ICGD-$\ell_2$): We have $\mathbf{h}_{N+1}^{(L+1)} = [\mathbf{x}_{N+1}; \widehat{y}_{N+1}; \widehat{\mathbf{w}}^L; \mathbf{0}_{D-2d-3}; 1; 0]$, where*

$$\left| \widehat{y}_{N+1} - \left\langle \mathbf{w}_{\mathrm{GD}}^L, \mathbf{x}_{N+1} \right\rangle \right| \leq \varepsilon \cdot (2L\eta B_x^2).$$

*Further, the transformer admits norm bound $\|\boldsymbol{\theta}\| \leq 2 + R + (2C + \lambda)\eta$.*

*Proof.* This construction is the same as in the proof of Theorem D.1, except that within each layer $\ell \in [L]$, we add one more attention head $(\mathbf{Q}^{(\ell)}, \mathbf{K}^{(\ell)}, \mathbf{V}^{(\ell)}) \subset \mathbb{R}^{D \times D}$ which when acting on its input $\mathbf{h}_i^{(\ell-1)} = [*; *; \widehat{\mathbf{w}}^{\ell-1}; 1; *]$ gives

$$\mathbf{Q}^{(\ell)}\mathbf{h}_i^{(\ell-1)} = \begin{bmatrix} 1 \\ \mathbf{0}_{D-1} \end{bmatrix}, \quad \mathbf{K}^{(\ell)}\mathbf{h}_j^{(\ell-1)} = \begin{bmatrix} 1 \\ \mathbf{0}_{D-1} \end{bmatrix}, \quad \mathbf{V}^{(\ell)}\mathbf{h}_j^{(\ell-1)} = \begin{bmatrix} \mathbf{0}_{d+1} \\ -\eta\lambda\widehat{\mathbf{w}}^{\ell-1} \\ \mathbf{0}_2 \end{bmatrix}$$

for all $i, j \in [N+1]$. Note that $\left\|\mathbf{Q}^{(\ell)}\right\|_{\mathrm{op}} = \left\|\mathbf{K}^{(\ell)}\right\|_{\mathrm{op}} = 1$, and $\left\|\mathbf{V}^{(\ell)}\right\|_{\mathrm{op}} = \eta\lambda$. Further, it is straightforward to check that the output of this attention head on every $\mathbf{h}_i^{(\ell)}$ is

$$\frac{1}{N+1} \sum_{j=1}^{N+1} \sigma\left(\left\langle \mathbf{Q}^{(\ell)}\mathbf{h}_i^{(\ell-1)}, \mathbf{K}^{(\ell)}\mathbf{h}_j^{(\ell-1)} \right\rangle\right) \mathbf{V}^{(\ell)}\mathbf{h}_j^{(\ell-1)} = \begin{bmatrix} \mathbf{0}_{d+1} \\ -\eta\lambda\widehat{\mathbf{w}}^{\ell-1} \\ \mathbf{0}_2 \end{bmatrix}.$$

Adding this onto the original output of the $\ell$-th layer exactly implements the gradient of the regularizer $\mathbf{w} \mapsto \frac{\lambda}{2}\|\mathbf{w}\|_2^2$. The rest of the proof follows by repeating the argument of Theorem D.1, and combining the norm bound for the additional attention head here with the norm bound therein. $\qquad\square$

### E.5 Proof of Theorem D.3

We only need to prove the following single-step version of Theorem D.3.

**Proposition E.2.** *Under the assumptions of Theorem D.3, there exists a 2-layer transformer $\mathrm{TF}_{\boldsymbol{\theta}}$ with the same bounds on the number of heads, hidden dimension and the norm, such that for any input data $(\mathcal{D}, \mathbf{x}_{N+1})$ and any $\mathbf{w} \in \mathbb{R}^d$, $\mathrm{TF}_{\boldsymbol{\theta}}$ maps*

$$\mathbf{h}_i = [\mathbf{x}_i; y_i'; \mathbf{w}; \mathbf{0}; 1; t_i] \quad \rightarrow \quad \mathbf{h}_i' = [\mathbf{x}_i; y_i'; \mathbf{w}_\eta^+; \mathbf{0}; 1; t_i],$$

*where*

$$\mathbf{w}_\eta^+ = \mathrm{Proj}_{\mathcal{W}}\left(\mathbf{w} - \eta\nabla\widehat{L}_N(\mathbf{w}) + \boldsymbol{\varepsilon}(\mathbf{w})\right), \qquad \|\boldsymbol{\varepsilon}(\mathbf{w})\|_2 \leq \eta\varepsilon.$$

Before we present the formal (and technical) proof of Proposition E.2, we first provide some intuitions. To begin with, we first note that

$$\nabla_{\mathbf{w}}\widehat{L}_N(\mathbf{w}) = \frac{1}{N} \sum_{i=1}^N \partial_1 \ell(\mathrm{pred}(\mathbf{x}_i; \mathbf{w}), y_i) \cdot \nabla_{\mathbf{w}}\mathrm{pred}(\mathbf{x}_i; \mathbf{w}), \tag{16}$$

where $\partial_1\ell$ is the partial derivative of $\ell$ with respect to the first component, and

$$\nabla_{\mathbf{w}}\mathrm{pred}(\mathbf{x}_i; \mathbf{w}) = \begin{bmatrix} u_1 \cdot r'(\langle \mathbf{v}_1, \mathbf{x}_i \rangle) \cdot \mathbf{x}_i \\ r(\langle \mathbf{v}_1, \mathbf{x}_i \rangle) \\ \vdots \\ u_K \cdot r'(\langle \mathbf{v}_K, \mathbf{x}_i \rangle) \cdot \mathbf{x}_i \\ r(\langle \mathbf{v}_K, \mathbf{x}_i \rangle) \end{bmatrix} \in \mathbb{R}^{K(d+1)}. \tag{17}$$

Therefore, the basic idea is that we can use an attention layer to approximate $(\mathbf{x}_i, \mathbf{w}) \mapsto \mathrm{pred}(\mathbf{x}_i; \mathbf{w})$, then use an MLP layer to implement $(\mathrm{pred}(\mathbf{x}_i; \mathbf{w}), y_i', t_i) \mapsto \mathbb{1}\{i < N+1\} \cdot \partial_1 \ell(\mathrm{pred}(\mathbf{x}_i; \mathbf{w}), y_i)$, and then use an attention layer to compute the gradient descent step $\mathbf{w} \mapsto \mathbf{w} - \eta \nabla L_N(\mathbf{w})$, and finally use an MLP layer to implement the projection into $\mathcal{W}$.

Based on the observations above, we now present the proof of Proposition E.2.

*Proof of Proposition E.2.* We write $D_0 = d + 1 + K(d+1)$ be the length of the vector $[\mathbf{x}_i; y_i; \mathbf{w}]$. We also define

$$B_r := \max_{|t| \leq B_x B_u} |r(t)|, \qquad B_g := \max_{|t| \leq K B_r, |y| \leq B_y} |\partial_t \ell(t, y)|.$$

Let us fix $\varepsilon_r, \varepsilon_p, \varepsilon_\ell > 0$ that will be specified later in proof (see (18)). By our assumption and Proposition B.1, the following facts hold.

(1) The function $r(t)$ is $(\varepsilon_r, R_1, M_1, C_1)$ for $R_1 = \max\{B_x B_u, 1\}$, $M_1 \leq \widetilde{\mathcal{O}}\left(C_1^2 \varepsilon_r^{-2}\right)$, where $C_1$ depends only on $R_1$ and the $C^2$-smoothness of $r$. Therefore, there exists

$$\bar{r}(t) = \sum_{m=1}^{M} c_m^1 \sigma(\langle \mathbf{a}_m^1, [t; 1] \rangle) \quad \text{with} \quad \sum_{m=1}^{M} |c_m^1| \leq C_1, \ \|\mathbf{a}_m^1\|_1 \leq 1, \ \forall m \in [M_1],$$

such that $\sup_{t \in [-R_1, R_1]} |r(t) - \bar{r}(t)| \leq \varepsilon_r$.

(2) The function $(t, y) \mapsto \partial_1 \ell(t, y)$ is $(\varepsilon_\ell, R_2, M_2, C_2)$ for $R_2 = \max\{K B_r, B_y, 1\}$ $M_2 \leq \widetilde{\mathcal{O}}\left(C_2^2 \varepsilon_\ell^{-2}\right)$, where $C_2$ depends only on $R_2$ and the $C^3$-smoothness of $\partial_1 \ell$. Therefore, there exists

$$g(t, y) = \sum_{m=1}^{M} c_m^2 \sigma(\langle \mathbf{a}_m^2, [t; y; 1] \rangle) \quad \text{with} \quad \sum_{m=1}^{M} |c_m^2| \leq C_2, \ \|\mathbf{a}_m^2\|_1 \leq 1, \ \forall m \in [M_2],$$

such that $\sup_{(t,y) \in [-R_2, R_2]^2} |g(t, y) - \partial_1 \ell(t, y)| \leq \varepsilon_\ell$.

(3) The function $(s, t) \mapsto s \cdot r'(t)$ is $(\varepsilon_p, R_3, M_3, C_3)$ for $R_3 = \max\{B_x B_u, B_g B_u, 1\}$, $M_3 \leq \widetilde{\mathcal{O}}\left(C_3^2 \varepsilon_p^{-2}\right)$, where $C_3$ depends only on $R_3$ and the $C^3$-smoothness of $r'$. Therefore, there exists

$$P(s, t) = \sum_{m=1}^{M} c_m^3 \sigma(\langle \mathbf{a}_m^3, [s; t; 1] \rangle) \quad \text{with} \quad \sum_{m=1}^{M} |c_m^3| \leq C_3, \ \|\mathbf{a}_m^3\|_1 \leq 1, \ \forall m \in [M_3],$$

such that $\sup_{(s,t) \in [-R_3, R_3]^2} |P(s, t) - s \cdot r'(t)| \leq \varepsilon_p$.

In the following, we proceed to construct the desired transformer step by step.

Step 1: construction of $\boldsymbol{\theta}_{\mathtt{attn}}^{(1)}$. We consider the matrices $\{\mathbf{Q}_{k,m}^{(1)}, \mathbf{K}_{k,m}^{(1)}, \mathbf{V}_{k,m}^{(1)}\}_{k \in [K], m \in [M_1]}$ so that for all $i, j \in [N+1]$, we have

$$\mathbf{Q}_{k,m}^{(1)} \mathbf{h}_i = \begin{bmatrix} \mathbf{a}_m^1[1] \cdot \mathbf{x}_i \\ \mathbf{a}_m^1[2] \\ \mathbf{0} \end{bmatrix}, \quad \mathbf{K}_{k,m}^{(1)} \mathbf{h}_j = \begin{bmatrix} \mathbf{v}_k \\ 1 \\ \mathbf{0} \end{bmatrix}, \quad \mathbf{V}_{k,m}^{(1)} \mathbf{h}_j = c_m^1 \cdot u_k \mathbf{e}_{D_0+1}.$$

As the input has structure $\mathbf{h}_i = [\mathbf{x}_i; y_i'; \mathbf{w}; \mathbf{0}; 1; t_i]$, these matrices indeed exist, and further it is straightforward to check that they have norm bounds

$$\max_{k,m} \left\|\mathbf{Q}_{k,m}^{(1)}\right\|_{\mathrm{op}} \leq 1, \quad \max_{k,m} \left\|\mathbf{K}_{k,m}^{(1)}\right\|_{\mathrm{op}} \leq 1, \quad \sum_{k,m} \left\|\mathbf{V}_{k,m}^{(1)}\right\|_{\mathrm{op}} \leq C_1.$$

A simple calculation shows that

$$\sum_{m \in [M_1], k \in [K]} \sigma\left(\left\langle \mathbf{Q}_{k,m}^{(1)} \mathbf{h}_i, \mathbf{K}_{k,m}^{(1)} \mathbf{h}_j \right\rangle\right) \mathbf{V}_{k,m}^{(1)} \mathbf{h}_j = \sum_{k=1}^{K} u_k \bar{r}(\langle \mathbf{v}_k, \mathbf{x}_i \rangle) \cdot \mathbf{e}_{D_0+1}.$$

For simplicity, we denote $\overline{\mathrm{pred}}(\mathbf{x};\mathbf{w}) := \sum_{k=1}^K u_k \bar{r}(\langle \mathbf{v}_k, \mathbf{x}\rangle)$ in the following analysis. Thus, letting the attention layer $\boldsymbol{\theta}_{\mathrm{attn}}^{(1)} = \{(\mathbf{V}_{k,m}^{(1)}, \mathbf{Q}_{k,m}^{(1)}, \mathbf{K}_{k,m}^{(1)})\}_{(k,m)}$, we have

$$\mathrm{Attn}_{\boldsymbol{\theta}_{\mathrm{attn}}^{(1)}} : \mathbf{h}_i \mapsto \mathbf{h}_i^{(0.5)} = [\mathbf{x}_i; y_i'; \mathbf{w}; \overline{\mathrm{pred}}(\mathbf{x}_i; \mathbf{w}); \mathbf{0}; 1; t_i].$$

Step 2: construction of $\boldsymbol{\theta}_{\mathrm{mlp}}^{(1)}$. We pick matrices $\mathbf{W}_1, \mathbf{W}_2$ so that $\mathbf{W}_1$ maps

$$\mathbf{W}_1 \mathbf{h}_i^{(0.5)} = \left[ \mathbf{a}_m^2[1] \cdot \overline{\mathrm{pred}}(\mathbf{x}_i; \mathbf{w}) + \mathbf{a}_m^2[2] \cdot y_i' + \mathbf{a}_m^2[3] - R_2(1 - t_i) \right]_{m \in [M_2]} \in \mathbb{R}^{M_2},$$

and $\mathbf{W}_2 \in \mathbb{R}^{D \times M_3}$ with entries being $(\mathbf{W}_2)_{(j,m)} = c_m^2 1\{j = D_0 + 2\}$. It is clear that $\|\mathbf{W}_1\|_{\mathrm{op}} \le R_2 + 1$, $\|\mathbf{W}_2\|_{\mathrm{op}} \le C_2$. Then we have

$$\mathbf{W}_2 \sigma(\mathbf{W}_1 \mathbf{h}_i^{(0.5)}) = \sum_{m \in [M_3]} \sigma\left(\langle \mathbf{a}_m^2, [\overline{\mathrm{pred}}(\mathbf{x}_i; \mathbf{w}); y_i'; 1]\rangle - R_2(1 - t_j)\right) \cdot c_m^2 \mathbf{e}_{D_0+2}$$

$$= 1\{t_j = 1\} \cdot g(\overline{\mathrm{pred}}(\mathbf{x}_i; \mathbf{w}), y_i') \cdot \mathbf{e}_{D_0+2}.$$

In the following, we abbreviate $g_i = 1\{t_j = 1\} \cdot g(\overline{\mathrm{pred}}(\mathbf{x}_i; \mathbf{w}), y_i')$. Hence, $\boldsymbol{\theta}_{\mathrm{mlp}}$ maps

$$\mathrm{MLP}_{\boldsymbol{\theta}_{\mathrm{mlp}}} : \mathbf{h}_i^{(0.5)} \mapsto \mathbf{h}_i^{(1)} = [\mathbf{x}_i; y_i'; \mathbf{w}; \overline{\mathrm{pred}}(\mathbf{x}_i; \mathbf{w}); g_i; \mathbf{0}; 1; t_i].$$

By the definition of the function $g$, for each $i \in [N]$,

$$|g_i - \partial_1 \ell(\mathrm{pred}(\mathbf{x}_i; \mathbf{w}), y_i)| \le \varepsilon_\ell + B_u L_\ell \varepsilon_r,$$

where $L_\ell := \max_{|t| \le KB_r, |y| \le B_y} |\partial_{tt}^2 \ell(t, y)|$ is the smoothness of $\partial_1 \ell$. Also, $g_{N+1} = 0$ by definition.

Step 3: construction of $\boldsymbol{\theta}_{\mathrm{attn}}^{(2)}$. We consider the matrices $\{\mathbf{Q}_{k,1,m}^{(2)}, \mathbf{K}_{k,1,m}^{(2)}, \mathbf{V}_{k,1,m}^{(2)}\}_{k \in [K], m \in [M_3]}$ so that for all $i, j \in [N+1]$, we have

$$\mathbf{Q}_{k,1,m}^{(2)} \mathbf{h}_i^{(1)} = \begin{bmatrix} \mathbf{a}_m^3[1] \cdot u_k \\ \mathbf{a}_m^3[2] \cdot \mathbf{v}_k \\ \mathbf{a}_m^3[3] \\ \mathbf{0} \end{bmatrix}, \qquad \mathbf{K}_{k,1,m}^{(2)} \mathbf{h}_j^{(1)} = \begin{bmatrix} g_j \\ \mathbf{x}_j \\ 1 \\ \mathbf{0} \end{bmatrix}, \qquad \mathbf{V}_{k,1,m}^{(2)} \mathbf{h}_j^{(1)} = -\frac{(N+1)\eta c_m^3}{N} \cdot \begin{bmatrix} \mathbf{0}_{k(d+1)} \\ \mathbf{x}_j \\ \mathbf{0} \end{bmatrix}.$$

We further consider the matrices $\{\mathbf{Q}_{k,2,m}^{(2)}, \mathbf{K}_{k,2,m}^{(2)}, \mathbf{V}_{k,2,m}^{(2)}\}_{k \in [K], m \in [M_1]}$ so that for all $i, j \in [N+1]$, we have

$$\mathbf{Q}_{k,2,m}^{(2)} \mathbf{h}_i^{(1)} = \begin{bmatrix} \mathbf{a}_m^1[1] \cdot \mathbf{v}_k \\ \mathbf{a}_m^1[2] \\ \mathbf{0} \end{bmatrix}, \qquad \mathbf{K}_{k,2,m}^{(2)} \mathbf{h}_j^{(1)} = \begin{bmatrix} \mathbf{x}_j \\ 1 \\ \mathbf{0} \end{bmatrix}, \qquad \mathbf{V}_{k,2,m}^{(2)} \mathbf{h}_j^{(1)} = -\frac{(N+1)\eta c_m^1}{N} \cdot \begin{bmatrix} \mathbf{0}_{k(d+1)+d} \\ g_j \\ \mathbf{0} \end{bmatrix}.$$

By the structure of the input $\mathbf{h}_i^{(1)}$, these matrices indeed exist, and further it is straightforward to check that they have norm bounds

$$\max_{(k,w,m)} \left\|\mathbf{Q}_{k,w,m}^{(2)}\right\|_{\mathrm{op}} \le 1, \qquad \max_{(k,w,m)} \left\|\mathbf{K}_{k,w,m}^{(2)}\right\|_{\mathrm{op}} \le 1, \qquad \sum_{(k,w,m)} \left\|\mathbf{V}_{k,w,m}^{(2)}\right\|_{\mathrm{op}} \le 2\eta C_1 + 2\eta C_3.$$

Furthermore, a simple calculation shows that

$$\mathbf{g}(\mathbf{w}) =: \frac{1}{N+1} \sum_{i=1}^{N+1} \sum_{(k,w,m)} \sigma\left(\langle \mathbf{Q}_{k,w,m}^{(2)} \mathbf{h}_i, \mathbf{K}_{k,w,m}^{(2)} \mathbf{h}_j\rangle\right) \mathbf{V}_{k,w,m}^{(2)} \mathbf{h}_j = -\frac{\eta}{N} \sum_{j=1}^{N+1} \begin{bmatrix} \mathbf{0}_{d+1} \\ P(u_1 g_j, \langle \mathbf{v}_1, \mathbf{x}_j\rangle) \cdot \mathbf{x}_j \\ \bar{r}(\langle \mathbf{v}_1, \mathbf{x}_j\rangle) \cdot g_j \\ \vdots \\ P(u_K g_j, \langle \mathbf{v}_K, \mathbf{x}_j\rangle) \cdot \mathbf{x}_j \\ \bar{r}(\langle \mathbf{v}_K, \mathbf{x}_j\rangle) \cdot g_j \\ \mathbf{0} \end{bmatrix},$$

where the summation is taken over all possibilities of the tuple $(k, w, m)$, i.e. over the union of $[K] \times \{1\} \times [M_3]$ and $[K] \times \{2\} \times [M_1]$.

By our definition, we have $|P(s,t) - sr'(t)| \leq \varepsilon_p$ for all $s,t \in [-R_3, R_3]$. Therefore, for each $i \in [N]$, $k \in [K]$,

$$|P(u_k g_j, \langle \mathbf{v}_k, \mathbf{x}_j \rangle) - \partial_1 \ell(\mathrm{pred}(\mathbf{x}_j; \mathbf{w}), y_j) \cdot u_k \cdot r'(\langle \mathbf{v}_k, \mathbf{x}_j \rangle)| \leq \varepsilon_p + |g_j - \partial_1 \ell(\mathrm{pred}(\mathbf{x}_i; \mathbf{w}), y_i)| \cdot |u_k| \cdot |r'(\langle \mathbf{v}_k, \mathbf{x}_j \rangle)|$$
$$\leq \varepsilon_p + B_u L_r (\varepsilon_\ell + B_u L_\ell \varepsilon_r),$$

where $L_r := \max_{|t| \leq B_x B_u} |r'(t)|$ is the upper bound of $r'$. Similarly, for each $i \in [N]$, $k \in [K]$, we have

$$|\overline{r}(\langle \mathbf{v}_k, \mathbf{x}_j \rangle) \cdot g_j - r(\langle \mathbf{v}_k, \mathbf{x}_j \rangle) \cdot \partial_1 \ell(\mathrm{pred}(\mathbf{x}_j; \mathbf{w}), y_j)| \leq 2B_g \varepsilon_r + 2B_r(\varepsilon_\ell + B_u L_\ell^2 \varepsilon_r).$$

As for the case $i = N+1$, we have $g_{N+1} = 0$ and $|P(u_k g_{N+1}, \langle \mathbf{v}_k, \mathbf{x}_{N+1} \rangle)| \leq \varepsilon_p$ for each $k \in [K]$ by defintion. Combining these estimations and using (16) and (17), we can conclude that

$$\left\| \eta^{-1} \mathbf{g}(\mathbf{w}) + \nabla \widehat{L}_N(\mathbf{w}) \right\|_2 \leq \sqrt{K} B_x \cdot [\varepsilon_p + B_u L_r(\varepsilon_\ell + B_u L_\ell \varepsilon_r)] + 2\sqrt{K}[B_g \varepsilon_r + B_r(\varepsilon_\ell + B_u L_\ell \varepsilon_r)].$$

Thus, to ensure $\left\| \eta^{-1} \mathbf{g}(\mathbf{w}) + \nabla \widehat{L}_N(\mathbf{w}) \right\|_2 \leq \varepsilon$, we only need to choose $\varepsilon_p, \varepsilon_\ell, \varepsilon_r$ as

$$\varepsilon_p = \frac{\varepsilon}{3\sqrt{K} B_x}, \qquad \varepsilon_\ell = \frac{\varepsilon}{9\sqrt{K} \max\{B_r, L_r B_x B_u\}}, \qquad \varepsilon_r = \frac{\varepsilon}{15\sqrt{K} \max\{B_g, L_\ell B_r B_u, L_r L_\ell B_x B_r B_u^2\}}.$$
(18)

Thus, letting the attention layer $\boldsymbol{\theta}_{\mathtt{attn}}^{(2)} = \{(\mathbf{V}_{k,w,m}^{(2)}, \mathbf{Q}_{k,w,m}^{(2)}, \mathbf{K}_{k,w,m}^{(2)})\}_{(k,w,m)}$, we have

$$\mathrm{Attn}_{\boldsymbol{\theta}_{\mathtt{attn}}^{(2)}} : \mathbf{h}_i^{(1)} \mapsto \mathbf{h}_i^{(1.5)} = [\mathbf{x}_i; y_i'; \mathbf{w} + \eta \mathbf{g}(\mathbf{w}); \overline{\mathrm{pred}}(\mathbf{x}_i; \mathbf{w}); g_i; \mathbf{0}; 1; t_i].$$

Step 4: construction of $\boldsymbol{\theta}_{\mathtt{mlp}}^{(2)}$. We only need to pick $\boldsymbol{\theta}_{\mathtt{mlp}}^{(2)}$ so that it maps

$$\mathbf{h}_i^{(1.5)} = [\mathbf{x}_i; y_i'; \mathbf{w} + \eta \mathbf{g}(\mathbf{w}); \overline{\mathrm{pred}}(\mathbf{x}_i; \mathbf{w}); g_i; \mathbf{0}; 1; t_i] \xrightarrow{\mathrm{MLP}_{\boldsymbol{\theta}_{\mathtt{mlp}}^{(2)}}} \mathbf{h}_i^{(2)} = [\mathbf{x}_i; y_i'; \mathrm{Proj}_{\mathcal{W}}(\mathbf{w} - \eta \mathbf{g}(\mathbf{w})); 0; 0; \mathbf{0}; 1; t_i].$$

By our assumption on the map $\mathrm{Proj}_{\mathcal{W}}$, this is easy.

Combining the four steps above and taking $\boldsymbol{\theta} = (\boldsymbol{\theta}_{\mathtt{attn}}^{(1)}, \boldsymbol{\theta}_{\mathtt{mlp}}^{(1)}, \boldsymbol{\theta}_{\mathtt{attn}}^{(2)}, \boldsymbol{\theta}_{\mathtt{mlp}}^{(2)})$ completes the proof. $\square$

### E.6 Proof of Lemma D.3

For every $\ell \geq 0$, define the intermediate iterates (before projection)

$$\widehat{\mathbf{w}}^{\ell + \frac{1}{2}} := \widehat{\mathbf{w}}^\ell - \eta(\nabla f(\widehat{\mathbf{w}}^\ell) + \boldsymbol{\varepsilon}^\ell), \quad \mathbf{w}_{\mathrm{GD}}^{\ell + \frac{1}{2}} := \mathbf{w}_{\mathrm{GD}}^\ell - \eta \nabla f(\mathbf{w}_{\mathrm{GD}}^\ell),$$

so that $\widehat{\mathbf{w}}^{\ell+1} = \mathrm{Proj}_{\mathcal{W}}(\widehat{\mathbf{w}}^{\ell + \frac{1}{2}})$ and $\mathbf{w}_{\mathrm{GD}}^{\ell+1} = \mathrm{Proj}_{\mathcal{W}}(\mathbf{w}_{\mathrm{GD}}^{\ell + \frac{1}{2}})$.

We first prove part (a). We begin by deriving a relation between $\left\| \widehat{\mathbf{w}}^{\ell+1} - \widehat{\mathbf{w}}^\ell \right\|_2^2$ and $\left\| \eta \mathsf{G}_{\mathcal{W},\eta}^f(\widehat{\mathbf{w}}^\ell) \right\|_2^2$. Let $\widetilde{\mathbf{w}}^{\ell + \frac{1}{2}} := \widehat{\mathbf{w}}^\ell - \eta \nabla f(\widehat{\mathbf{w}}^\ell)$ and $\widetilde{\mathbf{w}}^{\ell+1} := \mathrm{Proj}_{\mathcal{W}}(\widetilde{\mathbf{w}}^{\ell + \frac{1}{2}})$ denote the *exact* projected gradient iterate starting from $\widehat{\mathbf{w}}^\ell$. We have

$$\left\| \widehat{\mathbf{w}}^{\ell+1} - \widehat{\mathbf{w}}^\ell \right\|_2^2 \overset{(i)}{\geq} \frac{1}{2} \left\| \widetilde{\mathbf{w}}^{\ell+1} - \widehat{\mathbf{w}}^\ell \right\|_2^2 - \left\| \widehat{\mathbf{w}}^{\ell+1} - \widetilde{\mathbf{w}}^{\ell+1} \right\|_2^2 \overset{(ii)}{\geq} \frac{1}{2} \left\| \widetilde{\mathbf{w}}^{\ell+1} - \widehat{\mathbf{w}}^\ell \right\|_2^2 - \left\| \widehat{\mathbf{w}}^{\ell+\frac{1}{2}} - \widetilde{\mathbf{w}}^{\ell+\frac{1}{2}} \right\|_2^2$$
$$\overset{(iii)}{=} \frac{\eta^2}{2} \left\| \mathsf{G}_{\mathcal{W},\eta}^f(\widehat{\mathbf{w}}^\ell) \right\|_2^2 - \|\eta \boldsymbol{\varepsilon}\|_2^2 \geq \frac{\eta^2}{2} \left\| \mathsf{G}_{\mathcal{W},\eta}^f(\widehat{\mathbf{w}}^\ell) \right\|_2^2 - \eta^2 \varepsilon^2.$$
(19)

Above, (i) uses the inequality $\|a - b\|_2^2 \geq \frac{1}{2} \|a\|_2^2 - \|b\|_2^2$; (ii) uses the fact that projection to a convex set is a non-expansion; (iii) uses the definition of the gradient mapping.

By the $L_f$-smoothness of $f$ within $\mathcal{W}$, we have

$$f(\widehat{\mathbf{w}}^{\ell+1}) - f(\widehat{\mathbf{w}}^\ell) \leq \langle \nabla f(\widehat{\mathbf{w}}^\ell), \widehat{\mathbf{w}}^{\ell+1} - \widehat{\mathbf{w}}^\ell \rangle + \frac{L_f}{2} \left\| \widehat{\mathbf{w}}^{\ell+1} - \widehat{\mathbf{w}}^\ell \right\|_2^2$$

$$
= \left\langle \frac{\widehat{\mathbf{w}}^\ell - \widehat{\mathbf{w}}^{\ell+\frac{1}{2}}}{\eta} - \boldsymbol{\varepsilon}^\ell, \widehat{\mathbf{w}}^{\ell+1} - \widehat{\mathbf{w}}^\ell \right\rangle + \frac{L_f}{2} \left\| \widehat{\mathbf{w}}^{\ell+1} - \widehat{\mathbf{w}}^\ell \right\|_2^2
$$

$$
\overset{(i)}{\leq} \left\langle \frac{\widehat{\mathbf{w}}^\ell - \widehat{\mathbf{w}}^{\ell+1}}{\eta}, \widehat{\mathbf{w}}^{\ell+1} - \widehat{\mathbf{w}}^\ell \right\rangle + \left\langle \boldsymbol{\varepsilon}^\ell, \widehat{\mathbf{w}}^\ell - \widehat{\mathbf{w}}^{\ell+1} \right\rangle + \frac{L_f}{2} \left\| \widehat{\mathbf{w}}^{\ell+1} - \widehat{\mathbf{w}}^\ell \right\|_2^2
$$

$$
= \left( -\frac{1}{\eta} + \frac{L_f}{2} \right) \left\| \widehat{\mathbf{w}}^{\ell+1} - \widehat{\mathbf{w}}^\ell \right\|_2^2 + \frac{1}{4\eta} \left\| \widehat{\mathbf{w}}^{\ell+1} - \widehat{\mathbf{w}}^\ell \right\|_2^2 + \eta \left\| \boldsymbol{\varepsilon}^\ell \right\|_2^2
$$

$$
\overset{(ii)}{\leq} -\frac{1}{4\eta} \left\| \widehat{\mathbf{w}}^{\ell+1} - \widehat{\mathbf{w}}^\ell \right\|_2^2 + \eta \left\| \boldsymbol{\varepsilon}^\ell \right\|_2^2
$$

$$
\overset{(iii)}{\leq} -\frac{1}{4\eta} \left( \frac{\eta^2}{2} \left\| \mathsf{G}_{\mathcal{W},\eta}^f(\widehat{\mathbf{w}}^\ell) \right\|_2^2 - \left\| \eta \boldsymbol{\varepsilon}^\ell \right\|_2^2 \right) + \eta \left\| \boldsymbol{\varepsilon}^\ell \right\|_2^2
$$

$$
\leq -\frac{\eta}{8} \left\| \mathsf{G}_{\mathcal{W},\eta}^f(\widehat{\mathbf{w}}^\ell) \right\|_2^2 + \frac{5\eta}{4} \varepsilon^2.
$$

Above, (i) uses the property $\left\langle \widehat{\mathbf{w}}^{\ell+1} - \widehat{\mathbf{w}}^{\ell+\frac{1}{2}}, \widehat{\mathbf{w}}^{\ell+1} - \widehat{\mathbf{w}}^\ell \right\rangle \leq 0$ of the projection $\widehat{\mathbf{w}}^{\ell+1} = \mathrm{Proj}_{\mathcal{W}}(\widehat{\mathbf{w}}^{\ell+\frac{1}{2}})$ (using $\widehat{\mathbf{w}}^\ell \in \mathcal{W}$); (ii) uses $L_f/2 \leq 1/(2\eta)$ by our choice of $\eta \leq 1/L_f$; (iii) uses (19).

Rearranging and summing the above over $\ell = 0, \dots, L-1$, we obtain

$$
\frac{\eta}{8} \sum_{\ell=0}^{L-1} \left\| \mathsf{G}_{\mathcal{W},\eta}^f(\widehat{\mathbf{w}}^\ell) \right\|_2^2 \leq f(\mathbf{w}^0) - f(\widehat{\mathbf{w}}^L) + \frac{5\eta L}{4} \varepsilon^2.
$$

Dividing both sides by $\eta L/8$ yields part (a).

Next, we prove part (b). Let $C := 1 + \eta L_f$. We prove by induction that

$$
\left\| \widehat{\mathbf{w}}^\ell - \mathbf{w}_{\mathrm{GD}}^\ell \right\|_2 \leq \frac{C^\ell - 1}{C - 1} \cdot \eta \varepsilon \tag{20}
$$

for all $\ell \geq 0$. The base case of $\ell = 0$ follows by definition that $\widehat{\mathbf{w}}^0 = \mathbf{w}_{\mathrm{GD}}^0 = \mathbf{w}^0$. Suppose the result holds for $\ell$. Then for $\ell + 1$, we have

$$
\left\| \widehat{\mathbf{w}}^{\ell+1} - \mathbf{w}_{\mathrm{GD}}^{\ell+1} \right\|_2 \overset{(i)}{\leq} \left\| \widehat{\mathbf{w}}^{\ell+\frac{1}{2}} - \mathbf{w}_{\mathrm{GD}}^{\ell+\frac{1}{2}} \right\|_2 = \left\| \widehat{\mathbf{w}}^\ell - \eta(\nabla f(\widehat{\mathbf{w}}^\ell) - \boldsymbol{\varepsilon}^\ell) - (\mathbf{w}_{\mathrm{GD}}^\ell - \eta \nabla f(\mathbf{w}_{\mathrm{GD}}^\ell)) \right\|_2
$$

$$
\overset{(ii)}{\leq} C \left\| \widehat{\mathbf{w}}^\ell - \mathbf{w}_{\mathrm{GD}}^\ell \right\|_2 + \eta \varepsilon \overset{(iii)}{\leq} C \cdot \frac{C^\ell - 1}{C - 1} \cdot \eta \varepsilon + \eta \varepsilon = \frac{C^{\ell+1} - 1}{C - 1} \cdot \eta \varepsilon.
$$

Above, (i) uses again the non-expansiveness of the convex projection $\mathrm{Proj}_{\mathcal{W}}$; (ii) uses the fact that the operator $\mathbf{w} \mapsto \mathbf{w} - \eta \nabla f(\mathbf{w})$ is $(1 + \eta L_f) = C$-Lipschitz; and (iii) uses the inductive hypothesis. This proves the case for $\ell + 1$ and thus finishes the induction. We can further relax (20) into

$$
\left\| \widehat{\mathbf{w}}^\ell - \mathbf{w}_{\mathrm{GD}}^\ell \right\|_2 \leq \frac{C^\ell}{1 + \eta L_f - 1} \cdot \eta \varepsilon = L_f^{-1}(1 + \eta L_f)^\ell \varepsilon.
$$

This proves part (b). $\qquad\square$

# F   Proofs for Section 3.1

## F.1   Proof of Theorem 4

Fix $\lambda \geq 0$, $0 \leq \alpha \leq \beta$ with $\kappa := \frac{\beta + \lambda}{\alpha + \lambda}$, and $B_w > 0$, and consider any in-context data $\mathcal{D}$ such that the precondition of Theorem 4 holds. Let

$$
L_{\mathrm{ridge}}(\mathbf{w}) := \frac{1}{2N} \sum_{i=1}^N (\langle \mathbf{w}, \mathbf{x}_i \rangle - y_i)^2 + \frac{\lambda}{2} \|\mathbf{w}\|_2^2
$$

denote the ridge regression loss in (ICRidge), so that $\mathbf{w}_{\mathrm{ridge}}^{\lambda} = \arg\min_{\mathbf{w} \in \mathbb{R}^d} L_{\mathrm{ridge}}(\mathbf{w})$. It is a standard result that $\nabla^2 L_{\mathrm{ridge}}(\mathbf{w}) = \mathbf{X}^{\top}\mathbf{X}/N + \lambda \mathbf{I}_d$, so that $L_{\mathrm{ridge}}$ is $(\alpha + \lambda)$-strongly convex and $(\beta + \lambda)$-smooth over $\mathbb{R}^d$.

Consider the gradient descent algorithm on the ridge loss

$$\mathbf{w}_{\mathrm{GD}}^{t+1} = \mathbf{w}_{\mathrm{GD}}^{t} - \eta \nabla L_{\mathrm{ridge}}(\mathbf{w}_{\mathrm{GD}}^{t})$$

with initialization, learning rate, and number of steps

$$\mathbf{w}_{\mathrm{GD}}^{0} := \mathbf{0}_d, \quad \eta := \frac{1}{\beta + \lambda}, \quad T := \left\lceil 2\kappa \log\left(\frac{B_x B_w}{2\varepsilon}\right) \right\rceil.$$

By standard convergence results for strongly convex and smooth functions (Proposition B.2), we have for all $t \geq 1$ that

$$\left\|\mathbf{w}_{\mathrm{GD}}^{t} - \mathbf{w}_{\mathrm{ridge}}^{\lambda}\right\|_2^2 \leq \exp\left(-\frac{t}{\kappa}\right) \left\|\mathbf{w}_{\mathrm{GD}}^{0} - \mathbf{w}_{\mathrm{ridge}}^{\lambda}\right\|_2^2 = \exp\left(-\frac{t}{\kappa}\right) \left\|\mathbf{w}_{\mathrm{ridge}}^{\lambda}\right\|_2^2.$$

Further, we have

$$\left\|\mathbf{w}_{\mathrm{GD}}^{T} - \mathbf{w}_{\mathrm{ridge}}^{\lambda}\right\|_2 \leq \exp\left(-\frac{T}{2\kappa}\right) \left\|\mathbf{w}_{\mathrm{ridge}}^{\lambda}\right\|_2 \leq \frac{2\varepsilon}{B_x B_w} \cdot \frac{B_w}{2} \leq \frac{\varepsilon}{B_x}. \tag{21}$$

It remains to construct a transformer to approximate $\mathbf{w}_{\mathrm{GD}}^{T}$. Notice that the problem (ICRidge) corresponds to an $\ell_2$-regularized ERM with the square loss $\ell(s,t) := \frac{1}{2}(s-t)^2$, whose partial derivative $\partial_s \ell(s,t) = s - t$ is exactly a sum of two relus:

$$\partial_s \ell(s,t) = 2\sigma((s-t)/2) - 2\sigma(-(s-t)/2).$$

In particular, this shows that $\partial_s \ell(s,t)$ is $(0, R, 2, 4)$-approximable for any $R > 0$, in particular for $R = \max\{B_x B_w, B_y, 1\}$.

Therefore, we can apply Corollary E.1 with the square loss $\ell$, learning rate $\eta$, regularization strength $\lambda$ and accuracy parameter $\varepsilon = 0$ to obtain that there exists an attention-only transformer $\mathrm{TF}_{\boldsymbol{\theta}}^{0}$ with $(T+1) := L$ layers such that the final output $\mathbf{h}_{N+1}^{(L)} = [\mathbf{x}_{N+1}; \widehat{y}_{N+1}; *]$ with

$$\left|\widehat{y}_{N+1} - \left\langle\mathbf{w}_{\mathrm{GD}}^{T}, \mathbf{x}_{N+1}\right\rangle\right| = 0, \tag{22}$$

and number of heads $M^{(\ell)} = 3$ for all $\ell \in [L-1]$ (can be taken as 2 in the unregularized case $\lambda = 0$ directly by Theorem D.1), and $M^{(L)} = 2$. Further, $\boldsymbol{\theta}$ admits norm bound $\|\boldsymbol{\theta}\| \leq 2 + R + \frac{8+\lambda}{\beta+\lambda} \leq 3R + 8(\beta + \lambda)^{-1} + 1 \leq 4R + 8(\beta + \lambda)^{-1}$.

Combining (21) and (22), we obtain that

$$\left|\widehat{y}_{N+1} - \left\langle\mathbf{w}_{\mathrm{ridge}}^{\lambda}, \mathbf{x}_{N+1}\right\rangle\right| = \left|\left\langle\mathbf{w}_{\mathrm{GD}}^{T} - \mathbf{w}_{\mathrm{ridge}}^{\lambda}, \mathbf{x}_{N+1}\right\rangle\right| \leq (\varepsilon/B_x) \cdot B_x = \varepsilon.$$

Further, we have $\mathsf{read}_{\mathsf{w}}(\mathbf{h}_i^T) = \mathbf{w}_{\mathrm{GD}}^{T}$ for all $i \in [N+1]$, where $\mathsf{read}_{\mathsf{w}}(\mathbf{h}) := \mathbf{h}_{(d+2):(2d+1)}$ (cf. Corollary E.1), so that $\|\mathsf{read}_{\mathsf{w}}(\mathbf{h}_i^T) - \mathbf{w}_{\mathrm{ridge}}^{\lambda}\|_2 \leq \varepsilon/B_x$ as shown above. This finishes the proof. $\qquad\square$

### F.2 Statistical analysis of in-context least squares

Consider the standard least-squares algorithm $\mathcal{A}_{\mathrm{LS}}$ and least-squares estimator $\widehat{\mathbf{w}}_{\mathrm{LS}} \in \mathbb{R}^d$ defined as

$$\mathcal{A}_{\mathrm{LS}}(\mathcal{D})(\mathbf{x}_{N+1}) := \left\langle\widehat{\mathbf{w}}_{\mathrm{LS}}, \mathbf{x}_{N+1}\right\rangle, \quad \widehat{\mathbf{w}}_{\mathrm{LS}} = \left(\mathbf{X}^{\top}\mathbf{X}\right)^{-1}\mathbf{X}^{\top}\mathbf{y} \in \mathbb{R}^d. \tag{ICLS}$$

For any distribution $\mathsf{P}$ over $(\mathbf{x}, y) \in \mathbb{R}^d \times \mathbb{R}$ and any estimator $\mathbf{w} \in \mathbb{R}^d$, let

$$L_{\mathsf{P}}(\mathbf{w}) := \mathbb{E}_{(\mathbf{x}',y) \sim \mathsf{P}}\left[\frac{1}{2}(\langle\mathbf{w}, \mathbf{x}'\rangle - y')^2\right]$$

denote the expected risk of $\mathbf{w}$ over a new test example $(\mathbf{x}', y') \sim \mathsf{P}$.

**Assumption A** (Well-posedness for learning linear predictors). *We say a distribution $\mathsf{P}$ on $\mathbb{R}^d \times \mathbb{R}$ is well-posed for learning linear predictors, if $(\mathbf{x}, y) \sim \mathsf{P}$ satisfies*

*(1)* $\|\mathbf{x}\|_2 \leq B_x$ *and* $|y| \leq B_y$ *almost surely;*

*(2) The covariance* $\boldsymbol{\Sigma}_{\mathsf{P}} := \mathbb{E}_{\mathsf{P}}[\mathbf{x}\mathbf{x}^\top]$ *satisfies* $\lambda_{\min}\mathbf{I}_d \preceq \boldsymbol{\Sigma}_{\mathsf{P}} \preceq \lambda_{\max}\mathbf{I}_d$, *with* $0 < \lambda_{\min} \leq \lambda_{\max}$, *and* $\kappa := \lambda_{\max}/\lambda_{\min}$.

*(3) The whitened vector* $\boldsymbol{\Sigma}_{\mathsf{P}}^{-1/2}\mathbf{x}$ *is* $K^2$*-sub-Gaussian for some* $K \geq 1$.

*(4) The best linear predictor* $\mathbf{w}_{\mathsf{P}}^\star := \mathbb{E}_{\mathsf{P}}[\mathbf{x}\mathbf{x}^\top]^{-1}\mathbb{E}_{\mathsf{P}}[\mathbf{x}y]$ *satisfies* $\|\mathbf{w}_{\mathsf{P}}^\star\|_2 \leq B_w^\star$.

*(5) We have* $\mathbb{E}[(y - \langle\mathbf{x}, \mathbf{w}_{\mathsf{P}}^\star\rangle)^2|\mathbf{x}] \leq \sigma^2$ *with probability one (over* $\mathbf{x}$*).*

*Further, we say* $\mathsf{P}$ *is well-posed with* canonical parameters *if*

$$B_x = \Theta(\sqrt{d}), \quad B_y = \Theta(1), \quad B_w^\star = \Theta(1), \quad \sigma \leq \mathcal{O}(1), \quad \lambda_{\max} = \Theta(1), \quad K = \Theta(1), \quad (23)$$

*where* $\Theta(\cdot)$ *and* $\mathcal{O}(\cdot)$ *only hides absolute constants.*

The following result bounds the excess risk of least squares under Assumption A with a clipping operation on the predictor; the clipping allows the result to only depend on the second moment of the noise (cf. Assumption A(5)) instead of e.g. its sub-Gaussianity, and also makes the result convenient to be directly translated to a result for transformers.

**Proposition F.1** (Guarantees for in-context least squares)**.** *Suppose distribution* $\mathsf{P}$ *satisfies Assumption A. Then as long as* $N \geq \mathcal{O}(dK^4\log(1/\delta))$, *we have the following:*

*(a) The (clipped) least squares predictor achieves small expected excess risk (fast rate) over the best linear predictor: For any clipping radius* $R \geq B_y$,

$$\mathbb{E}_{\mathcal{D},\mathbf{x}_{N+1},y_{N+1}\sim\mathsf{P}}\left[\frac{1}{2}(\mathsf{clip}_R(\langle\widehat{\mathbf{w}}_{\mathrm{LS}}, \mathbf{x}_{N+1}\rangle) - y_{N+1})^2\right] \leq \underbrace{\inf_{\mathbf{w}\in\mathbb{R}^d}L_{\mathsf{P}}(\mathbf{w})}_{L_{\mathsf{P}}(\mathbf{w}_{\mathsf{P}}^\star)} + \mathcal{O}\left(R^2\delta + \frac{d\sigma^2}{N}\right).$$

$$(24)$$

*(b) We have* $\mathsf{P}(E_{\mathrm{cov}} \cap E_w) \geq 1 - \delta/10$, *where*

$$E_{\mathrm{cov}} = E_{\mathrm{cov}}(\mathcal{D}) := \left\{\frac{1}{2}\mathbf{I}_d \preceq \boldsymbol{\Sigma}_{\mathsf{P}}^{-1/2}\widehat{\boldsymbol{\Sigma}}\boldsymbol{\Sigma}_{\mathsf{P}}^{-1/2} \preceq 2\mathbf{I}_d\right\}, \quad (25)$$

$$E_w = E_w(\mathcal{D}) := \left\{\|\widehat{\mathbf{w}}_{\mathrm{LS}}\|_2 \leq B_w^\star + \sqrt{\frac{80d\sigma^2}{\delta N\lambda_{\min}}}\right\}. \quad (26)$$

*Proof.* We first show $\mathsf{P}(E_{\mathrm{cov}}) \geq 1 - \delta/20$. Let $\widehat{\boldsymbol{\Sigma}} := \frac{1}{N}\sum_{i=1}^N \mathbf{x}_i\mathbf{x}_i^\top$, and let the whitened covariance and noise variables be denoted as

$$\widetilde{\mathbf{x}}_i = \boldsymbol{\Sigma}_{\mathsf{P}}^{-1/2}\mathbf{x}_i, \quad \widetilde{\boldsymbol{\Sigma}} := \frac{1}{N}\sum_{i=1}^N \widetilde{\mathbf{x}}_i\widetilde{\mathbf{x}}_i^\top = \boldsymbol{\Sigma}_{\mathsf{P}}^{-1/2}\widehat{\boldsymbol{\Sigma}}\boldsymbol{\Sigma}_{\mathsf{P}}^{-1/2}.$$

Also let $z_i := y_i - \langle\mathbf{x}_i, \mathbf{w}_{\mathsf{P}}^\star\rangle$ denote the "noise" variables. Note that

$$E_{\mathrm{cov}} = \left\{\frac{1}{2}\mathbf{I}_d \preceq \widetilde{\boldsymbol{\Sigma}} \preceq 2\mathbf{I}_d\right\}$$

is exactly a covariance concentration of the whitened vectors $\{\widetilde{\mathbf{x}}_i\}_{i\in[N]}$. Recall that $\mathbb{E}[\widetilde{\mathbf{x}}_i\widetilde{\mathbf{x}}_i^\top] = \mathbf{I}_d$, and $\widetilde{\mathbf{x}}_i$ are $K^2$-sub-Gaussian by assumption. Therefore, we can apply [85, Theorem 4.6.1], we have with probability at least $1 - \delta/10$ that

$$\left\|\widetilde{\boldsymbol{\Sigma}} - \mathbf{I}_d\right\|_{\mathrm{op}} \leq \mathcal{O}\left(K^2\max\left\{\sqrt{\frac{d + \log(1/\delta)}{N}}, \frac{d + \log(1/\delta)}{N}\right\}\right).$$

Setting $N \geq \mathcal{O}(K^4(d + \log(1/\delta)))$ ensures that the right-hand side above is at most $1/2$, on which event we have

$$\frac{1}{2}\mathbf{I}_d \preceq \widetilde{\boldsymbol{\Sigma}} \preceq \frac{3}{2}\mathbf{I}_d \preceq 2\mathbf{I}_d, \quad (27)$$

i.e. $E_{\text{cov}}$ holds. This shows that $\mathsf{P}(E_{\text{cov}}^c) \leq \delta/10$.

Next, we show (24). Using $E_{\text{cov}}$, we decompose the risk as

$$\mathbb{E}\left[\frac{1}{2}(\mathsf{clip}_R(\langle \widehat{\mathbf{w}}_{\text{LS}}, \mathbf{x}_{N+1}\rangle) - y_{N+1})^2\right]$$

$$= \mathbb{E}\left[\frac{1}{2}(\mathsf{clip}_R(\langle \widehat{\mathbf{w}}_{\text{LS}}, \mathbf{x}_{N+1}\rangle) - y_{N+1})^2 1\{E_{\text{cov}}\}\right] + \mathbb{E}\left[\frac{1}{2}(\mathsf{clip}_R(\langle \widehat{\mathbf{w}}_{\text{LS}}, \mathbf{x}_{N+1}\rangle) - y_{N+1})^2 1\{E_{\text{cov}}^c\}\right]$$

$$\overset{(i)}{\leq} \mathbb{E}\left[\frac{1}{2}(\langle \widehat{\mathbf{w}}_{\text{LS}}, \mathbf{x}_{N+1}\rangle - y_{N+1})^2 1\{E_{\text{cov}}\}\right] + 2R^2 \cdot (\delta/20)$$

$$\overset{(ii)}{=} \mathbb{E}_{\mathcal{D}, \mathbf{x}_{N+1}}\left[\frac{1}{2}(\langle \widehat{\mathbf{w}}_{\text{LS}} - \mathbf{w}_{\mathsf{P}}^\star, \mathbf{x}_{N+1}\rangle)^2 1\{E_{\text{cov}}\}\right] + \mathbb{E}_{\mathbf{x}_{N+1}, y_{N+1}}\left[\frac{1}{2}(\langle \mathbf{w}_{\mathsf{P}}^\star, \mathbf{x}_{N+1}\rangle - y_{N+1})^2 1\{E_{\text{cov}}\}\right] + \mathcal{O}(R^2\delta)$$

$$\leq \mathbb{E}_{\mathcal{D}}\left[\frac{1}{2}\left\|\widehat{\mathbf{w}}_{\text{LS}} - \mathbf{w}_{\mathsf{P}}^\star\right\|_{\mathbf{\Sigma}_\mathsf{P}}^2 1\{E_{\text{cov}}\}\right] + \underbrace{\mathbb{E}_{\mathbf{x}_{N+1}, y_{N+1}}\left[\frac{1}{2}(\langle \mathbf{w}_{\mathsf{P}}^\star, \mathbf{x}_{N+1}\rangle - y_{N+1})^2\right]}_{L_\mathsf{P}(\mathbf{w}_{\mathsf{P}}^\star)} + \mathcal{O}(R^2\delta).$$

$$(28)$$

Above, (i) follows by assumption that $|y_{N+1}| \leq B_y \leq R$ almost surely, so that removing the clipping can only potentially increase the distance in the first term, and the square loss is upper bounded by $\frac{1}{2} \cdot (2R)^2$ almost surely in the second term; (ii) follows by the fact that $\mathbb{E}_{\mathbf{x}_{N+1}, y_{N+1}}[\langle \widehat{\mathbf{w}}_{\text{LS}} - \mathbf{w}_{\mathsf{P}}^\star, \mathbf{x}_{N+1}\rangle (\langle \mathbf{w}_{\mathsf{P}}^\star, \mathbf{x}_{N+1}\rangle - y_{N+1})] = 0$ by the definition of $\mathbf{w}_{\mathsf{P}}^\star$, as well as the fact that $1\{E_{\text{cov}}\}$ is independent of $(\mathbf{x}_{N+1}, y_{N+1})$.

It thus remains to bound $\mathbb{E}_{\mathcal{D}}\left[\frac{1}{2}\left\|\widehat{\mathbf{w}}_{\text{LS}} - \mathbf{w}_{\mathsf{P}}^\star\right\|_{\mathbf{\Sigma}_\mathsf{P}}^2 1\{E_{\text{cov}}\}\right]$. Note that on the event $E_{\text{cov}}$, we have

$$\mathbf{\Sigma}_\mathsf{P}^{1/2}\widehat{\mathbf{\Sigma}}^{-1}\mathbf{\Sigma}_\mathsf{P}^{1/2} = \left(\mathbf{\Sigma}_\mathsf{P}^{-1/2}\widehat{\mathbf{\Sigma}}\mathbf{\Sigma}_\mathsf{P}^{-1/2}\right)^{-1} \preceq 2\mathbf{I}_d.$$

Therefore,

$$\frac{1}{2}\left\|\widehat{\mathbf{w}}_{\text{LS}} - \mathbf{w}_{\mathsf{P}}^\star\right\|_{\mathbf{\Sigma}_\mathsf{P}}^2 1\{E_{\text{cov}}\} = \frac{1}{2}\left((\mathbf{X}^\top\mathbf{X})^{-1}\mathbf{X}^\top\mathbf{y} - \mathbf{w}_{\mathsf{P}}^\star\right)^\top \mathbf{\Sigma}_\mathsf{P}\left((\mathbf{X}^\top\mathbf{X})^{-1}\mathbf{X}^\top\mathbf{y} - \mathbf{w}_{\mathsf{P}}^\star\right)1\{E_{\text{cov}}\}$$

$$= \frac{1}{2}\mathbf{z}^\top\mathbf{X}(\mathbf{X}^\top\mathbf{X})^{-1}\mathbf{\Sigma}_\mathsf{P}(\mathbf{X}^\top\mathbf{X})^{-1}\mathbf{X}^\top\mathbf{z} \cdot 1\{E_{\text{cov}}\}$$

$$= \frac{1}{2N^2}\mathbf{z}^\top\mathbf{X}\mathbf{\Sigma}_\mathsf{P}^{-1/2}\left(\mathbf{\Sigma}_\mathsf{P}^{1/2}\widehat{\mathbf{\Sigma}}^{-1}\mathbf{\Sigma}_\mathsf{P}^{1/2}\right)^2 \mathbf{\Sigma}_\mathsf{P}^{-1/2}\mathbf{X}^\top\mathbf{z} \cdot 1\{E_{\text{cov}}\}$$

$$\leq \frac{2}{N^2}\left\|\mathbf{\Sigma}_\mathsf{P}^{-1/2}\mathbf{X}^\top\mathbf{z}\right\|_2^2 1\{E_{\text{cov}}\} = \frac{2}{N^2}\left\|\sum_{i=1}^N \widetilde{\mathbf{x}}_i z_i\right\|_2^2 1\{E_{\text{cov}}\} \leq \frac{2}{N^2}\left\|\sum_{i=1}^N \widetilde{\mathbf{x}}_i z_i\right\|_2^2.$$

Note that $\mathbb{E}[\widetilde{\mathbf{x}}_i z_i] = \mathbf{\Sigma}_\mathsf{P}^{-1/2}\mathbb{E}[\mathbf{x}_i(y_i - \langle \mathbf{w}_{\mathsf{P}}^\star, \mathbf{x}_i\rangle)] = 0$. Therefore, taking expectation on the above (over $\mathcal{D}$), we get

$$\mathbb{E}_{\mathcal{D}}\left[\frac{1}{2}\left\|\widehat{\mathbf{w}}_{\text{LS}} - \mathbf{w}_{\mathsf{P}}^\star\right\|_{\mathbf{\Sigma}_\mathsf{P}}^2 1\{E_{\text{cov}}\}\right] \leq \frac{2}{N^2}\mathbb{E}\left[\left\|\sum_{i=1}^N \widetilde{\mathbf{x}}_i z_i\right\|_2^2\right] = \frac{2}{N}\mathbb{E}\left[\|\widetilde{\mathbf{x}}_1 z_1\|_2^2\right] = \frac{2}{N}\mathbb{E}\left[z_1^2 \mathbf{x}_1^\top \mathbf{\Sigma}_\mathsf{P}^{-1}\mathbf{x}_1\right]$$

$$(29)$$

$$\overset{(i)}{\leq} \frac{2\sigma^2}{N}\mathbb{E}\left[\mathbf{x}_1^\top \mathbf{\Sigma}_\mathsf{P}^{-1}\mathbf{x}_1\right] = \frac{2d\sigma^2}{N}. \tag{30}$$

Above, (i) follows by conditioning on $\mathbf{x}_1$ and using Assumption A(5). Combining with (28), we obtain

$$\mathbb{E}\left[\frac{1}{2}(\mathsf{clip}_R(\langle \widehat{\mathbf{w}}_{\text{LS}}, \mathbf{x}_{N+1}\rangle) - y_{N+1})^2\right] \leq L_\mathsf{P}(\mathbf{w}_{\mathsf{P}}^\star) + \mathcal{O}\left(R^2\delta + \frac{d\sigma^2}{N}\right).$$

This proves (24).

Finally, we show $\mathbb{P}(E_{\text{cov}} \cap E_w) \geq 1 - \delta/10$. Using (29) and $\mathbf{\Sigma}_\mathsf{P} \succeq \lambda_{\min}\mathbf{I}_d$ by assumption, we get

$$\mathbb{E}\left[\left\|\widehat{\mathbf{w}}_{\text{LS}} - \mathbf{w}_{\mathsf{P}}^\star\right\|_2^2 1\{E_{\text{cov}}\}\right] \leq \frac{4d\sigma^2}{N\lambda_{\min}}.$$

Therefore, using an argument similar to Chebyshev's inequality,

$$\mathsf{P}(E_{\mathrm{cov}} \cap E_w^c) = \mathbb{E}\left[1\{E_{\mathrm{cov}}\} \times 1\{\|\widehat{\mathbf{w}}_{\mathrm{LS}}\|_2 > \sqrt{\frac{20}{\delta} \cdot \frac{4d\sigma^2}{N\lambda_{\min}}} + B_w^\star\}\right]$$

$$\leq \mathbb{E}\left[1\{E_{\mathrm{cov}}\} \times 1\{\|\widehat{\mathbf{w}}_{\mathrm{LS}} - \mathbf{w}_{\mathsf{P}}^\star\|_2 > \sqrt{\frac{20}{\delta} \cdot \frac{4d\sigma^2}{N\lambda_{\min}}}\}\right]$$

$$\leq \mathbb{E}\left[1\{E_{\mathrm{cov}}\} \times \frac{\|\widehat{\mathbf{w}}_{\mathrm{LS}} - \mathbf{w}_{\mathsf{P}}^\star\|_2^2}{\frac{20}{\delta} \cdot \frac{4d\sigma^2}{N\lambda_{\min}}}\right] \leq \delta/20.$$

This implies that

$$\mathsf{P}(E_{\mathrm{cov}} \cap E_w) = \mathsf{P}(E_{\mathrm{cov}}) - \mathsf{P}(E_{\mathrm{cov}} \cap E_w^c) \geq 1 - \delta/20 - \delta/20 \geq 1 - \delta/10.$$

This is the desired result. $\qquad\qquad\square$

### F.3   Proof of Corollary 5

The proof follows by first checking the well-conditionedness of the data $\mathcal{D}$ (cf. (5)) with high probability, then invoking Theorem 4 (for approximation least squares) and Proposition F.1 (for the statistical power of least squares).

First, as P satisfies Assumption A, by Proposition F.1, as long as $N \geq \mathcal{O}(K^4(d + \log(1/\delta)))$, we have with probability at least $1 - \delta/10$ that event $E_{\mathrm{cov}} \cap E_w$ holds. On this event, we have

$$\frac{1}{2}\lambda_{\min}\mathbf{I}_d \preceq \frac{1}{2}\boldsymbol{\Sigma}_{\mathsf{P}} \preceq \widehat{\boldsymbol{\Sigma}} = \mathbf{X}^\top\mathbf{X}/N \preceq 2\boldsymbol{\Sigma}_{\mathsf{P}} \preceq 2\lambda_{\max}\mathbf{I}_d,$$

$$\|\widehat{\mathbf{w}}_{\mathrm{LS}}\|_2 \leq B_w/2 := \mathcal{O}\left(B_w^\star + \sqrt{\frac{d\sigma^2}{\delta N\lambda_{\min}}}\right),$$

and thus the dataset $\mathcal{D}$ is well-conditioned (in the sense of (5)) with parameters $\alpha = \lambda_{\min}/2$, $\beta = 2\lambda_{\max}$, and $B_w$ defined as above. Note that the condition number of $\widehat{\boldsymbol{\Sigma}}$ is upper bounded by $\beta/\alpha = 4\lambda_{\max}/\lambda_{\min} \leq 4\kappa$, where $\kappa$ is the upper bound on the condition number of $\boldsymbol{\Sigma}_{\mathsf{P}}$ as in Assumption A(c).

Define parameters

$$\varepsilon = \sqrt{\frac{d\sigma^2}{N}}, \quad \delta = \frac{d\sigma^2}{B_y^2 N} \wedge 1. \tag{31}$$

Note that $B_w \leq \mathcal{O}(B_w^\star + \sqrt{B_y^2/\lambda_{\min}})$ by the above choice of $\delta$.

We can thus apply Theorem 4 in the unregularized case ($\lambda = 0$) to obtain that, there exists a transformer $\boldsymbol{\theta}$ with $\max_{\ell \in [L]} M^{(\ell)} \leq 3$, $\|\boldsymbol{\theta}\| \leq 4R + 4/\lambda_{\max}$ (with $R = \max\{B_x B_w, B_y, 1\}$), and number of layers

$$L \leq \mathcal{O}\left(\kappa \log\frac{B_x B_w}{\varepsilon}\right) \leq \mathcal{O}\left(\kappa \log\left(B_x\sqrt{\frac{N}{d\sigma^2}}\left(B_w^\star + \frac{B_y^2}{\sqrt{\lambda_{\min}}}\right)\right)\right),$$

such that on $E_{\mathrm{cov}} \cap E_w$ (so that $\mathcal{D}$ is well-conditioned), we have (choosing the clipping radius in $\widetilde{\mathrm{read}}_{\mathsf{y}}(\cdot) = \mathrm{clip}_{B_y}(\mathrm{read}_{\mathsf{y}}(\cdot))$ to be $B_y$):

$$\left|\widetilde{\mathrm{read}}_{\mathsf{y}}(\mathrm{TF}_{\boldsymbol{\theta}}^0(\mathbf{H})) - \mathrm{clip}_{B_y}(\langle\widehat{\mathbf{w}}_{\mathrm{LS}}, \mathbf{x}_{N+1}\rangle)\right| \leq \left|\mathrm{read}_{\mathsf{y}}(\mathrm{TF}_{\boldsymbol{\theta}}^0(\mathbf{H})) - \langle\widehat{\mathbf{w}}_{\mathrm{LS}}, \mathbf{x}_{N+1}\rangle\right| \leq \varepsilon = \sqrt{\frac{d\sigma^2}{N}}. \tag{32}$$

We now bound the excess risk of the above transformer. Combining Proposition F.1 and (32), we have

$$\mathbb{E}\left[\left(\widetilde{\mathrm{read}}_{\mathsf{y}}(\mathrm{TF}_{\boldsymbol{\theta}}^0(\mathbf{H})) - y_{N+1}\right)^2\right]$$

$$= \mathbb{E}\left[\left(\widetilde{\mathsf{read}}_y(\mathrm{TF}^0_{\boldsymbol{\theta}}(\mathbf{H})) - y_{N+1}\right)^2 1\{E_{\mathrm{cov}} \cap E_w\}\right] + \mathbb{E}\left[\left(\widetilde{\mathsf{read}}_y(\mathrm{TF}^0_{\boldsymbol{\theta}}(\mathbf{H})) - y_{N+1}\right)^2 1\{(E_{\mathrm{cov}} \cap E_w)^c\}\right]$$

$$\leq 2\mathbb{E}\left[\left(\widetilde{\mathsf{read}}_y(\mathrm{TF}^0_{\boldsymbol{\theta}}(\mathbf{H})) - \mathsf{clip}_{B_y}(\langle \widehat{\mathbf{w}}_{\mathrm{LS}}, \mathbf{x}_{N+1}\rangle)\right)^2 1\{E_{\mathrm{cov}} \cap E_w\}\right]$$

$$+ 2\mathbb{E}\left[\left(\mathsf{clip}_{B_y}(\langle \widehat{\mathbf{w}}_{\mathrm{LS}}, \mathbf{x}_{N+1}\rangle) - y_{N+1}\right)^2 1\{E_{\mathrm{cov}} \cap E_w\}\right] + 2B_y^2 \cdot \delta/10$$

$$\overset{(i)}{\leq} 2\varepsilon^2 + L_{\mathsf{P}}(\mathbf{w}_{\mathsf{P}}^\star) + \mathcal{O}\left(B_y^2 \delta + \frac{d\sigma^2}{N}\right) + \mathcal{O}(B_y^2 \delta)$$

$$\leq L_{\mathsf{P}}(\mathbf{w}_{\mathsf{P}}^\star) + \mathcal{O}\left(B_y^2 \delta + \frac{d\sigma^2}{N}\right) \leq \mathcal{O}\left(\frac{d\sigma^2}{N}\right).$$

Above, (i) uses the approximation guarantee (32) as well as Proposition F.1(a) (with clipping radius $B_y$). This proves the desired excess risk guarantee.

Finally, under the canonical choice of parameters (23), the bounds for $L, M, \|\boldsymbol{\theta}\|$ simplify to

$$L \leq \mathcal{O}\left(\kappa \log \frac{N\kappa}{\sigma}\right), \quad \max_{\ell \in [L]} M^{(\ell)} \leq 3, \quad \|\boldsymbol{\theta}\| \leq \mathcal{O}(\sqrt{\kappa d}), \tag{33}$$

and the requirement for $N$ simplifies to $N \geq \mathcal{O}(d + \log(1/\delta)) = \widetilde{\mathcal{O}}(d)$ (as $K = \Theta(1)$). This proves the claim about the required $N$ and $L$. $\qquad\square$

### F.4  Proof of Corollary 6

Fix parameters $\delta, \underline{\varepsilon} > 0$ to be specified later and a large universal constant $C_0$. Let us set

$$\alpha = \max\left\{0, 1/2 - \sqrt{d/N}\right\}^2, \qquad \beta = 25,$$

$$B_w^\star := 1 + 2\sqrt{\frac{\log(4/\delta)}{d}}, \qquad B_w = C_0(B_w^\star + \sigma),$$

$$B_x = C_0\sqrt{d\log(N/\delta)}, \qquad B_y = C_0(B_w^\star + \sigma)\sqrt{\log(N/\delta)}.$$

Consider the following good events (below $\boldsymbol{\varepsilon} = [\varepsilon_i]_{i \in [N]} \in \mathbb{R}^N$ is given by $\varepsilon_i = y_i - \langle \mathbf{w}_\star, \mathbf{x}_i\rangle$)

$$\mathcal{E}_\pi = \left\{\|\mathbf{w}_\star\|_2 \leq B_w^\star, \|\boldsymbol{\varepsilon}\|_2 \leq 2\sqrt{N}\sigma\right\},$$

$$\mathcal{E}_w = \left\{\alpha \leq \lambda_{\min}(\mathbf{X}^\top \mathbf{X}/N) \leq \lambda_{\max}(\mathbf{X}^\top \mathbf{X}/N) \leq \beta\right\},$$

$$\mathcal{E}_b = \{\forall i \in [N], \ \|\mathbf{x}_i\|_2 \leq B_x, \ |y_i| \leq B_y\},$$

$$\mathcal{E}_{b,N+1} = \{\|\mathbf{x}_{N+1}\|_2 \leq B_x, \ |y_{N+1}| \leq B_y\},$$

and we define $\mathcal{E} := \mathcal{E}_\pi \cap \mathcal{E}_w \cap \mathcal{E}_b \cap \mathcal{E}_{b,N+1}$. Under the event $\mathcal{E}$, the problem (ICRidge) is well-conditioned and $\|\mathbf{w}_{\mathrm{ridge}}^\lambda\| \leq B_w/2$ (by Lemma F.1).

Therefore, Theorem 4 implies that for $\kappa = \frac{\alpha + \lambda}{\beta + \lambda}$, there exists a $L = \lceil 2\kappa \log(B_w/\underline{\varepsilon}) \rceil + 1$-layer transformer $\boldsymbol{\theta}$ with prediction $\widehat{y}_{N+1} := \widetilde{\mathsf{read}}_y(\mathrm{TF}^0_{\boldsymbol{\theta}}(\mathbf{H}))$ (clipped by $B_y$), such that under the good event $\mathcal{E}$, we have $\widehat{y}_{N+1} = \mathsf{clip}_{B_y}(\langle \mathbf{x}_{N+1}, \widehat{\mathbf{w}}\rangle)$ and $\|\widehat{\mathbf{w}} - \mathbf{w}_{\mathrm{ridge}}^\lambda\| \leq \underline{\varepsilon}$.

In the following, we show that $\boldsymbol{\theta}$ is indeed the desired transformer (when $\underline{\varepsilon}$ and $\delta$ is suitably chosen). Notice that we have

$$\mathbb{E}(\widehat{y}_{N+1} - y_{N+1})^2 = \mathbb{E}\left[1\{\mathcal{E}\}(\widehat{y}_{N+1} - y_{N+1})^2\right] + \mathbb{E}\left[1\{\mathcal{E}^c\}(\widehat{y}_{N+1} - y_{N+1})^2\right],$$

and we analyze these two parts separately.

**Prediction risk under good event $\mathcal{E}$.**  We first note that

$$\mathbb{E}\left[1\{\mathcal{E}\}(\widehat{y}_{N+1} - y_{N+1})^2\right] = \mathbb{E}\left[1\{\mathcal{E}\}(\mathsf{clip}_{B_y}(\langle \mathbf{x}_{N+1}, \widehat{\mathbf{w}}\rangle) - y_{N+1})^2\right]$$

$$\leq \mathbb{E}\left[1\{\mathcal{E}\}(\langle \mathbf{x}_{N+1}, \widehat{\mathbf{w}}\rangle - y_{N+1})^2\right],$$

where the inequality is because $y_{N+1} \in [-B_y, B_y]$ under the good event $\mathcal{E}$. Notice that by our construction, under the good event $\mathcal{E}$, $\widehat{\mathbf{w}} = \widehat{\mathbf{w}}(\mathcal{D})$ depends only on the dataset $\mathcal{D}$[7]. Therefore, we have $\|\widehat{\mathbf{w}}(\mathcal{D}) - \mathbf{w}_{\mathrm{ridge}}^{\lambda}(\mathcal{D})\| \leq \underline{\varepsilon}$ as long as the event $\mathcal{E}_0 := \mathcal{E}_\pi \cap \mathcal{E}_w \cap \mathcal{E}_b$ holds for $(\mathbf{w}_\star, \mathcal{D})$. Thus, under $\mathcal{E}_0$,

$$
\begin{aligned}
\mathbb{E}\left[\mathbf{1}\{\mathcal{E}\}(\langle \mathbf{x}_{N+1}, \widehat{\mathbf{w}}\rangle - y_{N+1})^2 \,\middle|\, \mathbf{w}_\star, \mathcal{D}\right] &= \mathbb{E}\left[\mathbf{1}\{\mathcal{E}\}(\langle \mathbf{x}_{N+1}, \widehat{\mathbf{w}}(\mathcal{D})\rangle - y_{N+1})^2 \,\middle|\, \mathbf{w}_\star, \mathcal{D}\right] \\
&\leq \mathbb{E}\left[(\langle \mathbf{x}_{N+1}, \widehat{\mathbf{w}}(\mathcal{D})\rangle - y_{N+1})^2 \,\middle|\, \mathbf{w}_\star, \mathcal{D}\right] \\
&= \mathbb{E}\left[(\langle \mathbf{x}_{N+1}, \widehat{\mathbf{w}}(\mathcal{D})\rangle - \langle \mathbf{x}_{N+1}, \mathbf{w}_\star\rangle)^2 \,\middle|\, \mathbf{w}_\star, \mathcal{D}\right] + \sigma^2 \\
&= \|\widehat{\mathbf{w}}(\mathcal{D}) - \mathbf{w}_\star\|_2^2 + \sigma^2,
\end{aligned}
$$

and we also have

$$
\begin{aligned}
\|\widehat{\mathbf{w}}(\mathcal{D}) - \mathbf{w}_\star\|_2^2 &\leq \left\|\mathbf{w}_{\mathrm{ridge}}^{\lambda} - \mathbf{w}_\star\right\|_2^2 + 2\left\|\mathbf{w}_{\mathrm{ridge}}^{\lambda} - \mathbf{w}_\star\right\|_2 \left\|\widehat{\mathbf{w}}(\mathcal{D}) - \mathbf{w}_{\mathrm{ridge}}^{\lambda}\right\|_2 + \left\|\widehat{\mathbf{w}}(\mathcal{D}) - \mathbf{w}_{\mathrm{ridge}}^{\lambda}\right\|_2^2 \\
&\leq \left\|\mathbf{w}_{\mathrm{ridge}}^{\lambda} - \mathbf{w}_\star\right\|_2^2 + 2\underline{\varepsilon}\left\|\mathbf{w}_{\mathrm{ridge}}^{\lambda} - \mathbf{w}_\star\right\|_2 + \underline{\varepsilon}^2.
\end{aligned}
$$

Recall that $2\mathsf{BayesRisk}_\pi = \mathbb{E}_{\mathbf{w}_\star, \mathcal{D}}\|\mathbf{w}_{\mathrm{ridge}}^{\lambda} - \mathbf{w}_\star\|_2^2 + \sigma^2$. Note that $2\mathsf{BayesRisk}_\pi \leq 1 + \sigma^2$ by definition. Therefore, we can conclude that

$$
\mathbb{E}\left[\mathbf{1}\{\mathcal{E}\}(\widehat{y}_{N+1} - y_{N+1})^2\right] \leq 2\mathsf{BayesRisk}_\pi + 2\underline{\varepsilon} + \underline{\varepsilon}^2.
$$

**Prediction risk under bad event $\mathcal{E}^c$.** Notice that

$$
\mathbb{E}\left[\mathbf{1}\{\mathcal{E}^c\}(\widehat{y}_{N+1} - y_{N+1})^2\right] \leq \sqrt{\mathbb{P}(\mathcal{E}^c)\mathbb{E}[(\widehat{y}_{N+1} - y_{N+1})^4]}.
$$

We can upper bound $\mathbb{P}(\mathcal{E}^c) = \mathbb{P}(\mathcal{E}_\pi^c \cup \mathcal{E}_w^c \cup \mathcal{E}_b^c \cup \mathcal{E}_{b,N+1}^c)$ by Lemma B.1, Lemma B.2 and the sub-Gaussian tail bound:

$$
\mathbb{P}(\mathcal{E}_\pi^c) \leq \frac{\delta}{2} + \exp(-N/8), \qquad \mathbb{P}(\mathcal{E}_w^c) \leq 2\exp(-N/8), \qquad \mathbb{P}(\mathcal{E}_b^c \cup \mathcal{E}_{b,N+1}^c) \leq \frac{\delta}{4}.
$$

Thus, as long as $N \geq 8\log(12/\delta)$, we have $\mathbb{P}(\mathcal{E}^c) \leq \delta$. Further, a simple calculation yields

$$
\mathbb{E}(\widehat{y}_{N+1} - y_{N+1})^4 \leq 8\mathbb{E}\widehat{y}_{N+1}^4 + 8\mathbb{E}y_{N+1}^4 \leq 8B_y^2 + 8\mathbb{E}y_{N+1}^4.
$$

Notice that $y_{N+1}|\mathbf{w}_\star \sim \mathsf{N}(0, \|\mathbf{w}_\star\|_2^2 + \sigma^2)$, hence $\mathbb{E}y_{N+1}^4 = 3\mathbb{E}(\|\mathbf{w}_\star\|_2^2 + \sigma^2)^2 \leq 3(3 + 2\sigma^2 + \sigma^4) \leq B_y^4$. Thus, we can conclude that

$$
\mathbb{E}\left[\mathbf{1}\{\mathcal{E}^c\}(\widehat{y}_{N+1} - y_{N+1})^2\right] \leq 4\sqrt{\delta}B_y.
$$

**Choosing $\underline{\varepsilon}$ and $\delta$.** Combining the inequalities above, we have

$$
\mathbb{E}(\widehat{y}_{N+1} - y_{N+1})^2 \leq 2\mathsf{BayesRisk}_\pi + \left[2\underline{\varepsilon}\sqrt{2\mathsf{BayesRisk}_\pi} + \underline{\varepsilon}^2 + 4\sqrt{\delta}B_y\right].
$$

To ensure $\frac{1}{2}\mathbb{E}(\widehat{y}_{N+1} - y_{N+1})^2 \leq \mathsf{BayesRisk}_\pi + \varepsilon$, we only need to take $(\varepsilon, \delta)$ so that the following constraints are satisfied:

$$
\underline{\varepsilon} = \frac{1}{2}\min\{\varepsilon, \sqrt{\varepsilon}\}, \qquad 4\sqrt{\delta}B_y \leq \frac{\varepsilon}{2}, \qquad N \geq 8\log(12/\delta).
$$

Therefore, it suffices to take $\delta = \frac{c_0}{\log^2(N)}\left(\frac{\varepsilon^2}{1+\sigma^2}\right)^2$ for some small constant $c_0$, then as long as

$$
N \geq C\log\left(\frac{\sigma^2 + 1}{\varepsilon}\right) + C.
$$

our choice of $\underline{\varepsilon}$ and $\delta$ is feasible. Note that $\kappa \leq \mathcal{O}\left(1 + \sigma^{-2}\right)$, and hence under such choice of $(\underline{\varepsilon}, \delta)$, we have $L = O(\log(1/\varepsilon))$ and $\|\boldsymbol{\theta}\| = \widetilde{O}\left(\sqrt{d}\right)$. This is the desired result. $\qquad\square$

---

[7]We need this, as on $\mathcal{E}^c$, the transformer output at this location could in principle depend additionally on $\mathbf{x}_{N+1}$, as (15) may not hold due to the potential unboundedness of its input. A similar fact will also appear in later proofs (for generalized linear models and Lasso).

**Lemma F.1.** *Under the event $\mathcal{E}_\pi \cap \mathcal{E}_w$, we have $\left\|\mathbf{w}_{\mathrm{ridge}}^\lambda\right\|_2 \le \mathcal{O}\left(B_w^\star + \sigma\right)$.*

*Proof of Lemma F.1.* By the definition of $\mathbf{w}_{\mathrm{ridge}}^\lambda$ and recall that $\lambda = d\sigma^2/N$, we have $\mathbf{w}_{\mathrm{ridge}}^\lambda = (\mathbf{X}^\top\mathbf{X} + d\sigma^2\mathbf{I}_d)^{-1}\mathbf{X}^\top\mathbf{y}$.

Therefore, we only need to prove the following fact: for any $\gamma > 0$ and $\widehat{\boldsymbol{\beta}} = (\mathbf{X}^\top\mathbf{X} + d\gamma\mathbf{I}_d)^{-1}\mathbf{X}^\top\mathbf{y}$, we have

$$\|\widehat{\boldsymbol{\beta}}\|_2 \le B_w^\star + 10\sigma(1 + \gamma^{-1/2}). \tag{34}$$

We now prove (34). Note that we have

$$\|\widehat{\boldsymbol{\beta}}\|_2 = \|(\mathbf{X}^\top\mathbf{X} + d\gamma\mathbf{I}_d)^{-1}\mathbf{X}^\top(\mathbf{X}\mathbf{w}_\star + \boldsymbol{\varepsilon})\|_2 \le \|\mathbf{B}_1\|_{\mathrm{op}}\|\mathbf{w}_\star\|_2 + \|\mathbf{B}_2\|_{\mathrm{op}}\|\boldsymbol{\varepsilon}\|_2$$

where $\mathbf{B}_1 = \mathbf{X}^\top\mathbf{X}(\mathbf{X}^\top\mathbf{X} + d\gamma\mathbf{I}_d)^{-1}$, $\mathbf{B}_2 = (\mathbf{X}^\top\mathbf{X} + d\gamma\mathbf{I}_d)^{-1}\mathbf{X}^\top$. Note that $\|\mathbf{B}_1\|_{\mathrm{op}} \le 1$ clearly holds, and under $\mathcal{E}_\pi$ we also have $\|\boldsymbol{\varepsilon}\|_2 \le 2\sqrt{N}\sigma$. Therefore, it remains to bound the term $\|\mathbf{B}_2\|_{\mathrm{op}}$.

Consider the SVD decomposition of $\mathbf{X} = U\Sigma V$, $\Sigma = \mathrm{diag}(\lambda_1, \cdots, \lambda_d)$, and $U \in \mathbb{R}^{N \times d}$, $V \in \mathbb{R}^{d \times d}$ are orthonormal matrices. Then $\mathbf{B}_2 = V^\top(\Sigma^2 + d\gamma\mathbf{I}_d)^{-1}\Sigma U^\top$, and hence

$$\|\mathbf{B}_2\|_{\mathrm{op}} = \left\|(\Sigma^2 + d\gamma\mathbf{I}_d)^{-1}\Sigma\right\|_{\mathrm{op}} = \max_i \frac{\lambda_i}{\lambda_i^2 + d\gamma}.$$

When $N \le 36d$, we directly have $\|\mathbf{B}_2\|_{\mathrm{op}} \le \frac{1}{2}(d\gamma)^{-1/2} \le 3(N\gamma)^{-1/2}$. Otherwise, we have $N \ge 36d$, and then for each $i \in [d]$, $\lambda_i \ge \sqrt{\lambda_{\min}(\mathbf{X}^\top\mathbf{X})} \ge \sqrt{\alpha N} \ge \sqrt{N}/3$. Hence, in this case we also have $\|\mathbf{B}_2\|_{\mathrm{op}} \le \max_i \lambda_i^{-1} \le 3N^{-1/2}$. Combining the both cases completes the proof of (34). $\qquad\square$

## G    In-context learning of generalized linear models

As a natural generalization of linear regression, we now show that transformers can recover learn generalized linear models (GLMs) [53] (which includes logistic regression for linear classification as an important special case), by implementing the corresponding convex risk minimization algorithm in context, and achieve near-optimal excess risk under standard statistical assumptions.

Let $g : \mathbb{R} \to \mathbb{R}$ be a link function that is non-decreasing and $C^2$-smooth. We consider the following convex empirical risk minimization (ERM) problem

$$\mathbf{w}_{\mathrm{GLM}} := \underset{\mathbf{w}\in\mathbb{R}^d}{\arg\min}\, \widehat{L}_N(\mathbf{w}) := \frac{1}{N}\sum_{i=1}^N \ell(\langle\mathbf{x}_i, \mathbf{w}\rangle, y_i), \tag{ICGLM}$$

where $\ell(t, y) := -yt + \int_0^t g(s)ds$ is the convex (integral) loss associated with $g$. A canonical example of (ICGLM) is logistic regression, in which $g(t) = \sigma_{\log}(t) := (1 + e^{-t})^{-1}$ is the sigmoid function, and the resulting $\ell(t, y) = \ell_{\log}(t, y) = -yt + \log(1 + e^t)$ is the logistic loss.

The following result (proof in Appendix G.1) shows that, as long as the empirical risk $\widehat{L}_N$ satisfies strong convexity and bounded solution conditions (similar as in Theorem 4), transformers can approximately implement the ERM predictor $g(\langle\mathbf{x}_{N+1}, \mathbf{w}_{\mathrm{GLM}}\rangle)$, with $\mathbf{w}_{\mathrm{GLM}}$ given by (ICGLM).

**Theorem G.1** (Implementing convex risk minimization for GLMs). *For any $0 < \alpha < \beta$ with $\kappa := \frac{\beta}{\alpha}$, $B_w > 0, B_x > 0$, $\kappa_w := L_g B_x^2/\alpha + 1$ and $\varepsilon < B_w/2$, there exists an attention-only transformer $\mathrm{TF}_{\boldsymbol{\theta}}^0$ with*

$$L = \lceil 2\kappa\log(L_g B_w B_x/\varepsilon)\rceil + 1, \qquad \max_{\ell\in[L]} M^{(\ell)} \le \widetilde{\mathcal{O}}\left(C_g^2 \kappa_w^2 \varepsilon^{-2}\right), \qquad \|\boldsymbol{\theta}\| \le \mathcal{O}\left(R + \beta^{-1}C_g\right),$$

*(where $L_g := \sup_t |g'(t)|$, $R := \max\{B_x B_w, B_y, 1\}$, and $C_g > 0$ is a constant that depends only on $R$ and the $C^2$-smoothness of $g$ within $[-R, R]$), such that the following holds. On any input data $(\mathcal{D}, \mathbf{x}_{N+1})$ such that*

$$\alpha \le \lambda_{\min}(\nabla^2\widehat{L}_N(\mathbf{w})) \le \lambda_{\max}(\nabla^2\widehat{L}_N(\mathbf{w})) \le \beta \text{ for all } \mathbf{w} \in \mathsf{B}_2(B_w), \qquad \|\mathbf{w}_{\mathrm{GLM}}\|_2 \le B_w/2, \tag{35}$$

$\mathrm{TF}_{\boldsymbol{\theta}}^0(\mathbf{H}^{(0)})$ *approximately implements (ICGLM): We have* $\mathbf{h}_{N+1}^{(L+1)} := [\mathbf{x}_{N+1}; \widehat{y}_{N+1}; \widehat{\mathbf{w}}; 1; 1]$*, where*

$$|\widehat{y}_{N+1} - g(\langle \mathbf{x}_{N+1}, \mathbf{w}_{\mathrm{GLM}} \rangle)| \le \varepsilon.$$

In Theorem G.1, the number of heads scales as $\widetilde{\mathcal{O}}(1/\varepsilon^2)$ as opposed to $\Theta(1)$ as in ridge regression (Theorem 4), due to the fact that the gradient of the loss is in general a smooth function that can be only *approximately* expressed as a sum-of-relus (cf. Definition D.1 & Lemma B.5) rather than exactly expressed as in the case for the square loss.

**In-context prediction power** We next show that (proof in Appendix G.2) the transformer constructed in Theorem G.1 achieves desirable statistical power if the in-context data distribution satisfies standard statistical assumptions for learning GLMs. Let $L_{\mathsf{P}}(\mathbf{w}) := \mathbb{E}_{(\mathbf{x},y) \sim \mathsf{P}}[\ell(\langle \mathbf{w}, \mathbf{x} \rangle, y)]$ denote the corresponding population risk for any distribution $\mathsf{P}$ of $(\mathbf{x}, y)$. When $\mathsf{P}$ is *realizable* by a generalized linear model of link function $g$ and parameter $\boldsymbol{\beta}$ in the sense that $\mathbb{E}_{\mathsf{P}}[y|\mathbf{x}] = g(\langle \boldsymbol{\beta}, \mathbf{x} \rangle)$, it is a standard result that $\boldsymbol{\beta}$ is indeed a minimizer of $L_{\mathsf{P}}$ [42] (see also [6, Appendix A.3]).

**Theorem G.2** (Statistical guarantee for generalized linear models). *For any fixed set of parameters defined in Assumption B, there exists a transformer $\boldsymbol{\theta}$ with $L \le \mathcal{O}(\log(N))$ layers and $\max_{\ell \in [L]} M^{(\ell)} \le \widetilde{\mathcal{O}}(d^3 N)$, such that for any distribution $\mathsf{P}$ satisfying Assumption B with those parameters, as long as $N \ge \mathcal{O}(d)$, that outputs $\widehat{y}_{N+1} = \widetilde{\mathrm{read}}_{\mathsf{y}}(\mathrm{TF}_{\boldsymbol{\theta}}(\mathbf{H}))$ and $\widehat{\mathbf{w}} = \widetilde{\mathrm{read}}_{\mathsf{w}}(\mathrm{TF}_{\boldsymbol{\theta}}(\mathbf{H})) \in \mathbb{R}^d$ (for another read-out function $\widetilde{\mathrm{read}}_{\mathsf{w}}$) satisfying the following.*

(a) *$\widehat{\mathbf{w}}$ achieves small excess risk under the population loss, i.e. for the linear prediction $\widehat{y}_{N+1}^{\mathsf{lin}} := \langle \mathbf{x}_{N+1}, \widehat{\mathbf{w}} \rangle$,*

$$\mathbb{E}_{(\mathcal{D}, \mathbf{x}_{N+1}, y_{N+1}) \sim \mathsf{P}} \left[ \ell(\widehat{y}_{N+1}^{\mathsf{lin}}, y_{N+1}) \right] - \min_{\boldsymbol{\beta}} L_{\mathsf{P}}(\boldsymbol{\beta}) \le \mathcal{O}(d/N). \tag{36}$$

(b) *(Realizable setting) If there exists a $\boldsymbol{\beta} \in \mathbb{R}^d$ such that under $\mathsf{P}$, $\mathbb{E}[y|\mathbf{x}] = g(\langle \boldsymbol{\beta}, \mathbf{x} \rangle)$ almost surely, then*

$$\mathbb{E}_{(\mathcal{D}, \mathbf{x}_{N+1}, y_{N+1}) \sim \mathsf{P}} \left[ (\widehat{y}_{N+1} - y_{N+1})^2 \right] \le \mathbb{E}_{(\mathbf{x}_{N+1}, y_{N+1}) \sim \mathsf{P}} \left[ (g(\langle \boldsymbol{\beta}, \mathbf{x}_{N+1} \rangle) - y_{N+1})^2 \right] + \mathcal{O}(d/N),$$
$$\tag{37}$$

*or equivalently, $\mathbb{E}[(\widehat{y}_{N+1} - \mathbb{E}[y_{N+1}|\mathbf{x}_{N+1}])^2] \le \mathcal{O}(d/N)$.*

Above, $\mathcal{O}(\cdot)$ hides constants that depend polynomially on the parameters in Assumption B. Similar as in Corollary 5, the $\mathcal{O}(d/N)$ excess risk obtained here matches the optimal (fast) rate for typical learning problems with $d$ parameters and $N$ samples [87].

**Assumption B** (Well-posedness for learning GLMs). *We assume that there is some $B_\mu > 0$ such that for any $t \in [-B_\mu, B_\mu]$, $g'(t) \ge \mu_g > 0$.*

*We also assume that for each $i \in [N+1]$, $(\mathbf{x}_i, y_i)$ is independently sampled from $\mathsf{P}$ such that the following holds.*

(a) *Under the law $(\mathbf{x}, y) \sim \mathsf{P}$, We have $\mathbf{x} \sim \mathrm{SG}(K_x)$, $y \sim \mathrm{SG}(K_y)$ and $g(\langle \mathbf{w}, \mathbf{x} \rangle) \sim \mathrm{SG}(K_y) \; \forall \mathbf{w} \in \mathsf{B}_2(B_w)$.*

(b) *For some $\mu_x > 0$, it holds that*

$$\mathbb{E}[\mathbf{1}\{|\mathbf{x}^\top \mathbf{w}| \le B_\mu/2\}\mathbf{x}\mathbf{x}^\top] \succeq \mu_x \mathbf{I}_d \quad \forall \mathbf{w} \in \mathsf{B}_2(B_w).$$

(c) *For $\boldsymbol{\beta}^\star = \arg\min L_{\mathsf{P}}$, it holds $\|\boldsymbol{\beta}^\star\|_2 \le B_w/4$.*

Applying Theorem G.2 to logistic regression, we have the following result as a direct corollary. Below, the Gaussian input assumption is for convenience only and can be generalized to e.g. sub-Gaussian input.

**Corollary G.1** (In-context logistic regression). *Consider any in-context data distribution $\mathsf{P}$ satisfying*

$$\mathbf{x} \sim \mathsf{N}(0, \mathbf{I}_d), \qquad y \in \{0, 1\}, \qquad \arg\min_{\boldsymbol{\beta} \in \mathbb{R}^d} L_{\mathsf{P}}(\boldsymbol{\beta}) \in \mathsf{B}_2(B_w^\star).$$

*For the link function $g = \sigma_{\log}$ and $B_w^\star = \mathcal{O}(1)$, we can choose $B_w, B_\mu, \mu_g, L_g, \mu_x, K_x, K_y = \Theta(1)$ so that Assumption B holds. In that case, when $N \ge \mathcal{O}(d)$, there exists a transformer $\boldsymbol{\theta}$ with $L = \mathcal{O}(\log(N))$ layers, such that for any $\mathsf{P}$ considered above,*

(a) *The estimation $\widehat{\mathbf{w}} = \widetilde{\mathrm{read}}_w(\mathrm{TF}_{\boldsymbol{\theta}}(\mathbf{H}))$ outputted by $\boldsymbol{\theta}$ achieves excess risk bound (36).*

(b) *(Realizable setting) Consider the logistic in-context data distribution*

$$\mathsf{P}_{\boldsymbol{\beta}}^{\log} : \qquad \mathbf{x} \sim \mathsf{N}(0, \mathbf{I}_d), \qquad y|\mathbf{x} \sim \mathrm{Bernoulli}(g(\langle \boldsymbol{\beta}, \mathbf{x}\rangle)).$$

*Then, for any distribution $\mathsf{P} = \mathsf{P}_{\boldsymbol{\beta}}^{\log}$ with $\|\boldsymbol{\beta}\|_2 \leq B_w^\star$, the prediction $\widehat{y}_{N+1} = \widetilde{\mathrm{read}}_y(\mathrm{TF}_{\boldsymbol{\theta}}(\mathbf{H}))$ of $\boldsymbol{\theta}$ additionally achieves the square loss excess risk (37).*

### G.1 Proof of Theorem G.1

Let us fix parameters $\varepsilon_g > 0$ and $T > 0$ (that we specify later in proof).

Define $R = \max\{B_x B_w, B_y, 1\}$ and

$$C_g := \max_{i=0,1,2}\left( R^i \max_{s \in [-B,B]} \left|g^{(i)}(s)\right| \right).$$

By Proposition B.1, $g$ is $(\varepsilon_g, M, R, C)$ with

$$C \leq \mathcal{O}\left(C_g\right), \qquad M \leq \mathcal{O}\left(C_g^2 \varepsilon_g^{-2} \log(1 + C_g \varepsilon_g^{-1})\right).$$

Therefore, we can invoke Theorem D.1 to obtain that, as long as $2T\varepsilon_g \leq B_w$, there exists a $T$-layer attention-only transformer $\boldsymbol{\theta}^{(1:T)}$ with $M$ heads per layer, such that for any input $\mathbf{H}$ of format (3) and satisfies (35), its last layer outputs $\mathbf{h}_i^{(T)} = [\mathbf{x}_i; y_i'; \widehat{\mathbf{w}}^T; \mathbf{0}_{D-2d-3}; 1; t_i]$, such that

$$\left\|\widehat{\mathbf{w}}^T - \mathbf{w}_{\mathrm{GD}}^T\right\|_2 \leq \varepsilon_g \cdot (L\beta^{-1} B_x),$$

where $\{\mathbf{w}_{\mathrm{GD}}^\ell\}_{\ell \in [L]}$ is the sequence of gradient descent iterates with stepsize $\beta^{-1}$ and initialization $\mathbf{w}_{\mathrm{GD}}^0 = \mathbf{0}$. Notice that Proposition B.2 implies (with $\kappa := \beta/\alpha$)

$$\left\|\mathbf{w}_{\mathrm{GD}}^T - \mathbf{w}_{\mathrm{GLM}}\right\|_2 \leq \exp(-T/(2\kappa)) \left\|\mathbf{w}_{\mathrm{GLM}}\right\|_2 \leq \exp(-T/(2\kappa)) \cdot \frac{B_w}{2} := \varepsilon_o.$$

Furthermore, we can show that (similar to the proof of Theorem D.1 (b)), there exists a single attention layer $\boldsymbol{\theta}^{(T+1)}$ with $M$ heads such that it outputs $\mathbf{h}_{N+1}^{(T+1)} = [\mathbf{x}_{N+1}; \widehat{y}_{N+1}; \widehat{\mathbf{w}}^T; \mathbf{0}_{D-2d-3}; 1; 0]$, where $\left|\widehat{y}_{N+1} - g(\langle \mathbf{x}_{N+1}, \widehat{\mathbf{w}}^T\rangle)\right| \leq \varepsilon_g$.

In the following, we show that for suitably chosen $(T, \varepsilon_g)$, $\boldsymbol{\theta} = (\boldsymbol{\theta}^{(1:T)}, \boldsymbol{\theta}^{(T+1)})$ is the desired transformer. First notice that its output $\mathbf{h}_{N+1}^{(T+1)} = [\mathbf{x}_{N+1}; \widehat{y}_{N+1}; \widehat{\mathbf{w}}^T; \mathbf{0}_{D-2d-3}; 1; 0]$ satisfies

$$\begin{aligned}
\left|\widehat{y}_{N+1} - g(\langle \mathbf{x}_{N+1}, \mathbf{w}_{\mathrm{GLM}}\rangle)\right| &\leq \left|\widehat{y}_{N+1} - g(\langle \mathbf{x}_{N+1}, \widehat{\mathbf{w}}^T\rangle)\right| + L_g \left|\langle \mathbf{x}_{N+1}, \widehat{\mathbf{w}}^T\rangle - \langle \mathbf{x}_{N+1}, \mathbf{w}_{\mathrm{GLM}}\rangle\right| \\
&\leq \varepsilon_g + L_g B_x \left\|\widehat{\mathbf{w}}^T - \mathbf{w}_{\mathrm{GD}}^T\right\|_2 + L_g B_x \left\|\mathbf{w}_{\mathrm{GD}}^T - \mathbf{w}_{\mathrm{GLM}}\right\|_2 \\
&\leq \varepsilon_g(1 + L_g B_x \cdot T\beta^{-1} B_x) + L_g B_x \varepsilon_o.
\end{aligned}$$

Therefore, for any fixed $\varepsilon > 0$, we can take

$$T = \lceil 2\kappa \log(L_g B_x B_w/\varepsilon)\rceil, \qquad \varepsilon_g = \frac{1}{2}\frac{\varepsilon}{1 + T \cdot (L_g B_x^2 \beta^{-1})},$$

so that the $\boldsymbol{\theta}$ we construct above ensures $\left|\widehat{y}_{N+1} - g(\langle \mathbf{x}_{N+1}, \mathbf{w}_{\mathrm{GLM}}\rangle)\right| \leq \varepsilon$ for any input $\mathbf{H}$ that satisfies (35). The upper bound on $\|\boldsymbol{\theta}\|$ follows immediately from Theorem D.1. $\square$

### G.2 Proof of Theorem G.2

We summarize some basic and useful facts about GLM in the following theorem. Its proof is presented in Appendix G.3 - G.6.

**Theorem G.3.** *Under Assumption B, the following statements hold with universal constant $C_0$ and constant $C_1, C_2$ that depend only on the parameters $(K_x, K_y, B_\mu, B_w, \mu_x, L_g, \mu_g)$.*

*(a) As long as $N \geq C_1 \cdot d$, the following event happens with probability at least $1 - 2e^{-N/C_1}$:*

$$\mathcal{E}_w : \qquad \frac{1}{8}\mu_g\mu_x \leq \lambda_{\min}(\nabla^2 \widehat{L}_N(\mathbf{w})) \leq \lambda_{\max}(\nabla^2 \widehat{L}_N(\mathbf{w})) \leq 8L_g K_x^2, \quad \forall \mathbf{w} \in \mathsf{B}_2(B_w).$$

*(b) For any $\delta > 0$, we have with probability at least $1 - \delta$ that*

$$\varepsilon_{\text{stat}} := \sup_{\mathbf{w} \in \mathsf{B}_2(B_w)} \left\| \nabla_{\mathbf{w}} \widehat{L}_N(\mathbf{w}) - \nabla_{\mathbf{w}} \mathbb{E}[\widehat{L}_N(\mathbf{w})] \right\|_2 \leq C_0 K_x K_y \max\left\{ \sqrt{\frac{d\iota + \log(1/\delta)}{N}}, \frac{d\iota + \log(1/\delta)}{N} \right\},$$

*where we denote $\iota = \log(2 + L_g K_x^2 B_w / K_y)$.*

*(c) Condition on (a) holds and $N \geq C_2 \cdot d$, the event $\mathcal{E}_r := \{\|\mathbf{w}_{\text{GLM}}\|_2 \leq B_w/2\}$ happens with probability at least $1 - e^{N/C_2}$.*

*(d) For any $\mathbf{w} \in \mathsf{B}_2(B_w)$, it holds that*

$$L_p(\mathbf{w}) - L_p(\boldsymbol{\beta}) \leq \frac{4}{\mu_g\mu_x}\left( \varepsilon_{\text{stat}}^2 + \left\| \nabla \widehat{L}_N(\mathbf{w}) \right\|_2^2 \right).$$

*(e) (Realizable setting) As long as $\mathbf{w}_{\text{GLM}} \in \mathsf{B}_2(B_w)$, it holds that*

$$\mathbb{E}_{\mathbf{x}}(g(\langle \mathbf{x}, \mathbf{w}_{\text{GLM}} \rangle) - g(\langle \mathbf{x}, \boldsymbol{\beta} \rangle))^2 \leq \frac{L_g}{\mu_x\mu_g}\varepsilon_{\text{stat}}^2.$$

Therefore, we can set

$$\alpha = \frac{\mu_g\mu_x}{8}, \qquad \beta = 8L_g K_x^2,$$
$$B_x = C_0 K_x \sqrt{d\log(N/\delta)}, \qquad B_y = C_0 K_y \sqrt{\log(N/\delta)}.$$

Consider the following good events

$$\mathcal{E}_b = \{\forall i \in [N], \ \|\mathbf{x}_i\|_2 \leq B_x, \ |y_i| \leq B_y\},$$
$$\mathcal{E}_{b,N+1} = \{\|\mathbf{x}_{N+1}\|_2 \leq B_x, \ |y_{N+1}| \leq B_y\},$$
$$\mathcal{E} = \mathcal{E}_r \cap \mathcal{E}_w \cap \mathcal{E}_b \cap \mathcal{E}_{b,N+1}.$$

Under the event $\mathcal{E}$ and our choice of $\alpha, \beta$, the problem (ICGLM) is well-conditioned (i.e. (35) holds).

Theorem G.1 implies that there exists a transformer $\boldsymbol{\theta}$ such that for any input $\mathbf{H}$ of the form (3), $\text{TF}_{\boldsymbol{\theta}}$ outputs $\mathbf{h}'_{N+1} = [\mathbf{x}_{N+1}; \widetilde{y}_{N+1}; \widetilde{\mathbf{w}}; \mathbf{0}_{D-2d-3}; 1; 0]$, such that the output is given by $\widehat{y}_{N+1} = \widetilde{\text{read}}_y(\text{TF}_{\boldsymbol{\theta}}(\mathbf{H})) = \text{clip}_{B_y}(\widetilde{y}_{N+1})$ and $\widehat{\mathbf{w}} = \widetilde{\text{read}}_w(\text{TF}_{\boldsymbol{\theta}}(\mathbf{H})) := \text{Proj}_{\mathsf{B}_2(B_w)}(\widetilde{\mathbf{w}})$, and the following holds on the good event $\mathcal{E}$:

(a) $\widetilde{y}_{N+1} = f_{\mathcal{D}}(\mathbf{x}_{N+1})$, where $f_{\mathcal{D}} = \mathcal{A}(\mathcal{D})$ is a predictor such that $|f_{\mathcal{D}}(\mathbf{x}) - g(\langle \mathbf{x}, \mathbf{w}_{\text{GLM}} \rangle)| \leq \varepsilon$ for all $\mathbf{x} \in \mathsf{B}_2(B_x)$.

(b) $\widetilde{\mathbf{w}} = \widetilde{\mathbf{w}}(\mathcal{D}) \in \mathsf{B}_2(B_w)$ depends only on $\mathcal{D}$ (by the proof of Theorem G.1 and Theorem D.1), such that $\left\| \nabla \widehat{L}_N(\widetilde{\mathbf{w}}) \right\|_2 \leq \frac{\beta\varepsilon}{L_g B_w}$.

In the following, we show that $\boldsymbol{\theta}$ constructed above fulfills both (a) & (b) of Theorem G.2. The bounds on number of layers and heads and $\|\boldsymbol{\theta}\|$ follows from plugging our choice of $B_x, B_y$ in our proof of Theorem G.1.

**Proof of Theorem G.2 (a).** Notice that under the good event $\mathcal{E}$, we have $\widehat{\mathbf{w}} = \widetilde{\mathbf{w}} = \widetilde{\mathbf{w}}(\mathcal{D})$ depends only on $\mathcal{D}$. Then we have

$$\mathbb{E}_{(\mathcal{D},\mathbf{x}_{N+1},y_{N+1})}\left[ \ell(\widehat{y}_{N+1}^{\text{lin}}, y_{N+1}) \right]$$
$$= \mathbb{E}_{(\mathcal{D},\mathbf{x}_{N+1},y_{N+1})}\left[ \mathbb{1}\{\mathcal{E}\}\ell(\widehat{y}_{N+1}^{\text{lin}}, y_{N+1}) \right] + \mathbb{E}_{(\mathcal{D},\mathbf{x}_{N+1},y_{N+1})}\left[ \mathbb{1}\{\mathcal{E}^c\}\ell(\widehat{y}_{N+1}^{\text{lin}}, y_{N+1}) \right]$$
$$= \mathbb{E}_{(\mathcal{D},\mathbf{x}_{N+1},y_{N+1})}\left[ \mathbb{1}\{\mathcal{E}\}\ell(\langle \mathbf{x}_{N+1}, \widetilde{\mathbf{w}}(\mathcal{D}) \rangle, y_{N+1}) \right] + \mathbb{E}_{(\mathcal{D},\mathbf{x}_{N+1},y_{N+1})}\left[ \mathbb{1}\{\mathcal{E}^c\}\ell(\widehat{y}_{N+1}^{\text{lin}}, y_{N+1}) \right].$$

Thus, we can consider $\mathcal{E}_0 = \mathcal{E}_r \cap \mathcal{E}_w \cap \mathcal{E}_b$, and then

$$\mathbb{E}_{(\mathcal{D},\mathbf{x}_{N+1},y_{N+1})}\left[ \mathbb{1}\{\mathcal{E}\}\ell(\langle \mathbf{x}_{N+1}, \widetilde{\mathbf{w}}(\mathcal{D}) \rangle, y_{N+1}) \right]$$

$$= \mathbb{E}_{(\mathcal{D}, \mathbf{x}_{N+1}, y_{N+1})}[1\{\mathcal{E}_0\}\ell(\langle\mathbf{x}_{N+1}, \widetilde{\mathbf{w}}(\mathcal{D})\rangle, y_{N+1})] - \mathbb{E}_{(\mathcal{D}, \mathbf{x}_{N+1}, y_{N+1})}[1\{\mathcal{E}_0 - \mathcal{E}\}\ell(\langle\mathbf{x}_{N+1}, \widetilde{\mathbf{w}}(\mathcal{D})\rangle, y_{N+1})]$$

$$= \mathbb{E}_{(\mathcal{D}, \mathbf{x}_{N+1}, y_{N+1})}[1\{\mathcal{E}_0\}L_p(\widehat{\mathbf{w}}(\mathcal{D}))] - \mathbb{E}_{(\mathcal{D}, \mathbf{x}_{N+1}, y_{N+1})}[1\{\mathcal{E}_0 - \mathcal{E}\}\ell(\langle\mathbf{x}_{N+1}, \widetilde{\mathbf{w}}(\mathcal{D})\rangle, y_{N+1})],$$

where the second equality follows from $L_p(\widehat{\mathbf{w}}(\mathcal{D})) = \mathbb{E}_{(\mathbf{x}_{N+1}, y_{N+1})|\mathcal{D}}\ell(\langle\mathbf{x}_{N+1}, \widetilde{\mathbf{w}}(\mathcal{D})\rangle, y_{N+1})$. Therefore,

$$\mathbb{E}_{(\mathcal{D}, \mathbf{x}_{N+1}, y_{N+1})}\big[\ell(\widehat{y}_{N+1}^{\mathsf{lin}}, y_{N+1})\big] - \mathbb{E}_{\mathcal{D}}[1\{\mathcal{E}_0\}L_p(\widetilde{\mathbf{w}}(\mathcal{D}))]$$

$$= \mathbb{E}_{(\mathcal{D}, \mathbf{x}_{N+1}, y_{N+1})}\big[1\{\mathcal{E}^c\}\ell(\widehat{y}_{N+1}^{\mathsf{lin}}, y_{N+1})\big] - \mathbb{E}_{(\mathcal{D}, \mathbf{x}_{N+1}, y_{N+1})}[1\{\mathcal{E}_0 - \mathcal{E}\}\ell(\langle\mathbf{x}_{N+1}, \widetilde{\mathbf{w}}(\mathcal{D})\rangle, y_{N+1})]$$

$$\leq 2\sqrt{\mathbb{P}(\mathcal{E}^c) \cdot \max\big\{\mathbb{E}\big[\ell(\widehat{y}_{N+1}^{\mathsf{lin}}, y_{N+1})^4\big], \mathbb{E}[\ell(\langle\mathbf{x}_{N+1}, \widetilde{\mathbf{w}}(\mathcal{D})\rangle, y_{N+1})^4]\big\}} = \mathcal{O}\left(\frac{B_\ell^2}{N^5}\right),$$

where the last line follows from Cauchy inequality and the fact $\mathbb{P}(\mathcal{E}^c) = \mathcal{O}(N^{-10})$, and $B_\ell$ is defined in Lemma G.1.

Notice that by Theorem G.3 (d), we have

$$\mathbb{E}_{\mathcal{D}}[1\{\mathcal{E}_0\}(L_p(\widetilde{\mathbf{w}}) - \inf L_p)] \leq \frac{4}{\mu_g \mu_x}\left(\mathbb{E}[\varepsilon_{\mathrm{stat}}^2] + \mathbb{E}\left[1\{\mathcal{E}_0\}\big\|\nabla\widehat{L}_N(\widetilde{\mathbf{w}})\big\|_2^2\right]\right),$$

and by Theorem G.3 (b) and taking integration over $\delta > 0$, we have

$$\mathbb{E}[\varepsilon_{\mathrm{stat}}^2] \leq \mathcal{O}(1) \cdot K_x^2 K_y^2\left(\frac{d\iota}{N} + \left(\frac{d\iota}{N}\right)^2\right).$$

Also, we have $\inf L_p = L_p(\boldsymbol{\beta}^\star) \leq B_\ell$ by Lemma G.1. Therefore, we can conclude that

$$\mathbb{E}_{(\mathcal{D}, \mathbf{x}_{N+1}, y_{N+1})}\big[\ell(\widehat{y}_{N+1}^{\mathsf{lin}}, y_{N+1})\big] \leq \inf L_p + \mathcal{O}(1) \cdot \left(\frac{K_x^2 K_y^2 \iota}{\mu_g \mu_x}\frac{d}{N} + \frac{K_x^4}{\mu_g \mu_x B_w}\varepsilon^2 + \frac{B_\ell^2}{N^5}\right).$$

Taking $\varepsilon^2 \leq \frac{K_y^2 \iota}{B_w K_x^2}\frac{d}{N}$ completes the proof. $\qquad\square$

**Proof of Theorem G.2 (b).** Similar to the proof of Corollary 6, we have

$$\mathbb{E}(\widehat{y}_{N+1} - y_{N+1})^2 = \mathbb{E}\big[1\{\mathcal{E}\}(\widehat{y}_{N+1} - y_{N+1})^2\big] + \mathbb{E}\big[1\{\mathcal{E}^c\}(\widehat{y}_{N+1} - y_{N+1})^2\big]$$

$$\leq \mathbb{E}\big[1\{\mathcal{E}\}(\widetilde{y}_{N+1} - y_{N+1})^2\big] + \sqrt{\mathbb{P}(\mathcal{E}^c)\mathbb{E}(\widehat{y}_{N+1} - y_{N+1})^4},$$

where the inequality follows from $y_{N+1} \in [-B_y, B_y]$ on event $\mathcal{E}$. For the first part, we have

$$\mathbb{E}\left[1\{\mathcal{E}\}(\widetilde{y}_{N+1} - y_{N+1})^2\right] = \mathbb{E}\left[1\{\mathcal{E}\}(f_{\mathcal{D}}(\mathbf{x}_{N+1}) - y_{N+1})^2\right]$$

$$\leq \mathbb{E}_{\mathcal{D}}\left[1\{\mathcal{E}_0\} \cdot \mathbb{E}_{(\mathbf{x}, y)\sim\mathsf{P}}\left[1\{\|\mathbf{x}\|_2 \leq B_x\}(f_{\mathcal{D}}(\mathbf{x}) - y)^2\right]\right],$$

where we use the fact that the conditional distribution of $(\mathbf{x}_{N+1}, y_{N+1})|\mathcal{D}$ agrees with P. Thus,

$$\mathbb{E}\left[1\{\mathcal{E}\}(\widetilde{y}_{N+1} - y_{N+1})^2\right] - \mathbb{E}_{(\mathbf{x}, y)\sim\mathsf{P}}(g(\langle\mathbf{x}, \boldsymbol{\beta}\rangle) - y)^2$$

$$\leq \mathbb{E}_{\mathcal{D}}\left[1\{\mathcal{E}_0\} \cdot \left(\mathbb{E}_{(\mathbf{x}, y)\sim\mathsf{P}}1\{\|\mathbf{x}\|_2 \leq B_x\}(f_{\mathcal{D}}(\mathbf{x}) - y)^2 - \mathbb{E}_{(\mathbf{x}, y)\sim\mathsf{P}}(g(\langle\mathbf{x}, \boldsymbol{\beta}\rangle) - y)^2\right)\right]$$

$$\leq \mathbb{E}_{\mathcal{D}}\left[1\{\mathcal{E}_0\} \cdot \mathbb{E}_{\mathbf{x}}1\{\|\mathbf{x}\|_2 \leq B_x\}(f_{\mathcal{D}}(\mathbf{x}) - g(\langle\mathbf{x}, \boldsymbol{\beta}\rangle))^2\right]$$

$$\leq 2\mathbb{E}_{\mathcal{D}}\left[1\{\mathcal{E}_0\} \cdot \mathbb{E}_{\mathbf{x}}1\{\|\mathbf{x}\|_2 \leq B_x\}(f_{\mathcal{D}}(\mathbf{x}) - g(\langle\mathbf{x}, \mathbf{w}_{\mathrm{GLM}}\rangle))^2\right] + 2\mathbb{E}_{\mathcal{D}}\left[1\{\mathcal{E}_0\} \cdot \mathbb{E}_{\mathbf{x}}(g(\langle\mathbf{x}, \mathbf{w}_{\mathrm{GLM}}\rangle) - g(\langle\mathbf{x}, \boldsymbol{\beta}\rangle))^2\right]$$

$$\leq 2\varepsilon^2 + \frac{2L_g}{\mu_x \mu_g}\mathbb{E}[\varepsilon_{\mathrm{stat}}^2] \leq 2\varepsilon^2 + \mathcal{O}(1) \cdot \frac{L_g K_x^2 K_y^2 \iota}{\mu_x \mu_g}\frac{d}{N}.$$

For the second part, we know $\mathbb{P}(\mathcal{E}^c) = \mathcal{O}(N^{-10})$ and

$$\mathbb{E}(\widehat{y}_{N+1} - y_{N+1})^4 \leq 8\mathbb{E}\widehat{y}_{N+1}^2 + 8\mathbb{E}y_{N+1}^4 = \mathcal{O}(B_y^4).$$

In conclusion, we have

$$\mathbb{E}(\widehat{y}_{N+1} - y_{N+1})^2 \leq \mathbb{E}(y_{N+1} - g(\langle\mathbf{x}_{N+1}, \boldsymbol{\beta}\rangle))^2 + 2\varepsilon^2 + \mathcal{O}(1) \cdot \frac{L_g K_x^2 K_y^2 \iota}{\mu_x \mu_g}\frac{d}{N} + \mathcal{O}\left(\frac{B_y^2}{N^5}\right).$$

Taking $\varepsilon^2 \leq \frac{L_g K_x^2 K_y^2 \iota}{\mu_x \mu_g}\frac{d}{N}$ completes the proof. $\qquad\square$

**Lemma G.1.** *Suppose that* $\mathbf{x} \sim \mathrm{SG}(K_x)$, $y \sim \mathrm{SG}(K_y)$, *and* $\mathbf{w}$ *is a (possibly random) vector such that* $\|\mathbf{w}\|_2 \le B_w$. *Then*

$$\mathbb{E}\big[\ell(\langle \mathbf{x}, \mathbf{w}\rangle, y)^4\big]^{1/4} \le \mathcal{O}\big(L_g K_x^2 B_w^2 d + K_x K_y B_w d\big) =: B_\ell.$$

*Proof.* Notice that by our assumption, $|g(0)| \le 2K_y$. Therefore, by the definition of $\ell$,

$$|\ell(t,y)| = \left| -yt + \int_0^t g(s)ds \right| \le |t(g(0) - y)| + \left| \int_0^t (g(s) - g(0))ds \right| \le |t|\,(2K_y + |y|) + 2L_g t^2.$$

The proof is then done by bounding the moment by $\mathbb{E}\,|y|^8 \le \mathcal{O}\big(K_y^8\big)$ and $\mathbb{E}\,|\langle \mathbf{x}, \mathbf{w}\rangle|^8 \le B_w^8 \mathbb{E}\,\|\mathbf{x}\|_2^8 \le \mathcal{O}\big((\sqrt{d}B_w K_x)^8\big)$, which is standard (by utilizing the tail bound of sub-Gaussian/sub-Exponential random variable). $\qquad\square$

### G.3 Proof of Theorem G.3 (a)

We begin with the upper bound on $\lambda_{\max}(\nabla^2 \widehat{L}_N(\mathbf{w}))$. By Lemma B.3, as long as $N \ge C_0 \cdot d$, the following event

$$\mathcal{E}_{w,0}: \qquad \left\| \frac{1}{N}\sum_{i=1}^N \mathbf{x}_i \mathbf{x}_i^\top \right\|_{\mathrm{op}} \le 8K^2.$$

happens with probability at least $1 - \exp(-N/C_0)$. By the assumption that $\sup |g'| \le L_g$, it is clear that when $\mathcal{E}_{w,0}$ holds, we have $\lambda_{\max}(\nabla^2 \widehat{L}_N(\mathbf{w})) \le 8L_g K_x^2 \;\forall \mathbf{w} \in \mathbb{R}^d$.

In the following, we analyze the quantity $\lambda_{\max}(\nabla^2 \widehat{L}_N(\mathbf{w}))$. We have to invoke the following covering argument (see e.g. [85, Section 4.1.1]).

**Lemma G.2.** *Suppose that* $\mathcal{V}$ *is a* $\varepsilon$-*covering of* $\mathbb{S}^{d-1}$ *with* $\varepsilon \in [0,1)$. *Then the following holds:*

1. *For any* $d \times d$ *symmetric matrix* $A$, $\|A\|_{\mathrm{op}} \le \frac{1}{1-2\varepsilon} \max_{\mathbf{v} \in \mathcal{V}} |\mathbf{v}^\top A \mathbf{v}|$ *and*

$$\lambda_{\min}(A) \ge \min_{\mathbf{v} \in \mathcal{V}} \mathbf{v}^\top A \mathbf{v} - 2\varepsilon \|A\|_{\mathrm{op}}$$

2. *For any vector* $\mathbf{x} \in \mathbb{R}^d$, $\|\mathbf{x}\|_2 \le \frac{1}{1-\varepsilon} \max_{\mathbf{v} \in \mathcal{V}} |\langle \mathbf{v}, \mathbf{x}\rangle|$.

Notice that

$$\nabla^2 \widehat{L}_N(\mathbf{w}) = \frac{1}{N}\sum_{i=1}^N g'(\langle \mathbf{w}, \mathbf{x}_i\rangle)\mathbf{x}_i \mathbf{x}_i^\top \succeq \frac{1}{N}\sum_{i=1}^N \mu_g \mathbb{I}(|\langle \mathbf{w}, \mathbf{x}_i\rangle| \le B_\mu)\mathbf{x}_i \mathbf{x}_i^\top$$

$$\succeq \frac{1}{N}\sum_{i=1}^N \mu_g \left(1 - \frac{|\langle \mathbf{w}, \mathbf{x}_i\rangle|}{B_\mu}\right)_+ \mathbf{x}_i \mathbf{x}_i^\top.$$

Therefore, we can define $h(t) := (B_\mu - |t|)_+$ (which is a 1-Lipschitz function), and we have

$$\nabla^2 \widehat{L}_N(\mathbf{w}) \succeq \frac{\mu_g}{B_\mu} \underbrace{\frac{1}{N}\sum_{i=1}^N h(\langle \mathbf{w}, \mathbf{x}_i\rangle)\mathbf{x}_i \mathbf{x}_i^\top}_{=:A(\mathbf{w})}.$$

In the following, we pick a $\varepsilon_{\mathbf{v}}$-covering $\mathcal{V}$ of $\mathbb{S}^{d-1}$ such that $|\mathcal{V}| \le (3/\varepsilon_{\mathbf{v}})^d$ (we will specify $\varepsilon_{\mathbf{v}}$ later in proof). Then for any $\mathbf{w} \in \mathsf{B}_2(B_w)$,

$$\lambda_{\min}(A(\mathbf{w})) \ge \min_{\mathbf{v} \in \mathcal{V}} \mathbf{v}^\top A(\mathbf{w})\mathbf{v} - 2\varepsilon_{\mathbf{v}} \|A(\mathbf{w})\|_{\mathrm{op}}$$

By our definition of $A(\mathbf{w})$, we have (for any fixed $B_{xv}$)

$$\min_{\mathbf{v} \in \mathcal{V}} \mathbf{v}^\top A(\mathbf{w})\mathbf{v} = \min_{\mathbf{v} \in \mathcal{V}} \frac{1}{N}\sum_{i=1}^N h(\langle \mathbf{w}, \mathbf{x}_i\rangle) \langle \mathbf{v}, \mathbf{x}_i\rangle^2$$

$$\geq \min_{\mathbf{v} \in \mathcal{V}} \underbrace{\frac{1}{N} \sum_{i=1}^{N} h(\langle \mathbf{w}, \mathbf{x}_i \rangle) \min \left\{ \langle \mathbf{v}, \mathbf{x}_i \rangle^2, B_{xv}^2 \right\}}_{=:U_{\mathbf{v}}(\mathbf{w})}$$

$$\geq \min_{\mathbf{v} \in \mathcal{V}} \mathbb{E}[U_{\mathbf{v}}(\mathbf{w})] + \min_{\mathbf{v} \in \mathcal{V}} \left( U_{\mathbf{v}}(\mathbf{w}) - \mathbb{E}[U_{\mathbf{v}}(\mathbf{w})] \right).$$

By Lemma G.3, we can choose $B_{xv} = K_x(15 + \log(K_x^2/\mu_x))$, and then $\mathbb{E}[U_{\mathbf{v}}(\mathbf{w})] \geq 3B_\mu \mu_x/8$. Thus, combining the inequalities above, we can take $\varepsilon_{\mathbf{v}} = \frac{128K_x^2}{\mu_x}$ in the following, so that under event $\mathcal{E}_{w,0}$,

$$\lambda_{\min}(\nabla^2 \widehat{L}_N(\mathbf{w})) \geq \frac{\mu_g \mu_x}{8} + \frac{\mu_g}{B_\mu} \left( \frac{B_\mu \mu_x}{16} - \max_{\mathbf{v} \in \mathcal{V}} \left( \mathbb{E}[U_{\mathbf{v}}(\mathbf{w})] - U_{\mathbf{v}}(\mathbf{w}) \right) \right).$$

In the following, we consider the random process $\{\overline{U}_{\mathbf{v}}(\mathbf{w}) := U_{\mathbf{v}}(\mathbf{w}) - \mathbb{E}[U_{\mathbf{v}}(\mathbf{w})]\}_{\mathbf{w}}$, which is zero-mean and indexed by $\mathbf{w} \in \mathsf{B}_2(B_w)$. For any fixed $\mathbf{v}$, consider applying Proposition B.4 to the random process $\{\overline{U}_{\mathbf{v}}(\mathbf{w})\}_{\mathbf{w}}$. We need to verify the preconditions:

(a) With norm $\rho(\mathbf{w}, \mathbf{w}') = \|\mathbf{w} - \mathbf{w}'\|_2$, $\log \mathcal{N}(\mathsf{B}_\rho(\mathbf{w}, r), \delta) \leq d \log(2Ar/\delta)$ with constant $A = 2$;

(b) Let $f(\mathbf{x}; \mathbf{w}) := h(\langle \mathbf{w}, \mathbf{x}_i \rangle) \min \left\{ \langle \mathbf{v}, \mathbf{x}_i \rangle^2, B_{xv}^2 \right\}$, then $|f(\mathbf{x}; \mathbf{w})| \leq B_\mu B_{xv}^2$ and hence in $\mathrm{SG}(CB_\mu B_{xv}^2)$ for any random $\mathbf{x}$;

(c) For $\mathbf{w}, \mathbf{w}' \in \mathcal{W}$, we have $|h(\langle \mathbf{w}, \mathbf{x}_i \rangle) - h(\langle \mathbf{w}', \mathbf{x}_i \rangle)| \leq |\langle \mathbf{w} - \mathbf{w}', \mathbf{x}_i \rangle|$. Hence, because $\mathbf{x} \sim \mathrm{SG}(K_x)$, the random variable $h(\langle \mathbf{w}, \mathbf{x} \rangle) - h(\langle \mathbf{w}', \mathbf{x} \rangle)$ is $\mathrm{SG}(CK_x \|\mathbf{w} - \mathbf{w}'\|_2)$, and the random variable $f(\mathbf{x}; \mathbf{w}) - f(\mathbf{x}; \mathbf{w}')$ is $\mathrm{SG}(CK_x B_{xv}^2 \|\mathbf{w} - \mathbf{w}'\|_2)$.

Therefore, we can apply Proposition B.4 to obtain that with probability $1 - \delta_0$, it holds

$$\sup_{\mathbf{w}} |\overline{U}_{\mathbf{v}}(\mathbf{w})| \leq C' B_\mu B_{xv}^2 \left[ \sqrt{\frac{d \log(2\kappa_g) + \log(1/\delta_0)}{N}} \right],$$

where we denote $\kappa_g = 1 + K_x B_w / B_\mu$. Setting $\delta_0 = \delta / |\mathcal{V}|$ and taking the union bound over $\mathbf{v} \in \mathcal{V}$, we obtain that with probability at least $1 - \delta$,

$$\max_{\mathbf{v} \in \mathcal{V}} \sup_{\|\mathbf{w}\|_2 \leq B_w} |\overline{U}_{\mathbf{v}}(\mathbf{w})| \leq C' B_\mu B_{xv}^2 \left[ \sqrt{\frac{d \log(8\kappa_g/\varepsilon_{\mathbf{v}}) + \log(1/\delta)}{N}} \right],$$

where we use $\log |\mathcal{V}| \leq d \log(4/\varepsilon_{\mathbf{v}})$. Therefore, we plug in the definition of $\varepsilon_{\mathbf{v}}$ and $B_{xv}$ to deduce that, if we set

$$C_1 = \left( \frac{16C' B_{xv}^2}{\mu_x} \right)^2 \log(8\kappa_g/\varepsilon_{\mathbf{v}}), \qquad \varepsilon_{\mathbf{v}} = \frac{128K_x^2}{\mu_x}, \qquad B_{xv} = K_x(15 + \log(K_x^2/\mu_x)),$$

then as long as $N \geq C_1 \cdot d$, it holds $\max_{\mathbf{v} \in \mathcal{V}} \mathbb{E}[U_{\mathbf{v}}(\mathbf{w})] - U_{\mathbf{v}}(\mathbf{w}) \leq \frac{\mu_x B_\mu}{16}$ with probability at least $1 - \exp(-N/C_1)$. This is the desired result. $\qquad \square$

**Lemma G.3.** *Under Assumption B, for $B_{xv} = K_x(15 + \log(K_x^2/\mu_x))$, it holds*

$$\inf_{\mathbf{w} \in \mathsf{B}_2(B_w), \mathbf{v} \in \mathbb{S}^{d-1}} \mathbb{E}[\mathbf{1}\{|\mathbf{x}^\top \mathbf{w}| \leq B_\mu/2\}(\mathbf{x}^\top \mathbf{v})^2 \mathbf{1}\{|\mathbf{x}^\top \mathbf{v}| \leq B_{xv}\}] \geq 3\mu_x/4.$$

*Proof.* For any fixed $\mathbf{w} \in \mathsf{B}_2(B_w), \mathbf{v} \in \mathbb{S}^{d-1}$,

$$\mathbb{E}[\mathbf{1}\{|\mathbf{x}^\top \mathbf{w}| \leq B_\mu/2\}(\mathbf{x}^\top \mathbf{v})^2 \mathbf{1}\{|\mathbf{x}^\top \mathbf{v}| \leq B_{xv}\}]$$
$$= \mathbb{E}[\mathbf{1}\{|\mathbf{x}^\top \mathbf{w}| \leq B_\mu/2\}(\mathbf{x}^\top \mathbf{v})^2\}] - \mathbb{E}[\mathbf{1}\{|\mathbf{x}^\top \mathbf{w}| \leq B_\mu/2\}(\mathbf{x}^\top \mathbf{v})^2 \mathbf{1}\{|\mathbf{x}^\top \mathbf{v}| > B_{xv}\}]$$
$$\geq \mu_x - \mathbb{E}[(\mathbf{x}^\top \mathbf{v})^2 \mathbf{1}\{|\mathbf{x}^\top \mathbf{v}| > B_{xv}\}].$$

Because $\mathbf{x} \sim \mathrm{SG}(K_x)$, $\mathbf{x}^\top \mathbf{v} \sim \mathrm{SG}(K_x)$, and a simple calculation yields

$$\mathbb{E}[(\mathbf{x}^\top \mathbf{v})^2 \mathbf{1}\{|\mathbf{x}^\top \mathbf{v}| > tK_x\}] \leq 2K_x^2(t^2 + 1) \exp(-t^2).$$

Taking $t = 15 + \log(K_x^2/\mu_x)$ gives $\mathbb{E}[(\mathbf{x}^\top \mathbf{v})^2 \mathbf{1}\{|\mathbf{x}^\top \mathbf{v}| > B_{xv}\}] \leq \mu_x/4$, which completes the proof. $\qquad \square$

### G.4 Proof of Theorem G.3 (b)

Notice that

$$\nabla\widehat{L}_N(\mathbf{w}) = \frac{1}{N}\sum_{i=1}^{N}\left(g(\langle\mathbf{w},\mathbf{x}_i\rangle) - y_i\right)\mathbf{x}_i.$$

In the following, we pick a minimal $1/2$-covering of $\mathbb{S}^{d-1}$ (so $|\mathcal{V}| \leq 5^d$). Then by Lemma G.2, it holds

$$\left\|\nabla\widehat{L}_N(\mathbf{w}) - \mathbb{E}[\nabla\widehat{L}_N(\mathbf{w})]\right\|_2 \leq 2\max_{\mathbf{v}\in\mathcal{V}}\bigg|\underbrace{\langle\nabla\widehat{L}_N(\mathbf{w}),\mathbf{v}\rangle - \mathbb{E}[\langle\nabla\widehat{L}_N(\mathbf{w}),\mathbf{v}\rangle]}_{=:X_{\mathbf{v}}(\mathbf{w})}\bigg|$$

Fix a $\mathbf{v}\in\mathbb{S}^{d-1}$ and set $\delta' = \delta/|\mathcal{V}|$. We proceed to bound $\sup_{\mathbf{w}}|X_{\mathbf{v}}(\mathbf{w})|$ by applying Proposition B.4 to the random process $\{X_{\mathbf{v}}(\mathbf{w})\}_{\mathbf{w}}$. We need to verify the preconditions:

(a) With norm $\rho(\mathbf{w},\mathbf{w}') = \|\mathbf{w}-\mathbf{w}'\|_2$, $\log N(\delta;\mathsf{B}_\rho(r),\rho) \leq d\log(2Ar/\delta)$ with constant $A = 2$;

(b) For $\mathbf{z} = [\mathbf{x}; y]$, we let $f(\mathbf{z};\mathbf{w}) := (g(\langle\mathbf{w},\mathbf{x}\rangle) - y)\langle\mathbf{x},\mathbf{v}\rangle$, then $f(\mathbf{z};\mathbf{w}) \sim \text{SE}(CK_xK_y)$ for any $\mathbf{w}$ by our assumption on $(\mathbf{x}, y)$;

(c) For $\mathbf{w},\mathbf{w}'\in\mathcal{W}$, we have $|g(\langle\mathbf{w},\mathbf{x}\rangle) - g(\langle\mathbf{w}',\mathbf{x}\rangle)| \leq L_g|\langle\mathbf{w}-\mathbf{w}',\mathbf{x}\rangle|$. Hence, because $\mathbf{x}\sim\text{SG}(K_x)$, the random variable $g(\langle\mathbf{w},\mathbf{x}_i\rangle) - g(\langle\mathbf{w}',\mathbf{x}_i\rangle)$ is sub-Gaussian in $\text{SG}(K_xL_g\|\mathbf{w}-\mathbf{w}'\|_2)$. Thus, $f(\mathbf{z};\mathbf{w}) - f(\mathbf{z};\mathbf{w}')$ is sub-exponential in $\text{SE}(CK_x^2L_g\|\mathbf{w}-\mathbf{w}'\|_2)$.

Therefore, we can apply Proposition B.4 to obtain that with probability $1 - \delta_0$, it holds

$$\sup_{\mathbf{w}}|X_{\mathbf{v}}(\mathbf{w})| \leq C'K_xK_y\left[\sqrt{\frac{d\log(2\kappa_y)+\log(1/\delta_0)}{N}} + \frac{d\log(2\kappa_y)+\log(1/\delta_0)}{N}\right],$$

where we denote $\kappa_y = 1 + L_gK_x^2B_w/K_y$. Setting $\delta_0 = \delta/|\mathcal{V}|$ and taking the union bound over $\mathbf{v}\in\mathcal{V}$, we obtain that with probability at least $1 - \delta$,

$$\max_{\mathbf{v}\in\mathcal{V}}\sup_{\|\mathbf{w}\|_2\leq B_w}|X_{\mathbf{v}}(\mathbf{w})| \leq C'K_xK_y\left[\sqrt{\frac{d\log(10\kappa_y)+\log(1/\delta)}{N}} + \frac{d\log(10\kappa_y)+\log(1/\delta)}{N}\right].$$

This is the desired result. $\qquad\square$

### G.5 Proof of Theorem G.3 (c)

In the following, we condition on (a) holds, i.e. $\widehat{L}_N$ is $\alpha$-strongly-convex and $\beta$ smooth over $\mathsf{B}_2(B_w)$ with $\alpha = \mu_x\mu_g/8$ and $\beta = 8L_gK_x^2$. We define

$$\widetilde{\mathbf{w}} = \underset{\mathbf{w}\in\mathsf{B}_2(B_w)}{\arg\min}\ \widehat{L}_N(\mathbf{w}).$$

Then by standard convex analysis, we have

$$\alpha\|\widetilde{\mathbf{w}}-\boldsymbol{\beta}^\star\|_2^2 \leq \left\langle\nabla\widehat{L}_N(\widetilde{\mathbf{w}}) - \nabla\widehat{L}_N(\boldsymbol{\beta}^\star),\widetilde{\mathbf{w}}-\boldsymbol{\beta}^\star\right\rangle \leq \left\langle-\nabla\widehat{L}_N(\boldsymbol{\beta}^\star),\widetilde{\mathbf{w}}-\boldsymbol{\beta}^\star\right\rangle \leq \left\|\nabla\widehat{L}_N(\boldsymbol{\beta}^\star)\right\|_2\|\widetilde{\mathbf{w}}-\boldsymbol{\beta}^\star\|_2.$$

Notice that $\left\|\nabla\widehat{L}_N(\boldsymbol{\beta}^\star)\right\|_2 \leq \varepsilon_{\text{stat}}$, we can conclude that

$$\|\widetilde{\mathbf{w}}\|_2 \leq \|\boldsymbol{\beta}^\star\|_2 + \frac{\varepsilon_{\text{stat}}}{\alpha}.$$

Recall that we assume $\|\boldsymbol{\beta}^\star\|_2 \leq B_w/4$, we can then consider $\mathcal{E}_s := \{\varepsilon_{\text{stat}} < \alpha B_w/4\}$. Once $\mathcal{E}_s$ holds, our argument above yields $\|\widetilde{\mathbf{w}}\|_2 < B_w$, which implies $\nabla\widehat{L}_N(\widetilde{\mathbf{w}}) = 0$. Therefore, $\widetilde{\mathbf{w}} = \arg\min_{\mathbf{w}\in\mathbb{R}^d}\widehat{L}_N(\mathbf{w})$. Further, by Theorem G.3, we can set

$$C_2 := \max\left\{2\iota\left(\frac{32\alpha K_xK_y}{B_w}\right)^2, 2\iota\cdot\frac{32\alpha K_xK_y}{B_w}\right\},$$

so that as long as $N \geq C_2d$, the event $\mathcal{E}_s$ holds with probability at least $1 - \exp(-N/C_2)$. This is the desired result. $\qquad\square$

### G.6 Proof of Theorem G.3 (d) & (e)

We first prove Theorem G.3 (d). Notice that

$$\nabla^2 L_p(\mathbf{w}) = \mathbb{E}\big[g'(\langle \mathbf{x}, \mathbf{w}\rangle)\mathbf{x}\mathbf{x}^\top\big] \succeq \mathbb{E}\big[\mu_g \mathbb{I}(|\langle \mathbf{x}, \mathbf{w}\rangle| \leq B_\mu)\mathbf{x}\mathbf{x}^\top\big] \succeq \mu_g \mu_x \mathbf{I}_d, \forall \mathbf{w} \in \mathsf{B}_2(B_w).$$

Therefore, $L_p$ is $(\mu_g \mu_x)$-strongly-convex over $\mathsf{B}_2(B_w)$. Therefore, because $\boldsymbol{\beta}^\star \in \mathsf{B}_2(B_w)$ is the global minimum of $L_p$, it holds that for all $\mathbf{w} \in \mathsf{B}_2(B_w)$,

$$L_p(\mathbf{w}) - L_p(\boldsymbol{\beta}^\star) \leq \frac{1}{2\mu_g\mu_x} \|\nabla L_p(\mathbf{w})\|_2^2.$$

By the definition of $\varepsilon_{\text{stat}}$, $\|\nabla L_p(\mathbf{w})\|_2 \leq \varepsilon_{\text{stat}} + \|\nabla \widehat{L}_N(\mathbf{w})\|_2$, and hence the proof of Theorem G.3 (d) is completed.

We next prove Theorem G.3 (e), where we assume that $\mathbb{E}[y|\mathbf{x}] = g(\langle \mathbf{x}, \boldsymbol{\beta}\rangle)$ (which implies $\boldsymbol{\beta}^\star = \boldsymbol{\beta}$ directly) and $\mathbf{w}_{\text{GLM}} \in \mathsf{B}_2(B_w)$. Notice that

$$\nabla L_p(\mathbf{w}) = \mathbb{E}\Big[\nabla \widehat{L}_N(\mathbf{w})\Big] = \mathbb{E}[(g(\langle \mathbf{x}, \mathbf{w}\rangle) - y)\mathbf{x}] = \mathbb{E}[(g(\langle \mathbf{x}, \mathbf{w}\rangle) - g(\langle \mathbf{w}, \boldsymbol{\beta}\rangle))\mathbf{x}],$$

and hence

$$\langle \nabla L_p(\mathbf{w}_{\text{GLM}}), \mathbf{w}_{\text{GLM}} - \boldsymbol{\beta}\rangle = \mathbb{E}[(g(\langle \mathbf{x}, \mathbf{w}_{\text{GLM}}\rangle) - g(\langle \mathbf{w}, \boldsymbol{\beta}\rangle)) \cdot (\langle \mathbf{x}, \mathbf{w}_{\text{GLM}}\rangle - \langle \mathbf{w}, \boldsymbol{\beta}\rangle)]$$
$$\geq \frac{1}{L_g}\mathbb{E}\big[(g(\langle \mathbf{x}, \mathbf{w}_{\text{GLM}}\rangle) - g(\langle \mathbf{w}, \boldsymbol{\beta}\rangle))^2\big].$$

On the other hand, by the $(\mu_g \mu_x)$-strong-convexity of $L_p$ over $\mathsf{B}_2(B_w)$, it holds that

$$\langle \nabla L_p(\mathbf{w}_{\text{GLM}}), \mathbf{w}_{\text{GLM}} - \boldsymbol{\beta}\rangle \leq \frac{1}{\mu_g\mu_x} \|\nabla L_p(\mathbf{w}_{\text{GLM}})\|_2^2.$$

Finally, using the definition of $\mathbf{w}_{\text{GLM}}$, we have $\nabla \widehat{L}_N(\mathbf{w}_{\text{GLM}}) = 0$, and hence $\|\nabla L_p(\mathbf{w}_{\text{GLM}})\|_2 \leq \varepsilon_{\text{stat}}$, which completes the proof of Theorem G.3 (e). $\qquad\square$

## H Proofs for Section 3.2

### H.1 Proof of Theorem 7

Fix $\lambda_N \geq 0$, $\beta > 0$ and $B_w > 0$, and consider any in-context data $\mathcal{D}$ such that the precondition of Theorem 7 holds. Recall that

$$L_{\text{lasso}}(\mathbf{w}) := \frac{1}{2N} \sum_{i=1}^{N} (\langle \mathbf{w}, \mathbf{x}_i\rangle - y_i)^2 + \lambda_N \|\mathbf{w}\|_1$$

denotes the lasso regression loss in (ICLasso), so that $\mathbf{w}_{\text{lasso}} = \arg\min_{\mathbf{w} \in \mathbb{R}^d} L_{\text{lasso}}(\mathbf{w})$. We further write

$$\widehat{L}_N^0(\mathbf{w}) := \frac{1}{2N} \sum_{i=1}^{N} (\langle \mathbf{w}, \mathbf{x}_i\rangle - y_i)^2, \qquad \mathcal{R}(\mathbf{w}) := \lambda_N \|\mathbf{w}\|_1.$$

Note that $\nabla^2 \widehat{L}_N^0(\mathbf{w}) = \mathbf{X}^\top \mathbf{X}/N$ and thus $\widehat{L}_N^0$ is $\beta$-smooth over $\mathbb{R}^d$.

Consider the proximal gradient descent algorithm on the ridge loss

$$\mathbf{w}_{\text{PGD}}^{t+1} = \mathbf{prox}_{\eta\mathcal{R}}\Big(\mathbf{w}_{\text{PGD}}^t - \eta \nabla \widehat{L}_N^0(\mathbf{w}_{\text{PGD}}^t)\Big)$$

with initialization $\mathbf{w}_{\text{PGD}}^0 := \mathbf{0}_d$, learning rate $\eta := \beta^{-1}$, and number of steps $T$ to be specified later. Similar to the proof of Theorem 4, we can construct a transformer to approximate $\mathbf{w}_{\text{GD}}^T$. Consider $\ell(s,t) = \frac{1}{2}(s-t)^2$ and $\mathcal{R}(\mathbf{w}) = \lambda_N \|\mathbf{w}\|_1$, then $\partial_s \ell(s,t)$ is $(0, +\infty, 2, 4)$-approximable by sum of relus (cf. Definition D.1), and $\mathbf{prox}_{\eta\mathcal{R}}$ is $(0, +\infty, 4d, 4 + 2\eta\lambda_N)$-approximable by sum of relus (Proposition D.1). Therefore, we can apply Theorem D.2 with the square loss $\ell$, regularizer $\mathcal{R}$, learning rate $\eta$ and accuracy parameter $0$ to obtain that there exists a transformer $\text{TF}_{\boldsymbol{\theta}}$ with $(T + 1)$

layers, number of heads $M^{(\ell)} = 2$ for all $\ell \in [L]$, and hidden dimension $D' = 2d$, such that the final output $\mathbf{h}_{N+1}^{(L)} = [\mathbf{x}_{N+1}; \widehat{y}_{N+1}; \mathbf{w}_{\mathrm{PGD}}^T; *]$ with $\widehat{y}_{N+1} = \langle \mathbf{w}_{\mathrm{PGD}}^T, \mathbf{x}_{N+1} \rangle$. Further, the weight matrices have norm bounds $\|\boldsymbol{\theta}\| \le 10R + (8 + 2\lambda_N)\beta^{-1}$.

By the standard convergence result for proximal gradient descent (Proposition B.3), we have for all $t \ge 1$ that

$$L_{\mathrm{lasso}}(\mathbf{w}_{\mathrm{PGD}}^t) - L_{\mathrm{lasso}}(\mathbf{w}_{\mathrm{lasso}}) \le \frac{\beta}{2t} \|\mathbf{w}_{\mathrm{lasso}}\|_2^2.$$

Plugging in $\|\mathbf{w}_{\mathrm{lasso}}\|_2 \le B_w/2$ and $T = L - 1 = \lceil \beta B_w^2/\varepsilon \rceil$ finishes the proof. $\qquad\square$

## H.2 Sharper convergence analysis of proximal gradient descent for Lasso

**Collection of parameters** Throughout the rest of this section, we consider fixed $N \ge 1$, $\lambda_N = \sqrt{\frac{\rho\nu\log d}{N}}$ for $\rho \ge 0$, $\nu \ge 0$ fixed (and to be determined), fixed $0 < \alpha \le \beta$, and fixed $B_w^\star > 0$. We write $\kappa := \beta/\alpha$, $\kappa_s := \beta(B_w^\star)^2/\nu^2$, and $\omega_N := \frac{\rho}{\alpha}\frac{s\log d}{N}$.

Here we present a sharper convergence analysis on the proximal gradient descent algorithm for $L_{\mathrm{lasso}}$ under the following well-conditionedness assumption, which will be useful for proving Theorem 8 in the sequel.

**Assumption C** (Well-conditioned property for Lasso). *We say the (ICLasso) problem is well-conditioned with sparsity $s$ if the following conditions hold:*

1. *The $(\alpha, \rho)$-RSC condition holds:*

$$\frac{\|\mathbf{X}\mathbf{w}\|_2^2}{N} \ge \alpha \|\mathbf{w}\|_2^2 - \rho \frac{\log d}{N} \|\mathbf{w}\|_1^2, \qquad \forall \mathbf{w} \in \mathbb{R}^d. \tag{38}$$

   *Further, $\lambda_{\max}(\mathbf{X}^\top \mathbf{X}/N) \le \beta$.*

2. *The data $(\mathbf{X}, \mathbf{y})$ is "approximately generated from a $s$-sparse linear model": There exists a $\mathbf{w}_\star \in \mathbb{R}^d$ such that $\|\mathbf{w}_\star\|_2 \le B_w^\star$, $\|\mathbf{w}_\star\|_0 \le s$ and for the residue $\boldsymbol{\varepsilon} = \mathbf{y} - \mathbf{X}\mathbf{w}_\star$,*

$$\|\mathbf{X}^\top \boldsymbol{\varepsilon}\|_\infty \le \frac{1}{2}N\lambda_N.$$

3. *It holds that $N \ge 32\frac{\rho}{\alpha} \cdot s\log d$ (i.e. $32\omega_N \le 1$).*

Assumption C1 imposes the standard restricted strong convexity (RSC) condition for the feature matrix $\mathbf{X} \in \mathbb{R}^{N \times d}$, and Assumption C2 asserts that the data is approximately generated from a sparse linear model, with a bound on the $L_\infty$ norm of the error vector $\mathbf{X}^\top \boldsymbol{\varepsilon}$. Assumption C is entirely deterministic in nature, and suffices to imply the following convergence result. In the proof of Theorem 8, we show that Assumption C is satisfied with high probability when data is generated from the standard sparse linear model considered therein.

**Theorem H.1** (Sharper convergence guarantee for Lasso). *Under Assumption C, for the PGD iterates $\{\mathbf{w}^t\}_{t \ge 0}$ on loss function $\widehat{L}_{\mathrm{lasso}}$ with stepsize $\eta = 1/\beta$ and starting point $\mathbf{w}^0 = \mathbf{0}$, we have $\widehat{L}_{\mathrm{lasso}}(\mathbf{w}^T) - \widehat{L}_{\mathrm{lasso}}(\mathbf{w}_{\mathrm{lasso}}) \le \varepsilon$ for all*

$$T \ge C\left[\frac{\beta(B_w^\star)^2}{\nu} + \kappa \log\left(C \cdot \kappa \cdot \frac{\beta(B_w^\star)^2}{\nu} \cdot \frac{\nu}{\varepsilon}\right) + \kappa \frac{\nu\omega_N^2}{\varepsilon}\right],$$

*where $C$ is a universal constant.*

The proof can be found in Appendix H.4. Combining Theorem H.1 with the construction in Theorem 7, we directly obtain the following result as a corollary.

**Theorem H.2** (In-context Lasso with transformers with sharper convergence). *For any $N, d, s \ge 1$, $0 < \alpha \le \beta$, $\nu \ge 0$, $\rho \ge 0$, there exists a $L$-layer transformer $\mathrm{TF}_{\boldsymbol{\theta}}$ with*

$$L = \lceil C(\kappa_s + \kappa(\log(C\kappa_s/\varepsilon) + \nu\omega_N^2/\varepsilon)) \rceil, \quad \max_{\ell \in [L]} M^{(\ell)} \le 2, \quad \max_{\ell \in [L]} D^{(\ell)} \le 2d,$$

$$\|\boldsymbol{\theta}\| \le 3 + R + (8 + 2\lambda_N)\beta^{-1},$$

such that the following holds. On any input data $(\mathcal{D}, \mathbf{x}_{N+1})$ such that the (ICLasso) problem satisfies Assumption C (which implies $\|\mathbf{w}_{\mathrm{lasso}}\|_2 \le B_w/2$ with $B_w = 2B_w^\star + \sqrt{\nu/\alpha}$), $\mathrm{TF}_{\boldsymbol{\theta}}(\mathbf{H}^{(0)})$ approximately implements (ICLasso), in that it outputs $\widehat{y}_{N+1} = \mathrm{read}_{\mathsf{y}}(\mathrm{TF}_{\boldsymbol{\theta}}(\mathbf{H})) = \langle \mathbf{x}_{N+1}, \widehat{\mathbf{w}} \rangle$ with

$$\widehat{L}_{\mathrm{lasso}}(\widehat{\mathbf{w}}) - \widehat{L}_{\mathrm{lasso}}(\mathbf{w}_{\mathrm{lasso}}) \le \varepsilon.$$

### H.3 Basic properties for Lasso

**Lemma H.1** (Relaxed basic inequality). *Suppose that Assumption C2 holds. Then it holds that*

$$\|\mathbf{w} - \mathbf{w}_\star\|_1 \le 4\sqrt{s}\,\|\mathbf{w} - \mathbf{w}_\star\|_2 + \frac{2}{\lambda_N}\Big(\widehat{L}_{\mathrm{lasso}}(\mathbf{w}) - \widehat{L}_{\mathrm{lasso}}(\mathbf{w}_\star)\Big), \qquad \forall \mathbf{w} \in \mathbb{R}^d.$$

*As a corollary,* $\|\mathbf{w}_{\mathrm{lasso}} - \mathbf{w}_\star\|_1 \le 4\sqrt{s}\,\|\mathbf{w}_{\mathrm{lasso}} - \mathbf{w}_\star\|_2$.

*Proof.* Let us first fix any $\mathbf{w} \in \mathbb{R}^d$. Denote $\boldsymbol{\Delta} = \mathbf{w} - \mathbf{w}_\star$, and let $S = \mathrm{supp}(\mathbf{w}_\star)$ be the set of indexes of nonzero entries of $\mathbf{w}_\star$. Then by definition, $\mathbf{y} = \mathbf{X}\mathbf{w}_\star + \boldsymbol{\varepsilon}$ and $|S| \le s$, and hence

$$\|\mathbf{X}\mathbf{w} - \mathbf{y}\|_2^2 - \|\mathbf{X}\mathbf{w}_\star - \mathbf{y}\|_2^2 = \|\mathbf{X}\boldsymbol{\Delta} - \boldsymbol{\varepsilon}\|_2^2 - \|\boldsymbol{\varepsilon}\|_2^2 = \|\mathbf{X}\boldsymbol{\Delta}\|_2^2 - 2\boldsymbol{\varepsilon}^\top \mathbf{X}\boldsymbol{\Delta},$$

$$\|\mathbf{w}\|_1 - \|\mathbf{w}_\star\|_1 = \sum_{j \in S}(|\mathbf{w}[j]| - |\mathbf{w}_\star[j]|) + \sum_{j \notin S}|\mathbf{w}[j]|$$

$$\ge -\sum_{j \in S}|\mathbf{w}[j] - \mathbf{w}_\star[j]| + \sum_{j \notin S}|\mathbf{w}[j]| = \|\boldsymbol{\Delta}_{S^c}\|_1 - \|\boldsymbol{\Delta}_S\|_1.$$

Combining these inequalities, we obtain

$$0 \le \frac{1}{2N}\|\mathbf{X}\boldsymbol{\Delta}\|_2^2 \le \frac{\boldsymbol{\varepsilon}^\top \mathbf{X}\boldsymbol{\Delta}}{N} + \lambda_N(\|\boldsymbol{\Delta}_S\|_1 - \|\boldsymbol{\Delta}_{S^c}\|_1) + \widehat{L}_{\mathrm{lasso}}(\mathbf{w}) - \widehat{L}_{\mathrm{lasso}}(\mathbf{w}_\star)$$

$$\le \frac{\lambda_N}{2}\|\boldsymbol{\Delta}\|_1 + \lambda_N(\|\boldsymbol{\Delta}_S\|_1 - \|\boldsymbol{\Delta}_{S^c}\|_1) + \widehat{L}_{\mathrm{lasso}}(\mathbf{w}) - \widehat{L}_{\mathrm{lasso}}(\mathbf{w}_\star) \qquad (39)$$

$$= \frac{\lambda_N}{2}(3\|\boldsymbol{\Delta}_S\|_1 - \|\boldsymbol{\Delta}_{S^c}\|_1) + \widehat{L}_{\mathrm{lasso}}(\mathbf{w}) - \widehat{L}_{\mathrm{lasso}}(\mathbf{w}_\star),$$

where the second inequality follows from $\frac{\boldsymbol{\varepsilon}^\top \mathbf{X}\boldsymbol{\Delta}}{N} \le \frac{\|\mathbf{X}^\top \boldsymbol{\varepsilon}\|_\infty}{N}\|\boldsymbol{\Delta}\|_1$ and our assumption that $2\frac{\|\mathbf{X}^\top \boldsymbol{\varepsilon}\|_\infty}{N} \le \lambda_N$, and the last inequality is due to $\|\boldsymbol{\Delta}\|_1 = \|\boldsymbol{\Delta}_S\|_1 + \|\boldsymbol{\Delta}_{S^c}\|_1$. Therefore, we have

$$\|\boldsymbol{\Delta}\|_1 = \|\boldsymbol{\Delta}_S\|_1 + \|\boldsymbol{\Delta}_{S^c}\|_1 \le 4\|\boldsymbol{\Delta}_S\|_1 + \frac{2}{\lambda_N}\Big(\widehat{L}_{\mathrm{lasso}}(\mathbf{w}) - \widehat{L}_{\mathrm{lasso}}(\mathbf{w}_\star)\Big)$$

$$\le 4\sqrt{s}\,\|\boldsymbol{\Delta}\|_2 + \frac{2}{\lambda_N}\Big(\widehat{L}_{\mathrm{lasso}}(\mathbf{w}) - \widehat{L}_{\mathrm{lasso}}(\mathbf{w}_\star)\Big),$$

where the last inequality follows from $\|\boldsymbol{\Delta}_S\|_1 \le \sqrt{s}\,\|\boldsymbol{\Delta}_S\|_2 \le \sqrt{s}\,\|\boldsymbol{\Delta}\|_2$. This completes the proof of our main inequality. As for the corollary, we only need to use the definition that $\widehat{L}_{\mathrm{lasso}}(\mathbf{w}_{\mathrm{lasso}}) \le \widehat{L}_{\mathrm{lasso}}(\mathbf{w}_\star)$. $\qquad \square$

**Proposition H.1** (Gap to parameter estimation error). *Suppose that Assumption C holds. Then for all* $\mathbf{w} \in \mathbb{R}^d$,

$$\|\mathbf{w} - \mathbf{w}_\star\|_2^2 \le C\left[\frac{s\lambda_N^2}{\alpha^2} + \nu^{-1}\mathsf{gap}^2 + \mathsf{gap}\right],$$

*where we write* $\mathsf{gap} := \widehat{L}_{\mathrm{lasso}}(\mathbf{w}) - \widehat{L}_{\mathrm{lasso}}(\mathbf{w}_{\mathrm{lasso}})$, *and* $C = 120$ *is a universal constant. In particular, we have* $\|\mathbf{w}_{\mathrm{lasso}} - \mathbf{w}_\star\|_2^2 \le 10\frac{\rho\nu}{\alpha^2}\frac{s\log d}{N}$.

*Proof.* We follow the notation in the proof of Lemma H.1. By (39), we have

$$0 \leq \frac{1}{2N} \|\mathbf{X}\mathbf{\Delta}\|_2^2 \leq \frac{\lambda_N}{2} (3 \|\mathbf{\Delta}_S\|_1 - \|\mathbf{\Delta}_{S^c}\|_1) + \widehat{L}_{\text{lasso}}(\mathbf{w}) - \widehat{L}_{\text{lasso}}(\mathbf{w}_\star),$$

and hence $\|\mathbf{\Delta}\|_1 \leq 4\sqrt{s} \|\mathbf{\Delta}\|_2 + \frac{2\text{gap}}{\lambda_N}$ due to $\widehat{L}_{\text{lasso}}(\mathbf{w}) - \widehat{L}_{\text{lasso}}(\mathbf{w}_\star) \leq \text{gap}$. On the other hand, by the RSC condition (38), it holds that

$$\frac{\|\mathbf{X}\mathbf{\Delta}\|_2^2}{N} \geq \alpha \|\mathbf{\Delta}\|_2^2 - \rho \frac{\log d}{N} \|\mathbf{\Delta}\|_1^2.$$

Therefore, we have

$$\alpha \|\mathbf{\Delta}\|_2^2 \leq 3\lambda_N \sqrt{s} \|\mathbf{\Delta}\|_2 + \rho \frac{\log d}{N} \|\mathbf{\Delta}\|_1^2 + 2\text{gap}$$

$$\leq 3\lambda_N \sqrt{s} \|\mathbf{\Delta}\|_2 + \rho \frac{\log d}{N} \left( 4\sqrt{s} \|\mathbf{\Delta}\|_2 + \frac{2\text{gap}}{\lambda_N} \right)^2 + 2\text{gap}$$

$$\leq \frac{5s\lambda_N^2}{\alpha} + \frac{\alpha}{6} \|\mathbf{\Delta}\|_2^2 + \rho \frac{20s \log d}{\lambda_N^2 N} \|\mathbf{\Delta}\|_2^2 + \rho \frac{20 \log d}{N} \text{gap}^2 + 2\text{gap},$$

where the last inequality uses AM-GM inequality and Cauchy inequality. Notice that $\rho \frac{20s \log d}{N} \leq \frac{2}{3}\alpha$, we now derive that

$$\|\mathbf{\Delta}\|_2^2 \leq \frac{30s\lambda_N^2}{\alpha^2} + \rho \frac{120 \log d}{\lambda_N^2 N} \text{gap}^2 + 12\text{gap}.$$

Plugging in $\lambda_N = \sqrt{\frac{\rho\nu \log d}{N}}$ completes the proof. The corollary follows immediately by letting $\mathbf{w} = \mathbf{w}_{\text{lasso}}$ in above proof (hence $\text{gap} = 0$). $\qquad\square$

**Lemma H.2** (Growth). *It holds that*

$$\frac{1}{2N} \|\mathbf{X}(\mathbf{w} - \mathbf{w}_{\text{lasso}})\|_2^2 \leq \widehat{L}_{\text{lasso}}(\mathbf{w}) - \widehat{L}_{\text{lasso}}(\mathbf{w}_{\text{lasso}}), \qquad \forall \mathbf{w}.$$

*Proof.* For simplicity we denote $\mathbf{w}_{\text{lasso}} := \mathbf{w}_{\text{lasso}}$. By the first order optimality condition, it holds that

$$0 \in \frac{1}{N}\mathbf{X}^\top(\mathbf{X}\mathbf{w}_{\text{lasso}} - \mathbf{y}) + \partial R(\mathbf{w}_{\text{lasso}}),$$

where we write $R(\mathbf{w}) := \lambda_N \|\mathbf{w}\|_1$. Then by the convexity of $R$, we have

$$R(\mathbf{w}) - R(\mathbf{w}_{\text{lasso}}) \geq \langle \partial R(\mathbf{w}_{\text{lasso}}), \mathbf{w} - \mathbf{w}_{\text{lasso}} \rangle = \left\langle -\frac{1}{N}\mathbf{X}^\top(\mathbf{X}\mathbf{w}_{\text{lasso}} - \mathbf{y}), \mathbf{w} - \mathbf{w}_{\text{lasso}} \right\rangle$$

$$= -\frac{1}{N} \langle \mathbf{X}\mathbf{w}_{\text{lasso}} - \mathbf{y}, (\mathbf{X}\mathbf{w} - \mathbf{y}) - (\mathbf{X}\mathbf{w}_{\text{lasso}} - \mathbf{y}) \rangle$$

$$= -\frac{1}{2N} \|\mathbf{X}\mathbf{w} - \mathbf{y}\|_2^2 + \frac{1}{2N} \|\mathbf{X}\mathbf{w}_{\text{lasso}} - \mathbf{y}\|_2^2 + \frac{1}{2N} \|\mathbf{X}(\mathbf{w} - \mathbf{w}_{\text{lasso}})\|_2^2.$$

Rearranging completes the proof. $\qquad\square$

## H.4 Proof of Theorem H.1

For the simplicity of presentation, we write $\mathbf{w}_{\text{lasso}} = \mathbf{w}_{\text{lasso}}$ and we denote $\text{gap}^t := \widehat{L}_{\text{lasso}}(\mathbf{w}^t) - \widehat{L}_{\text{lasso}}(\mathbf{w}_{\text{lasso}})$.

By Lemma H.1, we have $\|\mathbf{w}^t - \mathbf{w}_\star\|_1 \leq 4\sqrt{s} \|\mathbf{w}^t - \mathbf{w}_\star\|_2 + \frac{2\text{gap}^t}{\lambda_N}$, which implies

$$\left\| \mathbf{w}^t - \mathbf{w}_{\text{lasso}} \right\|_1 \leq \left\| \mathbf{w}^t - \mathbf{w}_\star \right\|_1 + \left\| \mathbf{w}_{\text{lasso}} - \mathbf{w}_\star \right\|_1 \leq 4\sqrt{s} \left\| \mathbf{w}^t - \mathbf{w}_{\text{lasso}} \right\|_2 + 8\sqrt{s} \left\| \mathbf{w}_{\text{lasso}} - \mathbf{w}_\star \right\|_2 + \frac{2\text{gap}^t}{\lambda_N}.$$

We denote $\mu_N = \rho^2 \frac{\log d}{N}$. Using the assumption that $\mathbf{X}$ is $(\alpha, \rho)$-RSC, we obtain that

$$\frac{1}{N} \left\| X(\mathbf{w}^t - \mathbf{w}_{\text{lasso}}) \right\|_2^2 \geq \alpha \left\| \mathbf{w}^t - \mathbf{w}_{\text{lasso}} \right\|_2^2 - \mu_N \left\| \mathbf{w}^t - \mathbf{w}_{\text{lasso}} \right\|_1^2$$

$$\geq \alpha \left\| \mathbf{w}^t - \mathbf{w}_{\text{lasso}} \right\|_2^2 - \mu_N \left( 20s \left\| \mathbf{w}^t - \mathbf{w}_{\text{lasso}} \right\|_2^2 + 640s \left\| \mathbf{w}_{\text{lasso}} - \mathbf{w}_\star \right\|_2^2 + \frac{40}{\lambda_N^2} (\mathsf{gap}^t)^2 \right).$$

Thus, as long as $N \geq \frac{30\rho^2 s \log d}{\alpha}$, we have

$$\frac{\alpha}{3} \left\| \mathbf{w}^t - \mathbf{w}_{\text{lasso}} \right\|_2^2 \leq \frac{1}{N} \left\| \mathbf{X}(\mathbf{w}^t - \mathbf{w}_{\text{lasso}}) \right\|_2^2 + 640s\mu_N \left\| \mathbf{w}_{\text{lasso}} - \mathbf{w}_\star \right\|_2^2 + \frac{40\mu_N}{\lambda_N^2} (\mathsf{gap}^t)^2$$

$$\leq 2\mathsf{gap}^t + 40\nu^{-1}(\mathsf{gap}^t)^2 + 640s\mu_N \left\| \mathbf{w}_{\text{lasso}} - \mathbf{w}_\star \right\|_2^2,$$

where the last inequality follows from Lemma H.2 and the definition of $\lambda_N, \mu_N$.

We define $\varepsilon_{\text{stat}} := 640s\mu_N \left\| \mathbf{w}_{\text{lasso}} - \mathbf{w}_\star \right\|_2^2$, $T_0 := 10\beta\nu^{-1} \left\| \mathbf{w}_{\text{lasso}} \right\|_2^2$. By Proposition B.3(3), it holds that for $t \geq T_0$,

$$\mathsf{gap}^t \leq \frac{\beta}{2t} \left\| \mathbf{w}_{\text{lasso}} \right\|_2^2 \leq \frac{\beta}{2T_0} \left\| \mathbf{w}_{\text{lasso}} \right\|_2^2 = \frac{\nu}{20}.$$

Then for all $t \geq T_0 - 1$, we have (the second $\leq$ below uses Proposition B.3(2))

$$\frac{\alpha}{3} \left\| \mathbf{w}^{t+1} - \mathbf{w}_{\text{lasso}} \right\|_2^2 \leq 4\mathsf{gap}^{t+1} + \varepsilon_{\text{stat}} \leq 2\beta \left( \left\| \mathbf{w}^t - \mathbf{w}_{\text{lasso}} \right\|_2^2 - \left\| \mathbf{w}^{t+1} - \mathbf{w}_{\text{lasso}} \right\|_2^2 \right) + \varepsilon_{\text{stat}},$$

$$\Rightarrow \left\| \mathbf{w}^{t+1} - \mathbf{w}_{\text{lasso}} \right\|_2^2 - \frac{3\varepsilon_{\text{stat}}}{\alpha} \leq \left( 1 + \frac{\alpha}{6\beta} \right)^{-1} \left( \left\| \mathbf{w}^t - \mathbf{w}_{\text{lasso}} \right\|_2^2 - \frac{3\varepsilon_{\text{stat}}}{\alpha} \right).$$

Therefore, for $t \geq T_0 - 1$,

$$\left\| \mathbf{w}^t - \mathbf{w}_{\text{lasso}} \right\|_2^2 \leq \exp\left( -\frac{\alpha}{12\beta}(t - \lceil T_0 \rceil + 1) \right) \left\| \mathbf{w}^{\lceil T_0 \rceil - 1} - \mathbf{w}_{\text{lasso}} \right\|_2^2 + \frac{3\varepsilon_{\text{stat}}}{\alpha}$$

$$\leq \exp\left( -\frac{\alpha}{8\beta}(t - T_0) \right) \left\| \mathbf{w}_{\text{lasso}} \right\|_2^2 + \frac{3\varepsilon_{\text{stat}}}{\alpha},$$

where the last inequality follows from Proposition B.3(2). Further, by Proposition B.3(3), we have

$$\mathsf{gap}^{t+k} \leq \frac{\beta}{2k} \left\| \mathbf{w}^t - \mathbf{w}_{\text{lasso}} \right\|_2^2 \leq \frac{\beta}{2k} \left[ \exp\left( -\frac{\alpha}{8\beta}(t - T_0) \right) \left\| \mathbf{w}_{\text{lasso}} \right\|_2^2 + \frac{3\varepsilon_{\text{stat}}}{\alpha} \right], \quad \forall t \geq T_0 - 1, k \geq 0.$$

Hence, we can conclude that $\mathsf{gap}^T \leq \varepsilon$ for all $T$ such that

$$T \geq 10\beta\nu^{-1} \left\| \mathbf{w}_{\text{lasso}} \right\|_2^2 + 8\kappa \log\left( \frac{\beta \left\| \mathbf{w}_{\text{lasso}} \right\|_2^2}{\varepsilon} \right) + \frac{3\kappa\varepsilon_{\text{stat}}}{\varepsilon} + 1.$$

Now, by Proposition H.1, it holds that $\left\| \mathbf{w}_{\text{lasso}} - \mathbf{w}_\star \right\|_2^2 \leq 10\frac{\rho\nu}{\alpha^2} \frac{s \log d}{N}$, and hence

$$\left\| \mathbf{w}_{\text{lasso}} \right\|_2^2 \leq 2 \left\| \mathbf{w}_\star \right\|_2^2 + 2 \left\| \mathbf{w}_{\text{lasso}} - \mathbf{w}_\star \right\|_2^2 \leq 2(B_w^\star)^2 + \frac{20\rho\nu s \log d}{\alpha^2 N}.$$

Plugging in our definition of

$$\mu_N = \frac{\rho \log d}{N}, \qquad \varepsilon_{\text{stat}} := 400s\mu_N \left\| \mathbf{w}_{\text{lasso}} - \mathbf{w}_\star \right\|_2^2, \qquad \omega_N = \frac{\rho}{\alpha} \frac{s \log d}{N} \leq 1$$

completes the proof. $\qquad\square$

## H.5  Proof of Theorem 8

In this section, we present the proof of Theorem 8 based on Theorem H.2. We begin by recalling the following RSC property of a Gaussian random matrix [87, Theorem 7.16], a classical result in the high-dimensional statistics literature.

**Proposition H.2** (RSC for Gaussian random design). *Suppose that* $\mathbf{X} = [\mathbf{x}_1; \cdots ; \mathbf{x}_N]^\top \in \mathbb{R}^{N \times d}$ *is a random matrix with each row* $\mathbf{x}_i$ *being i.i.d. samples from* $\mathsf{N}(0, \boldsymbol{\Sigma})$. *Then there are universal constants* $c_1 = \frac{1}{8}, c_2 = 50$ *such that with probability at least* $1 - \frac{e^{-N/32}}{1 - e^{-N/32}}$,

$$\frac{\|\mathbf{X}\mathbf{w}\|_2^2}{N} \geq c_1 \|\mathbf{w}\|_{\boldsymbol{\Sigma}}^2 - c_2 \rho(\boldsymbol{\Sigma}) \frac{\log d}{N} \|\mathbf{w}\|_1^2, \qquad \forall \mathbf{w} \in \mathbb{R}^d, \tag{40}$$

*where* $\rho(\boldsymbol{\Sigma}) = \max_{i \in [d]} \Sigma_{ii}$ *is the maximum of diagonal entries of* $\boldsymbol{\Sigma}$.

Fix a parameter $\delta_1 \leq \delta$ (which we will specify in proof) and a large universal constant $C_0$. Let us set

$$\alpha = c_1 = \Theta(1), \qquad \beta = 8(1 + (d/N)), \qquad \rho = c_2 = \Theta(1),$$
$$B_x = C_0 \sqrt{d \log(N/\delta_1)}, \qquad B_y = C_0(B_w^\star + \sigma)\sqrt{\log(N/\delta_1)}.$$

Similar to the proof of Corollary 6 (Appendix F.4), we consider the following good events (where $\varepsilon = \mathbf{X}\mathbf{w}_\star - \mathbf{y}$)

$$\mathcal{E}_w = \left\{ \lambda_{\max}(\mathbf{X}^\top \mathbf{X}/N) \leq \beta \text{ and } \mathbf{X} \text{ is } (\alpha, \rho)\text{-RSC} \right\},$$
$$\mathcal{E}_r = \left\{ \|\mathbf{X}^\top \varepsilon\|_\infty \geq 4\sigma \sqrt{N \log(4d/\delta)} \right\},$$
$$\mathcal{E}_b = \{ \forall i \in [N], \ \|\mathbf{x}_i\|_2 \leq B_x, \ |y_i| \leq B_y \},$$
$$\mathcal{E}_{b,N+1} = \{ \|\mathbf{x}_{N+1}\|_2 \leq B_x, \ |y_{N+1}| \leq B_y \},$$

and we define $\mathcal{E} := \mathcal{E}_w \cap \mathcal{E}_r \cap \mathcal{E}_b \cap \mathcal{E}_{b,N+1}$.

Furthermore, we choose $\nu > 0$ that correspond to the choice $\lambda_N = 8\sigma \sqrt{\frac{\log(4d/\delta)}{N}}$, and we also assume $N \geq \frac{32c_2}{c_1} \cdot s \log d$. Then, Assumption C holds on the event $\mathcal{E}$.

Therefore, we can apply Theorem H.2 with $\varepsilon = \nu \omega_N$, which implies that there exists a $L$-layer transformer $\boldsymbol{\theta}$ such that its prediction $\widehat{y}_{N+1} := \widetilde{\mathsf{read}}_y(\mathrm{TF}_{\boldsymbol{\theta}}^0(\mathbf{H}))$, so that under the good event $\mathcal{E}$ we have $\widehat{y}_{N+1} = \mathsf{clip}_{B_y}(\langle \mathbf{x}_{N+1}, \widehat{\mathbf{w}} \rangle)$, where

$$L_{\mathrm{lasso}}(\widehat{\mathbf{w}}) - L_{\mathrm{lasso}}(\mathbf{w}_{\mathrm{lasso}}) \leq \nu \omega_N.$$

In the following, we show that $\boldsymbol{\theta}$ is indeed the desired transformer (similarly to the proof in Appendix F.4). Consider the conditional prediction error

$$\mathbb{E}\left[(\widehat{y}_{N+1} - y_{N+1})^2 \big| \mathcal{D}\right] = \mathbb{E}\left[1\{\mathcal{E}\}(\widehat{y}_{N+1} - y_{N+1})^2 \big| \mathcal{D}\right] + \mathbb{E}\left[1\{\mathcal{E}^c\}(\widehat{y}_{N+1} - y_{N+1})^2 \big| \mathcal{D}\right],$$

and we analyze these two parts separately under the good event $\mathcal{E}_0 := \mathcal{E}_w \cap \mathcal{E}_r \cap \mathcal{E}_b$ of $\mathcal{D}$.

**Part I.** We first note that

$$\mathbb{E}\left[1\{\mathcal{E}\}(\widehat{y}_{N+1} - y_{N+1})^2 \big| \mathcal{D}\right] = \mathbb{E}\left[1\{\mathcal{E}\}(\mathsf{clip}_{B_y}(\langle \mathbf{x}_{N+1}, \widehat{\mathbf{w}} \rangle) - y_{N+1})^2 \big| \mathcal{D}\right]$$
$$\leq \mathbb{E}\left[1\{\mathcal{E}\}(\langle \mathbf{x}_{N+1}, \widehat{\mathbf{w}} \rangle - y_{N+1})^2 \big| \mathcal{D}\right],$$

where the inequality is because $y_{N+1} \in [-B_y, B_y]$ under the good event $\mathcal{E}$. Notice that by our construction, under the good event $\mathcal{E}$, $\widehat{\mathbf{w}} = \widehat{\mathbf{w}}(\mathcal{D})$ depends only on the dataset $\mathcal{D}$ (because it is the $(L-1)$-th iterate of PGD on (ICLasso) problem). Applying Proposition H.1 to $\widehat{\mathbf{w}}(\mathcal{D})$ and using the definition of $\omega_N$ and our choice of $\lambda_N$, we obtain that (under $\mathcal{E}_0$)

$$\|\widehat{\mathbf{w}}(\mathcal{D}) - \mathbf{w}_\star\|_2^2 \leq C \cdot \left[\frac{s\lambda_N^2}{\alpha^2} + \nu \omega_N^2 + \nu \omega_N\right] = \mathcal{O}\left(\frac{\sigma^2 s \log(d/\delta)}{N}\right).$$

Therefore, under $\mathcal{E}_0$,

$$\mathbb{E}\left[1\{\mathcal{E}\}(\langle \mathbf{x}_{N+1}, \widehat{\mathbf{w}} \rangle - y_{N+1})^2 \big| \mathcal{D}\right] = \mathbb{E}\left[1\{\mathcal{E}\}(\langle \mathbf{x}_{N+1}, \widehat{\mathbf{w}}(\mathcal{D}) \rangle - y_{N+1})^2 \big| \mathcal{D}\right]$$
$$\leq \mathbb{E}\left[(\langle \mathbf{x}_{N+1}, \widehat{\mathbf{w}}(\mathcal{D}) \rangle - y_{N+1})^2 \big| \mathcal{D}\right]$$
$$= \mathbb{E}\left[(\langle \mathbf{x}_{N+1}, \widehat{\mathbf{w}}(\mathcal{D}) \rangle - \langle \mathbf{x}_{N+1}, \mathbf{w}_\star \rangle)^2 \big| \mathcal{D}\right] + \sigma^2$$
$$= \|\widehat{\mathbf{w}}(\mathcal{D}) - \mathbf{w}_\star\|_2^2 + \sigma^2$$
$$= \sigma^2 \left[1 + \mathcal{O}\left(\frac{s \log(d/\delta)}{N}\right)\right].$$

**Part II.** Notice that under good event $\mathcal{E}_0$, the bad event $\mathcal{E}^c$ holds if and only if $\mathcal{E}_{b,N+1}^c$ holds, and hence

$$\mathbb{E}\left[1\{\mathcal{E}^c\}(\widehat{y}_{N+1} - y_{N+1})^2 \big| \mathcal{D}\right] = \mathbb{E}\left[1\{\mathcal{E}_{b,N+1}^c\}(\widehat{y}_{N+1} - y_{N+1})^2 \big| \mathcal{D}\right]$$
$$\leq \sqrt{\mathbb{P}(\mathcal{E}_{b,N+1}^c)\mathbb{E}[(\widehat{y}_{N+1} - y_{N+1})^4]}.$$

With a large enough constant $C_0$, we clearly have $\mathbb{P}(\mathcal{E}_{b,N+1}^c) \leq (\delta_1/N)^{10}$. Further, a simple calculation yields

$$\mathbb{E}(\widehat{y}_{N+1} - y_{N+1})^4 \leq 8\mathbb{E}(\widehat{y}_{N+1}^4 + y_{N+1}^4) \leq 8B_y^4 + 8\mathbb{E}y_{N+1}^4 \leq 16B_y^4,$$

where the last inequality is because the marginal distribution of $y_{N+1}$ is simply $\mathsf{N}(0, \sigma^2 + \|\mathbf{w}_\star\|_2^2)$. Combining these yields

$$\mathbb{E}\left[1\{\mathcal{E}^c\}(\widehat{y}_{N+1} - y_{N+1})^2 \big| \mathcal{D}\right] \leq \mathcal{O}\left(\frac{\delta_1^5 B_y^2}{N^5}\right) \leq \mathcal{O}\left(\frac{\delta_1^5((B_w^\star)^2 + \sigma^2)\log(1/\delta_1)}{N^4}\right).$$

Therefore, choosing $\delta_1 = \min\{\delta, \frac{\sigma}{B_w^\star}\}$ is enough for our purpose, and under such choice of $\delta_1$,

$$\mathbb{E}\left[1\{\mathcal{E}^c\}(\widehat{y}_{N+1} - y_{N+1})^2 \big| \mathcal{D}\right] \leq \mathcal{O}\left(\frac{\sigma^2}{N^4}\right).$$

**Conclusion.** Combining the inequalities above, we can conclude that under $\mathcal{E}_0$,

$$\mathbb{E}\left[(\widehat{y}_{N+1} - y_{N+1})^2 \big| \mathcal{D}\right] \leq \sigma^2\left[1 + \mathcal{O}\left(\frac{s\log(d/\delta)}{N}\right)\right].$$

It remains to show that $\mathbb{P}(\mathcal{E}_0) \geq 1 - \delta$. By Proposition H.2, Lemma B.2 and Lemma B.4, we have

$$\mathbb{P}(\mathcal{E}_w) \leq 3\exp(-N/32), \qquad \mathbb{P}(\mathcal{E}_r) \leq \frac{\delta}{2}, \qquad \mathbb{P}(\mathcal{E}_b) \leq \frac{\delta}{4}.$$

Therefore, as long as $N \geq 32\log(12/\delta)$, we have $\mathbb{P}(\mathcal{E}_0) \geq 1 - \delta$. This completes the proof. $\qquad\square$

We also remark that in the construction above,

$$R = \mathcal{O}\left((B_w^\star + \sigma)\sqrt{d}\log(N \cdot (1 + B_w^\star/\sigma))\right),$$

which would be useful for bounding $\|\boldsymbol{\theta}\|$.

# I Proofs for Section 4

## I.1 Proof of Proposition 10

We begin by restating Proposition 10 into the following version, which contains additional size bounds on $\boldsymbol{\theta}$.

**Theorem I.1** (Full statement of Proposition 10). *Suppose that for*

$$\widehat{L}_{\mathsf{val}}(f) := \frac{1}{|\mathcal{D}_{\mathsf{val}}|} \sum_{(\mathbf{x}_i, y_i) \in \mathcal{D}_{\mathsf{val}}} \ell(f(\mathbf{x}_i), y_i),$$

$\ell(\cdot, \cdot)$ *is* $(\gamma/3, R, M, C)$-*approximable by sum of relus (Definition D.1). Then there exists a 3-layer transformer* $\mathrm{TF}_{\boldsymbol{\theta}}$ *with*

$$\max_{\ell \in [3]} M^{(\ell)} \leq (M+3)K, \quad \max_{\ell \in [3]} D^{(\ell)} \leq K^2 + K + 1, \quad \|\boldsymbol{\theta}\| \leq \frac{2NKC}{|\mathcal{D}_{\mathsf{val}}|} + 3\gamma^{-1} + 7KR.$$

*that maps*

$$\mathbf{h}_i = [*; f_1(\mathbf{x}_i); \cdots ; f_K(\mathbf{x}_i); \mathbf{0}_{K+1}; 1; t_i] \quad \rightarrow \quad \mathbf{h}_i' = [*; \widehat{f}(\mathbf{x}_i); 1; t_i], \ i \in [N+1],$$

*where the predictor* $\widehat{f} : \mathbb{R}^d \to \mathbb{R}$ *is a convex combination of* $\{f_k : \widehat{L}_{\mathsf{val}}(f_k) \leq \min_{k_\star \in [K]} \widehat{L}_{\mathsf{val}}(f_{k_\star}) + \gamma\}$. *As a corollary, for any convex risk* $L : (\mathbb{R}^d \to \mathbb{R}) \to \mathbb{R}$, $\widehat{f}$ *satisfies*

$$L(\widehat{f}) \leq \min_{k_\star \in [K]} L(f_{k_\star}) + \max_{k \in [K]} \left|\widehat{L}_{\mathsf{val}}(f_k) - L(f_k)\right| + \gamma.$$

To prove Theorem I.1, we first state and prove the following two propositions.

**Proposition I.1** (Evaluation layer). *There exists a 1-layer transformer* $\mathrm{TF}_{\boldsymbol{\theta}}$ *with* $MK$ *heads and* $\|\boldsymbol{\theta}\| \leq 3R + 2NKC/|\mathcal{D}_{\mathsf{val}}|$ *such that for all* $\mathbf{H}$ *such that* $\max_i\{|y_i'|\} \leq R, \max_{i,k}\{|f_k(\mathbf{x}_i)|\} \leq R,$ $\mathrm{TF}_{\boldsymbol{\theta}}$ *maps*

$$\mathbf{h}_i = [\mathbf{x}_i; y_i'; *; f_1(\mathbf{x}_i); \cdots; f_K(\mathbf{x}_i); \mathbf{0}_{K+1}; 1; t_i]$$

$$\rightarrow \quad \mathbf{h}_i' = [\mathbf{x}_i; y_i'; *; f_1(\mathbf{x}_i); \cdots; f_K(\mathbf{x}_i); \widetilde{L}_{\mathsf{val}}(f_1); \cdots; \widetilde{L}_{\mathsf{val}}(f_K); 0; 1; t_i], \qquad i \in [N+1],$$

*where* $\widetilde{L}_{\mathsf{val}}(\cdot)$ *is a functional such that* $\max_k \left|\widetilde{L}_{\mathsf{val}}(f_k) - \widehat{L}_{\mathsf{val}}(f_k)\right| \leq \varepsilon.$

*Proof of Proposition I.1.* As $\ell$ is $(\varepsilon, R, M, C)$-approximable by sum of relus, there exists a function $g : \mathbb{R}^2 \to \mathbb{R}$ of form

$$g(s,t) = \sum_{m=1}^M c_m \sigma(a_m s + b_m t + d_m) \quad \text{with} \quad \sum_{m=1}^M |c_m| \leq C, \ |a_m| + |b_m| + |d_m| \leq 1, \ \forall m \in [M],$$

such that $\sup_{(s,t)\in[-R,R]^2} |g(s,t) - \ell(s,t)| \leq \varepsilon.$ We define

$$\widetilde{L}_{\mathsf{val}}(f) := \frac{1}{|\mathcal{D}_{\mathsf{val}}|} \sum_{(\mathbf{x}_i, y_i) \in \mathcal{D}_{\mathsf{val}}} g(f(\mathbf{x}_i), y_i),$$

Next, for every $m \in [M]$ and $k \in [K]$, we define matrices $\mathbf{Q}_{m,k}, \mathbf{K}_{m,k}, \mathbf{V}_{m,k} \in \mathbb{R}^{D \times D}$ such that for all $i, j \in [N+1]$,

$$\mathbf{Q}_{m,k}\mathbf{h}_i = \begin{bmatrix} a_m \\ b_m \\ d_m \\ -2 \\ \mathbf{0} \end{bmatrix}, \quad \mathbf{K}_{m,k}\mathbf{h}_j = \begin{bmatrix} f_k(\mathbf{x}_j) \\ y_j \\ 1 \\ R(1+t_j) \\ \mathbf{0} \end{bmatrix}, \quad \mathbf{V}_{m,k}\mathbf{h}_j = \frac{(N+1)c_m}{|\mathcal{D}_{\mathsf{val}}|} \cdot \mathbf{e}_{D-(K-k)-3}$$

where $\mathbf{e}_s \in \mathbb{R}^D$ is the vector with $s$-th entry being $1$ and others being $0$. As the input has structure $\mathbf{h}_i = [\mathbf{x}_i; y_i'; *; f_1(\mathbf{x}_i); \cdots; f_K(\mathbf{x}_i); \mathbf{0}_{K+1}; 1; t_i]$, these matrices indeed exist, and further it is straightforward to check that they have norm bounds

$$\max_{m\in[M],k\in[K]} \|\mathbf{Q}_{m,k}\|_{\mathrm{op}} \leq 3, \quad \max_{m\in[M],k\in[K]} \|\mathbf{K}_{m,k}\|_{\mathrm{op}} \leq 2 + R, \quad \sum_{m\in[M],k\in[K]} \|\mathbf{V}_{m,k}\|_{\mathrm{op}} \leq \frac{K(N+1)C}{|\mathcal{D}_{\mathsf{val}}|}.$$

Now, for every $i, j \in [N+1]$, we have

$$\sigma(\langle \mathbf{Q}_{m,k}\mathbf{h}_i, \mathbf{K}_{m,k}\mathbf{h}_j \rangle) = \sigma(a_m f_k(\mathbf{x}_j) + b_m y_j + d_m - 2R(1+t_j))$$
$$= \sigma(a_m \mathbf{w}^\top \mathbf{x}_j + b_m y_j + d_m) \mathbb{1}\{t_j = -1\},$$

where the last equality follows from the bound $|a_m f_k(\mathbf{x}_j) + b_m y_j + d_m| \leq R(|a_m| + |b_m|) + d_m \leq 2R$, so that the above relu equals $0$ if $t_j \leq 0$. Therefore, for each $i \in [N+1]$ and $k \in [K]$,

$$\sum_{m=1}^M \sigma(\langle \mathbf{Q}_{m,k}\mathbf{h}_i, \mathbf{K}_{m,k}\mathbf{h}_j \rangle) \mathbf{V}_{m,k}\mathbf{h}_j$$

$$= \left( \sum_{m=1}^M c_m \sigma(a_m \mathbf{w}^\top \mathbf{x}_j + b_m y_j + d_m) \right) \cdot \frac{(N+1)}{|\mathcal{D}_{\mathsf{val}}|} \mathbb{1}\{t_j = -1\} \mathbf{e}_{D-(K-k)-3}$$

$$= g(f_k(\mathbf{x}_j), y_j) \cdot \frac{(N+1)}{|\mathcal{D}_{\mathsf{val}}|} \mathbb{1}\{t_j = -1\} \mathbf{e}_{D-(K-k)-3}.$$

Thus letting the attention layer $\boldsymbol{\theta} = \{(\mathbf{V}_{m,k}, \mathbf{Q}_{m,k}, \mathbf{K}_{m,k})\}_{(m,k)\in[M]\times[K]}$, we have

$$\widetilde{\mathbf{h}}_i = [\mathrm{Attn}_{\boldsymbol{\theta}}(\mathbf{H})]_i = \mathbf{h}_i + \frac{1}{N+1} \sum_{j=1}^{N+1} \sum_{m,k} \sigma(\langle \mathbf{Q}_{m,k}\mathbf{h}_i, \mathbf{K}_{m,k}\mathbf{h}_j \rangle) \mathbf{V}_{m,k}\mathbf{h}_j$$

$$= \mathbf{h}_i + \frac{1}{|\mathcal{D}_{\mathsf{val}}|} \sum_{j=1}^{N+1} \sum_{k=1}^{K} g(f_k(\mathbf{x}_j), y_j) \cdot 1\{t_j = -1\} \mathbf{e}_{D-(K-k)-3}$$

$$= \mathbf{h}_i + \sum_{k=1}^{K} \left( \frac{1}{|\mathcal{D}_{\mathsf{val}}|} \sum_{(\mathbf{x}_j, y_j) \in \mathcal{D}_{\mathsf{val}}} g(f_k(\mathbf{x}_j), y_j) \right) \mathbf{e}_{D-(K-k)-3}$$

$$= \mathbf{h}_i + \sum_{k=1}^{K} \widetilde{L}_{\mathsf{val}}(f_k) \cdot \mathbf{e}_{D-(K-k)-3}$$

$$= [\mathbf{x}_i; y_i'; *; f_1(\mathbf{x}_i); \cdots ; f_K(\mathbf{x}_i); \mathbf{0}_{K+1}; 1; t_i] + [\mathbf{0}_{D-K-3}; \widetilde{L}_{\mathsf{val}}(f_1); \cdots ; \widetilde{L}_{\mathsf{val}}(f_K); 0; 0; 0]$$

$$= [\mathbf{x}_i; y_i'; *; f_1(\mathbf{x}_i); \cdots ; f_K(\mathbf{x}_i); \widetilde{L}_{\mathsf{val}}(f_1); \cdots ; \widetilde{L}_{\mathsf{val}}(f_K); 0; 1; t_i], \qquad i \in [N+1].$$

This is the desired result. $\qquad \square$

**Proposition I.2** (Selection layer). *There exists a 3-layer transformer* $\mathrm{TF}_{\boldsymbol{\theta}}$ *with*

$$\max_{\ell \in [3]} M^{(\ell)} \leq 2K + 2, \quad \max_{\ell \in [3]} D^{(\ell)} \leq K^2 + K + 1, \quad \|\boldsymbol{\theta}\| \leq \gamma^{-1} + 3KR + 2.$$

*such that* $\mathrm{TF}_{\boldsymbol{\theta}}$ *maps*

$$\mathbf{h}_i = [*; f_1(\mathbf{x}_i); \cdots ; f_K(\mathbf{x}_i); \mathbb{L}_1; \cdots ; \mathbb{L}_K; 0; 1; t_i]$$

$$\rightarrow \quad \mathbf{h}_i' = [*; f_1(\mathbf{x}_i); \cdots ; f_K(\mathbf{x}_i); *; \cdots ; *; \widehat{f}(\mathbf{x}_i); 1; t_i], \qquad i \in [N+1],$$

*where* $\widehat{f} = \sum_{k=1}^{K} \lambda_k f_k$ *is an aggregated predictor, where the weights* $\lambda_1, \cdots, \lambda_K \geq 0$ *are functions only on* $\mathbb{L}_1, \cdots, \mathbb{L}_k$ *such that*

$$\sum_{k=1}^{K} \lambda_k = 1, \qquad \lambda_k > 0 \text{ only if } \mathbb{L}_k \leq \min_{k^\star \in [K]} \mathbb{L}_{k^\star} + \gamma.$$

*Proof of Proposition I.2.* We construct a $\boldsymbol{\theta}$ which is a composition of 2 MLP layers followed by an attention layer $(\boldsymbol{\theta}_{\mathtt{mlp}}^{(1)}, \boldsymbol{\theta}_{\mathtt{mlp}}^{(2)}, \boldsymbol{\theta}_{\mathtt{attn}}^{(3)})$.

Step 1: construction of $\boldsymbol{\theta}_{\mathtt{mlp}}^{(1)}$. We consider matrix $\mathbf{W}_1^{(1)}$ that maps

$$\mathbf{h} = [*_{D-K-3}; \mathbb{L}_1; \cdots ; \mathbb{L}_K; *; *; *]$$

$$\mapsto \mathbf{W}_1^{(1)} \mathbf{h} = [\mathbb{L}_1 - \mathbb{L}_2; \cdots ; \mathbb{L}_1 - \mathbb{L}_K; \cdots ; \mathbb{L}_K - \mathbb{L}_{K-1}; \mathbb{L}_1; -\mathbb{L}_1; \cdots ; \mathbb{L}_K; -\mathbb{L}_K],$$

i.e. $\mathbf{W}_1^{(1)} \mathbf{h}$ is a $K^2 + K$ dimensional vector so that its entry contains $\{\mathbb{L}_k - \mathbb{L}_l\}_{k,l \in [K]}$ and $\{\mathbb{L}_k, -\mathbb{L}_k\}_{k \in [K]}$. Clearly, such $\mathbf{W}_1^{(1)}$ exists and can be chosen so that $\left\| \mathbf{W}_1^{(1)} \right\|_{\mathrm{op}} \leq 2K$. We then consider a matrix $\mathbf{W}_2^{(1)}$ that maps

$$\sigma(\mathbf{W}_1^{(1)} \mathbf{h}) \mapsto \mathbf{W}_2^{(1)} \sigma(\mathbf{W}_1^{(1)} \mathbf{h}) = [\mathbf{0}_{D-K-3}; c_1 - \mathbb{L}_1; \cdots ; c_K - \mathbb{L}_K; \mathbf{0}_3] \in \mathbb{R}^D,$$

where $c_k = c_k(\mathbb{L}) := \sum_{l \neq k} \sigma(\mathbb{L}_k - \mathbb{L}_l)$. Notice that

$$c_k - \mathbb{L}_k = -\sigma(\mathbb{L}_k) + \sigma(-\mathbb{L}_k) + \sum_{l \neq k} \sigma(\mathbb{L}_k - \mathbb{L}_l),$$

and hence such $\mathbf{W}_2^{(1)}$ exists and can be chosen so that $\left\| \mathbf{W}_2^{(1)} \right\|_{\mathrm{op}} \leq K + 1$. We set $\boldsymbol{\theta}_{\mathtt{mlp}}^{(1)} = (\mathbf{W}_1^{(1)}, \mathbf{W}_2^{(1)})$, then $\mathrm{MLP}_{\boldsymbol{\theta}_{\mathtt{mlp}}^{(1)}}$ maps $\mathbf{h}_i$ to

$$\mathbf{h}_i^{(1)} = [*; f_1(\mathbf{x}_i); \cdots ; f_K(\mathbf{x}_i); c_1; \cdots ; c_K; 0; 1; t_i].$$

The basic property of $\{c_k\}_{k \in [K]}$ is that, if $c_k \leq \gamma$, then $\mathbb{L}_k \leq \min_{k^\star \in [K]} \mathbb{L}_{k^\star} + \gamma$.

Step 2: construction of $\boldsymbol{\theta}_{\mathtt{mlp}}^{(2)}$. We consider matrix $\mathbf{W}_1^{(2)}$ that maps

$$\mathbf{h} = [*_{D-K-3}; c_1; \cdots ; c_K; *; 1; *]$$

$$\mapsto \mathbf{W}_1^{(2)}\mathbf{h} = [1 - \gamma^{-1}c_1; c_1; -c_1; \cdots; 1 - \gamma^{-1}c_K; c_K; -c_K] \in \mathbb{R}^{3K},$$

and $\mathbf{W}_1^{(2)}$ can be chosen so that $\left\|\mathbf{W}_1^{(2)}\right\|_{\mathrm{op}} \leq K + 1 + \gamma^{-1}$. We then consider a matrix $\mathbf{W}_2^{(2)}$ that maps

$$\sigma(\mathbf{W}_1^{(2)}\mathbf{h}) \mapsto \mathbf{W}_2^{(2)}\sigma(\mathbf{W}_1^{(1)}\mathbf{h}) = [\mathbf{0}_{D-K-3}; \sigma(1 - \gamma^{-1}c_1) - c_1; \cdots; \sigma(1 - \gamma^{-1}c_K) - c_K; \mathbf{0}_3] \in \mathbb{R}^D,$$

which exists and can be chosen so that $\left\|\mathbf{W}_2^{(1)}\right\|_{\mathrm{op}} \leq 2$. We set $\boldsymbol{\theta}_{\mathtt{mlp}}^{(2)} = (\mathbf{W}_1^{(2)}, \mathbf{W}_2^{(2)})$, then $\mathrm{MLP}_{\boldsymbol{\theta}_{\mathtt{mlp}}^{(2)}}$ maps $\mathbf{h}_i^{(1)}$ to

$$\mathbf{h}_i^{(2)} = [*; f_1(\mathbf{x}_i); \cdots; f_K(\mathbf{x}_i); u_1; \cdots; u_K; 0; 1; t_i],$$

where $u_k = \sigma(1 - \gamma^{-1}c_k) \forall k \in [K]$. Clearly, $u_k \in [0, 1]$, and $u_k > 0$ if and only if $c_k \leq \gamma$.

Step 3: construction of $\boldsymbol{\theta}_{\mathtt{attn}}^{(3)}$. We define

$$\lambda_1 = 1 - \sigma(1 - u_1), \qquad \lambda_k = \sigma(1 - u_1 - \cdots - u_{k-1}) - \sigma(1 - u_1 - \cdots - u_k) \, \forall k \geq 2.$$

Clearly, $\lambda_k \geq 0$, and $\sum_k \lambda_k = 1$. Further,

$$\lambda_k > 0 \Rightarrow u_k > 0 \Rightarrow c_k \leq \gamma \Rightarrow \mathbb{L}_k \leq \min_{k^\star \in [K]} \mathbb{L}_{k^\star} + \gamma.$$

Therefore, it remains to construct $\boldsymbol{\theta}_{\mathtt{attn}}^{(3)}$ that implements $\widehat{f} = \sum_{k=1}^K \lambda_k f_k$ based on $[\mathbf{h}_i^{(2)}]_i$. Notice that

$$\widehat{f}(\mathbf{x}_i) = \sigma(1) \cdot f_1(\mathbf{x}_i) + \sum_{k=1}^{K-1} \sigma(1 - u_1 - \cdots - u_{k-1}) \cdot (f_k(\mathbf{x}_i) - f_{k-1}(\mathbf{x}_i)) \tag{41}$$
$$- \sigma(1 - u_1 - \cdots - u_K) \cdot f_K(\mathbf{x}_i),$$

and hence we construct $\boldsymbol{\theta}_{\mathtt{attn}}^{(3)}$ as follows: for every $k \in [K+1]$ and $w \in \{0, 1\}$, we define matrices $\mathbf{Q}_{k,w}, \mathbf{K}_{k,w}, \mathbf{V}_{k,w} \in \mathbb{R}^{D \times D}$ such that for all $k \in [K+1]$

$$\mathbf{Q}_{k,0}\mathbf{h}_i^{(2)} = \begin{bmatrix} (f_k(\mathbf{x}_i) + R) \cdot \mathbf{1}_k \\ \mathbf{0} \end{bmatrix}, \quad \mathbf{Q}_{k,1}\mathbf{h}_i^{(2)} = \begin{bmatrix} (f_{k-1}(\mathbf{x}_i) + R) \cdot \mathbf{1}_k \\ \mathbf{0} \end{bmatrix},$$

$$\mathbf{K}_{k,0}\mathbf{h}_j^{(2)} = \mathbf{K}_{k,1}\mathbf{h}_j^{(2)} = \begin{bmatrix} 1 \\ -u_1 \\ \vdots \\ -u_{k-1} \\ \mathbf{0} \end{bmatrix}, \qquad \mathbf{V}_{k,0}\mathbf{h}_j^{(2)} = \mathbf{e}_{D-2} = -\mathbf{V}_{k,1}\mathbf{h}_j^{(2)},$$

for all $i, j \in [N+1]$, where we understand $f_0 = f_{K+1} = 0$ and $\mathbf{1}_k$ is the $k$-dimensional vector with all entries being 1. By the structure of $\mathbf{h}_i^{(2)}$, these matrices indeed exist, and further it is straightforward to check that they have norm bounds

$$\max_{k \in [K+1], w \in \{0,1\}} \|\mathbf{Q}_{k,w}\|_{\mathrm{op}} \leq KR, \quad \max_{k \in [K+1], w \in \{0,1\}} \|\mathbf{K}_k\|_{\mathrm{op}} \leq 1, \quad \sum_{k \in [K+1], w \in \{0,1\}} \|\mathbf{V}_{k,w}\|_{\mathrm{op}} \leq 2K + 2.$$

Now, for every $i, j \in [N+1]$, $k \in [K+1]$, $w \in \{0, 1\}$, we have

$$\sigma\left(\left\langle \mathbf{Q}_{k,w}\mathbf{h}_i^{(2)}, \mathbf{K}_{k,w}\mathbf{h}_j^{(2)} \right\rangle\right) = \sigma((1 - u_1 - \cdots - u_{k-1})(f_{k-w}(\mathbf{x}_i) + R))$$
$$= \sigma(1 - u_1 - \cdots - u_{k-1}) \cdot (f_{k-w}(\mathbf{x}_i) + R),$$

where the last equality follows from $f_k(\mathbf{x}_i) + R \geq 0 \forall k \in [K]$. Therefore,

$$\sum_{k \in [K+1], w \in \{0,1\}} \sigma\left(\left\langle \mathbf{Q}_{m,k}\mathbf{h}_i^{(2)}, \mathbf{K}_{m,k}\mathbf{h}_j^{(2)} \right\rangle\right) \mathbf{V}_{m,k}\mathbf{h}_j^{(2)}$$

$$= \sum_{k=1}^K \left[\sigma(1 - u_1 - \cdots - u_{k-1}) \cdot (f_k(\mathbf{x}_i) + R) - \sigma(1 - u_1 - \cdots - u_{k-1}) \cdot (f_{k-1}(\mathbf{x}_i) + R)\right] \cdot \mathbf{e}_{D-2}$$

$$= \widehat{f}(\mathbf{x}_i) \cdot \mathbf{e}_{D-2},$$

where the last equality is due to (41). Thus letting the attention layer $\boldsymbol{\theta}_{\mathtt{attn}}^{(3)} = \{(\mathbf{V}_{k,w}, \mathbf{Q}_{k,w}, \mathbf{K}_{k,w})\}_{(k,w) \in [K+1] \times \{0,1\}}$, we have

$$\mathbf{h}_i^{(3)} = \left[\mathrm{Attn}_{\boldsymbol{\theta}}(\mathbf{H}^{(2)})\right]_i = \mathbf{h}_i + \frac{1}{N+1} \sum_{j=1}^{N+1} \sum_{k,w} \sigma\left(\left\langle \mathbf{Q}_{k,w} \mathbf{h}_i^{(2)}, \mathbf{K}_{k,w} \mathbf{h}_j^{(2)} \right\rangle\right) \mathbf{V}_{k,w} \mathbf{h}_j^{(2)}$$

$$= \mathbf{h}_i^{(2)} + \widehat{f}(\mathbf{x}_i) \cdot \mathbf{e}_{D-2}$$

$$= [*; f_1(\mathbf{x}_i); \cdots ; f_K(\mathbf{x}_i); u_1; \cdots ; u_K; \widehat{f}(\mathbf{x}_i); 1; t_i].$$

This is the desired result. $\qquad\square$

Now, we are ready to prove Theorem I.1.

**Proof of Theorem I.1** As $\ell(\cdot, \cdot)$ is $(\gamma/3, R, M, C)$-approximable by sum of relus, we can invoke Proposition I.1 to show that there exists a single attention layer $\boldsymbol{\theta}_{\mathtt{attn}}^{(1)}$ so that $\mathrm{Attn}_{\boldsymbol{\theta}_{\mathtt{attn}}^{(1)}}$ maps

$$\mathbf{h}_i \quad \rightarrow \quad \mathbf{h}_i' = [\mathbf{x}_i; y_i'; *; f_1(\mathbf{x}_i); \cdots ; f_K(\mathbf{x}_i); \widetilde{L}_{\mathsf{val}}(f_1); \cdots ; \widetilde{L}_{\mathsf{val}}(f_K); 0; 1; t_i], \qquad i \in [N+1],$$

for any input $\mathbf{H} = [\mathbf{h}_i]_i$ of the form described in Theorem I.1, and $\widetilde{L}_{\mathsf{val}}(\cdot)$ is a functional such that $\max_k \left|\widetilde{L}_{\mathsf{val}}(f_k) - \widehat{L}_{\mathsf{val}}(f_k)\right| \le \gamma/3$.

Next, by the proof of Proposition I.2, there exists $(\boldsymbol{\theta}_{\mathtt{mlp}}^{(1)}, \boldsymbol{\theta}_{\mathtt{mlp}}^{(2)}, \boldsymbol{\theta}_{\mathtt{attn}}^{(3)})$ that maps

$$\mathbf{h}_i' \quad \rightarrow \quad \mathbf{h}_i^{(3)} = \left[\mathbf{x}_i; y_i'; *; f_1(\mathbf{x}_i); \cdots ; f_K(\mathbf{x}_i); *; \sum_{k=1}^{K} \lambda_k f_k(\mathbf{x}_i); 1; t_i\right], \qquad i \in [N+1],$$

where $\lambda = (\lambda_1, \cdots, \lambda_K) \in \Delta([K])$ and $\lambda_k > 0$ only when $\widetilde{L}_{\mathsf{val}}(f_k) \le \min_{k^\star} \widetilde{L}_{\mathsf{val}}(f_{k^\star}) + \gamma/3$. Using the fact that $\max_k |\widetilde{L}_{\mathsf{val}}(f_k) - \widehat{L}_{\mathsf{val}}(f_k)| \le \gamma/3$, we deduce that $\lambda$ is supported on $\{k : \widehat{L}_{\mathsf{val}}(f_k) \le \min_{k_\star \in [K]} \widehat{L}_{\mathsf{val}}(f_{k_\star}) + \gamma\}$.

Therefore, $\boldsymbol{\theta} = (\boldsymbol{\theta}_{\mathtt{attn}}^{(1)}, \boldsymbol{\theta}_{\mathtt{mlp}}^{(1)}, \boldsymbol{\theta}_{\mathtt{mlp}}^{(2)}, \boldsymbol{\theta}_{\mathtt{attn}}^{(3)})$ is the desired transformer, with

$$\max_{\ell \in [3]} M^{(\ell)} \le (M+3)K, \quad \max_{\ell \in [3]} D^{(\ell)} \le K^2 + K + 1,$$

and

$$\|\boldsymbol{\theta}\| \le \max\left\{3R + \frac{2NKC}{|\mathcal{D}_{\mathsf{val}}|} + 3K + 1, K + 3 + \gamma^{-1}, KR + 2K + 2\right\}$$

$$\le 7KR + \frac{2NKC}{|\mathcal{D}_{\mathsf{val}}|} + \gamma^{-1}.$$

This completes the proof. $\qquad\square$

## I.2 Proof of Theorem 11

We first restate Theorem 11 into the following version which provides additional size bounds for $\boldsymbol{\theta}$. For the simplicity of presentation, throughout this subsection and Appendix J, we denote $\mathcal{I}_t = \{i : (\mathbf{x}_i, y_i) \in \mathcal{D}_{\mathsf{train}}\}$, $\mathcal{I}_v = \{i : (\mathbf{x}_i, y_i) \in \mathcal{D}_{\mathsf{val}}\}$, $\mathbf{X}_{\mathsf{train}} = [\mathbf{x}_i]_{i \in \mathcal{I}_t}$ to be the input matrix corresponding to the training split only, and $N_{\mathsf{train}} = |\mathcal{D}_{\mathsf{train}}|$, $N_{\mathsf{val}} = |\mathcal{D}_{\mathsf{val}}|$.

**Theorem I.2.** *For any sequence of regularizations $\{\lambda_k\}_{k \in [K]}$, $0 \le \alpha \le \beta$ with $\kappa := \max_k \frac{\beta + \lambda_k}{\alpha + \lambda_k}$, $B_w > 0$, $\gamma > 0$, and $\varepsilon < B_w/2$, suppose in input format (3) we have $D \ge \Theta(Kd)$. Then there exists an $L$-layer transformer $\mathrm{TF}_{\boldsymbol{\theta}}$ with*

$$L = \lceil 2\kappa \log(B_w/(2\varepsilon)) \rceil + 4, \quad \max_{\ell \in [L]} M^{(\ell)} \le 3K + 1, \quad \max_{\ell \in [L]} D^{(\ell)} \le K^2 + K + 1,$$

$$\|\boldsymbol{\theta}\| \leq \mathcal{O}\left(KR + (\beta + \lambda)^{-1} + \frac{N}{N_{\mathsf{val}}} + \gamma^{-1}\right), \qquad R := \max\{B_x B_w, B_y, 1\},$$

*such that the following holds. On any input data* $(\mathcal{D}, \mathbf{x}_{N+1})$ *such that the problem (ICRidge) is well-conditioned and has a bounded solution:*

$$\alpha \leq \lambda_{\min}(\mathbf{X}_{\mathsf{train}}^{\top} \mathbf{X}_{\mathsf{train}} / N_{\mathsf{train}}) \leq \lambda_{\max}(\mathbf{X}_{\mathsf{train}}^{\top} \mathbf{X}_{\mathsf{train}} / N_{\mathsf{train}}) \leq \beta, \quad \max_{k \in [K]} \left\| \mathbf{w}_{\mathsf{ridge}}^{\lambda_k}(\mathcal{D}_{\mathsf{train}}) \right\|_2 \leq B_w/2,$$
(42)

$\mathrm{TF}_{\boldsymbol{\theta}}^0$ *approximately implements ridge selection: its prediction*

$$\widehat{y}_{N+1} = \mathsf{read}_{\mathsf{y}}(\mathrm{TF}_{\boldsymbol{\theta}}^0(\mathbf{H})) = \langle \widehat{\mathbf{w}}, \mathbf{x}_{N+1} \rangle, \qquad \widehat{\mathbf{w}} = \sum_{k=1}^{K} \lambda_k \widehat{\mathbf{w}}_k$$

*satisfies the following.*

1. *For each* $k \in [K]$, $\widehat{\mathbf{w}}_k = \widehat{\mathbf{w}}_k(\mathcal{D}_{\mathsf{train}})$ *approximates the ridge estimator* $\mathbf{w}_{\mathsf{ridge}}^{\lambda_k}(\mathcal{D}_{\mathsf{train}})$, *i.e.* $\left\| \widehat{\mathbf{w}}_k - \mathbf{w}_{\mathsf{ridge}}^{\lambda_k}(\mathcal{D}_{\mathsf{train}}) \right\|_2 \leq \varepsilon$.

2. $\lambda = (\lambda_1, \cdots, \lambda_K) \in \Delta([K])$ *so that*

$$\lambda_k > 0 \text{ only if } \widehat{L}_{\mathsf{val}}(\widehat{\mathbf{w}}_k) \leq \min_{k^\star \in [K]} \widehat{L}_{\mathsf{val}}(\widehat{\mathbf{w}}_{k^\star}) + \gamma.$$

In particular, if we set $\gamma' = 2(B_x B_w + B_y)B_x \varepsilon + \gamma$, then it holds that[8]

$$\mathrm{dist}\left( \widehat{\mathbf{w}}, \mathrm{conv}\{\widehat{\mathbf{w}}_{\mathsf{ridge},\mathsf{train}}^{\lambda_k} : \widehat{L}_{\mathsf{val}}(\widehat{\mathbf{w}}_{\mathsf{ridge},\mathsf{train}}^{\lambda_k}) \leq \min_{k_\star \in [K]} \widehat{L}_{\mathsf{val}}(\widehat{\mathbf{w}}_{\mathsf{ridge},\mathsf{train}}^{\lambda_{k_\star}}) + \gamma'\} \right) \leq \varepsilon,$$

*where we denote* $\widehat{\mathbf{w}}_{\mathsf{ridge},\mathsf{train}}^{\lambda_k} := \mathbf{w}_{\mathsf{ridge}}^{\lambda_k}(\mathcal{D}_{\mathsf{train}})$.

To prove Theorem I.2, we first show that, for the squared validation loss, there exists a 3-layer transformer that performs predictor selection based on the *exactly* evaluated $\widehat{L}_{\mathsf{val}}(f_k)$ for each $k \in [K]$. (Proof in Appendix I.2.1.)

**Theorem I.3** (Square-loss version of Theorem I.1). *Consider the squared validation loss*

$$\widehat{L}_{\mathsf{val}}(f) := \frac{1}{2|\mathcal{D}_{\mathsf{val}}|} \sum_{(x_i, y_i) \in \mathcal{D}_{\mathsf{val}}} (f(\mathbf{x}_i) - y_i)^2.$$

*Then there exists a 3-layer transformer* $\mathrm{TF}_{\boldsymbol{\theta}}$ *with*

$$\max_{\ell \in [3]} M^{(\ell)} \leq 2K + 2, \quad \max_{\ell \in [3]} D^{(\ell)} \leq K^2 + K + 1, \quad \|\boldsymbol{\theta}\| \leq 7KR + \frac{2N}{|\mathcal{D}_{\mathsf{val}}|} + \gamma^{-1},$$

*such that for any input* $\mathbf{H}$ *that takes form*

$$\mathbf{h}_i = [\mathbf{x}_i; y_i'; *; f_1(\mathbf{x}_i); \cdots; f_K(\mathbf{x}_i); \mathbf{0}_K; *; 1; t_i],$$

*where* $\mathrm{TF}_{\boldsymbol{\theta}}$ *outputs* $\mathbf{h}_{N+1} = [\mathbf{x}_{N+1}; \widehat{f}(\mathbf{x}_{N+1}); *; 1; 0]$, *where the predictor* $\widehat{f} : \mathbb{R}^d \to \mathbb{R}$ *is a convex combination of* $\{f_k : \widehat{L}_{\mathsf{val}}(f_k) \leq \min_{k_\star \in [K]} \widehat{L}_{\mathsf{val}}(f_{k_\star}) + \gamma\}$. *As a corollary, for any convex risk* $L : (\mathbb{R}^d \to \mathbb{R}) \to \mathbb{R}$, $\widehat{f}$ *satisfies*

$$L(\widehat{f}) \leq \min_{k_\star \in [K]} L(f_{k_\star}) + \max_{k \in [K]} \left| \widehat{L}_{\mathsf{val}}(f_k) - L(f_k) \right| + \gamma.$$

**Proof of Theorem I.2** First, by the proof[9] of Theorem 4 and Proposition B.6, for each $k \in [K]$, there exists a $T = L - 3$ layer transformer $\boldsymbol{\theta}^{(1:T)}$ such that $\mathrm{TF}_{\boldsymbol{\theta}^{(1:T)}}$ maps

$$\mathbf{h}_i \quad \to \quad \mathbf{h}_i^{(T)} = [\mathbf{x}_i; y_i'; *; \langle \widehat{\mathbf{w}}_1, \mathbf{x}_i \rangle; \cdots; \langle \widehat{\mathbf{w}}_K, \mathbf{x}_i \rangle; \mathbf{0}_K; 1; t_i],$$

---

[8]This is because $\widehat{L}_{\mathsf{val}}(\mathbf{w})$ is $(B_x B_w + B_y)B_x$-Lipschitz w.r.t. $\mathbf{w} \in \mathsf{B}_2(B_w)$.

[9]Technically, an adapted version where the underlying ICGD mechanism operates on the training split (with $t_i = 1$) with size $N_{\mathsf{train}}$ instead of on all $N$ training examples, which only changes $\|\boldsymbol{\theta}\|$ by at most a constant factor, and does not change the number of layers and heads.

so that if (42) holds, we have $\left\| \widehat{\mathbf{w}}_k - \mathbf{w}_{\mathrm{ridge}}^{\lambda_k} \right\|_2 \le \varepsilon$ and $\widehat{\mathbf{w}}_k \in \mathsf{B}_2(B_w)$.

Next, by Theorem I.3, there exists a 3-layer transformer $\boldsymbol{\theta}^{(T+1:T+3)}$ that outputs

$$\mathbf{h}_{N+1}^{(T+3)} = [\mathbf{x}_{N+1}; \langle \widehat{\mathbf{w}}, \mathbf{x}_{N+1} \rangle; *; 1; t_i],$$

where $\widehat{\mathbf{w}} = \sum_{k=1}^K \lambda_k \widehat{\mathbf{w}}_k$, $\lambda = (\lambda_1, \cdots, \lambda_K) \in \Delta([K])$ so that

$$\lambda_k > 0 \text{ only if } \widehat{L}_{\mathsf{val}}(\widehat{\mathbf{w}}_k) \le \min_{k^\star \in [K]} \widehat{L}_{\mathsf{val}}(\widehat{\mathbf{w}}_{k^\star}) + \gamma.$$

This is the desired result. $\qquad\square$

### I.2.1   Proof of Theorem I.3

Similar to the proof of Proposition 10, Theorem I.3 is a direct corollary by combining Proposition I.3 with Proposition I.2.

**Proposition I.3** (Evaluation layer for the squared loss). *There exists an attention layer $\mathrm{TF}_{\boldsymbol{\theta}}$ with $2K$ heads and $\|\boldsymbol{\theta}\| \le 3R + 2NK/|\mathcal{D}_{\mathsf{val}}|$ such that $\mathrm{TF}_{\boldsymbol{\theta}}$ maps*

$$\mathbf{h}_i = [*; f_1(\mathbf{x}_i); \cdots; f_K(\mathbf{x}_i); \mathbf{0}_K; *; 1; t_i]$$
$$\rightarrow \quad \mathbf{h}_i' = [*; f_1(\mathbf{x}_i); \cdots; f_K(\mathbf{x}_i); \widehat{L}_{\mathsf{val}}(f_1); \cdots; \widehat{L}_{\mathsf{val}}(f_K); *; 1; t_i], \qquad i \in [N+1].$$

*Proof of Proposition I.3.* For every $k \in [K]$, we define matrices $\mathbf{Q}_{m,k}, \mathbf{K}_{m,k}, \mathbf{V}_{m,k} \in \mathbb{R}^{D \times D}$ such that for all $i, j \in [N+1]$,

$$\mathbf{Q}_{k,0}\mathbf{h}_i = \begin{bmatrix} 1 \\ -1 \\ -2 \\ \mathbf{0} \end{bmatrix}, \quad \mathbf{Q}_{k,1}\mathbf{h}_i = \begin{bmatrix} -1 \\ 1 \\ -2 \\ \mathbf{0} \end{bmatrix}, \quad \mathbf{K}_{k,0}\mathbf{h}_j = \mathbf{K}_{k,1}\mathbf{h}_j = \begin{bmatrix} f_k(\mathbf{x}_j) \\ y_j \\ R(1 + t_j) \\ \mathbf{0} \end{bmatrix},$$

$$\mathbf{V}_{k,0}\mathbf{h}_j = -\mathbf{V}_{k,1}\mathbf{h}_j = \frac{(N+1)}{2|\mathcal{D}_{\mathsf{val}}|} \cdot (f_k(\mathbf{x}_j) - y_j)\mathbf{e}_{D-(K-k)-3}.$$

As the input has structure $\mathbf{h}_i = [\mathbf{x}_i; y_i'; *; f_1(\mathbf{x}_i); \cdots; f_K(\mathbf{x}_i); \mathbf{0}_{K+1}; 1; t_i]$, these matrices indeed exist, and further it is straightforward to check that they have norm bounds

$$\max_{k \in [K], w \in \{0,1\}} \|\mathbf{Q}_{k,w}\|_{\mathrm{op}} \le 3, \quad \max_{k \in [K], w \in \{0,1\}} \|\mathbf{K}_{k,w}\|_{\mathrm{op}} \le 1 + R, \quad \sum_{k \in [K], w \in \{0,1\}} \|\mathbf{V}_{k,w}\|_{\mathrm{op}} \le \frac{K(N+1)}{|\mathcal{D}_{\mathsf{val}}|}.$$

Now, for every $i, j \in [N+1]$, we have

$$\sum_{w \in \{0,1\}} \sigma(\langle \mathbf{Q}_{k,w}\mathbf{h}_i, \mathbf{K}_{k,w}\mathbf{h}_j \rangle)\mathbf{V}_{k,w}\mathbf{h}_j$$

$$= [\sigma(f_k(\mathbf{x}_j) - y_j - 2R(1 + t_j)) - \sigma(y_j - f_k(\mathbf{x}_j) - 2R(1 + t_j))] \cdot \frac{(N+1)}{2|\mathcal{D}_{\mathsf{val}}|}(f_k(\mathbf{x}_j) - y_j)\mathbf{e}_{D-(K-k)-3}$$

$$= \mathbb{1}\{t_j = -1\} \cdot [\sigma(f_k(\mathbf{x}_j) - y_j) - \sigma(y_j - f_k(\mathbf{x}_j))] \cdot \frac{(N+1)}{2|\mathcal{D}_{\mathsf{val}}|}(f_k(\mathbf{x}_j) - y_j)\mathbf{e}_{D-(K-k)-3}$$

$$= \mathbb{1}\{t_j = -1\} \cdot \frac{(N+1)}{2|\mathcal{D}_{\mathsf{val}}|}(f_k(\mathbf{x}_j) - y_j)^2 \mathbf{e}_{D-(K-k)-3},$$

where the second equality follows from the bound $|f_k(\mathbf{x}_j) - y_j| \le 2R$, so that the relus equals 0 if $t_j \le 0$. Thus letting the attention layer $\boldsymbol{\theta} = \{(\mathbf{V}_{k,w}, \mathbf{Q}_{k,w}, \mathbf{K}_{k,w})\}_{(k,w) \in [K] \times \{0,1\}}$, we have

$$\widetilde{\mathbf{h}}_i = [\mathrm{Attn}_{\boldsymbol{\theta}}(\mathbf{H})]_i = \mathbf{h}_i + \frac{1}{N+1} \sum_{j=1}^{N+1} \sum_{k,w} \sigma(\langle \mathbf{Q}_{k,w}\mathbf{h}_i, \mathbf{K}_{k,w}\mathbf{h}_j \rangle)\mathbf{V}_{k,w}\mathbf{h}_j$$

$$= \mathbf{h}_i + \frac{1}{2|\mathcal{D}_{\text{val}}|} \sum_{j=1}^{N+1} \sum_{k=1}^{K} (f_k(\mathbf{x}_j) - y_j)^2 \cdot 1\{t_j = -1\} \mathbf{e}_{D-(K-k)-3}$$

$$= \mathbf{h}_i + \sum_{k=1}^{K} \left( \frac{1}{2|\mathcal{D}_{\text{val}}|} \sum_{(\mathbf{x}_j, y_j) \in \mathcal{D}_{\text{val}}} (f_k(\mathbf{x}_j) - y_j)^2 \right) \mathbf{e}_{D-(K-k)-3}$$

$$= \mathbf{h}_i + \sum_{k=1}^{K} \widehat{L}_{\text{val}}(f_k) \cdot \mathbf{e}_{D-(K-k)-3}$$

$$= [\mathbf{x}_i; y_i'; *; f_1(\mathbf{x}_i); \cdots; f_K(\mathbf{x}_i); \mathbf{0}_{K+1}; 1; t_i] + [\mathbf{0}_{D-K-3}; \widehat{L}_{\text{val}}(f_1); \cdots; \widehat{L}_{\text{val}}(f_K); 0; 0; 0]$$

$$= [\mathbf{x}_i; y_i'; *; f_1(\mathbf{x}_i); \cdots; f_K(\mathbf{x}_i); \widehat{L}_{\text{val}}(f_1); \cdots; \widehat{L}_{\text{val}}(f_K); 0; 1; t_i], \qquad i \in [N+1].$$

This is the desired result. $\qquad\square$

## I.3 Proofs for Section 4.2

### I.3.1 Proof of Lemma 13

It is straightforward to check that the binary type check $\psi : \mathbb{R} \to \mathbb{R}$ can be expressed as a linear combination of 6 relu's (recalling $\sigma(\cdot) = \text{ReLU}(\cdot)$):

$$\psi(y) = \sigma\left(\frac{y+\varepsilon}{\varepsilon}\right) - 2\sigma\left(\frac{y}{\varepsilon}\right) + \sigma\left(\frac{y-\varepsilon}{\varepsilon}\right) + \sigma\left(\frac{y-(1-\varepsilon)}{\varepsilon}\right) - 2\sigma\left(\frac{y-1}{\varepsilon}\right) + \sigma\left(\frac{y-(1+\varepsilon)}{\varepsilon}\right)$$

$$=: \sum_{m=1}^{6} a_m \sigma(b_m y + c_m),$$

with $\sum_m |a_m| = 8/\varepsilon$, $\max_m \max\{|b_m|, |c_m|\} \le 2$. We can thus construct an attention layer $\boldsymbol{\theta} = \{(\mathbf{Q}_m, \mathbf{K}_m, \mathbf{V}_m)\}_{m=1}^{6}$ with 6 heads such that

$$\mathbf{Q}_m \mathbf{h}_i = [b_m; c_m; \mathbf{0}_{D-2}], \quad \mathbf{K}_m \mathbf{h}_j = [y_j; 1; \mathbf{0}_{D-2}], \quad \mathbf{V}_m \mathbf{h}_j = \left[\frac{N+1}{N} a_m \cdot t_j; \mathbf{0}_{D-1}\right],$$

which gives that for every $i \in [N+1]$,

$$\sum_{m=1}^{6} \frac{1}{N+1} \sum_{j \in [N+1]} \sigma(\langle \mathbf{Q}_m \mathbf{h}_i, \mathbf{K}_m \mathbf{h}_j \rangle)[\mathbf{V}_m \mathbf{h}_j]_1$$

$$= \sum_{m=1}^{6} \frac{1}{N} \sum_{j=1}^{N} \sigma(b_m y_j + c_m) a_m = \frac{1}{N} \sum_{j=1}^{N} \psi(y_j) = \Psi^{\text{binary}}(\mathcal{D}).$$

Further, we have $\|\boldsymbol{\theta}\| \le 18/\varepsilon = \mathcal{O}(1/\varepsilon)$. This is the desired result. $\qquad\square$

By composing the above attention layer with one additional layer (with 2 heads) that implement the following function

$$\sigma(2(t - 1/2)) - \sigma(2(t - 1)),$$

on the output $\Psi^{\text{binary}}(\mathcal{D})$, we directly obtain the following corollary.

**Corollary I.1** (Thresholded binary test). *There exists a two-layer attention-only transformer with* $\max_{\ell \in [2]} M^{(\ell)} \le 6$ *and* $\|\boldsymbol{\theta}\| \le \mathcal{O}(1/\varepsilon)$ *that exactly implements the thresholded binary test*

$$\Psi^{\text{binary}}_{\text{thres}}(\mathcal{D}) := \begin{cases} 1, & \text{if } \Psi^{\text{binary}}(\mathcal{D}) \ge 1, \\ 0, & \text{if } \Psi^{\text{binary}}(\mathcal{D}) \le \frac{1}{2}, \\ \text{linear interpolation}, & \text{o.w.} \end{cases} \tag{43}$$

*at every token* $i \in [N+1]$, *where we recall the definition of* $\Psi^{\text{binary}}$ *in Lemma 13.*

### I.3.2 Formal statement and proof of Proposition 14

We say a distribution $\mathsf{P}_y$ on $\mathbb{R}$ is $(C, \varepsilon_0)$-not-concentrated around $\{0, 1\}$ if

$$\mathsf{P}_y([-\varepsilon, \varepsilon] \cup [1 - \varepsilon, 1 + \varepsilon]) \leq C\varepsilon$$

for all $\varepsilon \in (0, \varepsilon_0]$. A sufficient condition is that the density $\mathsf{p}_y$ is upper bounded by $C$ within $[-\varepsilon_0, \varepsilon_0] \cup [1 - \varepsilon_0, 1 + \varepsilon_0]$.

Throughout this section, let $\sigma_{\log}(t) := (1 + e^{-t})^{-1}$ denote the sigmoid activation, and let $\widehat{\mathbf{w}}_{\log}$ denote the solution to the in-context logistic regression problem, i.e. (ICGLM) with $g(\cdot) = \sigma_{\log}(\cdot)$.

**Proposition I.4** (Adaptive regression or classification; Formal version of Proposition 14). *For any $B_w > 0$, $\varepsilon \leq B_x B_w / 10$, $0 < \alpha \leq \beta$ with $\kappa := \beta/\alpha$, and any $(C, \varepsilon_0)$, there exists a $L$-layer attention-only transformer with*

$$L \leq \mathcal{O}\left(\kappa \log \frac{B_x B_w}{\varepsilon}\right), \quad \max_{\ell \in [L]} M^{(\ell)} \leq \mathcal{O}\left(\left(1 + \frac{B_x^4}{\alpha^2}\right)\varepsilon^{-2}\right), \quad \|\boldsymbol{\theta}\| \leq \mathcal{O}\left(R + \frac{1}{\beta} + \frac{1}{\varepsilon}\right)$$

*(with $R := \max\{B_x B_w, B_y, 1\}$, and $\varepsilon$ depending only on $(C, \varepsilon_0)$) such that the following holds. Suppose the input format is (3) with dimension $D \geq 3d + 4$.*

*On any* classification *instance $(\mathcal{D}, \mathbf{x}_{N+1})$ (such that $\{y_i\}_{i \in [N]} \subset \{0, 1\}$) that is well-conditioned for logistic regression in the sense of (35), it outputs $\widehat{y}_{N+1}$ that $\varepsilon$-approximates the prediction of in-context logistic regression:*

$$|\widehat{y}_{N+1} - \sigma_{\log}(\langle \mathbf{x}_{N+1}, \widehat{\mathbf{w}}_{\log} \rangle)| \leq \varepsilon.$$

*On the contrary, for* regression *problems, i.e. any in-context distribution $\mathsf{P}$ whose marginal $\mathsf{P}_y$ is $(C, \varepsilon_0)$-not-concentrated around $\{0, 1\}$, with probability at least $1 - \exp(-cN)$ over $\mathcal{D}$ (where $c > 0$ depends only on $(C, \varepsilon_0)$), $\widehat{y}_{N+1}$ $\varepsilon$-approximates the prediction of in-context least squares if the data is well-conditioned:*

$$|\widehat{y}_{N+1} - \langle \mathbf{x}_{N+1}, \widehat{\mathbf{w}}_{\mathrm{LS}} \rangle| \leq \varepsilon \quad \text{whenever } \mathcal{D} \text{ satisfies (5) with } \lambda = 0,$$

*where $\widehat{\mathbf{w}}_{\mathrm{LS}}$ denotes the in-context least squares estimator, i.e. (ICRidge) with $\lambda = 0$.*

*Proof.* The result follows by combining the binary test in Corollary I.1 with Theorem 4 and Theorem G.1. By those results, there exists three attention-only transformers $\boldsymbol{\theta}_{\mathrm{LS}}, \boldsymbol{\theta}_{\log}, \boldsymbol{\theta}_{\mathrm{bin}}$, with (below $L_g, C_g = \Theta(1)$ for $g = \sigma_{\log}(\cdot)$)

$$L_{\mathrm{LS}} \leq \mathcal{O}\left(\kappa \log \frac{B_x B_w}{\varepsilon}\right), \quad \max_{\ell \in [L_{\mathrm{LS}}]} M_{\mathrm{LS}}^{(\ell)} \leq 3, \quad \|\boldsymbol{\theta}_{\mathrm{LS}}\| \leq \mathcal{O}\left(R + \frac{1}{\beta}\right),$$

$$L_{\log} \leq \mathcal{O}\left(\kappa \log \frac{L_g B_x B_w}{\varepsilon}\right), \quad \max_{\ell \in [L_{\log}]} M_{\log}^{(\ell)} \leq \mathcal{O}\left(C_g^2\left(1 + \frac{L_g^2 B_x^4}{\alpha^2}\right)\varepsilon^{-2}\right), \quad \|\boldsymbol{\theta}_{\log}\| \leq \mathcal{O}\left(R + \frac{C_g}{\beta}\right),$$

$$L_{\mathrm{bin}} = 2, \quad \max_{\ell \in [2]} M_{\mathrm{bin}}^{(\ell)} \leq 6, \quad \|\boldsymbol{\theta}_{\mathrm{bin}}\| \leq \mathcal{O}(1/\varepsilon),$$

that outputs prediction $\widehat{y}_{N+1}^{\mathrm{LS}}$, $\widehat{y}_{N+1}^{\log}$ (at the $(N + 1)$-th token) and $\Psi_{\mathrm{thres}}^{\mathrm{binary}}(\mathcal{D})$ (at every token) respectively, which satisfy

$$\left|\widehat{y}_{N+1}^{\log} - \sigma_{\log}(\langle \mathbf{x}_{N+1}, \widehat{\mathbf{w}}_{\log} \rangle)\right| \leq \varepsilon,$$

$$\left|\widehat{y}_{N+1}^{\mathrm{LS}} - \langle \mathbf{x}_{N+1}, \widehat{\mathbf{w}}_{\mathrm{LS}} \rangle\right| \leq \varepsilon.$$

when the corresponding well-conditionednesses are satisfied. In particular, we can make $\widehat{\mathbf{w}}_{\log}$ well-defined on non-binary data, by multiplying $\Psi_{\mathrm{thres}}^{\mathrm{binary}}(\mathcal{D})$ onto the $\mathbf{x}_i$'s (which can be implemented by slightly modifying $\boldsymbol{\theta}_{\log}$ without changing the order of the number of layers, heads, and norms) so that $\widehat{\mathbf{w}}_{\log} = \mathbf{0}$ on any data where $\Psi_{\mathrm{thres}}^{\mathrm{binary}}(\mathcal{D}) = 0$.

By joining $\boldsymbol{\theta}_{\mathrm{LS}}$ and $\boldsymbol{\theta}_{\log}$ using Proposition B.6, concatenating with $\boldsymbol{\theta}_{\mathrm{bin}}$ before, and concatenating with one additional attention layer with 2 heads after to implement

$$\Psi_{\mathrm{thres}}^{\mathrm{binary}}(\mathcal{D})\widehat{y}_{N+1}^{\log} + \left(1 - \Psi_{\mathrm{thres}}^{\mathrm{binary}}(\mathcal{D})\right)\widehat{y}_{N+1}^{\mathrm{LS}}, \tag{44}$$

we obtain a single transformer $\boldsymbol{\theta}$ with

$$L \leq \mathcal{O}\left(\kappa \log \frac{B_x B_w}{\varepsilon}\right), \quad \max_{\ell \in [L]} M^{(\ell)} \leq \mathcal{O}\left(\left(1 + \frac{B_x^4}{\alpha^2}\right)\varepsilon^{-2}\right), \quad \|\boldsymbol{\theta}_{\mathrm{LS}}\| \leq \mathcal{O}\left(R + \frac{1}{\beta} + \frac{1}{\varepsilon}\right),$$

which outputs (44) as its prediction (at the location for $\widehat{y}_{N+1}$).

It remains to show that (44) reduces to either one of $\widehat{y}_{N+1}^{\log}$ or $\widehat{y}_{N+1}^{\mathrm{LS}}$. When the data are binary ($y_i \in \{0, 1\}$), we have $\Psi^{\mathsf{binary}}(\mathcal{D}) = 1$ and $\Psi_{\mathsf{thres}}^{\mathsf{binary}}(\mathcal{D}) = 1$, in which case (44) becomes exactly $\widehat{y}_{N+1}^{\log}$. By contrast, when data is sampled from a distribution that is $(C, \varepsilon_0)$-not-concentrated around $\{0, 1\}$, we have for any fixed $\varepsilon \leq \varepsilon_0 \wedge \frac{1}{4C}$ that, letting $B_\varepsilon := [-\varepsilon, \varepsilon] \cup [1 - \varepsilon, 1 + \varepsilon]$ and $\mathsf{p}_\varepsilon := \mathsf{P}_y(B_\varepsilon) \leq C\varepsilon \leq \frac{1}{4}$, by Hoeffding's inequality,

$$\mathsf{P}(\Psi_{\mathsf{thres}}^{\mathsf{binary}}(\mathcal{D}) \neq 0) = \mathsf{P}\left(\Psi_{\mathsf{thres}}^{\mathsf{binary}}(\mathcal{D}) \geq \frac{1}{2}\right) = \mathsf{P}\left(\frac{1}{N}\sum_{i=1}^{N} 1\{y_j \in B_\varepsilon\} \geq \frac{1}{2}\right)$$

$$\leq \exp\left(-c(1/2 - \mathsf{p}_\varepsilon)^2 N\right) \leq \exp(-c'N),$$

where $c' > 0$ is an absolute constant. On the event $\Psi_{\mathsf{thres}}^{\mathsf{binary}}(\mathcal{D}) = 0$ (which happens with probability at least $1 - \exp(-c'N)$), (44) becomes exactly $\widehat{y}_{N+1}^{\mathrm{LS}}$. This finishes the proof. $\qquad\square$

### I.4 Linear correlation test and application

In this section, we give another instantiation of the pre-ICL testing mechanism by showing that the transformer can implement a *linear correlation test* that tests whether the correlation vector $\mathbb{E}[\mathbf{x}y]$ has a large norm. We then use this test to construct a transformer to perform "confident linear regression", i.e. output a prediction from linear regression only when the signal-to-noise ratio is high.

For any fixed parameters $\lambda_{\min}, B_w^\star > 0$, consider the linear correlation test over data $\mathcal{D}$ defined as

$$
\begin{aligned}
\Psi^{\mathsf{lin}}(\mathcal{D}) &:= \frac{1}{\lambda_{\min}^2 (B_w^\star)^2 / 2} \cdot \left[\sigma\left(\|\widehat{\mathbf{t}}\|_2^2 - (\lambda_{\min} B_w^\star / 4)^2\right) - \sigma\left(\|\widehat{\mathbf{t}}\|_2^2 - (3\lambda_{\min} B_w^\star / 4)^2\right)\right] \\
&= \begin{cases} 0, & \|\widehat{\mathbf{t}}\|_2^2 \leq (\lambda_{\min} B_w^\star / 4)^2, \\ 1, & \|\widehat{\mathbf{t}}\|_2^2 \geq (3\lambda_{\min} B_w^\star / 4)^2, \\ \text{linear interpolation}, & \text{o.w.}, \end{cases}
\end{aligned}
\tag{45}
$$

where $\widehat{\mathbf{t}} = \mathbf{T}(\mathcal{D}) := \frac{1}{N}\sum_{i=1}^{N} \mathbf{x}_i y_i$.

Recall that $\sigma(\cdot) = \mathrm{ReLU}(\cdot)$ above denotes the relu activation.

We show that $\Psi^{\mathsf{lin}}$ can be exactly implemented by a 3-layer transformer.

**Lemma I.1** (Expressing $\Psi^{\mathsf{lin}}$ by transformer). *There exists a 3-layer attention-only transformer $\mathrm{TF}_{\boldsymbol{\theta}}$ with at most 2 heads per layer and $\|\boldsymbol{\theta}\| \leq \mathcal{O}(1 + \lambda_{\min}^2 (B_w^\star)^2)$ such that on input sequence $\mathbf{H}$ of the form (3) with $D \geq 2d + 4$, the transformer* exactly *implements $\Psi^{\mathsf{lin}}$: it outputs $\widetilde{\mathbf{H}}$ such that $\widetilde{\mathbf{h}}_i = [\mathbf{x}_i; y_i t_i; *; \Psi^{\mathsf{lin}}(\mathcal{D}); 1]$ for all $i \in [N + 1]$.*

*Proof.* We begin by noting the following basic facts:

- Identity function can be implemented exactly by two ReLUs: $t = \sigma(t) - \sigma(-t)$.

- Squared $\ell_2$ norm can be implemented exactly by a single attention head (assuming every input $\mathbf{h}_i$ contains the same vector $\mathbf{g}$): $\|\mathbf{g}\|_2^2 = \sigma(\langle \mathbf{g}, \mathbf{g} \rangle)$.

We construct the transformer $\boldsymbol{\theta}$ as follows.

Layer 1: Use 2 heads to implement $\widehat{\mathbf{t}} = \frac{1}{N}\sum_{i=1}^{N} \mathbf{x}_i y_i$, where $\mathbf{V}_{\{1,2\}}^{(1)} \mathbf{h}_j = [\pm \mathbf{x}_j; \mathbf{0}_{D-d}]$, $\mathbf{Q}_{\{1,2\}}^{(1)} \mathbf{h}_i = [\frac{N+1}{N}; \mathbf{0}_{D-1}]$, and $\mathbf{K}_{\{1,2\}}^{(1)} \mathbf{h}_j = [\pm y_j t_j; \mathbf{0}_{D-1}] = [\pm y_j 1\{j < N + 1\}; \mathbf{0}_{D-1}]$ (where we recall $t_j = 1\{j < N + 1\}$ and note that $y_j t_j$ corresponds exactly to the location for $y_j$ in $\mathbf{H}$, cf. (3)).

By manipulating the output dimension in $\mathbf{V}^{(1)}$, write the result $\widehat{\mathbf{t}}$ into blank memory space with dimension $d$ at every token $i \in [N+1]$.

Layer 2: Use a single head to compute $\|\widehat{\mathbf{t}}\|_2^2$: $\mathbf{Q}_1^{(2)}\mathbf{h}_i^{(1)} = [\widehat{\mathbf{t}}; \mathbf{0}_{D-d}]$, $\mathbf{K}_1^{(2)}\mathbf{h}_j^{(1)} = [\widehat{\mathbf{t}}; \mathbf{0}_{D-d}]$, and $\mathbf{V}_1^{(2)}\mathbf{h}_j^{(1)} = [1; \mathbf{0}_{D-1}]$. By manipulating the output dimension in $\mathbf{V}^{(2)}$, write the result $\|\widehat{\mathbf{t}}\|_2^2$ into blank memory space with dimension 1 at every token $i \in [N+1]$. After layer 2, we have $\mathbf{h}_i^{(3)} = [\mathbf{x}_i; y_i t_i; *; \|\widehat{\mathbf{t}}\|_2^2; *; 1]$.

Layer 3: Use 2 heads to implement two ReLU functions with bias: $\|\widehat{\mathbf{t}}\|_2^2 \mapsto \frac{1}{B-A}(\sigma(\|\widehat{\mathbf{t}}\|_2^2 - A) - \sigma(\|\widehat{\mathbf{t}}\|_2^2 - B))$. The two query (or key) matrices contain values $A$ and $B$. In our problem we take

$$A = (\lambda_{\min} B_w^\star/4)^2, \quad B = (3\lambda_{\min} B_w^\star/4)^2,$$

so that the above ReLU function implements $\Psi^{\mathrm{lin}}(\mathcal{D})$ exactly. Write the result into a blank memory space with dimension 1. We finish the proof by noting that $\|\boldsymbol{\theta}\| \leq \mathcal{O}(1 + \lambda_{\min}^2(B_w^\star)^2)$. $\qquad\square$

**Statistical guarantee for $\Psi^{\mathrm{lin}}$** We consider the following well-posedness assumption for the linear correlation test $\Psi^{\mathrm{lin}}$. Note that, similar as Assumption A, the assumption does not require the data to be generated from any true linear model, but rather only requires some properties about the best linear fit $\mathbf{w}_P^\star$, as well as sub-Gaussianity conditions.

**Assumption D** (Well-posedness for linear correlation test). *We say a distribution P on $\mathbb{R}^d \times \mathbb{R}$ is well-posed for linear independence tests, if $(\mathbf{x}, y) \sim$ P satisfies*

(1) *$\|\mathbf{x}\|_2 \leq B_x$ and $|y| \leq B_y$ almost surely;*

(2) *The covariance $\boldsymbol{\Sigma}_P := \mathbb{E}_P[\mathbf{x}\mathbf{x}^\top]$ satisfies $\lambda_{\min}\mathbf{I}_d \preceq \boldsymbol{\Sigma}_P \preceq \lambda_{\max}\mathbf{I}_d$, with $0 < \lambda_{\min} \leq \lambda_{\max}$, and $\kappa := \lambda_{\max}/\lambda_{\min}$.*

(3) *The whitened vector $\boldsymbol{\Sigma}_P^{-1/2}\mathbf{x}$ is $K^2$-sub-Gaussian for some $K \geq 1$.*

(4) *The best linear predictor $\mathbf{w}_P^\star := \mathbb{E}_P[\mathbf{x}\mathbf{x}^\top]^{-1}\mathbb{E}_P[\mathbf{x}y]$ satisfies $\|\mathbf{w}_P^\star\|_2 \leq \overline{B_w^\star}$.*

(5) *The label $y$ is $\sigma^2$-sub-Gaussian.*

(6) *The residual $z := y - \langle \mathbf{x}, \mathbf{w}_P^\star \rangle$ is $\sigma^2$-sub-Gaussian with probability one (over $\mathbf{x}$).*

The following results states that $\Psi^{\mathrm{lin}}$ achieves high power as long as the sample size is high enough, and the signal $\|\mathbf{w}_P^\star\|_2$ is either sufficiently high or sufficiently low.

**Proposition I.5** (Power of linear correlation test). *Suppose distribution P satisfies Assumption D with parameters $\lambda_{\min}, \lambda_{\max}, \overline{B_w^\star}$. Then, for the linear correlation test $\Psi^{\mathrm{lin}}$ with parameters $(\lambda_{\min}, B_w^\star)$ with $B_w^\star \leq \overline{B_w^\star}$ and any $N \geq \widetilde{\mathcal{O}}\left(\max\{K^4, \frac{\lambda_{\max}K^2\sigma^2}{(B_w^\star)^2\lambda_{\min}^2}\} \cdot d\right)$, we have*

1. *If $\|\mathbf{w}_P^\star\|_2 \geq B_w^\star$, then with probability at least $1 - \delta$ over $\mathcal{D}$, we have $\Psi^{\mathrm{lin}}(\mathcal{D}) = 1$.*

2. *If $\|\mathbf{w}_P^\star\|_2 \leq \frac{\lambda_{\min}}{10\lambda_{\max}} B_w^\star$, then with probability at least $1 - \delta$ over $\mathcal{D}$, we have $\Psi^{\mathrm{lin}}(\mathcal{D}) = 0$.*

*Proof.* For any P satisfying Assumption D, note that $\mathbb{E}[\mathbf{x}z] = \mathbb{E}[\mathbf{x}(y - \langle \mathbf{w}_P^\star, \mathbf{x}\rangle)] = \mathbf{0}$ by construction. Therefore, by standard sub-Gaussian and sub-exponential concentration combined with union bound, the following events hold simultaneously with probability at least $1 - \delta$:

$$0.9\boldsymbol{\Sigma}_P \preceq \widehat{\boldsymbol{\Sigma}} = \frac{1}{N}\sum_{i=1}^N \mathbf{x}_i\mathbf{x}_i^\top \preceq 1.1\boldsymbol{\Sigma}_P \quad \text{as } N \geq \widetilde{\mathcal{O}}(dK^4) \text{ by (27)},$$

$$\left\|\frac{1}{N}\sum_{i=1}^N \mathbf{x}_i z_i\right\|_2 \leq \lambda_{\max}^{1/2} \cdot \left\|\frac{1}{N}\sum_{i=1}^N \boldsymbol{\Sigma}_P^{-1/2}\mathbf{x}_i z_i\right\|_2 \leq \widetilde{\mathcal{O}}\left(\lambda_{\max}^{1/2}\left(\frac{K\sigma\sqrt{d}}{\sqrt{N}} + \frac{K\sigma d}{N}\right)\right)$$

$$\leq \widetilde{\mathcal{O}}\left(\lambda_{\max}^{1/2}K\sigma\sqrt{\frac{d}{N}}\right) \leq \frac{\lambda_{\min}B_w^\star}{8}, \quad \text{as } N \geq \widetilde{\mathcal{O}}\left(\frac{\lambda_{\max}dK^2\sigma^2}{(B_w^\star)^2\lambda_{\min}^2}\right).$$

On the above event, we have

$$\left\|\widehat{\mathbf{t}}\right\|_2 = \left\|\frac{1}{N}\sum_{i=1}^{N}\mathbf{x}_i(\langle\mathbf{x}_i,\mathbf{w}_\mathsf{P}^\star\rangle + \mathbf{z}_i)\right\|_2 = \left\|\widehat{\boldsymbol{\Sigma}}\mathbf{w}_\mathsf{P}^\star + \frac{1}{N}\sum_{i=1}^{N}\mathbf{x}_i z_i\right\|_2.$$

Therefore, in case 1, we have

$$\left\|\widehat{\mathbf{t}}\right\|_2 \geq \left\|\widehat{\boldsymbol{\Sigma}}\mathbf{w}_\mathsf{P}^\star\right\|_2 - \left\|\frac{1}{N}\sum_{i=1}^{N}\mathbf{x}_i z_i\right\|_2 \geq 0.9\lambda_{\min}\left\|\mathbf{w}_\mathsf{P}^\star\right\|_2 - \frac{\lambda_{\min}B_w^\star}{8} \geq \frac{3\lambda_{\min}B_w^\star}{4}.$$

In case 2, we have

$$\left\|\widehat{\mathbf{t}}\right\|_2 \leq \left\|\widehat{\boldsymbol{\Sigma}}\mathbf{w}_\mathsf{P}^\star\right\|_2 + \left\|\frac{1}{N}\sum_{i=1}^{N}\mathbf{x}_i z_i\right\|_2 \leq \lambda_{\max}\cdot\frac{\lambda_{\min}B_w^\star}{10\lambda_{\max}} + \frac{\lambda_{\min}B_w^\star}{8} \leq \frac{\lambda_{\min}B_w^\star}{4}.$$

The proof is finished by recalling the definition of $\Psi^{\mathsf{lin}}$ in (45), so that $\Psi^{\mathsf{lin}}(\mathcal{D}) = 1$ if $\|\widehat{\mathbf{t}}\|_2 \geq 3\lambda_{\min}B_w^\star/4$, and $\Psi^{\mathsf{lin}}(\mathcal{D}) = 0$ if $\|\widehat{\mathbf{t}}\|_2 \leq \lambda_{\min}B_w^\star/4$. $\qquad\square$

**Application: Confident linear regression** By directly composing the linear correlation test in Lemma I.1 with the transformer construction in Corollary 5 (using an argument similar as the proof of Proposition I.4), and using the power of the linear correlation test Proposition I.5, we immediately obtain the following result, which outputs a prediction from (approximately) least squares if $\widehat{\psi} := \Psi^{\mathsf{lin}}(\mathcal{D}) = 1$, and abstains from predicting if $\widehat{\psi} = 0$. This can be viewed as a form of "confident linear regression", where the model predicts only if it thinks the linear signal is strong enough.

**Proposition I.6** (Confident linear regression). *For any $B_w > 0$, $0 < B_w^\star \leq \overline{B_w^\star}$, $0 \leq \lambda_{\min} \leq \lambda_{\max}$, $\varepsilon \leq B_x B_w/10$, $0 < \alpha \leq \beta$ with $\kappa := \beta/\alpha$, there exists a $L$-layer attention-only transformer with*

$$L \leq \mathcal{O}\left(\kappa\log\frac{B_x B_w}{\varepsilon}\right), \quad \max_{\ell\in[L]}M^{(\ell)} \leq \mathcal{O}(1), \quad \|\boldsymbol{\theta}\| \leq \mathcal{O}\left(R + \frac{1}{\beta} + \lambda_{\min}^2(B_w^\star)^2\right)$$

*(with $R := \max\{B_x B_w, B_y, 1\}$) such that the following holds. Let $N \geq \widetilde{\mathcal{O}}\left(\max\{K^4, \frac{\lambda_{\max}K^2\sigma^2}{(B_w^\star)^2\lambda_{\min}^2}\}\cdot d\right)$. Suppose the input format is (3) with dimension $D \geq 2d + 4$. Let ICL instance $(\mathcal{D},\mathbf{x}_{N+1})$ be drawn from any distribution $\mathsf{P}$ satisfying Assumption D. Then the transformer outputs a 2-dimensional prediction (within the test token $\widetilde{\mathbf{h}}_{N+1}$)*

$$(\widehat{y}_{N+1},\widehat{\psi}) \in \mathbb{R}\times\{0,1\}$$

*such that the following holds:*

1. *If $\|\mathbf{w}_\mathsf{P}^\star\|_2 \geq B_w^\star$, then with probability at least $1-\delta$ over $\mathcal{D}$, we have $|\widehat{y}_{N+1} - \langle\widehat{\mathbf{w}}_{\mathrm{LS}},\mathbf{x}_{N+1}\rangle| \leq \varepsilon$, and $\widehat{\psi} = 1$ if $\mathcal{D}$ is in addition well-conditioned for least squares (in the sense of (5) with $\lambda = 0$).*

2. *If $\|\mathbf{w}_\mathsf{P}^\star\|_2 \leq \frac{\lambda_{\min}}{10\lambda_{\max}}B_w^\star$, then with probability at least $1-\delta$ over $\mathcal{D}$, we have $\widehat{y}_{N+1} = 0$ and $\widehat{\psi} = 0$.*

## J   Proof of Theorem 12: Noisy linear model with mixed noise levels

For each fixed $k \in [K]$, we consider the following data generating model $\mathbb{P}_k$, where we first sample $\mathsf{P} = \mathsf{P}_{\mathbf{w}_\star,\sigma_k} \sim \pi$ from $\mathbf{w}_\star \sim \mathsf{N}(\mathbf{0},\mathbf{I}_d/d)$, and then sample data $\{(\mathbf{x}_i,y_i)\}_{i\in[N+1]} \overset{\mathrm{iid}}{\sim} \mathsf{P}_{\mathbf{w}_\star,\sigma_k}$ as

$$\mathsf{P}_{\mathbf{w}_\star,\sigma_k} : \mathbf{x}_i \sim \mathsf{N}(\mathbf{0},\mathbf{I}_d), \quad y_i = \langle\mathbf{x}_i,\mathbf{w}_\star\rangle + \varepsilon_i, \quad \varepsilon_i \sim \mathsf{N}(0,\sigma_k^2).$$

Also, recall that the Bayes optimal estimator on $\mathbb{P}_k$ is given by $\widehat{y}_{N+1}^{\mathsf{Bayes}} = \left\langle\mathbf{w}_{\mathrm{ridge}}^{\lambda_k}(\mathcal{D}),\mathbf{x}_{N+1}\right\rangle$ with ridge $\lambda_k = \sigma_k^2 d/N$, and the Bayes risk on $\mathbb{P}_k$ is given by

$$\mathsf{BayesRisk}_k := \inf_{\mathcal{A}}\mathbb{E}_k\left[\tfrac{1}{2}(\mathcal{A}(\mathcal{D})(\mathbf{x}_{N+1}) - y_{N+1})^2\right] = \mathbb{E}_k\left[\tfrac{1}{2}\left(\widehat{y}_{N+1}^{\mathsf{Bayes}} - y_{N+1}\right)^2\right].$$

Recall that in Section 4.1.1, we consider a mixture law $\mathbb{P}_\pi$ that generates data from $\mathbb{P}_k$ with $k \sim \Lambda$. It is clear that we have (pushing $\inf_{\mathcal{A}}$ into $\mathbb{E}_{k\sim\Lambda}$ does not increase the value) we have

$$\mathsf{BayesRisk}_\pi \geq \mathbb{E}_{k\sim\Lambda}[\mathsf{BayesRisk}_k],$$

i.e., the Bayes risk can only be greater if we consider a mixture of models. In other words, if a transformer can achieve near-Bayes ICL on each meta-task $\mathbb{P}_k$, then it can perform near-Bayes ICL on any meta-task $\pi$ which is a mixture of $\mathbb{P}_k$ with $k \sim \Lambda$. Therefore, to prove Theorem 12, it suffices to show the following (strengthened) result.

**Theorem J.1** (Formal version of Theorem 12). *Suppose that $N \geq 0.1d$ and we write $\sigma_{\max} = \max_k\{\sigma_k, 1\}, \sigma_{\min} = \min_k\{\sigma_k, 1\}$. Suppose in input format (3) we have $D \geq \Theta(Kd)$. Then there exists a transformer $\boldsymbol{\theta}$ with*

$$L \leq \mathcal{O}\left(\sigma_{\min}^{-2}\log(N/\sigma_{\min})\right), \qquad \max_{\ell\in[L]} M^{(\ell)} \leq \mathcal{O}(K), \qquad \max_{\ell\in[L]} D^{(\ell)} \leq \mathcal{O}(K^2),$$

$$\|\boldsymbol{\theta}\| \leq \mathcal{O}\left(\sigma_{\max}Kd\log(N)\right),$$

*such that for any $k \in [K]$, it holds that*

$$\mathbb{E}_k\left[\frac{1}{2}(y_{N+1} - \widehat{y}_{N+1})^2\right] \leq \mathsf{BayesRisk}_k + \widetilde{\mathcal{O}}\left(\frac{\sigma_{\max}^2}{\sigma_{\min}^{2/3}}\left(\frac{\log K}{N}\right)^{1/3}\right)$$

*if we choose $N_{\mathsf{val}} := |\mathcal{D}_{\mathsf{val}}| \asymp N^{2/3}[\log K]^{1/3}$.*

The core of the proof of Theorem J.1 is to show that any estimator $\widehat{\mathbf{w}}$ that achieves small validation loss $\widehat{L}_{\mathsf{val}}$ must achieve small population loss.

Throughout the rest of this section, recall that we define $N_{\mathsf{train}} = |\mathcal{D}_{\mathsf{train}}|$, $N_{\mathsf{val}} = |\mathcal{D}_{\mathsf{val}}|$, $\mathcal{I}_{\mathsf{train}} = \{i : (\mathbf{x}_i, y_i) \in \mathcal{D}_{\mathsf{train}}\}$, $\mathcal{I}_{\mathsf{val}} = \{i : (\mathbf{x}_i, y_i) \in \mathcal{D}_{\mathsf{val}}\}$, and $\mathbf{X}_{\mathsf{train}} = [\mathbf{x}_i]_{i\in\mathcal{I}_{\mathsf{train}}}$.

## J.1 Proof of Theorem J.1

Fix parameters $\delta, \varepsilon, \gamma > 0$ and a large universal constant $C_0$. Let us set

$$\alpha = \max\left\{0, 1/2 - \sqrt{d/N_{\mathsf{train}}}\right\}^2, \qquad \beta = 25,$$

$$B_w^\star = 1 + C_0\sqrt{\frac{\log(N)}{d}}, \qquad B_w = C_0(B_w^\star + \sigma_{\max}/\sigma_{\min}),$$

$$B_x = C_0\sqrt{d\log(N)}, \qquad B_y = C_0(B_w^\star + \sigma_{\max})\sqrt{\log(N)},$$

Then, we define good events similarly to the proof of Corollary 6 (Appendix F.4):

$$\mathcal{E}_\pi = \{\|\mathbf{w}_\star\|_2 \leq B_w^\star, \|\boldsymbol{\varepsilon}\|_2 \leq 2\sigma_{\max}\sqrt{N}\},$$
$$\mathcal{E}_w = \{\alpha \leq \lambda_{\min}(\mathbf{X}_{\mathsf{train}}^\top\mathbf{X}_{\mathsf{train}}/N_{\mathsf{train}}) \leq \lambda_{\max}(\mathbf{X}_{\mathsf{train}}^\top\mathbf{X}_{\mathsf{train}}/N_{\mathsf{train}}) \leq \beta\},$$
$$\mathcal{E}_{b,\mathsf{train}} = \{\forall(\mathbf{x}_i, y_i) \in \mathcal{D}_{\mathsf{train}}, \|\mathbf{x}_i\|_2 \leq B_x, |y_i| \leq B_y\},$$
$$\mathcal{E}_{b,\mathsf{val}} = \{\forall(\mathbf{x}_i, y_i) \in \mathcal{D}_{\mathsf{val}}, \|\mathbf{x}_i\|_2 \leq B_x, |y_i| \leq B_y\},$$
$$\mathcal{E}_{b,N+1} = \{\|\mathbf{x}_{N+1}\|_2 \leq B_x, |y_{N+1}| \leq B_y\}.$$

For the good event $\mathcal{E} := \mathcal{E}_\pi \cap \mathcal{E}_w \cap \mathcal{E}_{b,\mathsf{train}} \cap \mathcal{E}_{b,\mathsf{test}} \cap \mathcal{E}_{b,N+1}$, we can show that $\mathbb{P}(\mathcal{E}^c) \leq \mathcal{O}(N^{-10})$. Further, by the proof of Lemma F.1 (see e.g. (34)), we know that $\max_{k\in[K]}\left\|\mathbf{w}_{\mathsf{ridge}}^{\lambda_k}(\mathcal{D}_{\mathsf{train}})\right\|_2 \leq B_w/2$ holds under the good event $\mathcal{E}$.

For the ridge $\lambda_k = \frac{d\sigma_k^2}{N_{\mathsf{train}}}$ and parameters $(\alpha, \beta, \gamma, \varepsilon)$, we consider the transformer $\boldsymbol{\theta}$ constructed in Theorem I.2, with a clipped prediction $\widehat{y}_{N+1} = \widetilde{\mathsf{read}}_{\mathsf{y}}(\mathrm{TF}_{\boldsymbol{\theta}}(\mathbf{H}))$.

In the following, we upper bound the quantity $\mathbb{E}_k(\widehat{y}_{N+1} - y_{N+1})^2$ for any fixed $k$. Similar to the proof of Corollary 6 (Appendix F.4), we decompose

$$\mathbb{E}_k(\widehat{y}_{N+1} - y_{N+1})^2 = \mathbb{E}_k\left[1\{\mathcal{E}\}(\widehat{y}_{N+1} - y_{N+1})^2\right] + \mathbb{E}_k\left[1\{\mathcal{E}^c\}(\widehat{y}_{N+1} - y_{N+1})^2\right],$$

and we analyze these two parts separately.

**Part I.** Recall that by our construction, when $\mathcal{E}$ holds, we have $\widehat{y}_{N+1} = \mathsf{clip}_{B_y}(\langle \widehat{\mathbf{w}}, \mathbf{x}_{N+1} \rangle)$ and the statements of Theorem I.2 hold for $\widehat{\mathbf{w}}$. Thus, we have

$$\mathbb{E}_k\left[1\{\mathcal{E}\}(\widehat{y}_{N+1} - y_{N+1})^2\right] = \mathbb{E}_k\left[1\{\mathcal{E}\}(\mathsf{clip}_{B_y}(\langle \mathbf{x}_{N+1}, \widehat{\mathbf{w}} \rangle) - y_{N+1})^2\right]$$

$$\leq \mathbb{E}_k\left[1\{\mathcal{E}\}(\langle \mathbf{x}_{N+1}, \widehat{\mathbf{w}} \rangle - y_{N+1})^2\right].$$

Let us consider the following risk functional

$$L_{\mathsf{val},\mathbf{w}_\star}(\mathbf{w}) = \mathbb{E}_{(\mathbf{x},y)\sim\mathsf{P}_{\mathbf{w}_\star,\sigma_k}}\left[\tfrac{1}{2}(\langle \mathbf{w}, \mathbf{x} \rangle - y)^2\right] = \tfrac{1}{2}\left(\|\mathbf{w} - \mathbf{w}_\star\|_2^2 + \sigma_k^2\right).$$

Then, under the good event $\mathcal{E}_0 := \mathcal{E}_\pi \cap \mathcal{E}_w \cap \mathcal{E}_{b,\mathsf{train}} \cap \mathcal{E}_{b,\mathsf{test}}$ of $(\mathbf{w}_\star, \mathcal{D})$,

$$\mathbb{E}_k\left[1\{\mathcal{E}\}(\langle \mathbf{x}_{N+1}, \widehat{\mathbf{w}} \rangle - y_{N+1})^2 \,\middle|\, \mathbf{w}_\star, \mathcal{D}\right] = \mathbb{E}_k\left[1\{\mathcal{E}\}(\langle \mathbf{x}_{N+1}, \widehat{\mathbf{w}}(\mathcal{D}) \rangle - y_{N+1})^2 \,\middle|\, \mathbf{w}_\star, \mathcal{D}\right]$$

$$\leq \mathbb{E}_k\left[(\langle \mathbf{x}_{N+1}, \widehat{\mathbf{w}}(\mathcal{D}) \rangle - y_{N+1})^2 \,\middle|\, \mathbf{w}_\star, \mathcal{D}\right]$$

$$= \mathbb{E}_{(\mathbf{x},y)\sim\mathsf{P}_{\mathbf{w}_\star,\sigma_k}}\left[(\langle \mathbf{x}_{N+1}, \widehat{\mathbf{w}}(\mathcal{D}) \rangle - y_{N+1})^2\right]$$

$$= L_{\mathsf{val},\mathbf{w}_\star}(\widehat{\mathbf{w}}(\mathcal{D})).$$

By our construction, under the good event $\mathcal{E}_0$, we have

$$L_{\mathsf{val},\mathbf{w}_\star}(\widehat{\mathbf{w}}(\mathcal{D})) \leq L_{\mathsf{val},\mathbf{w}_\star}(\widehat{\mathbf{w}}_k(\mathcal{D}_{\mathsf{train}})) + \max_{l\in[K]}\left|\widehat{L}_{\mathsf{val}}(\widehat{\mathbf{w}}_l(\mathcal{D}_{\mathsf{train}})) - L_{\mathsf{val},\mathbf{w}_\star}(\widehat{\mathbf{w}}_l(\mathcal{D}_{\mathsf{train}}))\right| + \gamma,$$

where $\left\|\widehat{\mathbf{w}}_l(\mathcal{D}_{\mathsf{train}})) - \mathbf{w}_{\mathrm{ridge}}^{\lambda_l}(\mathcal{D}_{\mathsf{train}})\right\|_2 \leq \varepsilon$ for each $l \in [K]$. Clearly,

$$2\mathbb{E}_k[1\{\mathcal{E}_0\}L_{\mathsf{val},\mathbf{w}_\star}(\widehat{\mathbf{w}}_k(\mathcal{D}_{\mathsf{train}}))] = \mathbb{E}_k\left[1\{\mathcal{E}_0\}\left(\|\widehat{\mathbf{w}}_k(\mathcal{D}_{\mathsf{train}}) - \mathbf{w}_\star\|_2^2 + \sigma_k^2\right)\right]$$

$$\leq \mathbb{E}_k\left[1\{\mathcal{E}_0\}\left(\left\|\mathbf{w}_{\mathrm{ridge}}^{\lambda_k}(\mathcal{D}_{\mathsf{train}}) - \mathbf{w}_\star\right\|_2^2 + 2\varepsilon\left\|\mathbf{w}_{\mathrm{ridge}}^{\lambda_k}(\mathcal{D}_{\mathsf{train}}) - \mathbf{w}_\star\right\|_2 + \varepsilon^2\right)\right] + \sigma_k^2$$

$$\leq \mathbb{E}_k\left[\left\|\mathbf{w}_{\mathrm{ridge}}^{\lambda_k}(\mathcal{D}_{\mathsf{train}}) - \mathbf{w}_\star\right\|_2^2 + 2\varepsilon\left\|\mathbf{w}_{\mathrm{ridge}}^{\lambda_k}(\mathcal{D}_{\mathsf{train}}) - \mathbf{w}_\star\right\|_2 + \varepsilon^2\right] + \sigma_k^2$$

$$\leq 2\mathsf{Risk}_{k,\mathsf{train}} + 2\varepsilon\sqrt{2\mathsf{Risk}_{k,\mathsf{train}}} + \varepsilon^2,$$

where we denote $2\mathsf{Risk}_{k,\mathsf{train}} = \mathbb{E}_k\left\|\mathbf{w}_{\mathrm{ridge}}^{\lambda_k}(\mathcal{D}_{\mathsf{train}}) - \mathbf{w}_\star\right\|_2^2 + \sigma_k^2$, and we also note that $\mathsf{Risk}_{k,\mathsf{train}} \leq 1 + \sigma_k^2$ by definition. By Lemma J.1, we have

$$\mathsf{Risk}_{k,\mathsf{train}} \leq \mathsf{BayesRisk}_k + \mathcal{O}\left((\sigma_k^2 + 1)\frac{N_{\mathsf{val}}}{N}\right).$$

We next deal with the term $\varepsilon_{\mathsf{val}} := \max_{l\in[K]}\left|\widehat{L}_{\mathsf{val}}(\widehat{\mathbf{w}}_l(\mathcal{D}_{\mathsf{train}})) - L_{\mathsf{val},\mathbf{w}_\star}(\widehat{\mathbf{w}}_l(\mathcal{D}_{\mathsf{train}}))\right|$. Note that for the good event $\mathcal{E}_{\mathsf{train}} := \mathcal{E}_\pi \cap \mathcal{E}_w \cap \mathcal{E}_{b,\mathsf{train}}$ of $(\mathbf{w}_\star, \mathcal{D}_{\mathsf{train}})$, we have

$$\mathbb{E}_k[1\{\mathcal{E}_0\}\varepsilon_{\mathsf{val}}] \leq \mathbb{E}_k[1\{\mathcal{E}_{\mathsf{train}}\}\varepsilon_{\mathsf{val}}] \leq \mathbb{E}_{\mathbf{w}_\star,\mathcal{D}_{\mathsf{train}}\sim\mathbb{P}_k}[1\{\mathcal{E}_{\mathsf{train}}\} \cdot \mathbb{E}_{\mathcal{D}_{\mathsf{val}}}[\varepsilon_{\mathsf{val}}| \mathbf{w}_\star, \mathcal{D}_{\mathsf{train}}]].$$

Thus, Lemma J.2 yields

$$\mathbb{E}_k[1\{\mathcal{E}_0\}\varepsilon_{\mathsf{val}}] \leq \mathcal{O}\left(B_w^2\right) \cdot \left[\sqrt{\frac{\log K}{N_{\mathsf{val}}}} + \frac{\log K}{N_{\mathsf{val}}}\right].$$

Therefore, we can conclude that

$$\mathbb{E}_k\left[1\{\mathcal{E}\}(\widehat{y}_{N+1} - y_{N+1})^2\right] \leq 2\mathsf{BayesRisk}_k + \mathcal{O}\left(\varepsilon\sigma_{\max} + \varepsilon^2 + \frac{\sigma_{\max}^2 N_{\mathsf{val}}}{N} + B_w^2\sqrt{\frac{\log K}{N_{\mathsf{val}}}} + \frac{B_w^2 \log K}{N_{\mathsf{val}}}\right).$$

Therefore, we can choose $(\varepsilon, N_{\mathsf{val}})$ so that $N_{\mathsf{val}} \leq N/2$ as

$$N_{\mathsf{val}} = \max\left\{\left(\frac{B_w^2}{\sigma_{\max}^2}N\right)^{2/3}\log^{1/3}(K), \log K\right\}, \qquad \varepsilon = \frac{\sigma_{\max}}{N}.$$

It is worth noting that such choice of $N_{\mathsf{val}}$ is feasible as long as $N \gtrsim \frac{B_w^4}{\sigma_{\max}^4}\log K$. Under such choice, we obtain

$$\frac{1}{2}\mathbb{E}_k\left[1\{\mathcal{E}\}(\widehat{y}_{N+1} - y_{N+1})^2\right] \leq \mathsf{BayesRisk}_k + \mathcal{O}\left(\sigma_{\max}^{4/3}B_w^{2/3}\left(\frac{\log K}{N}\right)^{1/3}\right).$$

**Part II.** Similar to the proof of Corollary 6, we have

$$\mathbb{E}\big[1\{\mathcal{E}^c\}(\widehat{y}_{N+1} - y_{N+1})^2\big] \leq \mathcal{O}\left(\frac{B_y^2}{N^5}\right) \leq \mathcal{O}\left(\frac{\sigma_{\max}^2}{N^4}\right).$$

**Conclusion.** Combining the both cases, we obtain

$$\mathbb{E}_k\left[\tfrac{1}{2}(y_{N+1} - \widehat{y}_{N+1})^2\right] \leq \mathsf{BayesRisk}_k + \mathcal{O}\left(\sigma_{\max}^{4/3} B_w^{2/3}\left(\tfrac{\log K}{N}\right)^{1/3}\right)$$

$$\leq \mathsf{BayesRisk}_k + \mathcal{O}\left(\frac{\sigma_{\max}^2}{\sigma_{\min}^{2/3}}\left(\tfrac{\log K}{N}\right)^{1/3} + \sigma_{\max}^{4/3}\frac{\log^{2/3}(N)\log^{1/3}(K)}{d^{2/3} N^{1/3}}\right)$$

$$\leq \mathsf{BayesRisk}_k + \widetilde{\mathcal{O}}\left(\frac{\sigma_{\max}^2}{\sigma_{\min}^{2/3}}\left(\tfrac{\log K}{N}\right)^{1/3}\right),$$

where we plug in our choice of $B_y$. The bounds on $M^{(\ell)}, D^{(\ell)}$ and $\|\boldsymbol{\theta}\|$ follows immediately from Theorem I.2. This completes the proof. □

## J.2 Derivation of the exact Bayes predictor

Let $(\mathcal{D}, \mathbf{x}_{N+1}, y_{N+1})$ be $(N+1)$ observations from the data generating model $\pi$ considered in Section 4.1.1. On observing $(\mathcal{D}, \mathbf{x}_{N+1})$, the Bayes predictor of $y_{N+1}$ is given by its posterior mean:

$$\mathbb{E}_\pi[y_{N+1}|\mathcal{D}, \mathbf{x}_{N+1}] = \mathbb{E}_\pi[\langle \mathbf{x}_{N+1}, \mathbf{w}_\star\rangle + \varepsilon_{N+1}|\mathcal{D}, \mathbf{x}_{N+1}] = \langle \mathbf{x}_{N+1}, \mathbb{E}_\pi[\mathbf{w}_\star|\mathcal{D}]\rangle.$$

It thus remains to derive $\mathbb{E}_\pi[\mathbf{w}_\star|\mathcal{D}]$. Recall that our data generating model is given by $k \sim \Lambda$, By Bayes' rule, we have

$$\mathbb{E}_\pi[\mathbf{w}_\star|\mathcal{D}] = \sum_{k'\in[K]} \mathbb{P}_\pi(k = k'|\mathcal{D}) \cdot \mathbb{E}_\pi[\mathbf{w}_\star|\mathcal{D}, k = k']. \tag{46}$$

On $k = k'$, the data is generated from the noisy linear model $\mathbf{w}_\star \sim \mathsf{N}(\mathbf{0}, \mathbf{I}_d/d)$, and $\mathbf{y} = \mathbf{X}\mathbf{w}_\star + \boldsymbol{\varepsilon}$ where $\varepsilon_i \overset{\text{iid}}{\sim} \mathsf{N}(0, \sigma_{k'}^2)$. It is a standard result that $\mathbb{E}_\pi[\mathbf{w}_\star|\mathcal{D}, k = k']$ is given by the ridge estimator

$$\mathbb{E}_\pi[\mathbf{w}_\star|\mathcal{D}, k = k'] = \underbrace{\left(\mathbf{X}^\top\mathbf{X} + d\sigma_{k'}^2\right)^{-1}}_{\widehat{\boldsymbol{\Sigma}}_{k'}^{-1}} \mathbf{X}^\top\mathbf{y} =: \widehat{\mathbf{w}}_{k'}$$

$$= \left(\frac{\mathbf{X}^\top\mathbf{X}}{N} + \frac{d\sigma_{k'}^2}{N}\right)^{-1}\frac{\mathbf{X}^\top\mathbf{y}}{N}.$$

(Note that the sample covariance within $\widehat{\boldsymbol{\Sigma}}_{k'}$ is not normalized by $N$, which is not to be confused with remaining parts within the paper.) Therefore, the posterior mean (46) is exactly a weighted combination of $K$ ridge regression estimators, each with regularization $d\sigma_k^2/N$.

It remains to derive the mixing weights $\mathbb{P}_\pi(k = k'|\mathcal{D})$ for all $k' \in [K]$. By Bayes' rule, we have

$$\mathbb{P}_\pi(k = k'|\mathcal{D}) \propto_{k'} \mathbb{P}_\pi(k = k') \cdot \int_{\mathbf{w}_\star} p(\mathbf{w}_\star) \cdot \mathsf{p}_{k',\mathbf{w}_\star}(\mathcal{D}|\mathbf{w}_\star)d\mathbf{w}_\star$$

$$\propto \Lambda_{k'} \cdot \int_{\mathbf{w}} \frac{1}{(2\pi d)^{d/2}(2\pi\sigma_{k'}^2)^{N/2}} \exp\left(-\frac{d\|\mathbf{w}\|_2^2}{2} - \frac{\|\mathbf{X}\mathbf{w} - \mathbf{y}\|_2^2}{2\sigma_{k'}^2}\right)d\mathbf{w}$$

$$\propto \Lambda_{k'} \cdot \int_{\mathbf{w}} \frac{1}{(2\pi\sigma_{k'}^2)^{N/2}} \exp\left(-\frac{1}{2}\mathbf{w}^\top\left(\frac{\mathbf{X}^\top\mathbf{X}}{\sigma_{k'}^2} + d\mathbf{I}_d\right)\mathbf{w} + \left\langle\mathbf{w}, \frac{\mathbf{X}^\top\mathbf{y}}{\sigma_{k'}^2}\right\rangle - \frac{\|\mathbf{y}\|_2^2}{2\sigma_{k'}^2}\right)d\mathbf{w}$$

$$\propto \Lambda_{k'} \cdot \int_{\mathbf{w}} \frac{1}{(2\pi\sigma_{k'}^2)^{N/2}} \exp\left(-\frac{1}{2\sigma_{k'}^2}(\mathbf{w} - \widehat{\mathbf{w}}_{k'})^\top\widehat{\boldsymbol{\Sigma}}_{k'}(\mathbf{w} - \widehat{\mathbf{w}}_{k'}) - \frac{1}{2\sigma_{k'}^2}\left(\|\mathbf{y}\|_2^2 - \mathbf{y}^\top\mathbf{X}\widehat{\boldsymbol{\Sigma}}_{k'}^{-1}\mathbf{X}^\top\mathbf{y}\right)\right)d\mathbf{w}$$

$$\propto \Lambda_{k'} \cdot \frac{\det(\widehat{\boldsymbol{\Sigma}}_{k'}/\sigma_{k'}^2)^{-1/2}}{\sigma_{k'}^N} \exp\left(-\frac{1}{2\sigma_{k'}^2}\left(\|\mathbf{y}\|_2^2 - \mathbf{y}^\top\mathbf{X}\widehat{\boldsymbol{\Sigma}}_{k'}^{-1}\mathbf{X}^\top\mathbf{y}\right)\right)$$

$$\propto \Lambda_{k'} \cdot \frac{1}{\sigma_{k'}^{N-d}\det(\mathbf{X}^\top\mathbf{X} + d\sigma_{k'}^2\mathbf{I}_d)^{1/2}} \exp\left(-\frac{1}{2\sigma_{k'}^2}\left(\|\mathbf{y}\|_2^2 - \langle\mathbf{y}, \mathbf{X}\widehat{\mathbf{w}}_{k'}\rangle\right)\right).$$

Note that such mixing weights involve the determinant of the matrix $\widehat{\mathbf{\Sigma}}_{k'} = \mathbf{X}^\top\mathbf{X} + d\sigma_{k'}^2\mathbf{I}_d$, which depends on the data $\mathbf{X}$ in a non-trivial fashion; Any transformer has to approximate these weights if their mechanism is to directly approximate the exact Bayesian predictor (46).

### J.3  Useful lemmas

**Lemma J.1.** *For* $2\mathsf{Risk}_{k,\text{train}} = \mathbb{E}_k\big\|\mathbf{w}_{\text{ridge}}^{\lambda_k}(\mathcal{D}_{\text{train}}) - \mathbf{w}_\star\big\|_2^2 + \sigma_k^2$, *there exists universal constant* $C$ *such that*

$$\mathsf{Risk}_{k,\text{train}} \leq \mathsf{BayesRisk}_k + C(\sigma_k^2 + 1)\frac{N_{\text{val}}}{N}.$$

*Proof.* Recall that under $\mathbb{P}_k$, we have

$$\mathbf{w}_\star \sim \mathsf{N}(0, \mathbf{I}_d/d), \qquad y_i = \langle\mathbf{x}_i, \mathbf{w}_\star\rangle + \varepsilon_i, \qquad \varepsilon_i \sim \mathsf{N}(0, \sigma^2).$$

We denote $\mathbf{y}_t = [y_i]_{i\in\mathcal{I}_{\text{train}}}$, then by definition $\mathbf{w}_{\text{ridge}}^{\lambda_k}(\mathcal{D}_{\text{train}}) = (\mathbf{X}_{\text{train}}^\top\mathbf{X}_{\text{train}} + d\sigma_k^2)^{-1}\mathbf{X}_{\text{train}}\mathbf{y}_t$ (with $\lambda_k = d\sigma_k^2/N_{\text{train}}$). Thus, a simple calculation yields

$$2\mathsf{Risk}_{k,\text{train}} = \mathbb{E}_k\big\|\mathbf{w}_{\text{ridge}}^{\lambda_k}(\mathcal{D}_{\text{train}}) - \mathbf{w}_\star\big\|_2^2 + \sigma_k^2 = \sigma_k^2\mathbb{E}\mathrm{tr}\big((\mathbf{X}_{\text{train}}^\top\mathbf{X}_{\text{train}} + d\sigma_k^2)^{-1}\big) + \sigma_k^2,$$

and analogously, $2\mathsf{BayesRisk}_k = \sigma_k^2\mathbb{E}\mathrm{tr}\big((\mathbf{X}^\top\mathbf{X} + d\sigma_k^2\mathbf{I}_d)^{-1}\big) + \sigma_k^2$. Therefore,

$$2\mathsf{Risk}_{k,\text{train}} - 2\mathsf{BayesRisk}_k = \sigma_k^2\mathbb{E}\mathrm{tr}\big((\mathbf{X}_{\text{train}}^\top\mathbf{X}_{\text{train}} + d\sigma_k^2\mathbf{I}_d)^{-1}\big) - \sigma_k^2\mathbb{E}\mathrm{tr}\big((\mathbf{X}^\top\mathbf{X} + d\sigma_k^2\mathbf{I}_d)^{-1}\big)$$
$$\leq \sigma_k^2 N_{\text{val}}\mathbb{E}_k[\lambda_{\min}(\mathbf{\Sigma})^{-1}],$$

where in the above inequality we denote $\mathbf{\Sigma} := \mathbf{X}_{\text{train}}^\top\mathbf{X}_{\text{train}} + d\sigma_k^2\mathbf{I}_d$ and use the following fact:

$$\mathrm{tr}\big(\mathbf{\Sigma}^{-1}\big) - \mathrm{tr}\big((\mathbf{\Sigma} + \mathbf{X}_v^\top\mathbf{X}_v)^{-1}\big) = \mathrm{tr}\Big(\mathbf{\Sigma}^{-1/2}(\mathbf{I}_d - (\mathbf{I}_d + \mathbf{\Sigma}^{-1/2}\mathbf{X}_v^\top\mathbf{X}_v\mathbf{\Sigma}^{-1/2})^{-1})\mathbf{\Sigma}^{-1/2}\Big)$$
$$= \mathrm{tr}\Big(\mathbf{\Sigma}^{-1/2}(\mathbf{I}_d + \mathbf{\Sigma}^{-1/2}\mathbf{X}_v^\top\mathbf{X}_v\mathbf{\Sigma}^{-1/2})^{-1}\mathbf{\Sigma}^{-1/2}\mathbf{X}_v^\top\mathbf{X}_v\mathbf{\Sigma}^{-1}\Big)$$
$$= \Big\langle(\mathbf{I}_d + \mathbf{\Sigma}^{-1/2}\mathbf{X}_v^\top\mathbf{X}_v\mathbf{\Sigma}^{-1/2})^{-1}\mathbf{\Sigma}^{-1/2}\mathbf{X}_v^\top\mathbf{X}_v\mathbf{\Sigma}^{-1/2}, \mathbf{\Sigma}^{-1}\Big\rangle$$
$$\leq \mathrm{rank}(\mathbf{\Sigma}^{-1/2}\mathbf{X}_v^\top\mathbf{X}_v\mathbf{\Sigma}^{-1/2})\lambda_{\max}(\mathbf{\Sigma}^{-1}) \leq N_{\text{val}}\lambda_{\min}(\mathbf{\Sigma})^{-1}.$$

**Case 1.** We first suppose that $N_{\text{train}} \leq 16d$. Then by definition $\mathbf{\Sigma} \succeq d\sigma_k^2\mathbf{I}_d$, and hence

$$\sigma_k^2 N_{\text{val}}\mathbb{E}_k[\lambda_{\min}(\mathbf{\Sigma})^{-1}] \leq \frac{\sigma_k^2 N_{\text{val}}}{d\sigma_k^2} \leq \frac{16N_{\text{val}}}{N_{\text{train}}} \leq \frac{32N_{\text{val}}}{N}.$$

**Case 2.** When $N_{\text{train}} \geq 9d$, then we consider the event $\mathcal{E}_t := \{\lambda_{\min}(\mathbf{X}_{\text{train}}^\top\mathbf{X}_{\text{train}}/N_{\text{train}}) \geq \frac{1}{16}\}$. By Lemma B.2 we have $\mathbb{P}(\mathcal{E}_t^c) \leq \exp(-N_{\text{train}}/8)$. Therefore,

$$\sigma_k^2 N_{\text{val}}\mathbb{E}_k[\lambda_{\min}(\mathbf{\Sigma})^{-1}] = \sigma_k^2 N_{\text{val}}\mathbb{E}_k[1\{\mathcal{E}_t\}\lambda_{\min}(\mathbf{\Sigma})^{-1}] + \sigma_k^2 N_{\text{val}}\mathbb{E}_k[1\{\mathcal{E}_t^c\}\lambda_{\min}(\mathbf{\Sigma})^{-1}]$$
$$\leq \frac{16\sigma_k^2 N_{\text{val}}}{N_{\text{train}}} \cdot \mathbb{P}(\mathcal{E}_t) + \frac{N_{\text{val}}}{d} \cdot \mathbb{P}(\mathcal{E}_t^c)$$
$$\leq \frac{32\sigma_k^2 N_{\text{val}}}{N} + \frac{N_{\text{val}}}{d} \cdot \exp(-N/16) = \mathcal{O}\left(\frac{(\sigma_k^2 + 1)N_{\text{val}}}{N}\right).$$

Combining these two cases finishes the proof. $\qquad\square$

**Lemma J.2.** *Condition on the event* $\mathcal{E}_{\text{train}}$, *we have*

$$\mathbb{E}_{\mathcal{D}_{\text{val}}\sim\mathbb{P}_k|\mathbf{w}_\star,\mathcal{D}_{\text{train}}}\left[\max_{l\in[K]}\left|\widehat{L}_{\text{val}}(\widehat{\mathbf{w}}_l) - L_{\text{val},\mathbf{w}_\star}(\widehat{\mathbf{w}}_l)\right|\right] \leq CB_w^2\left[\frac{\log(2K)}{N_{\text{val}}} + \sqrt{\frac{\log(2K)}{N_{\text{val}}}}\right],$$

*where we denote* $\widehat{\mathbf{w}}_l = \widehat{\mathbf{w}}_l(\mathcal{D}_{\text{train}})$.

*Proof.* We only need to work with a fixed pair of $(\mathbf{w}_\star, \mathcal{D}_{\text{train}})$ such that $\mathcal{E}_{\text{train}}$ holds. Hence, in the following we only consider the randomness of $\mathcal{D}_{\text{val}}$ conditional on such a $(\mathbf{w}_\star, \mathcal{D}_{\text{train}})$.

Recall that for any $\mathbf{w}$,

$$\widehat{L}_{\text{val}}(\mathbf{w}) = \frac{1}{2\left|\mathcal{D}_{\text{val}}\right|} \sum_{(\mathbf{x}_i, y_i) \in \mathcal{D}_{\text{val}}} \left(\langle \mathbf{x}_i, \mathbf{w} \rangle - y_i\right)^2,$$

and we have $\mathbb{E}_{\mathcal{D}_v}[\widehat{L}_{\text{val}}(\mathbf{w})] = L_{\text{val}, \mathbf{w}_\star}(\mathbf{w})$. For each $i \in \mathcal{I}_{\text{val}}$,

$$y_i - \langle \mathbf{x}_i, \widehat{\mathbf{w}}_l \rangle = \varepsilon_i - \langle \mathbf{x}_i, \mathbf{w}_\star - \widehat{\mathbf{w}}_l \rangle \sim \text{SG}(\sigma_k^2 + \|\mathbf{w}_\star - \widehat{\mathbf{w}}_l\|^2).$$

Note that under $\mathcal{E}_{\text{train}}$, we have $\widehat{\mathbf{w}}_l \in \mathsf{B}_2(B_w)$ for all $l \in [K]$, and hence $\sigma_k^2 + \|\mathbf{w}_\star - \widehat{\mathbf{w}}_l\|^2 \leq 5B_w^2$. We then have $(y_i - \langle \mathbf{x}_i, \widehat{\mathbf{w}}_l \rangle)^2$'s are (conditional) i.i.d random variables in $\text{SE}(CB_w^4)$. Then, by Bernstein's inequality, we have

$$\mathbb{P}_{\mathcal{D}_{\text{val}}}\left(\left|\widehat{L}_{\text{val}}(\widehat{\mathbf{w}}_l) - L_{\text{val}, \mathbf{w}_\star}(\widehat{\mathbf{w}}_l)\right| \geq t\right) \leq 2\exp\left(-cN_{\text{val}} \min\left\{\frac{t^2}{B_w^2}, \frac{t}{B_w}\right\}\right),$$

where $c$ is a universal constant. Applying the union bound, we obtain

$$\mathbb{P}_{\mathcal{D}_{\text{val}}}\left(\max_{l \in [K]}\left|\widehat{L}_{\text{val}}(\widehat{\mathbf{w}}_l) - L_{\text{val}, \mathbf{w}_\star}(\widehat{\mathbf{w}}_l)\right| \geq t\right) \leq K\exp\left(-cN_{\text{val}} \min\left\{\frac{t^2}{B_w^2}, \frac{t}{B_w}\right\}\right).$$

Taking integration completes the proof. $\qquad\square$

### J.4 Generalized linear models with adaptive link function selection

Suppose that $(g_k : \mathbb{R} \to \mathbb{R})_{k \in [K]}$ is a set of link functions such that $g_k$ is non-decreasing and $C^2$-smooth for each $k \in [K]$. We consider the input format we introduce in Section 4.1 with $|\mathcal{D}_{\text{train}}| = \lceil N/2 \rceil, |\mathcal{D}_{\text{val}}| = \lfloor N/2 \rfloor$.

**Theorem J.2** (GLMs with adaptive link function selection). *For any fixed set of parameters defined in Assumption B, as long as $N \geq \mathcal{O}(d)$, there exists a transformer $\boldsymbol{\theta}$ with $L \leq \mathcal{O}(\log(N))$ layers, input dimension $D = \Theta(dK)$ and $\max_{\ell \in [L]} M^{(\ell)} \leq \widetilde{\mathcal{O}}(d^3 N)$, such that the following holds.*

*For any $k^\star \in [K]$ and any distribution $\mathsf{P}$ that is a generalized linear model of the link function $g_{k^\star}$ and some parameter $\boldsymbol{\beta}$, if Assumption B holds for each pair $(\mathsf{P}, g_k)$, then*

$$\mathbb{E}_{(\mathcal{D}, \mathbf{x}_{N+1}, y_{N+1}) \sim \mathsf{P}}\left[(\widehat{y}_{N+1} - y_{N+1})^2\right] \leq \mathbb{E}_{(\mathbf{x}, y) \sim \mathsf{P}}\left[(g_{k^\star}(\langle \mathbf{x}, \boldsymbol{\beta} \rangle) - y)^2\right] + \mathcal{O}\left(\frac{d}{N} + \sqrt{\frac{\log(K)}{N}}\right),$$

*or equivalently, $\mathbb{E}_{(\mathcal{D}, \mathbf{x}_{N+1}) \sim \mathsf{P}}[(\widehat{y}_{N+1} - \mathbb{E}[y_{N+1}|\mathbf{x}_{N+1}])^2] \leq \mathcal{O}\left(d/N + \sqrt{\log(K)/N}\right)$.*

*Proof.* For each $k \in [K]$, we consider optimizing the following training loss:

$$\mathbf{w}_{\text{GLM}}^{(k)} := \arg\min_{\mathbf{w}} \widehat{L}_{\text{train}}^{(k)}(\mathbf{w}) := \frac{1}{N_{\text{train}}} \sum_{(\mathbf{x}_i, y_i) \in \mathcal{D}_{\text{train}}} \ell_k(\langle \mathbf{x}_i, \mathbf{w} \rangle, y_i),$$

where $\ell_k(t, y) := -yt + \int_0^t g_k(s)ds$ is the convex (integral) loss associated with $g_k$ (as in Section 3.1).

Also, for each predictor $f : \mathbb{R}^d \to \mathbb{R}$, we consider the squared validation loss $\widehat{L}_{\text{val}}$:

$$\widehat{L}_{\text{val}}(f) := \frac{1}{2N_{\text{val}}} \sum_{(\mathbf{x}_i, y_i) \in \mathcal{D}_{\text{val}}} (f(\mathbf{x}_i) - y_i)^2.$$

Fix a large universal constant $C_0$. Let us set

$$\alpha = \mu_g \mu_x / 8, \qquad \beta = 8L_g K_x,$$

$$B_x = C_0 K_x \sqrt{d \log(N)}, \qquad B_y = C_0 K_y \sqrt{\log(N)},$$

Then, we define good events similarly to the proof of Corollary 6 (Appendix F.4):

$$\mathcal{E}_w = \left\{ \forall k \in [K], \ \forall \mathbf{w} \in \mathsf{B}_2(B_w), \ \alpha \leq \lambda_{\min}(\nabla^2 \widehat{L}_{\mathsf{train}}^{(k)}(\mathbf{w})) \leq \lambda_{\max}(\nabla^2 \widehat{L}_{\mathsf{train}}^{(k)}(\mathbf{w})) \leq \beta, \right\},$$

$$\mathcal{E}_r = \left\{ \forall k \in [K], \big\| \mathbf{w}_{\mathsf{GLM}}^{(k)} \big\|_2 \leq B_w/2 \right\},$$

$$\mathcal{E}_{b,\mathsf{train}} = \{ \forall (\mathbf{x}_i, y_i) \in \mathcal{D}_{\mathsf{train}}, \|\mathbf{x}_i\|_2 \leq B_x, |y_i| \leq B_y \},$$

$$\mathcal{E}_{b,\mathsf{val}} = \{ \forall (\mathbf{x}_i, y_i) \in \mathcal{D}_{\mathsf{val}}, \|\mathbf{x}_i\|_2 \leq B_x, |y_i| \leq B_y \},$$

$$\mathcal{E}_{b,N+1} = \{ \|\mathbf{x}_{N+1}\|_2 \leq B_x, |y_{N+1}| \leq B_y \}.$$

Similar to the proof of Theorem G.2 (Appendix G.2), we know the good event $\mathcal{E} := \mathcal{E}_w \cap \mathcal{E}_r \cap \mathcal{E}_{b,\mathsf{train}} \cap \mathcal{E}_{b,\mathsf{test}} \cap \mathcal{E}_{b,N+1}$ holds with high probability: $\mathsf{P}(\mathcal{E}^c) \leq \mathcal{O}\left(N^{-10}\right)$.

Similar to the proof of Theorem I.2, we can show that there exists a transformer $\boldsymbol{\theta}$ with prediction $\widehat{y}_{N+1} = \widetilde{\mathsf{read}}_y(\mathrm{TF}_{\boldsymbol{\theta}}(\mathbf{H}))$ (clipped by $B_y$), such that (for any P) the following holds under $\mathcal{E}$:

(a) For each $k \in [K]$, $f_k = \mathcal{A}_k(\mathcal{D}_{\mathsf{train}})$ is a predictor such that $\left| f_k(\mathbf{x}_i) - g_k(\langle \mathbf{x}_i, \mathbf{w}_{\mathsf{GLM}}^{(k)} \rangle) \right| \leq \varepsilon$ for all $i \in [N+1]$ (where $\varepsilon$ is chosen as in Appendix G.2).

(b) $\widehat{y}_{N+1} = \mathsf{clip}_{B_y}(\widehat{f}(\mathbf{x}_{N+1}))$, where $\widehat{f} = \mathcal{A}_{\mathsf{TF}}(\mathcal{D})$ is an aggregated predictor given by $\widehat{f} = \sum_k \lambda_k f_k$, such that $(\lambda_k)$ is a distribution supported on $k \in [K]$ such that $\widehat{L}_{\mathsf{val}}(f_k) \leq \min_{k' \in [K]} \widehat{L}_{\mathsf{val}}(f_{k'}) + \gamma$.

Similar to the proof of Theorem G.2, for $\mathcal{E}_0 := \mathcal{E}_w \cap \mathcal{E}_r \cap \mathcal{E}_{b,\mathsf{train}} \cap \mathcal{E}_{b,\mathsf{test}}$, we have

$$\mathbb{E}_{(\mathcal{D},\mathbf{x}_{N+1},y_{N+1})\sim\mathsf{P}}(\widehat{y}_{N+1} - y_{N+1})^2 \leq \mathbb{E}_{\mathcal{D}\sim\mathsf{P}}\left[ \mathbb{1}\{\mathcal{E}_0\} L_{\mathsf{val}}(\widehat{f}) \right] + \mathcal{O}\left( \frac{B_y^2}{N^5} \right),$$

where we denote $L_{\mathsf{val}}(f) := \mathbb{E}_{(\mathbf{x},y)\sim\mathsf{P}}\left[ \mathbb{1}\{\|\mathbf{x}\|_2 \leq B_x\}(f(\mathbf{x}) - y)^2 \right]$ for each predictor $f$. By the definition of $\widehat{f}$, we then have (under $\mathcal{E}_0$)

$$L_{\mathsf{val}}(\widehat{f}) \leq L_{\mathsf{val}}(f_{k^\star}) + \max_l \left| \widehat{L}_{\mathsf{val}}(f_l) - L_{\mathsf{val}}(f_l) \right| + \gamma.$$

For the first term, repeating the argument in the proof of Theorem G.2 directly yields that for $\mathcal{E}_{\mathsf{train}} := \mathcal{E}_w \cap \mathcal{E}_r \cap \mathcal{E}_{b,\mathsf{train}}$,

$$\mathbb{E}_{\mathcal{D}_{\mathsf{train}}\sim\mathsf{P}}[\mathbb{1}\{\mathcal{E}_{\mathsf{train}}\} L_{\mathsf{val}}(f_{k^\star})] \leq \mathbb{E}_{(\mathbf{x},y)\sim\mathsf{P}}(g_{k^\star}(\langle \mathbf{x}, \boldsymbol{\beta} \rangle) - y)^2 + \mathcal{O}\left( d/N_{\mathsf{train}} \right).$$

For the second term, similar to Lemma J.2, we can show that conditional on $\mathcal{D}_{\mathsf{train}}$ such that $\mathcal{E}_{\mathsf{train}}$ holds, it holds

$$\mathbb{E}_{\mathcal{D}_{\mathsf{val}}\sim\mathsf{P}|\mathcal{D}_{\mathsf{train}}}\left[ \mathbb{1}\{\mathcal{E}_0\} \max_l \left| \widehat{L}_{\mathsf{val}}(f_l) - L_{\mathsf{val}}(f_l) \right| \right] \leq \mathcal{O}\left( K_y^2 \right) \cdot \left( \sqrt{\frac{\log K}{N_{\mathsf{val}}}} + \frac{\log K}{N_{\mathsf{val}}} \right).$$

Combining these inequalities and suitably choosing $\gamma$ complete the proof. $\qquad\square$

# K   Analysis of pretraining

Thus far, we have established the existence of transformers for performing various ICL tasks with good in-context statistical performance. We now analyze the sample complexity of pretraining these transformers from a finite number of training ICL instances.

## K.1   Generalization guarantee for pretraining

**Setup**   At pretraining time, each training ICL instance has form $\mathbf{Z} := (\mathbf{H}, y_{N+1})$, where $\mathbf{H} := \mathbf{H}(\mathcal{D}, \mathbf{x}_{N+1}) \in \mathbb{R}^{D \times (N+1)}$ denote the input sequence formatted as in (3). We consider the square loss between the in-context prediction and the ground truth label:

$$\ell_{\mathsf{icl}}(\boldsymbol{\theta}; \mathbf{Z}) := \frac{1}{2}\left( y_{N+1} - \underbrace{\mathsf{clip}_{B_y}\left( \mathsf{read}_y(\mathrm{TF}_{\boldsymbol{\theta}}^R(\mathbf{H})) \right)}_{\widetilde{\mathsf{read}}_y} \right)^2.$$

Above, $\mathsf{clip}_{B_y}(t) := \max\left\{\min\left\{t, B_y\right\}, -B_y\right\}$ is the standard clipping operator onto $[-B_y, B_y]$, and $\mathrm{TF}_{\boldsymbol{\theta}}^R$ the transformer architecture as in Definition 3 with clipping operators after each layer: let $\mathbf{H}^{(0)} = \mathsf{clip}_{\mathsf{R}}(\mathbf{H})$,

$$\mathbf{H}^{(\ell)} = \mathsf{clip}_{\mathsf{R}}\left(\mathrm{MLP}_{\boldsymbol{\theta}_{\mathtt{mlp}}^{(\ell)}}\left(\mathrm{Attn}_{\boldsymbol{\theta}_{\mathtt{attn}}^{(\ell)}}\left(\mathbf{H}^{(\ell-1)}\right)\right)\right) \text{ for all } \ell \in [L], \quad \mathsf{clip}_{\mathsf{R}}(\mathbf{H}) := [\mathrm{Proj}_{\|\mathbf{h}\|_2 \le \mathsf{R}}(\mathbf{h}_i)]_i.$$

The clipping operator is used to control the Lipschitz constant of $\mathrm{TF}_{\boldsymbol{\theta}}$ with respect to $\boldsymbol{\theta}$, and we typically choose a sufficiently large clipping radius $\mathsf{R}$ so that it does not modify the behavior of the transformer on any input sequence of our concern.

We draw ICL instances $\mathbf{Z} := (\mathbf{H}, y_{N+1}) = (\mathcal{D}, (\mathbf{x}_{N+1}, y_{N+1}))$ from a (meta-)distribution denoted as $\pi$, which first sample an in-context data distribution $\mathsf{P} \sim \pi$, then sample iid examples $(\mathbf{x}_i, y_i)_{i=1}^{N+1} \overset{\mathrm{iid}}{\sim} \mathsf{P}^{\otimes(N+1)}$ and form $\mathcal{D} = \{(\mathbf{x}_i, y_i)\}_{i \in [N]}$. Our pretraining loss is the average ICL loss on $n$ pretraining instances $\mathbf{Z}^{(1:n)} \overset{\mathrm{iid}}{\sim} \pi$, and we consider the corresponding test ICL loss on a new test instance:

$$\widehat{L}_{\mathsf{icl}}(\boldsymbol{\theta}) := \frac{1}{n}\sum_{j=1}^{n} \ell_{\mathsf{icl}}(\boldsymbol{\theta}; \mathbf{Z}^j), \quad L_{\mathsf{icl}}(\boldsymbol{\theta}) := \mathbb{E}_{\mathsf{P} \sim \pi, \mathbf{Z} \sim \mathsf{P}^{\otimes(N+1)}}[\ell_{\mathsf{icl}}(\boldsymbol{\theta}; \mathbf{Z})].$$

Our pretraining algorithm is to solve a standard constrained empirical risk minimization (ERM) problem over transformers with $L$ layers, $M$ heads, and norm bound $B$ (recall the definition of the $\|\cdot\|$ norm in (2)):

$$\widehat{\boldsymbol{\theta}} := \underset{\boldsymbol{\theta} \in \Theta_{L,M,D',B}}{\arg\min} \; \widehat{L}_{\mathsf{icl}}(\boldsymbol{\theta}),$$

$$\Theta_{L,M,D',B} := \left\{\boldsymbol{\theta} = (\boldsymbol{\theta}_{\mathtt{attn}}^{(1:L)}, \boldsymbol{\theta}_{\mathtt{mlp}}^{(1:L)}) : \max_{\ell \in [L]} M^{(\ell)} \le M, \; \max_{\ell \in [L]} D^{(\ell)} \le D', \; \|\boldsymbol{\theta}\| \le B\right\}. \tag{TF-ERM}$$

**Generalization guarantee** By standard uniform concentration analysis via chaining arguments (Proposition B.4; see also [87, Chapter 5] for similar arguments), we have the following excess loss guarantee for (TF-ERM). The proof can be found in Appendix L.2.

**Theorem K.1** (Generalization for pretraining). *With probability at least $1 - \xi$ (over the pretraining instances $\{\mathbf{Z}^j\}_{j \in [n]}$), the solution $\widehat{\boldsymbol{\theta}}$ to (TF-ERM) satisfies*

$$L_{\mathsf{icl}}(\widehat{\boldsymbol{\theta}}) \le \inf_{\boldsymbol{\theta} \in \Theta_{L,M,D',B}} L_{\mathsf{icl}}(\boldsymbol{\theta}) + \mathcal{O}\left(B_y^2 \sqrt{\frac{L^2(MD^2 + DD')\iota + \log(1/\xi)}{n}}\right),$$

*where $\iota = \log(2 + \max\{B, \mathsf{R}, B_y\})$ is a log factor.*

## K.2 Examples of pretraining for in-context regression problems

In Theorem K.1, the comparator $\inf_{\boldsymbol{\theta} \in \Theta_{L,M,D',B}} L_{\mathsf{icl}}(\boldsymbol{\theta})$ is simply the smallest expected ICL loss for ICL instances drawn from $\pi$, among all transformers within the norm ball $\Theta_{L,M,D',B}$. Using our constructions in Section 3 & 4, we show that this comparator loss is small on various (meta-)distribution $\pi$'s, by which we obtain end-to-end guarantees for pretraining transformers with small ICL loss at test time. Here we showcase this argument on several representative regression problems.

**Linear regression** For any in-context data distribution $\mathsf{P}$, let $\mathbf{w}_{\mathsf{P}}^\star := \mathbb{E}_{\mathsf{P}}[\mathbf{x}\mathbf{x}^\top]^{-1}\mathbb{E}_{\mathsf{P}}[\mathbf{x}y]$ denote the best linear predictor for $\mathsf{P}$. We show that with mild choices of $L, M, B$, the learned transformer can perform in-context linear regression with near-optimal statistical power, in that on the sampled $\mathsf{P} \sim \pi$ and ICL instance $\{(\mathbf{x}_i, y_i)\}_{i \in [N+1]} \overset{\mathrm{iid}}{\sim} \mathsf{P}$, it competes with the best linear predictor $\mathbf{w}_{\mathsf{P}}^\star$ *for this particular* $\mathsf{P}$. The proof follows directly by on combining Corollary 5 with Theorem K.1, and can be found in Appendix L.3.

**Theorem K.2** (Pretraining transformers for in-context linear regression). *Suppose $\mathsf{P} \sim \pi$ is almost surely well-posed for in-context linear regression (Assumption A) with the canonical parameters. Then, for $N \ge \widetilde{\mathcal{O}}(d)$, with probability at least $1 - \xi$ (over the training instances $\mathbf{Z}^{(1:n)}$), the solution*

$\widehat{\boldsymbol{\theta}}$ of (TF-ERM) with $L = \mathcal{O}(\kappa \log(\kappa N / \sigma))$ layers, $M = 3$ heads, $D' = 0$ (attention-only), and $B = \mathcal{O}(\sqrt{\kappa d})$ achieves small excess ICL risk over $\mathbf{w}_\mathsf{P}^\star$:

$$L_{\mathsf{icl}}(\widehat{\boldsymbol{\theta}}) - \mathbb{E}_{\mathsf{P} \sim \pi} \mathbb{E}_{(\mathbf{x},y) \sim \mathsf{P}} \left[ \frac{1}{2} (y - \langle \mathbf{w}_\mathsf{P}^\star, \mathbf{x} \rangle)^2 \right] \leq \widetilde{\mathcal{O}} \left( \sqrt{\frac{\kappa^2 d^2 + \log(1/\xi)}{n}} + \frac{d\sigma^2}{N} \right),$$

where $\widetilde{\mathcal{O}}(\cdot)$ only hides polylogarithmic factors in $\kappa, N, 1/\sigma$.

To our best knowledge, Theorem K.2 offers the first end-to-end result for pretraining a transformer to perform in-context linear regression with explicit excess loss bounds. The $\widetilde{\mathcal{O}}(\sqrt{\kappa^2 d^2/n})$ term originates from the generalization of pretraining (Theorem K.1), where as the $\widetilde{\mathcal{O}}(d\sigma^2/N)$ term agrees with the standard fast rate for the excess loss of linear regression [38]. Further, as long as $n \geq \widetilde{\mathcal{O}}(\kappa^2 N/\sigma^2)$, the excess risk achieves the optimal rate $\widetilde{\mathcal{O}}(d\sigma^2/N)$ (up to log factors).

**Additional examples** By similar arguments as in the proof of Theorem K.2, we can directly turn most of our other expressivity results into results on the pretrained transformers. Here we present three such additional examples (proofs in Appendix L.4-L.6). The first example is for the sparse linear regression problem considered in Theorem 8.

**Theorem K.3** (Pretraining transformers for in-context sparse linear regression). *Suppose each $\mathsf{P} \sim \pi$ is almost surely an instance of the sparse linear model specified in Theorem 8 with parameters $B_w^\star$ and $\sigma$. Suppose $N \geq \widetilde{\mathcal{O}}(s \log((d \vee N)/\sigma))$ and let $\kappa := B_w^\star/\sigma$.*

*Then with probability at least $1 - \xi$ (over the training instances $\mathbf{Z}^{(1:n)}$), the solution $\widehat{\boldsymbol{\theta}}$ of (TF-ERM) with $L = \widetilde{\mathcal{O}}(\kappa^2(1 + d/N))$ layers, $M = 2$ heads, $D' = 2d$, and $B = \widetilde{\mathcal{O}}(\mathrm{poly}(d, B_w^\star, \sigma))$ achieves small excess ICL risk:*

$$L_{\mathsf{icl}}(\widehat{\boldsymbol{\theta}}) - \sigma^2 \leq \widetilde{\mathcal{O}} \left( \sqrt{\frac{\kappa^4 d^2 (1 + d/N)^2 + \log(1/\xi)}{n}} + \sigma^2 \frac{s \log d}{N} \right),$$

where $\widetilde{\mathcal{O}}(\cdot)$ only hides polylogarithmic factors in $d, N, 1/\sigma$.

Our next example is for the problem of noisy linear regression with mixed noise levels considered in Theorem 12 and Theorem J.1. There, the constructed transformer uses the post-ICL validation mechanism to perform ridge regression with an adaptive regulariation strength depending on the particular input sequence.

**Theorem K.4** (Pretraining transformers for in-context noisy linear regression with algorithm selection). *Suppose $\pi$ is the data generating model (noisy linear model with mixed noise levels) considered in Theorem J.1, with $\sigma_{\max} \leq \mathcal{O}(1)$. Let $N \geq d/10$.*

*Then, with probability at least $1 - \xi$ (over the training instances $\mathbf{Z}^{(1:n)}$), the solution $\widehat{\boldsymbol{\theta}}$ of (TF-ERM) with input dimension $D = \Theta(dK)$, $L = \mathcal{O}(\sigma_{\min}^{-2} \log(N/\sigma_{\min}))$ layers, $M = \mathcal{O}(K)$ heads, $D' = \mathcal{O}(K^2)$, and $B = \mathcal{O}(\mathrm{poly}(K, \sigma_{\min}^{-1}, d, N))$ achieves small excess ICL risk:*

$$L_{\mathsf{icl}}(\widehat{\boldsymbol{\theta}}) - \mathsf{BayesRisk}_\pi \leq \widetilde{\mathcal{O}} \left( \sqrt{\frac{\sigma_{\min}^{-4} K^3 d^2 + \log(1/\xi)}{n}} + \frac{\sigma_{\max}^2}{\sigma_{\min}^{2/3}} \left( \frac{\log K}{N} \right)^{1/3} \right),$$

where $\widetilde{\mathcal{O}}(\cdot)$ only hides polylogarithmic factors in $d, N, K, 1/\sigma_{\min}$.

Our final example is for in-context logistic regression. For simplicity we consider the realizable case.

**Theorem K.5** (Pretraining transformers for in-context logistic regression; square loss guarantee). *Suppose for $\mathsf{P} \sim \pi$, $\mathsf{P}$ is almost surely a realizable logistic model (i.e. $\mathsf{P} = \mathsf{P}_{\boldsymbol{\beta}}^{\log}$ with $\|\boldsymbol{\beta}\|_2 \leq B_w^\star$ as in Corollary G.1). Suppose that $B_w^\star = \mathcal{O}(1)$ and $N \geq \mathcal{O}(d)$.*

*Then, with probability at least $1 - \xi$ (over the training instances $\mathbf{Z}^{(1:n)}$), the solution $\widehat{\boldsymbol{\theta}}$ of (TF-ERM) with $L = \mathcal{O}(\log(N))$ layers, $M = \widetilde{\mathcal{O}}(d^3 N)$ heads, $D' = 0$, and $B = \mathcal{O}(\mathrm{poly}(d, N))$ achieves small excess ICL risk:*

$$L_{\mathsf{icl}}(\widehat{\boldsymbol{\theta}}) - \mathbb{E}_{\mathsf{P}_{\boldsymbol{\beta}}^{\log} \sim \pi} \mathbb{E}_{(\mathbf{x},y) \sim \mathsf{P}_{\boldsymbol{\beta}}^{\log}} \left[ \frac{1}{2} (y - \sigma_{\log}(\langle \boldsymbol{\beta}, \mathbf{x} \rangle))^2 \right] \leq \widetilde{\mathcal{O}} \left( \sqrt{\frac{d^5 N + \log(1/\xi)}{n}} + \frac{d}{N} \right),$$

where $\widetilde{\mathcal{O}}(\cdot)$ only hides polylogarithmic factors in $d, N$.

**Remark on generality of transformer** All results above are established by the expressivity results in Section 3 & 4 for transformers to implement various ICL procedures (such as least squares, Lasso, GLM, and ridge regression with in-context algorithm selection), combined with the generalization bound (Theorem K.1). However, the transformer itself was not specified to encode any actual structure about the problem at hand in any result above, other than having sufficiently large number of layers, number of heads, and weight norms, which illustrates the flexibility of the transformer architecture.

## L  Proofs for Section K

### L.1  Lipschitzness of transformers

For any $p \in [1, \infty]$, let $\|\mathbf{H}\|_{2,p} := (\sum_{i=1}^{N} \|\mathbf{h}_i\|_2^p)^{1/p}$ denote the column-wise $(2, p)$-norm of $\mathbf{H}$. For any radius $\mathsf{R} > 0$, we denote $\mathcal{H}_\mathsf{R} := \{\mathbf{H} : \|\mathbf{H}\|_{2,\infty} \leq \mathsf{R}\}$ be the ball of radius $\mathsf{R}$ under norm $\|\cdot\|_{2,\infty}$.

**Lemma L.1.** *For a single MLP layer $\boldsymbol{\theta}_{\mathtt{mlp}} = (\mathbf{W}_1, \mathbf{W}_2)$, we introduce its norm (as in (2))*

$$\|\|\boldsymbol{\theta}_{\mathtt{mlp}}\|\| = \|\mathbf{W}_1\|_{\mathrm{op}} + \|\mathbf{W}_2\|_{\mathrm{op}}.$$

*For any fixed hidden dimension $D'$, we consider*

$$\Theta_{\mathtt{mlp}, B} := \{\boldsymbol{\theta}_{\mathtt{mlp}} : \|\|\boldsymbol{\theta}_{\mathtt{mlp}}\|\| \leq B\}.$$

*Then for $\mathbf{H} \in \mathcal{H}_\mathsf{R}$, $\boldsymbol{\theta}_{\mathtt{mlp}} \in \Theta_{\mathtt{mlp}, B}$, the function $(\boldsymbol{\theta}_{\mathtt{mlp}}, \mathbf{H}) \mapsto \mathrm{MLP}_{\boldsymbol{\theta}_{\mathtt{mlp}}}(\mathbf{H})$ is $(B\mathsf{R})$-Lipschitz w.r.t. $\boldsymbol{\theta}_{\mathtt{mlp}}$ and $(1 + B^2)$-Lipschitz w.r.t. $\mathbf{H}$.*

*Proof.* Recall that by our definition, for the parameter $\boldsymbol{\theta}_{\mathtt{mlp}} = (\mathbf{W}_1, \mathbf{W}_2) \in \Theta_{\mathtt{mlp}, B}$ and the input $\mathbf{H} = [\mathbf{h}_i] \in \mathbb{R}^{D \times N}$, the output $\mathrm{MLP}_{\boldsymbol{\theta}_{\mathtt{mlp}}}(\mathbf{H}) = \mathbf{H} + \mathbf{W}_2\sigma(\mathbf{W}_1\mathbf{H}) = [\mathbf{h}_i + \mathbf{W}_2\sigma(\mathbf{W}_1\mathbf{h}_i)]_i$. Therefore, for $\theta'_{\mathtt{mlp}} = (\mathbf{W}'_1, \mathbf{W}'_2) \in \Theta_{\mathtt{mlp}, B}$, we have

$$\left\| \mathrm{MLP}_{\boldsymbol{\theta}_{\mathtt{mlp}}}(\mathbf{H}) - \mathrm{MLP}_{\theta'_{\mathtt{mlp}}}(\mathbf{H}) \right\|_{2,\infty}$$
$$= \max_i \|\mathbf{W}_2\sigma(\mathbf{W}_1\mathbf{h}_i) - \mathbf{W}'_2\sigma(\mathbf{W}'_1\mathbf{h}_i)\|_2$$
$$= \max_i \|(\mathbf{W}_2 - \mathbf{W}'_2)\sigma(\mathbf{W}_1\mathbf{h}_i) + \mathbf{W}'_2(\sigma(\mathbf{W}_1\mathbf{h}_i) - \sigma(\mathbf{W}'_1\mathbf{h}_i))\|_2$$
$$\leq \max_i \|\mathbf{W}_2 - \mathbf{W}'_2\|_{\mathrm{op}} \|\sigma(\mathbf{W}_1\mathbf{h}_i)\|_2 + \|\mathbf{W}'_2\|_{\mathrm{op}} \|\sigma(\mathbf{W}_1\mathbf{h}_i) - \sigma(\mathbf{W}'_1\mathbf{h}_i)\|_2$$
$$\leq \max_i \|\mathbf{W}_2 - \mathbf{W}'_2\|_{\mathrm{op}} \|\mathbf{W}_1\mathbf{h}_i\|_2 + \|\mathbf{W}'_2\|_{\mathrm{op}} \|\mathbf{W}_1\mathbf{h}_i - \mathbf{W}'_1\mathbf{h}_i\|_2$$
$$\leq B\mathsf{R} \|\mathbf{W}_2 - \mathbf{W}'_2\|_{\mathrm{op}} + B\mathsf{R} \|\mathbf{W}_1 - \mathbf{W}'_1\|_{\mathrm{op}},$$

where the second inequality follows from the 1-Lipschitznees of $\sigma = [\cdot]_+$. Similarly, for $\mathbf{H}' = [\mathbf{h}'_i] \in \mathbb{R}^{D \times N}$,

$$\left\| \mathrm{MLP}_{\boldsymbol{\theta}_{\mathtt{mlp}}}(\mathbf{H}) - \mathrm{MLP}_{\boldsymbol{\theta}_{\mathtt{mlp}}}(\mathbf{H}') \right\|_{2,\infty} = \max_i \|\mathbf{h}_i + \mathbf{W}_1\sigma(\mathbf{W}_2\mathbf{h}_i) - \mathbf{h}'_i - \mathbf{W}_1\sigma(\mathbf{W}_2\mathbf{h}'_i)\|_2$$
$$\leq \|\mathbf{H} - \mathbf{H}'\|_{2,\infty} + \max_i \|\mathbf{W}_1(\sigma(\mathbf{W}_2\mathbf{h}_i) - \sigma(\mathbf{W}_2\mathbf{h}'_i))\|_2$$
$$\leq \|\mathbf{H} - \mathbf{H}'\|_{2,\infty} + \max_i B \|\sigma(\mathbf{W}_2\mathbf{h}_i) - \sigma(\mathbf{W}_2\mathbf{h}'_i)\|_2$$
$$\leq \|\mathbf{H} - \mathbf{H}'\|_{2,\infty} + B^2 \|\mathbf{H} - \mathbf{H}'\|_{2,\infty}.$$
$\square$

**Lemma L.2.** *For a single attention layer $\boldsymbol{\theta}_{\mathtt{attn}} = \{(\mathbf{V}_m, \mathbf{Q}_m, \mathbf{K}_m)\}_{m \in [M]} \subset \mathbb{R}^{D \times D}$, we introduce its norm (as in (2))*

$$\|\|\boldsymbol{\theta}_{\mathtt{attn}}\|\| := \max_{m \in [M]} \left\{ \|\mathbf{Q}_m\|_{\mathrm{op}}, \|\mathbf{K}_m\|_{\mathrm{op}} \right\} + \sum_{m=1}^{M} \|\mathbf{V}_m\|_{\mathrm{op}}.$$

*For any fixed dimension $D$, we consider*

$$\Theta_{\mathtt{attn}, B} := \{\boldsymbol{\theta}_{\mathtt{attn}} : \|\|\boldsymbol{\theta}_{\mathtt{attn}}\|\| \leq B\}.$$

*Then for $\mathbf{H} \in \mathcal{H}_\mathsf{R}$, $\boldsymbol{\theta}_{\mathtt{attn}} \in \Theta_{\mathtt{attn}, B}$, the function $(\boldsymbol{\theta}_{\mathtt{attn}}, \mathbf{H}) \mapsto \mathrm{Attn}_{\boldsymbol{\theta}_{\mathtt{attn}}}(\mathbf{H})$ is $(B^2\mathsf{R}^3)$-Lipschitz w.r.t. $\boldsymbol{\theta}_{\mathtt{attn}}$ and $(1 + B^3\mathsf{R}^2)$-Lipschitz w.r.t. $\mathbf{H}$.*

*Proof.* Recall that by our definition, for the parameter $\boldsymbol{\theta}_{\mathtt{attn}} = \{(\mathbf{V}_m, \mathbf{Q}_m, \mathbf{K}_m)\}_{m \in [M]} \in \Theta_{\mathtt{attn}, B}$ and the input $\mathbf{H} = [\mathbf{h}_i] \in \mathbb{R}^{D \times N}$, the output $\mathrm{Attn}_{\boldsymbol{\theta}_{\mathtt{attn}}}(\mathbf{H}) = [\widetilde{\mathbf{h}}_i]$ is given by

$$\widetilde{\mathbf{h}}_i = \mathbf{h}_i + \sum_{m=1}^{M} \frac{1}{N} \sum_{j=1}^{N} \sigma(\langle \mathbf{Q}_m \mathbf{h}_i, \mathbf{K}_m \mathbf{h}_j \rangle) \cdot \mathbf{V}_m \mathbf{h}_j.$$

Now, for $\theta'_{\mathtt{attn}} = \{(\mathbf{V}'_m, \mathbf{Q}'_m, \mathbf{K}'_m)\}_{m \in [M]}$, we consider

$$\widetilde{\mathbf{h}}'_i = \left[\mathrm{Attn}_{\theta'_{\mathtt{attn}}}(\mathbf{H})\right]_i = \mathbf{h}_i + \sum_{m=1}^{M} \frac{1}{N} \sum_{j=1}^{N} \sigma(\langle \mathbf{Q}'_m \mathbf{h}_i, \mathbf{K}'_m \mathbf{h}_j \rangle) \cdot \mathbf{V}'_m \mathbf{h}_j, \qquad \forall i \in [N].$$

Clearly $\left\|\mathrm{Attn}_{\boldsymbol{\theta}_{\mathtt{attn}}}(\mathbf{H}) - \mathrm{Attn}_{\theta'_{\mathtt{attn}}}(\mathbf{H})\right\|_{2,\infty} = \max_i \left\|\widetilde{\mathbf{h}}_i - \widetilde{\mathbf{h}}'_i\right\|_2$. For any $i \in [N]$, we have

$$
\begin{aligned}
\left\|\widetilde{\mathbf{h}}_i - \widetilde{\mathbf{h}}'_i\right\|_2 &= \left\|\sum_{m=1}^{M} \frac{1}{N} \sum_{j=1}^{N} \left[\sigma(\langle \mathbf{Q}_m \mathbf{h}_i, \mathbf{K}_m \mathbf{h}_j \rangle) \mathbf{V}_m \mathbf{h}_j - \sigma(\langle \mathbf{Q}'_m \mathbf{h}_i, \mathbf{K}'_m \mathbf{h}_j \rangle) \mathbf{V}'_m \mathbf{h}_j\right]\right\|_2 \\
&\leq \sum_{m=1}^{M} \frac{1}{N} \sum_{j=1}^{N} \left\|\sigma(\langle \mathbf{Q}_m \mathbf{h}_i, \mathbf{K}_m \mathbf{h}_j \rangle) \mathbf{V}_m - \sigma(\langle \mathbf{Q}'_m \mathbf{h}_i, \mathbf{K}'_m \mathbf{h}_j \rangle) \mathbf{V}'_m\right\|_{\mathrm{op}} \|\mathbf{h}_j\|_2 \\
&\leq \sum_{m=1}^{M} \frac{1}{N} \sum_{j=1}^{N} \|\mathbf{h}_j\|_2 \left\{\left|\sigma(\langle \mathbf{Q}_m \mathbf{h}_i, \mathbf{K}_m \mathbf{h}_j \rangle)\right| \cdot \|\mathbf{V}_m - \mathbf{V}'_m\|_{\mathrm{op}} \right. \\
&\qquad\qquad + \left|\sigma(\langle \mathbf{Q}_m \mathbf{h}_i, \mathbf{K}_m \mathbf{h}_j \rangle) - \sigma(\langle \mathbf{Q}'_m \mathbf{h}_i, \mathbf{K}_m \mathbf{h}_j \rangle)\right| \cdot \|\mathbf{V}'_m\|_{\mathrm{op}} \\
&\qquad\qquad \left. + \left|\sigma(\langle \mathbf{Q}'_m \mathbf{h}_i, \mathbf{K}_m \mathbf{h}_j \rangle) - \sigma(\langle \mathbf{Q}'_m \mathbf{h}_i, \mathbf{K}'_m \mathbf{h}_j \rangle)\right| \cdot \|\mathbf{V}'_m\|_{\mathrm{op}}\right\} \\
&\leq \sum_{m=1}^{M} \frac{1}{N} \sum_{j=1}^{N} \mathsf{R}\left\{B^2 \mathsf{R}^2 \cdot \|\mathbf{V}_m - \mathbf{V}'_m\|_{\mathrm{op}} + \|\mathbf{Q}_m \mathbf{h}_i - \mathbf{Q}'_m \mathbf{h}_i\|_2 \cdot \|\mathbf{K}_m \mathbf{h}_j\|_2 \cdot \|\mathbf{V}'_m\|_{\mathrm{op}} \right. \\
&\qquad\qquad \left. + \|\mathbf{Q}'_m \mathbf{h}_i\|_2 \cdot \|\mathbf{K}_m \mathbf{h}_j - \mathbf{K}'_m \mathbf{h}_j\|_2 \cdot \|\mathbf{V}'_m\|_{\mathrm{op}}\right\} \\
&\leq \sum_{m=1}^{M} \mathsf{R}\left\{B^2 \mathsf{R}^2 \|\mathbf{V}_m - \mathbf{V}'_m\|_{\mathrm{op}} + B\mathsf{R}^2 \|\mathbf{Q}_m - \mathbf{Q}'_m\|_{\mathrm{op}} \cdot \|\mathbf{V}'_m\|_{\mathrm{op}} + B\mathsf{R}^2 \|\mathbf{K}_m - \mathbf{K}'_m\|_{\mathrm{op}} \cdot \|\mathbf{V}'_m\|_{\mathrm{op}}\right\} \\
&\leq B^2 \mathsf{R}^3 \left\{\sum_{m=1}^{M} \|\mathbf{V}_m - \mathbf{V}'_m\|_{\mathrm{op}} + \max_m \|\mathbf{Q}_m - \mathbf{Q}'_m\|_{\mathrm{op}} + \max_m \|\mathbf{K}_m - \mathbf{K}'_m\|_{\mathrm{op}}\right\} \\
&= B^2 \mathsf{R}^3 \|\boldsymbol{\theta}_{\mathtt{attn}} - \theta'_{\mathtt{attn}}\|,
\end{aligned}
$$

where the second inequality uses the definition of operator norm, the third inequality follows from the triangle inequality, the forth inequality is because $\|\mathbf{Q}_m \mathbf{h}_i\|_2 \leq B\mathsf{R}$, $\|\mathbf{K}_m \mathbf{h}_j\|_2 \leq B\mathsf{R}$, and $\sigma$ is 1-Lipschitz. This completes the proof the Lipschitzness w.r.t. $\boldsymbol{\theta}_{\mathtt{attn}}$.

Similarly, we consider $\mathbf{H}' = [\mathbf{h}'_i]$, and

$$\widetilde{\mathbf{h}}'_i = \left[\mathrm{Attn}_{\theta'_{\mathtt{attn}}}(\mathbf{H})\right]_i = \mathbf{h}'_i + \sum_{m=1}^{M} \frac{1}{N} \sum_{j=1}^{N} \sigma\left(\langle \mathbf{Q}_m \mathbf{h}'_i, \mathbf{K}_m \mathbf{h}'_j \rangle\right) \cdot \mathbf{V}_m \mathbf{h}'_j, \qquad \forall i \in [N].$$

By definition, we can similarly bound

$$
\begin{aligned}
&\left\|\left(\widetilde{\mathbf{h}}'_i - \mathbf{h}'_i\right) - \left(\widetilde{\mathbf{h}}_i - \mathbf{h}_i\right)\right\|_2 \\
&= \left\|\sum_{m=1}^{M} \frac{1}{N} \sum_{j=1}^{N} \left[\sigma(\langle \mathbf{Q}_m \mathbf{h}_i, \mathbf{K}_m \mathbf{h}_j \rangle) \mathbf{V}_m \mathbf{h}_j - \sigma\left(\langle \mathbf{Q}_m \mathbf{h}'_i, \mathbf{K}_m \mathbf{h}'_j \rangle\right) \mathbf{V}_m \mathbf{h}'_j\right]\right\|_2 \\
&\leq \sum_{m=1}^{M} \frac{1}{N} \sum_{j=1}^{N} \|\mathbf{V}_m\|_{\mathrm{op}} \left\|\sigma(\langle \mathbf{Q}_m \mathbf{h}_i, \mathbf{K}_m \mathbf{h}_j \rangle) \mathbf{h}_j - \sigma\left(\langle \mathbf{Q}_m \mathbf{h}'_i, \mathbf{K}_m \mathbf{h}'_j \rangle\right) \mathbf{h}'_j\right\|_2
\end{aligned}
$$

$$
\leq \sum_{m=1}^{M} \frac{1}{N} \sum_{j=1}^{N} \|\mathbf{V}_m\|_{\mathrm{op}} \Big\{ \big|\sigma(\langle \mathbf{Q}_m \mathbf{h}_i, \mathbf{K}_m \mathbf{h}_j\rangle)\big| \cdot \big\|\mathbf{h}_j - \mathbf{h}_j'\big\|_2
$$

$$
+ \big|\sigma(\langle \mathbf{Q}_m \mathbf{h}_i, \mathbf{K}_m \mathbf{h}_j\rangle) - \sigma(\langle \mathbf{Q}_m \mathbf{h}_i', \mathbf{K}_m \mathbf{h}_j\rangle)\big| \cdot \big\|\mathbf{h}_j'\big\|_2
$$

$$
+ \big|\sigma(\langle \mathbf{Q}_m \mathbf{h}_i', \mathbf{K}_m \mathbf{h}_j\rangle) - \sigma\big(\langle \mathbf{Q}_m \mathbf{h}_i', \mathbf{K}_m \mathbf{h}_j'\rangle\big)\big| \cdot \big\|\mathbf{h}_j'\big\|_2 \Big\}
$$

$$
\leq \sum_{m=1}^{M} \frac{1}{N} \sum_{j=1}^{N} \|\mathbf{V}_m\|_{\mathrm{op}} \cdot 3 \|\mathbf{Q}_m\|_{\mathrm{op}} \|\mathbf{K}_m\|_{\mathrm{op}} \mathsf{R}^2 \big\|\mathbf{h}_j - \mathbf{h}_j'\big\|_2
$$

$$
\leq \mathsf{R}^2 \|\mathbf{H} - \mathbf{H}'\|_{2,\infty} \cdot 3 \max_{m \in [M]} \|\mathbf{Q}_m\|_{\mathrm{op}} \|\mathbf{K}_m\|_{\mathrm{op}} \cdot \sum_{m=1}^{M} \|\mathbf{V}_m\|_{\mathrm{op}}
$$

$$
\leq B^3 \mathsf{R}^2 \|\mathbf{H} - \mathbf{H}'\|_{2,\infty},
$$

where the last inequality uses $\|\!|\boldsymbol{\theta}_{\mathtt{attn}}|\!\| \leq B$ and the AM-GM inequality. This completes the proof the Lipschitzness w.r.t. $\mathbf{H}$. $\qquad\square$

**Corollary L.1.** *For a fixed number of heads $M$ and hidden dimension $D'$, we consider*

$$
\Theta_{\mathrm{TF},1,B} = \big\{ \boldsymbol{\theta} = (\boldsymbol{\theta}_{\mathtt{attn}}, \boldsymbol{\theta}_{\mathtt{mlp}}) : M \text{ heads, hidden dimension } D', \|\!|\boldsymbol{\theta}|\!\| \leq B \big\}.
$$

*Then for the function $\mathrm{TF}^{\mathsf{R}}$ given by*

$$
\mathrm{TF}^{\mathsf{R}} : (\boldsymbol{\theta}, \mathbf{H}) \mapsto \mathsf{clip}_{\mathsf{R}}\big(\mathrm{MLP}_{\boldsymbol{\theta}_{\mathtt{mlp}}}(\mathrm{Attn}_{\boldsymbol{\theta}_{\mathtt{attn}}}(\mathbf{H}))\big), \qquad \boldsymbol{\theta} \in \Theta_{\mathrm{TF},1,B}, \mathbf{H} \in \mathcal{H}_{\mathsf{R}}
$$

*$\mathrm{TF}^{\mathsf{R}}$ is $B_\Theta$-Lipschitz w.r.t $\boldsymbol{\theta}$ and $L_H$-Lipschitz w.r.t. $\mathbf{H}$, where $B_\Theta := B\mathsf{R}(1 + B\mathsf{R}^2 + B^3\mathsf{R}^2)$ and $B_H := (1 + B^2)(1 + B^2\mathsf{R}^3)$.*

*Proof.* For any $\boldsymbol{\theta} = (\boldsymbol{\theta}_{\mathtt{attn}}, \boldsymbol{\theta}_{\mathtt{mlp}})$, $\mathbf{H} \in \mathcal{H}_{\mathsf{R}}$, and $\theta' = (\theta'_{\mathtt{attn}}, \theta'_{\mathtt{mlp}})$, we have

$$
\|\mathrm{TF}_{\boldsymbol{\theta}}(\mathbf{H}) - \mathrm{TF}_{\theta'}(\mathbf{H})\|_{2,\infty} \leq \big\|\mathrm{MLP}_{\boldsymbol{\theta}_{\mathtt{mlp}}}(\mathrm{Attn}_{\boldsymbol{\theta}_{\mathtt{attn}}}(\mathbf{H})) - \mathrm{MLP}_{\boldsymbol{\theta}_{\mathtt{mlp}}}\big(\mathrm{Attn}_{\theta'_{\mathtt{attn}}}(\mathbf{H})\big)\big\|_{2,\infty}
$$

$$
+ \Big\|\mathrm{MLP}_{\boldsymbol{\theta}_{\mathtt{mlp}}}\big(\mathrm{Attn}_{\theta'_{\mathtt{attn}}}(\mathbf{H})\big) - \mathrm{MLP}_{\theta'_{\mathtt{mlp}}}\big(\mathrm{Attn}_{\theta'_{\mathtt{attn}}}(\mathbf{H})\big)\Big\|_{2,\infty}
$$

$$
\leq (1 + B^2) \big\|\mathrm{Attn}_{\boldsymbol{\theta}_{\mathtt{attn}}}(\mathbf{H}) - \mathrm{Attn}_{\theta'_{\mathtt{attn}}}(\mathbf{H})\big\|_{2,\infty} + B\overline{\mathsf{R}}\|\!|\boldsymbol{\theta}_{\mathtt{mlp}} - \theta'_{\mathtt{mlp}}|\!\|
$$

$$
\leq (1 + B^2)B^2\mathsf{R}^3\|\!|\boldsymbol{\theta}_{\mathtt{attn}} - \boldsymbol{\theta}'_{\mathtt{attn}}|\!\| + B\overline{\mathsf{R}}\|\!|\boldsymbol{\theta}_{\mathtt{mlp}} - \theta'_{\mathtt{mlp}}|\!\|
$$

$$
\leq B_\Theta\|\!|\boldsymbol{\theta} - \theta'|\!\|,
$$

where the second inequality follows from Lemma L.2 and Lemma L.1 and the fact that $\|\mathrm{Attn}_{\boldsymbol{\theta}_{\mathtt{attn}}}(\mathbf{H})\|_{2,\infty} \leq \overline{\mathsf{R}} := \mathsf{R} + B^3\mathsf{R}^3$ for all $\mathbf{H} \in \mathcal{H}_{\mathsf{R}}$.

Furthermore, for $\mathbf{H}' \in \mathcal{H}_{\mathsf{R}}$, we have

$$
\|\mathrm{TF}_{\boldsymbol{\theta}}(\mathbf{H}) - \mathrm{TF}_{\boldsymbol{\theta}}(\mathbf{H}')\|_{2,\infty} \leq (1 + B^2) \|\mathrm{Attn}_{\boldsymbol{\theta}_{\mathtt{attn}}}(\mathbf{H}) - \mathrm{Attn}_{\boldsymbol{\theta}_{\mathtt{attn}}}(\mathbf{H}')\|_{2,\infty}
$$

$$
\leq (1 + B^2)(1 + B^3\mathsf{R}^2) \|\mathbf{H} - \mathbf{H}'\|_{2,\infty},
$$

which also follows from Lemma L.2 and Lemma L.1. $\qquad\square$

**Proposition L.1** (Lipschitzness of transformers). *For a fixed number of heads $M$ and hidden dimension $D'$, we consider*

$$
\Theta_{\mathrm{TF},L,B} = \Big\{ \boldsymbol{\theta} = (\boldsymbol{\theta}_{\mathtt{attn}}^{(1:L)}, \boldsymbol{\theta}_{\mathtt{mlp}}^{(1:L)}) : M^{(\ell)} = M, D^{(\ell)} = D', \|\!|\boldsymbol{\theta}|\!\| \leq B \Big\}.
$$

*Then the function $\mathrm{TF}^{\mathsf{R}}$ is $(LB_H^{L-1}B_\Theta)$-Lipschitz w.r.t $\boldsymbol{\theta} \in \Theta_{\mathrm{TF},L,B}$ for any fixed $\mathbf{H}$.*

*Proof.* For $\boldsymbol{\theta} = \boldsymbol{\theta}^{(1:L)} \in \Theta_{\mathrm{TF},L,B}, \widetilde{\boldsymbol{\theta}} = \widetilde{\boldsymbol{\theta}}^{(1:L)} \in \Theta_{\mathrm{TF},L,B}$, we have

$$
\Big\|\mathrm{TF}_{\boldsymbol{\theta}}^{\mathsf{R}}(\mathbf{H}) - \mathrm{TF}_{\widetilde{\boldsymbol{\theta}}}^{\mathsf{R}}(\mathbf{H})\Big\|_{2,\infty}
$$

$$\leq \sum_{\ell=1}^{L} \left\| \mathrm{TF}^{\mathsf{R}}_{\boldsymbol{\theta}^{(\ell+1:L)}} \left( \mathrm{TF}^{\mathsf{R}}_{\boldsymbol{\theta}^{(\ell)}} \left( \mathrm{TF}^{\mathsf{R}}_{\widetilde{\boldsymbol{\theta}}^{(1:\ell-1)}}(\mathbf{H}) \right) \right) - \mathrm{TF}^{\mathsf{R}}_{\boldsymbol{\theta}^{(\ell+1:L)}} \left( \mathrm{TF}^{\mathsf{R}}_{\widetilde{\boldsymbol{\theta}}^{(\ell)}} \left( \mathrm{TF}^{\mathsf{R}}_{\widetilde{\boldsymbol{\theta}}^{(1:\ell-1)}}(\mathbf{H}) \right) \right) \right\|_{2,\infty}$$

$$\leq \sum_{\ell=1}^{L} B_{\Theta}^{L-\ell} \left\| \mathrm{TF}^{\mathsf{R}}_{\boldsymbol{\theta}^{(\ell)}} \left( \mathrm{TF}^{\mathsf{R}}_{\widetilde{\boldsymbol{\theta}}^{(1:\ell-1)}}(\mathbf{H}) \right) - \mathrm{TF}^{\mathsf{R}}_{\widetilde{\boldsymbol{\theta}}^{(\ell)}} \left( \mathrm{TF}^{\mathsf{R}}_{\widetilde{\boldsymbol{\theta}}^{(1:\ell-1)}}(\mathbf{H}) \right) \right\|_{2,\infty}$$

$$\leq \sum_{\ell=1}^{L} B_H^{L-\ell} B_{\Theta} \cdot \left\| \boldsymbol{\theta}^{(\ell)} - \widetilde{\boldsymbol{\theta}}^{(\ell)} \right\| \leq L B_H^{L-1} B_{\Theta} \cdot \left\| \boldsymbol{\theta} - \widetilde{\boldsymbol{\theta}} \right\|,$$

where the second inequality follows from Corollary L.1, and the last inequality is because $B_H \geq 1$. $\qquad\square$

## L.2 Proof of Theorem K.1

In this section, we prove a slightly more general result by considering the general ICL loss

$$\ell_{\mathsf{icl}}(\boldsymbol{\theta}; \mathbf{Z}) := \ell(\widetilde{\mathsf{read}}_{\mathsf{y}}(\mathrm{TF}^{\mathsf{R}}_{\boldsymbol{\theta}}(\mathcal{H})), y_{N+1}).$$

We assume that the loss function $\ell$ satisfies $\sup |\ell| \leq B_{\ell}^0$ and $\sup |\partial_1 \ell| \leq B_{\ell}^1$. For the special case $\ell(s,t) = \frac{1}{2}(s-t)^2$, we can take $B_{\ell}^0 = 4B_y^2, B_{\ell}^1 = 2B_y$.

We then consider

$$X_{\boldsymbol{\theta}} := \frac{1}{n} \sum_{j=1}^{n} \ell_{\mathsf{icl}}(\boldsymbol{\theta}; \mathbf{Z}^j) - \mathbb{E}_{\mathbf{Z}}[\ell_{\mathsf{icl}}(\boldsymbol{\theta}; \mathbf{Z})],$$

where $\mathbf{Z}^{(1:n)}$ are i.i.d copies of $\mathbf{Z} \sim \mathsf{P}, \mathsf{P} \sim \pi$. It remains to apply Proposition B.4 to the random process $\{X_{\boldsymbol{\theta}}\}$. We verify the preconditions:

(a) By [87, Example 5.8], it holds that $\log N(\delta; \mathsf{B}_{\|\cdot\|}(r), \|\cdot\|) \leq L(3MD^2 + 2DD') \log(1 + 2r/\delta)$, where $\mathsf{B}_{\|\cdot\|}(r)$ is any ball of radius $r$ under norm $\|\cdot\|$.

(b) $|\ell_{\mathsf{icl}}(\boldsymbol{\theta}; \mathbf{Z})| \leq B_{\ell}^0$ and hence $B_{\ell}^0$-sub-Gaussian.

(c) $\left| \ell_{\mathsf{icl}}(\boldsymbol{\theta}; \mathbf{Z}) - \ell_{\mathsf{icl}}(\widetilde{\boldsymbol{\theta}}; \mathbf{Z}) \right| \leq B_{\ell}^1 \cdot (L B_H^{L-1} B_{\Theta}) \cdot \left\| \boldsymbol{\theta} - \widetilde{\boldsymbol{\theta}} \right\|$, by Proposition L.1.

Therefore, we can apply the uniform concentration result in Proposition B.4 to obtain that, with probability at least $1 - \xi$,

$$\sup_{\boldsymbol{\theta}} |X_{\boldsymbol{\theta}}| \leq C B_{\ell}^0 \sqrt{\frac{L(MD^2 + DD')\iota + \log(1/\xi)}{n}},$$

where $\iota = \log(2 + B \cdot L B_H^{L-1} B_{\Theta} B_{\ell}^1 / B_{\ell}^0) \leq 20L \log(2 + \max\{B, \mathsf{R}, B_{\ell}^1 / B_{\ell}^0\})$. Recalling that

$$L_{\mathsf{icl}}(\widehat{\boldsymbol{\theta}}) \leq \inf_{\boldsymbol{\theta}} L_{\mathsf{icl}}(\boldsymbol{\theta}) + 2 \sup_{\boldsymbol{\theta}} |X_{\boldsymbol{\theta}}|$$

completes the proof. $\qquad\square$

## L.3 Proof of Theorem K.2

By Corollary 5, there exists a transformer $\mathrm{TF}_{\boldsymbol{\theta}}$ such that for every $\mathsf{P}$ satisfying Assumption A with canonical parameters (and thus in expectation over $\mathsf{P} \sim \pi$) and every $N \geq \widetilde{\mathcal{O}}(d)$, it outputs prediction $\widehat{y}_{N+1} = \widetilde{\mathsf{read}}_{\mathsf{y}}(\mathrm{TF}_{\boldsymbol{\theta}}(\mathbf{H}))$ such that

$$L_{\mathsf{icl}}(\boldsymbol{\theta}) = \mathbb{E}_{\mathsf{P} \sim \pi, (\mathcal{D}, \mathbf{x}_{N+1}, y_{N+1} \sim \mathsf{P})} \left[ \frac{1}{2} (\widehat{y}_{N+1} - y_{N+1})^2 \right] \leq \mathbb{E}_{\mathsf{P} \sim \pi}[L_{\mathsf{P}}(\mathbf{w}_{\mathsf{P}}^{\star})] + \mathcal{O}\left( \frac{d\sigma^2}{N} \right),$$

where we recall that $L_{\mathsf{P}}(\mathbf{w}_{\mathsf{P}}^{\star}) := \frac{1}{2} \mathbb{E}_{(\mathbf{x},y) \sim \mathsf{P}} \left[ (y - \langle \mathbf{w}_{\mathsf{P}}^{\star}, \mathbf{x} \rangle)^2 \right]$. By inspecting the proof, the same result holds if we change $\mathrm{TF}_{\boldsymbol{\theta}}$ to the clipped version $\mathrm{TF}^{\mathsf{R}}_{\boldsymbol{\theta}}$ if we choose $\mathsf{R}^2 = \mathcal{O}(B_x^2 + B_y^2 + B_w^2 + 1) = \mathcal{O}(d + \kappa)$, so that on the good event $E_{\mathrm{cov}} \cap E_w$ considered therein, all intermediate outputs within

$\mathrm{TF}_{\boldsymbol{\theta}}$ has $\|\cdot\|_{2,\infty} \leq \mathsf{R}$ and thus the clipping does not modify the transformer output on $E_{\mathrm{cov}} \cap E_w$. Further, recall by (33) that $\boldsymbol{\theta}$ has size bounds

$$L \leq \mathcal{O}\left(\kappa \log \frac{N\kappa}{\sigma}\right), \quad \max_{\ell \in [L]} M^{(\ell)} \leq 3, \quad \|\boldsymbol{\theta}\| \leq \mathcal{O}(\sqrt{\kappa d}).$$

We can thus apply Theorem K.1 to obtain that the solution $\widehat{\boldsymbol{\theta}}$ to (TF-ERM) with the above choice of $(L, M, B)$ and $D' = 0$ (attention-only) satisfies the following with probability at least $1 - \xi$:

$$L_{\mathrm{icl}}(\widehat{\boldsymbol{\theta}}) \leq \inf_{\boldsymbol{\theta}' \in \Theta_{L,M,D',B}} L_{\mathrm{icl}}(\boldsymbol{\theta}') + \mathcal{O}\left(\sqrt{\frac{L^2 M D^2 \iota + \log(1/\xi)}{n}}\right)$$

$$\leq L_{\mathrm{icl}}(\boldsymbol{\theta}) + \widetilde{\mathcal{O}}\left(\sqrt{\frac{L^2 M D^2 + \log(1/\xi)}{n}}\right) \leq \widetilde{\mathcal{O}}\left(\sqrt{\frac{\kappa^2 d^2 + \log(1/\xi)}{n}} + \frac{d\sigma^2}{N}\right).$$

Above, $\iota = \mathcal{O}(\log(1 + \max\{B_y, \mathsf{R}, B\})) = \widetilde{\mathcal{O}}(1)$. This finishes the proof. $\qquad \square$

## L.4 Proof of Theorem K.3

We invoke Theorem 8 (using the construction in Theorem 7 with a different choice of $L$) with the following parameters:

$$L = \widetilde{\mathcal{O}}\big((B_w^\star)^2/\sigma^2 \times (1 + d/N)\big) = \widetilde{\mathcal{O}}\big(\kappa^2(1 + d/N)\big), \quad M = \Theta(1), \quad D' = 2d,$$

$$B_x = \widetilde{\mathcal{O}}(\sqrt{d}), \quad B_y = \widetilde{\mathcal{O}}(B_w^\star + \sigma), \quad \delta = \left(\sigma^2 \frac{1}{B_y^2 N}\right)^2,$$

$$\|\boldsymbol{\theta}\| \leq B = \mathcal{O}\big(\mathsf{R} + (1 + \lambda_N)\beta^{-1}\big) \leq \mathcal{O}\big(\mathsf{R} + \sigma\sqrt{\log d}\big) \leq \widetilde{\mathcal{O}}(\mathrm{poly}(d, B_w^\star, \sigma)),$$

where $\widetilde{\mathcal{O}}(\cdot)$ hides polylogarithmic factors in $d, N, B_w^\star, \kappa$.

Then, Theorem 8 shows that there exists a transformer $\boldsymbol{\theta}$ with $L$ layers, $\max_{\ell \in [L]} M^{(\ell)} \leq M$ heads, $D'$ hidden dimension for the MLP layers, and $\|\boldsymbol{\theta}\| \leq B$ such that, on almost surely every $\mathsf{P} \sim \pi$, it returns a prediction $\widehat{y}_{N+1}$ such that, on the good event $\mathcal{E}_0$ considered therein (over $\mathcal{D} \sim \mathsf{P}$) which satisfies $\mathsf{P}(\mathcal{E}_0) \geq 1 - \delta$,

$$\mathbb{E}_{(\mathbf{x}_{N+1}, y_{N+1}) \sim \mathsf{P}}\left[(\widehat{y}_{N+1} - y_{N+1})^2\right] \leq \sigma^2[1 + \mathcal{O}(s \log(d/\delta)/N)].$$

By inspecting the proof, the same result holds if we change $\mathrm{TF}_{\boldsymbol{\theta}}$ to the clipped version $\mathrm{TF}_{\boldsymbol{\theta}}^{\mathsf{R}}$ if we choose $\mathsf{R}^2 = \mathcal{O}(B_x^2 + B_y^2 + (B_w^\star)^2 + 1) = \mathcal{O}(d + (B_w^\star)^2 + \sigma^2)$, so that on the good event $\mathcal{E}_0$ considered therein, all intermediate outputs within $\mathrm{TF}_{\boldsymbol{\theta}}$ has $\|\cdot\|_{2,\infty} \leq \mathsf{R}$ and thus the clipping does not modify the transformer output on the good event. On the bad event $\mathcal{E}_0^c$, using the same argument as in the proof of Theorem 8, we have

$$\mathbb{E}_{\mathcal{D},(\mathbf{x}_{N+1},y_{N+1}) \sim \mathsf{P}}\big[\mathbf{1}\{\mathcal{E}_0^c\}(\widehat{y}_{N+1} - y_{N+1})^2\big] \leq \sqrt{\mathsf{P}_{\mathcal{D}}(\mathcal{E}_0^c)} \cdot \big(8\mathbb{E}_{y_{N+1} \sim \mathsf{P}}\big[B_y^4 + y_{N+1}^4\big]\big)^{1/2} \leq \widetilde{\mathcal{O}}\left(\frac{\sigma^2}{N}\right).$$

Combining the above two bounds and further taking expectation over $\mathsf{P} \sim \pi$ gives

$$L_{\mathrm{icl}}(\boldsymbol{\theta}) = \mathbb{E}_{\mathsf{P} \sim \pi, (\mathcal{D}, \mathbf{x}_{N+1}, y_{N+1}) \sim \mathsf{P}}\left[\frac{1}{2}(\widehat{y}_{N+1} - y_{N+1})^2\right] \leq \sigma^2 + \widetilde{\mathcal{O}}(\sigma^2 s \log d/N).$$

We can thus apply Theorem K.1 to obtain that the solution $\widehat{\boldsymbol{\theta}}$ to (TF-ERM) with the above choice of $(L, M, B, D')$ satisfies the following with probability at least $1 - \xi$:

$$L_{\mathrm{icl}}(\widehat{\boldsymbol{\theta}}) \leq \inf_{\boldsymbol{\theta}' \in \Theta_{L,M,D',B}} L_{\mathrm{icl}}(\boldsymbol{\theta}') + \mathcal{O}\left(\sqrt{\frac{L^2(MD^2 + DD')\iota + \log(1/\xi)}{n}}\right)$$

$$\leq L_{\mathrm{icl}}(\boldsymbol{\theta}) + \widetilde{\mathcal{O}}\left(\sqrt{\frac{L^2(MD^2 + DD') + \log(1/\xi)}{n}}\right)$$

$$\leq \sigma^2 + \widetilde{\mathcal{O}}\left(\sqrt{\frac{\kappa^4 d^2(1 + d/N)^2 + \log(1/\xi)}{n}} + \sigma^2 \frac{s \log d}{N}\right).$$

Above, $\iota = \mathcal{O}(\log(1 + \max\{B_y, \mathsf{R}, B\})) = \widetilde{\mathcal{O}}(1)$. This finishes the proof. $\qquad \square$

## L.5  Proof of Theorem K.4

We invoke Theorem 12 and Theorem J.1, which shows that (recalling the input dimension $D = \Theta(Kd)$) there exists a transformer $\boldsymbol{\theta}$ with the following size bounds:

$$L \leq \mathcal{O}\left(\sigma_{\min}^{-2}\log(N/\sigma_{\min})\right), \qquad \max_{\ell \in [L]} M^{(\ell)} \leq M = \mathcal{O}\left(K\right), \qquad \max_{\ell \in [L]} D^{(\ell)} \leq D' = \mathcal{O}(K^2),$$

$$\|\boldsymbol{\theta}\| \leq \mathcal{O}\left(\sigma_{\max}Kd\log(N)\right),$$

such that it outputs $\widehat{y}_{N+1}$ that satisfies

$$\mathbb{E}_\pi\left[\frac{1}{2}(y_{N+1} - \widehat{y}_{N+1})^2\right] \leq \mathsf{BayesRisk}_\pi + \widetilde{\mathcal{O}}\left(\frac{\sigma_{\max}^2}{\sigma_{\min}^{2/3}}\left(\frac{\log K}{N}\right)^{1/3}\right).$$

By inspecting the proof, the same result holds if we change $\mathrm{TF}_{\boldsymbol{\theta}}$ to the clipped version $\mathrm{TF}_{\boldsymbol{\theta}}^{\mathsf{R}}$ if we choose $\mathsf{R}^2 = \mathcal{O}(B_x^2 + B_y^2 + (B_w^\star)^2 + 1) = \mathcal{O}(d + \sigma_{\max}^2)$, so that on the good event considered therein, all intermediate outputs within $\mathrm{TF}_{\boldsymbol{\theta}}$ has $\|\cdot\|_{2,\infty} \leq \mathsf{R}$ and thus the clipping does not modify the transformer output on the good event. Using this clipping radius, we obtain

$$L_{\mathsf{icl}}(\boldsymbol{\theta}) = \mathbb{E}_{\mathsf{P} \sim \pi, (\mathcal{D}, \mathbf{x}_{N+1}, y_{N+1}) \sim \mathsf{P}}\left[\frac{1}{2}(\widehat{y}_{N+1} - y_{N+1})^2\right] \leq \mathsf{BayesRisk}_\pi + \widetilde{\mathcal{O}}\left(\frac{\sigma_{\max}^2}{\sigma_{\min}^{2/3}}\left(\frac{\log K}{N}\right)^{1/3}\right).$$

We can thus apply Theorem K.1 to obtain that the solution $\widehat{\boldsymbol{\theta}}$ to (TF-ERM) with the above choice of $(L, M, B, D')$ satisfies the following with probability at least $1 - \xi$:

$$L_{\mathsf{icl}}(\widehat{\boldsymbol{\theta}}) \leq \inf_{\boldsymbol{\theta}' \in \Theta_{L,M,D',B}} L_{\mathsf{icl}}(\boldsymbol{\theta}') + \mathcal{O}\left(\sqrt{\frac{L^2(MD^2 + DD')\iota + \log(1/\xi)}{n}}\right)$$

$$\leq L_{\mathsf{icl}}(\boldsymbol{\theta}) + \widetilde{\mathcal{O}}\left(\sqrt{\frac{L^2(MD^2 + DD') + \log(1/\xi)}{n}}\right)$$

$$\leq \mathsf{BayesRisk}_\pi + \widetilde{\mathcal{O}}\left(\sqrt{\frac{\sigma_{\min}^{-4}K^3d^2 + \log(1/\xi)}{n}} + \frac{\sigma_{\max}^2}{\sigma_{\min}^{2/3}}\left(\frac{\log K}{N}\right)^{1/3}\right).$$

Above, $\iota = \mathcal{O}(\log(1 + \max\{B_y, \mathsf{R}, B\})) = \widetilde{\mathcal{O}}(1)$. This finishes the proof. $\qquad\square$

## L.6  Proof of Theorem K.5

The proof follows from similar arguments as of Theorem K.3 and Theorem K.4, where we plug in the size bounds (number of layers, heads, and weight norms) from Theorem G.2 and Corollary G.1. $\quad\square$

# M  Experimental details and additional studies

## M.1  Additional details for Section 6

**Architecture and optimization**  We train a 12-layer encoder-only transformer, where each layer consists of an attention layer as in Definition 1 with $M = 8$ heads, hidden dimension $D = 64$, and ReLU activation (normalized by the sequence length), as well as an MLP layer as in Definition 2 hidden dimension $D' = 64$. We add Layer Normalization [3] after each attention and MLP layer to help optimization, as in standard implementations [84]. We append linear read-in layer and linear read-out layer before and after the transformer respectively, both applying a same affine transform to all tokens in the sequence and are trainable. The read-in layer maps any input vector to a $D$-dimensional hidden state, and the read-out layer maps a $D$-dimensional hidden state to a 1-dimensional scalar.

Each training sequence corresponds to a single ICL instance with $N$ in-context training examples $\{(\mathbf{x}_i, y_i)\}_{i=1}^N \subset \mathbb{R}^d \times \mathbb{R}$ and test input $\mathbf{x}_{N+1} \in \mathbb{R}^d$. The input to the transformer is formatted as

in (3) where each token has dimension $d + 1$ (no zero-paddings). The transformer is trained by minimizing the following loss with fresh mini-batches:

$$L(\boldsymbol{\theta}) = \mathbb{E}_{\mathsf{P} \sim \pi, (\mathbf{H}, y_{N+1}) \sim \mathsf{P}}[\ell_\mathsf{P}(\mathrm{read}_\mathsf{y}(\mathrm{TF}_{\boldsymbol{\theta}}(\mathbf{H})), y_{N+1})], \qquad (47)$$

where the loss function $\ell_\mathsf{P} : \mathbb{R}^2 \to \mathbb{R}$ may depend on the training data distribution $\mathsf{P}$ in general; we use the square loss when $\mathsf{P}$ is regression data, and the logistic loss when $\mathsf{P}$ is classification data. We use the Adam optimizer with a fixed learning rate $10^{-4}$, which we find works well for all our experiments. Throughout all our experiments except for the sparse linear regression experiment in Figure 3a, we train the model for 300K steps, where each step consists of a (fresh) minibatch with batch size $64$ in the base mode, and $K$ minibatches each with batch size $64$ in the mixture mode.

For the sparse linear regression experiment, we find that minimizing the training objective (47) alone was not enough, e.g. for the learned transformer to achieve better loss than the least squares algorithm (which achieves much higher test loss than the Lasso; cf. Figure 3a). To help optimization, we augment (47) with another loss that encourages the second-to-last hidden states to recover the true (sparse) coefficient $\mathbf{w}_\star$:

$$L_{\mathsf{fit-w}}(\boldsymbol{\theta}) = \frac{1}{N_0} \sum_{j=1}^{N_0} \mathbb{E}_{\mathsf{P} = \mathsf{P}_{\mathbf{w}^\star} \sim \pi, (\mathbf{H}, y_{N+1}) \sim \mathsf{P}} \left[ \left\| \left[ \mathrm{TF}_{\boldsymbol{\theta}}^{(1:L-1)}(\mathbf{H}) \right]_{j,(D-d+1):D} - \mathbf{w}^\star \right\|_2^2 \right]. \qquad (48)$$

Specifically, the above loss encourages the first $N_0 \leq N$ tokens within the second-to-last layer to be close to $\mathbf{w}^\star$. We choose $N_0 = 5$ (recall that the total number of tokens is $N = 10$ and sequence length is $N + 1 = 11$ for this experiment). We minimize the loss $L(\boldsymbol{\theta}) + \lambda L_{\mathsf{fit-w}}(\boldsymbol{\theta})$ with $\lambda = 0.1$ for 2M steps for this task.

**Evaluation**  All evaluations are done on the trained transformer with 6400 test instances. We use the square loss for regression tasks, and the classification error $(1-\text{accuracy})$ between the true label $y_{N+1} \in \{0, 1\}$ and the predicted label $1\{\widehat{y}_{N+1} \geq 1/2\}$. We report the means in all experiments, as well as their standard deviations (using one-std error bars) in Figure 2a, 2b, 5a, 5b. In Figure 2c, 3b, 3c 5c, all standard deviations are sufficiently small (not significantly exceeding the width of the markers), thus we did not show error bars in those plots.

**Baseline algorithms**  We implement various baseline machine learning algorithms to compare with the learned transformers. A superset of the algorithms is shown in Figure 3a:

- `Least squares`, `Logistic regression`: Standard algorithms for linear regression and linear classification, respectively. Note that least squares is also a valid algorithm for classification.

- `Averaging`: The simple algorithm which computes the linear predictor $\widehat{\mathbf{w}} = \frac{1}{N} \sum_{i=1}^N y_i \mathbf{x}_i$ and predicts $\widehat{y}_{N+1} = \langle \widehat{\mathbf{w}}, \mathbf{x}_{N+1} \rangle$;

- `3-NN`: 3-Nearest Neighbors.

- `Ridge`: Standard ridge regression as in (ICRidge). We specifically consider two $\lambda$'s (denoted as `lam_1` and `lam_2`): $\lambda_1, \lambda_2 = (0.005, 0.125)$. These are the Bayes-optimal regularization strengths for the noise levels $(\sigma_1, \sigma_2) = (0.1, 0.5)$ respectively under the noisy linear model (cf. Corollary 6), using the formula $\lambda^\star = d\sigma^2/N$, with $(d, N) = (20, 40)$.

- `Lasso`: Standard Lasso as in (ICLasso) with $\lambda \in \{1, 0.1, 0.01, 0.001\}$.

In Figure 2c, the `ridge_analytical` curve plots the expected risk of ridge regression under the noisy linear model over 20 geometrically spaced values of $\lambda$'s in between $(\lambda_1, \lambda_2)$, using analytical formulae (with Monte Carlo simulations). The `Bayes_err_{1,2}` indicate the expected risks of $\lambda_1$ on task 1 (with noise $\sigma_1$) and $\lambda_2$ on task 2 (with noise $\sigma_2$), respectively.

## M.2  Decoder-based architecture

ICL capabilities have also been demonstrated in the literature for decoder-based architectures [31, 2, 47]. There, the transformer can do in-context predictions at every token $\mathbf{x}_i$ using past tokens $\{(\mathbf{x}_j, \mathbf{y}_j)\}_{j \leq i-1}$ as training examples. Here we show that such architectures is also able to perform in-context algorithm selection *at every token*; For results for this architecture on "base" ICL tasks (such as those considered in Figure 3a), we refer the readers to Garg et al. [31].

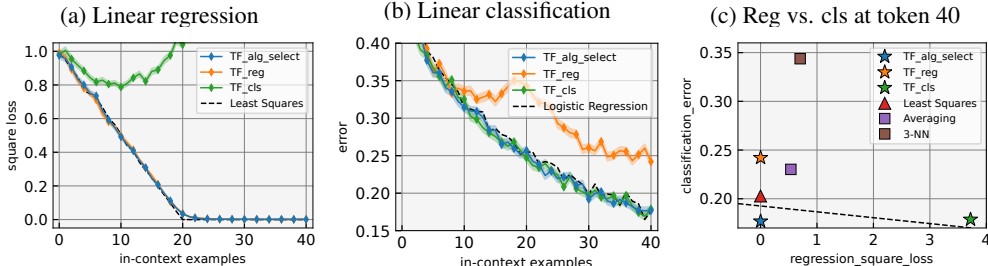

Figure 5: In-context algorithm selection abilities of transformers between linear regression and linear classification. *(a,b)* On these two tasks, a **single transformer** `TF_alg_select` **simultaneously approaches the performance of the strongest baseline algorithm** `Least Squares` on linear regression and `Logistic Regression` on linear classification. *(c)* At token 40 (using example $\{0, \dots, 39\}$ for training), `TF_alg_select` matches the performance of the best baseline algorithm for both tasks. *(a,b,c)* Note that transformers pretrained on a single task (`TF_reg`, `TF_cls`) perform near-optimally on their pretraining task but suboptimally on the other task.

**Setup** Our setup is the same as the two "mixture" modes (linear model + linear classification model, and noisy linear models with two different noise levels) as in Section 6, except that the architecture is GPT-2 following Garg et al. [31], and the input format is changed to (11) (so that the input sequence has $2N + 1$ tokens) without positional encodings. For every $i \in [N + 1]$, we extract the prediction $\widehat{y}_i$ using a linear read-out function applied on output token $2i - 1$, and the (learnable) linear read-out function is the same across all tokens, similar as in Appendix M.1. The rest of the setup (optimization, training, and evaluation) is the same as in Section 6 & M.1. Note that we also train on the objective (47) for all tokens averaged, instead of for the last test token as in Section 6.

**Result** Figure 2 shows the results for noisy linear models with two different noise levels, and Figure 5 shows the results for linear model + linear classification model. We observe that at every token, In both cases, `TF_alg_select` nearly matches the strongest baseline for both tasks simultaneously, whereas transformers trained on a single task perform suboptimally on the other task. Further, this phenomenon consistently shows up at every token. For example, in Figure 2a & 2b, `TF_alg_select` matches ridge regression with the optimal $\lambda$ on all tokens $i \in \{1, \dots, N\}$ ($N = 40$). In Figure 5a & 5b, `TF_alg_select` matches least squares on the regression task and logistic regression on the classification task on all tokens $i \in [N]$. This demonstrates the in-context algorithm selection capabilities of standard decoder-based transformer architectures.

### M.3 Computational resource

All our experiments are performed on 8 Nvidia Tesla A100 GPUs (40GB memory). The total GPU time is approximately 5 days (on 8 GPUs), with the largest individual training run taking about a single day on a single GPU.

