# OpenReview forum: "Transformers as Statisticians: Provable In-Context Learning with In-Context Algorithm Selection"
_NeurIPS.cc/2023/Conference — NeurIPS 2023 oral_

### Official Review · Reviewer_Utbg · 2023-06-13

**Soundness:** 4 excellent
**Presentation:** 4 excellent
**Contribution:** 4 excellent
**Rating:** 9
**Confidence:** 3

**Summary:**

1. The authors prove that transformer models can perform several machine learning tasks, such as least squares, ridge regression, lasso and learning of GLMs.
2. The authors prove that transformer models can select an algorithm from a pool of possible ones, in two scenarios, post ICL validation and pre ICL testing, and thus can perform in context learning.
3. The authors construct a transformer model with near Bayes-optimal performance on linear models with mixed noise levels
4. The authors prove polynomial sample complexity results for pretraining transformers to perform ICL.
5. Experimentally, the authors show that transformer models can indeed perform algorithm selection in context.

**Strengths:**

1. The level of clarity and writing is very good.
2. The theoretical contributions are insightful
3. The experimental results support the theoretical arguments
4. The scope of the work is broad and deep.

**Weaknesses:**

I do not recognize any major weakness

**Questions:**

--

**Limitations:**

--

---

> ### Author Rebuttal · Authors · 2023-08-07
>
> We thank the reviewer for the strong positive feedback on our paper!

---

### Official Review · Reviewer_RiCR · 2023-07-03

**Soundness:** 3 good
**Presentation:** 3 good
**Contribution:** 3 good
**Rating:** 7
**Confidence:** 3

**Summary:**

They take an in-depth empirical and also theoretical analysis of the in-context learning abilities of transformers on various tasks. Their theoretical analysis of the learned algorithms is especially strong. The work in my eyes presents an extension of previous works in this field to more complex ICL problems (like least squares, ridge regression, Lasso). They identify mechanisms which transformers can learn by which a number of ICL tasks can be performed.

**Strengths:**

The submission is technically sound and very detailed in its explanations and supplements.
The work extends previous studies on in-context learning. They provide in-depth and very relevant analyses for a number of simple ICL tasks. I do, however, wonder in which ways this work can be of practical relevance. For language modelling, the tasks and architectural sizes are quite different to these evaluated toy tasks. For learning optimization methods, it would be interesting to provide more general bounds for ICL analysis which could be used to transfer to more relevant tasks. Also, it would be interesting to see if the provided analyses could be used to make modifications to the transformer architecture. I understand the scope of this work is huge already and this work might exactly enable these kinds of follow-ups.
(Capability of transformers for ICL) In my eyes it seems rather obvious that the demonstrated capabilities can be learned through transformers, given that these architectures are able to model far more complex tasks.
(Architectural choices for ICL) From a practical perspective the suggestions for optimal choices seem less relevant as of now (e.g. which architectural parameters work well in practice for modelling certain ICL aspects), since architectural choices are studied in isolation and only for toy problems. However, theoretical upper bounds on architectural choices seem interesting and seem to provide interesting ways in which to modify and analyse architectures for enhanced ICL.


**Weaknesses:**

I do at times wonder if the theoretical derivations on the capabilities of in-context learning do hold up in practice. Take for example Pre-ICL. Authors motivate this with an example where the hyperparamter in a linear regression varies. I would expect the transformer to be able to learn to generalize these parameters, given a number of different hyperparamters and pretraining rounds. This would not be akin to algorithm selection but rather a learned mechanism that can learn across hyperparameters. While the work might just give upper bounds here, I imagine these are quite hard to transfer into practice.

The capability of transformers to learn various tasks in in-context learning has been known for quite some time. I think some relevant prior work on "Prior-data fitted networks" is missing, that demonstrates in-context learning with transformers before the term "in-context learning" came up:
Transformers Can Do Bayesian Inference (ICLR 2022): This work studies if Transformers can in-context learn algorithmic tasks (such as Gaussian Processes and MLPs), to my knowledge the first work to do so (e.g. prior to Garg and Ayurek) and with an eye of the Bayesian approximation learned in ICL. They show e.g. Gaussian Processes and MLPs can be approximated by transformers, demonstrating hyperparameter selection and deriving the Bayesian approximation capabilities of transformers.
Statistical Foundations of Prior-Data Fitted Networks (ICML 2023)
TabPFN: A Transformer That Solves Small Tabular Classification Problems in a Second (ICLR 2023): Studies the ability of in-context learning for tabular datasets. It also evaluates task mixtures and shows that transformers are able to model complex algorithmic tasks through ICL.

**Questions:**

(Pretraining rounds for excess risk) the authors provide derivations of multiple tasks in terms of the pretraining rounds needed to train transformers with an excess risk to a given baseline. These results are interesting and demonstrate that for a variety of tasks such bounds can be found. I wonder weather such rules can be made up for more general classes of tasks, such that it would be easier to transfer these results to other tasks? Can polynomial guarantees, e.g. be derived given some computational complexity of the baseline algorithm and task?

Did you study what happens when the pretrained transformer is applied to out-of-distribution data, e.g. linear pretraining applied to sparse linear tasks (algorithmically out of distribution) or when the input data is drawn from another distribiution (e.g. higher variance, colinearity, ..)

Are there limits to the number of algorithmic tasks selected by transformers in the selection algorithm?

**Limitations:**

Limitations are not clearly addressed, those could include: Theoretical analyses are hard to apply to more realtistic and complex real world tasks. The analyses are difficult to perform even for the simple tasks shown in this work, probably for most complex tasks infeasable?

---

> ### Author Rebuttal · Authors · 2023-08-07
>
> We thank the reviewer for the valuable feedback on our paper. We respond to the specific questions as follows.
>
> > …Authors motivate this with an example where the hyperparamter in a linear regression varies. I would expect the transformer to be able to learn to generalize these parameters, given a number of different hyperparamters and pretraining rounds. This would not be akin to algorithm selection but rather a learned mechanism that can learn across hyperparameters. While the work might just give upper bounds here, I imagine these are quite hard to transfer into practice.
>
> We would appreciate it if the reviewer can clarify this question or provide some concrete examples, as we were unsure if the following interpretation was correct.
>
> By “generalizing these hyperparameters”, did the reviewer mean e.g. generalizing from a finite set of $\lambda$ (as in Theorem 11)  to a continuous range of $\lambda$’s? In other words, can transformer pretrained on finitely many $\lambda$’s perform near-optimal algorithm selection over continuous $\lambda$.
>
> In that case, we believe it may be possible to construct a transformer that achieves this. The technical difficulty would be in the statistical analysis of ridge regression about best $\lambda$ selection (rather than the transformer construction). While we did not pursue this, we believe this could be an interesting direction for future work.
>
> > …relevant prior work on "Prior-data fitted networks" is missing, that demonstrates in-context learning with transformers before the term "in-context learning" came up. …
>
> We thank the reviewer for pointing out the missing references on Prior-data Fitted Networks (PFNs), and will properly cite them in our revision.
>
> Overall, these work on PFNs indeed demonstrates the in-context learning capability (Bayesian optimality) of transformers in various settings. We believe the experiments in these work and the theory in our paper complement each other: Our results essentially show that transformers can efficiently approximate a broad class of ICL algorithms, and we give theoretical constructions for concrete ICL algorithms, such as linear regression, Lasso, and gradient descent on neural networks. Such results for the expressive power and the Bayes-optimality of the resulting transformers were not established in the PFN literature.
>
> Also, our results are not restricted to the Bayesian setting and can be broaderly applicable for providing frequentist in-context prediction guarantees for transformers.
>
> > (Pretraining rounds for excess risk) the authors provide derivations of multiple tasks in terms of the pretraining rounds needed to train transformers with an excess risk to a given baseline… whether such rules can be made up for more general classes of tasks, such that it would be easier to transfer these results to other tasks? Can polynomial guarantees, e.g. be derived given some computational complexity of the baseline algorithm and task?
>
> We believe there may indeed be some general conditions for tasks that are learnable in-context by transformers. For example, for any task that i) can be efficiently learnable by gradient descent, and ii) the gradient is approximable by attention layers, we should be able to construct transformers to do ICL and obtain similar polynomial sample complexity guarantees as in our paper. Concretely spelling out such general conditions, and identifying alternative conditions (different from GD) would be an important question for future work.
>
> > … what happens when the pretrained transformer is applied to out-of-distribution data? e.g. linear pretraining applied to sparse linear tasks (algorithmically out of distribution) or when the input data is drawn from another distribiution (e.g. higher variance, colinearity, ..)
>
> Most of our approximation results (e.g. Theorem 4, Corollary 5, Theorem 7, Theorem 11) apply to any such OOD scenario, since the transformers we construct work on any in-context dataset with mild boundedness and non-degeneracy assumptions, with no distributional assumptions required. On the other hand, our pretraining results (Section 5) and experiments focus on the in-distribution setting.
>
> Experimentally, Garg et al. (2022) have conducted extensive empirical studies of ICL in OOD scenarios, with mixed results. Theoretically, we believe characterizing the OOD behavior of pretrained transformers is a major open problem and may require new insights and techniques.
>
> > Are there limits to the number of algorithmic tasks selected by transformers in the selection algorithm?
>
> Our transformer constructions do not have a hard limit on the number of tasks. A larger number of tasks $K$ would require a larger hidden token dimension $D$ to store all the intermediate results, as well as a larger $N$ in order for the validation losses to be faithful estimates of the true losses. See Theorem 11 (formal version in Theorem J.1) for an example.
>
> > Limitations are not clearly addressed, those could include: Theoretical analyses are hard to apply to more realtistic and complex real world tasks…
>
> We agree that complex real-world tasks could be much more challenging than our setting, especially if the task distributions are more complicated or even consist of language data rather than real-valued data (as in our setting).
>
> We will add a discussion on this and other limitations of our work in our revision.

---

> > ### Comment · Reviewer_RiCR · 2023-08-10
> >
> > Thank you for addressing my questions.
> >
> > Concerning "generalizing these hyperparameters": That is what I meant indeed. I imagine when a transformer is tested for lambda_1, lambda_2, .. it might be able to generalize to lambda_new atleast within the bounds of lambdas seen. This might not work perfectly but an approximation should be learned.
> > I have to say this is less of a criticism, this would be another work and does not fit the scope of yours.

---

> > > ### Author Response · Authors · 2023-08-15
> > > **Response**
> > >
> > > Thank you for the response. Yes, we agree this generalizing to \lambda_new would be an interesting direction for future work.

---

### Official Review · Reviewer_oV7P · 2023-07-05

**Soundness:** 3 good
**Presentation:** 3 good
**Contribution:** 2 fair
**Rating:** 4
**Confidence:** 4

**Summary:**

This paper theoretically investigate the in-context learning ability from the approximation view. The authors derive the error bound of approximating  least squares, ridge regression, and Lasso algorithms with transformer. The corresponding statistical properties, i.e., the convergence rate and Bayes suboptimality, are derived for transformers. In addition, the authors show that transformers are also able to implement the algorithm selection. Simulation results are provided to verify the theoretical findings.

**Strengths:**

This paper presents solid error bound for transformer to approximate different kinds of algorithms, which are interesting for the approximation theory of neural networks. The corresponding convergence rates for different algorithms are relevant for explaining the in-context learning behavior of transformer. In addition, the approximation results for the algorithm selection unit are noval in the investigation of in-context learning.

**Weaknesses:**

1. Although this paper provides extensive parameters assignments to show that transformer can implement a series of algorithms, rare experimental results are provided to corroborate these constructions. For example, the authors construct transformers that implement one step gradient descent with several layers and repest this pattern. However, such periodic pattern of network parameters and intermediate results is not shown in the simulation section. Consequently, it becomes challenging to assert that transformers indeed implement the algorithms as stated in the paper.

2.  The authors analyze the attention with ReLu activation function, although this particular activation function is rarely employed in practical designs of attentions. To justify the necessity of analyzing ReLU instead of softmax, it would be beneficial to provide reasoning considering that previous studies, such as [1] and [2], predominantly adhere to the softmax activation function.

3. Given the approximation results in the paper of transformers with ReLu activation and the fact that Multi-Layer Perceptrons (MLP) are universal approximators of continuous functions, it is beneficial to discuss that wether the results in this paper imply that MLP can also implement the algorithms in the paper and consequently possess the in-context learning ability.

4. In Corollary 6, the paper derives the estimation error of approximate ridge regression in Bayesian setting, but such Bayesian result is missing for Lasso. It will be helpful to discussion such absence of  Bayesia analysis for Lasso.

5. The generalization error bound in section 5 needs more discussions. Previous works, such as [3] and [4], have derived generalization error bounds for transformers. The novalty of the generalization bound in this paper needs more discussion.

I am happy to change my evaluation if the authors can answer the above questions.

[1] Von Oswald J, Niklasson E, Randazzo E, et al. Transformers learn in-context by gradient descent[J]. arXiv preprint arXiv:2212.07677, 2022.

[2] Akyürek E, Schuurmans D, Andreas J, et al. What learning algorithm is in-context learning? investigations with linear models[J]. arXiv preprint arXiv:2211.15661, 2022.

[3] Edelman B L, Goel S, Kakade S, et al. Inductive biases and variable creation in self-attention mechanisms[C]//International Conference on Machine Learning. PMLR, 2022: 5793-5831.

[4] Zhang Y, Liu B, Cai Q, et al. An Analysis of Attention via the Lens of Exchangeability and Latent Variable Models[J]. arXiv preprint arXiv:2212.14852, 2022.



**Questions:**

The questions are listed in the previous section.

**Limitations:**

The limitation section is missing in this paper.

---

> ### Author Rebuttal · Authors · 2023-08-07
>
> We thank the reviewer for the valuable feedback on our paper. We respond to the specific questions as follows.
>
> > …rare experimental results are provided to corroborate these constructions. For example, the authors construct transformers that implement one step gradient descent with several layers and repeat this pattern. However, such periodic pattern of network parameters and intermediate results is not shown in the simulation section. Consequently, it becomes challenging to assert that transformers indeed implement the algorithms as stated in the paper.
>
> We did not mean to show that empirically learned transformers implement gradient descent (GD) in its intermediate layers, and we suspect that alternative mechanisms could exist. GD is rather **one possible mechanism** we use in theory for constructing the transformers.
>
>
> > The authors analyze the attention with ReLu activation function, although this particular activation function is rarely employed in practical designs of attentions. To justify the necessity of analyzing ReLU instead of softmax, it would be beneficial to provide reasoning considering that previous studies, such as [1] and [2], predominantly adhere to the softmax activation function.
>
> For the purpose of our theory, we believe there is no essential difference between ReLU and softmax. We believe our constructions can be generalized to standard softmax attention with some additional technical treatments. Our choice of ReLU was merely to simplify certain arguments.
>
> Experimentally, we tried out exactly the ReLU architecture we used in our theory (cf. Line 289-291 & Appendix M.1) and it performs well. Other recent studies (e.g. Shen et al. 2023) have also found transformers with ReLU attentions to perform well.
>
> K. Shen, J. Guo, X. Tan, S. Tang, R. Wang, and J. Bian. A Study on ReLU and Softmax in Transformer. arXiv preprint arXiv:2302.06461, 2023.
>
> Re [1,2], we remark that [1] only used **linear attentions** instead of softmax, and the constructions in [2] used softmax in its “saturating regime” where it is approximately a **hard max** (cf. their arXiv version Appendix C.4.1, Page 19). Therefore, neither paper should count as "predominantly using the softmax activation" in a strict sense.
>
> > whether the results in this paper imply that MLP can also implement the algorithms in the paper and consequently possess the in-context learning ability.
>
> By “MLP”, did the reviewer mean an MLP over the concatenated input (a vector in $\mathbb{R}^{dN}$)? In that case, by the universal approximation property, MLPs can also approximate the ICL algorithms we considered.
>
> However, such an MLP would be **significantly more inefficient** (with much larger depth and width) compared with our constructions, as i) the input dimension (and thus the number of parameters) already scales linearly with the sequence length, whereas the size of our transformers do not; and ii) we used the attention layers in efficient ways to implement various operations such as in-context gradient descent (see e.g. Proposition E.1 for an example where the attention structure is crucial), which are unclear how to implement efficiently by MLPs. We will add a discussion about this point in our revision.
>
> Feel free to let us know if we understood the meaning of the "MLP" correctly.
>
> > In Corollary 6, the paper derives the estimation error of approximate ridge regression in Bayesian setting, but such Bayesian result is missing for Lasso. It will be helpful to discussion such absence of Bayesian analysis for Lasso.
>
> A Bayesian result for Lasso analogous to Corollary 6 holds directly for linear models with the Laplacian prior (by applying Theorem 7). We stated the result for ridge regression for convenience only (instead of for any fundamental reason), as our main focus was on the later results on adaptive algorithm selection building on these “base” algorithms.
>
> > The generalization error bound in section 5 needs more discussions. Previous works, such as [3] and [4], have derived generalization error bounds for transformers. The novalty of the generalization bound in this paper needs more discussion.
>
> The sample complexity results in Section 5 (& Appendix M) can be viewed as corollaries of our efficient transformer constructions in Section 3 & 4, where we combine the constructions with standard generalization analysis to derive the sample complexities for pretraining.
>
> We did not claim technical novelty in the generalization analysis part. Our techniques are indeed standard (by controlling Lipschitz constants + chaining arguments), and alternative techniques like [3,4] may also work here.
>
> The novelty is rather in the efficient transformer constructions in Section 3 & 4, which ensures that the final sample complexity in Section 5 is mild. Results like [3,4] did not provide such concrete constructions for ICL algorithms, and thus they themselves cannot deduce our sample complexity results.
>
> We will add a discussion of these points (space permitted) in our revision.
>
> > The limitation section is missing in this paper.
>
> We appreciate the suggestion, and will properly discuss the limitations of our work in our revision.
>
> ---
> We thank the reviewer again for reading our response. We appreciate it if the reviewer could consider raising the score, if our rebuttal has addressed your concerns.

---

> > ### Comment · Reviewer_oV7P · 2023-08-12
> >
> > Thanks for the author's reply. Given the rebuttal, I have the following concerns:
> >
> > (Explanations of possible mechanisms and ReLU activation)
> >
> > Thanks for explaining the perspective where the results in this paper are meaningful. I now accept the approximation view in this paper to study ICL. Given this perspective, the explanation in the rebuttal of the softmax in [2] seems pale. Although they mainly use the saturation part of softmax, this is one possible mechanism to understand softmax, viewed from the perspective explained in the rebuttal. I think a detailed discussion about this will be very helpful.
> >
> > In addition, the authors mentioned that the results can be generalized to softmax attention. Could the authors provide the intuitions about how to do this? I think this is unclear from the techniques in the current paper.
> >
> > (ICL in MLP)
> >
> > Thanks for the detailed explanations of the potential problems in MLP. I am sorry for not providing a clear definition of MLP. In fact, I would like to discuss the RNN, which is an universal approximator. Could authors discuss the ICL ability of RNN?
> >
> > In my personal opinion, a very important question related to the main theme of this paper is that: the constructions in the paper are a sufficient condition for ICL, i.e., any NN that can approximate these algorithms will have ICL ability,  or they are only a hypothesis to explain the ICL ability of transformer? If the latter is correct, could authors comment on how to verify this hypothesis? The answer to this question is not clear after reading this paper.
> >
> > (Novelty of Pretraining results)
> >
> > The authors mentioned that `We provide the first line of results for pretraining transformers to perform the various ICL tasks
> > 75 above, from polynomially many training sequences (Section 5 & Appendix K).' in the introduction. It seems that the pretraining result is one of the main contributions of this paper. If the novelty is not the technical part, as mentioned in the rebuttal, I think the contribution here needs more discussion.

---

> > > ### Author Response · Authors · 2023-08-15
> > > **Response to further questions**
> > >
> > > We thank the reviewer for the response. We respond to the additional questions as follows.
> > >
> > > **(Explanations of possible mechanisms and ReLU activation)**
> > >
> > > >  the explanation in the rebuttal of the softmax in [2] seems pale… this is one possible mechanism to understand softmax… a detailed discussion about this will be very helpful.
> > >
> > > Apart from the “saturating regime”, the way [2] used softmax is very different from the way we used the ReLU. They used softmax attention, in combination with the MLP layers, to approximate various low-level operations such as “aff, mul, mov”, and used them to approximate the gradient of the square loss. A single gradient step requires multiple low-level operations concatenated, and consequently, they need a 9-layer transformer (cf. their Appendix A) to approximate a single SGD step.
> > >
> > > By contrast, we directly use a single attention layer with ReLU activation to approximate a full-batch GD step, and our construction works for a broader class of convex losses (cf. our Proposition E.1). Technically, it directly uses the attention structure efficiently to approximate gradients, different from [2].
> > >
> > > We will add a discussion about this in our revision.
> > >
> > > > the authors mentioned that the results can be generalized to softmax attention. Could the authors provide the intuitions about how to do this?
> > >
> > > Our construction can be generalized to softmax attention as follows: We can use the softmax—in conjunction with a specific positional encoding in the input—to implement (tokenwise) sigmoid activation, and then approximate the gradients using sigmoid in place of the ReLU. See, for example, Giannou et al. (ICML 2023; Lemma 5) for implementing sigmoids from softmax attention. The argument would be in essence the same as ours after obtaining the sigmoid, but overall more tedious compared with our construction using the ReLU.
> > >
> > > A. Giannou, S. Rajput, J.-y. Sohn, K. Lee, J. D. Lee, and D. Papailiopoulos. Looped transformers as programmable computers. arXiv preprint arXiv:2301.13196, 2023.
> > >
> > > **(ICL in MLP)**
> > > > In fact, I would like to discuss the RNN, which is an universal approximator. Could authors discuss the ICL ability of RNN?
> > >
> > > RNNs could approximate ICL algorithms better than vanilla MLPs, due to their suitability for processing sequential inputs.
> > >
> > > However, for implementing the ICL algorithms in our paper, we believe **RNNs would still be much more inefficient than transformers**. One reason is that RNNs (in its basic form) only consist of matrix-vector products with fixed weight matrices, and lack the attention mechanism where input tokens themselves can interact with each other. Consequently, key mechanisms we used such as in-context gradient descent (cf. Theorem 9) would be much harder to implement by RNNs than transformers.
> > >
> > > A simple analogy would be the dot product $\langle x_1, x_2\rangle$: RNNs with layers of the form $\sigma(W[x_1; x_2])$ may approximate this function very inefficiently (by incurring universal approximation results in high dimension), whereas transformers can approximate this function efficiently using the attention mechanism when $x_1$ is the key and $x_2$ is the value.
> > >
> > > > the constructions in the paper are a sufficient condition for ICL, i.e., any NN that can approximate these algorithms will have ICL ability, or they are only a hypothesis to explain the ICL ability of transformer?
> > >
> > > Our paper merely provides upper bounds for Transformers to do ICL. While our constructions *suggest* that MLPs/RNNs are unlikely to match transformers in the efficiency of doing ICL, strictly speaking, our upper bounds don’t imply how these alternative architectures will do.
> > >
> > > One way to investigate this further is to establish formal lower bounds for these alternative architectures. This would be an interesting direction for future work, but we believe are out of scope for this paper and do not undermine our contributions, as transformers themselves are already widely used and have demonstrated remarkable ICL capabilities.

---

> > > > ### Author Response · Authors · 2023-08-15
> > > > **Response to further questions (cont'd)**
> > > >
> > > > **(Novelty of Pretraining results)**
> > > >
> > > > We re-emphasize that our pretraining results in Section 5 follow from two parts of results, with the **generalization techniques being only one part of it**:
> > > > * Efficient transformer constructions (Section 3 & 4) for performing various ICL algorithms, with mild bounds on the size of the transformer (number of layers, heads, and weight norms).
> > > > * Generalization bounds for pretraining transformers of a given size (Theorem K.1). However, to apply such bounds, the constructed transformers need to have bounded sizes in the first place.
> > > >
> > > > We only meant that the techniques for the generalization part are standard; The efficient transformer constructions required new techniques such as new efficient implementation of in-context gradient descent, which we did explain in the paper at length (see, e.g. Section 3.3).
> > > >
> > > > > If the novelty is not the technical part, as mentioned in the rebuttal, I think the contribution here needs more discussion.
> > > >
> > > > In terms of contributions, the result statements themselves (Theorem K.2-K.4) are already new; no such precise quantitative statements about learning ICL algorithms with transformers have been spelled out in the literature to our knowledge. Further, the setting is interesting (learning an ICL algorithm by transformers on a “meta”-distribution of ICL data distributions; cf. Appendix K.1), and the sample complexities are polynomial and depend mildly on all problem parameters.
> > > >
> > > > We firmly believe that **these statements themselves are already interesting contributions in their own right** and could motivate follow-up works, even with proof techniques aside.

---

> > > > ### Comment · Reviewer_oV7P · 2023-08-19
> > > >
> > > > Thank the authors for the detailed response. For the question
> > > >
> > > > "the constructions in the paper are a sufficient condition for ICL, i.e., any NN that can approximate these algorithms will have ICL ability, or they are only a hypothesis to explain the ICL ability of transformer? If the latter is correct, could authors comment on how to verify this hypothesis?"
> > > >
> > > > It seems that the authors prefer the latter claim is correct from the provided response, and the results in this paper cannot serve as sufficient conditions for ICL ability of other networks. In my personal understanding, this means that this paper proposes the hypothesis, i.e., the construction result, to explain the ICL ability of the transformer. I am wondering if the authors could provide some methods to verify these hypotheses.

---

> > > > > ### Author Response · Authors · 2023-08-19
> > > > > **Response**
> > > > >
> > > > > Thank you for the reply! Regarding the question
> > > > >
> > > > > > It seems that the authors prefer the latter claim is correct from the provided response, and the results in this paper cannot serve as sufficient conditions for ICL ability of other networks. In my personal understanding, this means that this paper proposes the hypothesis, i.e., the construction result, to explain the ICL ability of the transformer. I am wondering if the authors could provide some methods to verify these hypotheses.
> > > > >
> > > > > Our paper indeed focuses on the transformer architecture. In terms of "explaining" ICL ability of transformers, we believe there could be (at least) 3 typical methods:
> > > > >
> > > > > 1. Theoretical constructions: Prove there exists a transformer with certain size, that does ICL with certain performance guarantee. But the transformer is not guaranteed to be the one learned in experiments.
> > > > > 2. Experiments: Find a transformer that achieves strong ICL performance, e.g. one pretrained on massive data by gradient-based optimization.
> > > > > 3. Further mechanistic understanding about the experimentally learned transformer, for example, whether it indeed implements the theoretically constructed mechanisms.
> > > > >
> > > > > Our paper already provides results in both 1 and 2. We also answered 3 in one aspect in our algorithm selection experiments, where we showed that learned transformers do implement the (high-level) mechanism of selecting different algorithms for different input data (cf. Figure  2, 3, 5). A more comprehensive study of question 3 would be an important question for future work.
> > > > >
> > > > > >  ... this paper proposes the hypothesis, i.e., the construction result, to explain the ICL ability of the transformer. I am wondering if the authors could provide some methods to verify these hypotheses.
> > > > >
> > > > > We additionally remark that, for our theoretical constructions, the theorem statements are already self-contained results (there exists transformers that achieve good ICL performance...) rather than hypotheses. Our experiments verify a different and stronger hypothesis: gradient-based optimization in practice can find such a transformer.
> > > > >
> > > > > ---
> > > > >
> > > > > We thank the reviewer again for the fruitful rounds of discussions. We would love to hear whether you have any additional concerns about our paper, and engage with you in the remaining of the discussion period. Otherwise, we would appreciate if the reviewer can reconsider our contributions and the evaluation of our paper accordingly, based on our discussions.

---

> > > > > > ### Comment · Reviewer_oV7P · 2023-08-20
> > > > > >
> > > > > > Thank the authors for the detailed response. I think our main disagreement is about the main hypothesis of this paper.
> > > > > >
> > > > > > In my opinion, the main hypothesis of this paper is that transformers use the constructions provided to implement ICL.  Previous papers also provide different constructions for different algorithms. To make this paper novel, the main hypothesis should be on the new constructions. Otherwise, this is no new message except the algorithm selection part.
> > > > > >
> > > > > > I agree that the theorem statements are already self-contained results, but here the hypothesis means that the theory can be verified by the experiments. The experiments verify a potentially different hypothesis: the gradient-based optimization in practice can find a transformer, which may be different from the constructed transformer here.
> > > > > >
> > > > > > I think a very simple way to solve this concern is to check whether the trained transformer in the experiment part demonstrates the periodic property in the construction.

---

> > > > > > > ### Author Response · Authors · 2023-08-20
> > > > > > > **Response on concerns about "main hypothesis" and "no new message except the algorithm selection part"**
> > > > > > >
> > > > > > > Thank you for the further reply. We briefly remark on your opinions as follows, and would be happy to further engage.
> > > > > > >
> > > > > > > Our main point here is that "whether transformers use the constructions provided to implement ICL" is *not our main hypothesis*, and we believe *not the only important question* (though certainly being one) in the body of work on ICL.
> > > > > > >
> > > > > > > > I think our main disagreement is about the main hypothesis of this paper. In my opinion, the main hypothesis of this paper is that transformers use the constructions provided to implement ICL... The experiments verify a potentially different hypothesis: the gradient-based optimization in practice can find a transformer, which may be different from the constructed transformer here.
> > > > > > >
> > > > > > > We indeed disagree with the reviewer that "transformers use the constructions provided to implement ICL" is our main hypothesis. This is not our intended hypothesis, and we did not phrase our results like that in our paper. If any claim in our paper sounded like that to the reviewer, feel free to point out and we would be happy to clarify in our revision.
> > > > > > >
> > > > > > > Our ICL results (apart from algorithm selection) showed that transformers (i) in theory *can* achieve good ICL performance, and (ii) experimentally *does* achieve good ICL performance. The focus is on the efficiency of the theoretical constructions, as well as the ICL performance of learned transformers in experiments (with no restriction on the mechanisms).
> > > > > > >
> > > > > > > > Previous papers also provide different constructions for different algorithms. To make this paper novel, the main hypothesis should be on the new constructions.
> > > > > > >
> > > > > > > We believe "novelty" of neural network constructions could be a subjective measure; we rather focused on objective measures like concrete bounds on the size (number of layers, heads, etc), which we improved significantly over existing work.
> > > > > > >
> > > > > > > Also, our algorithm selection constructions are arguably "novel", as the target ICL algorithm we approximate there is new (involving selection between multiple base algorithms).
> > > > > > >
> > > > > > > > Otherwise, this is no new message except the algorithm selection part.
> > > > > > >
> > > > > > > We believe our results on algorithm selection are already interesting contributions on their own.
> > > > > > >
> > > > > > > > I think a very simple way to solve this concern is to check whether the trained transformer in the experiment part demonstrates the periodic property in the construction.
> > > > > > >
> > > > > > > We agree this is an interesting future direction, and the periodicity experiment you suggested would be a good starting point! However, we believe this is out of the scope of the current paper, and is not necessarily the only important question in the body of work on ICL.

---

> > > > > > > > ### Comment · Reviewer_oV7P · 2023-08-20
> > > > > > > >
> > > > > > > > ``Our ICL results (apart from algorithm selection) showed that transformers (i) in theory can achieve good ICL performance, and (ii) experimentally does achieve good ICL performance. The focus is on the efficiency of the theoretical constructions, as well as the ICL performance of learned transformers in experiments (with no restriction on the mechanisms).''
> > > > > > > >
> > > > > > > > It seems that these two points have been demonstrated in previous works, like [1]. Although the authors state they need more layers for approximation, the constructions in this paper need more positional embedding. It is hard to state which is more efficient in theory. A comparison of the experimental efficiency between the previous work and the existing work should be provided to support the author's claim.
> > > > > > > >
> > > > > > > >
> > > > > > > > [1] Akyürek E, Schuurmans D, Andreas J, et al. What learning algorithm is in-context learning? investigations with linear models[J]. arXiv preprint arXiv:2211.15661, 2022.
> > > > > > > >
> > > > > > > > [2] Von Oswald J, Niklasson E, Randazzo E, et al. Transformers learn in-context by gradient descent[J]. arXiv preprint arXiv:2212.07677, 2022.

---

> > > > > > > > > ### Author Response · Authors · 2023-08-21
> > > > > > > > >
> > > > > > > > > > "Our ICL results (apart from algorithm selection) showed that transformers (i) in theory can achieve good ICL performance... The focus is on the efficiency of the theoretical constructions... "
> > > > > > > > > It seems that these two points have been demonstrated in previous works, like [1]. Although the authors state they need more layers for approximation, the constructions in this paper need more positional embedding. It is hard to state which is more efficient in theory.
> > > > > > > > >
> > > > > > > > > [1] uses a 9-layer transformer to approximate a single SGD step for ridge regression, where our construction use a single-layer transformer to approximate a full-batch GD step, and hence $O(\log(1/\epsilon))$ layers to approximately solve the full ERM problem for ridge regression (Theorem 4). Using [1]'s SGD construction, the number of transformer layers for solving the same ERM would be ${\rm poly}(1/\epsilon)$ by standard optimization theory, much higher than ours.
> > > > > > > > >
> > > > > > > > > We additionally provided transformer constructions for Lasso, generalized linear models, and gradient descent for two-layer neural networks, all of which are not considered in [1].
> > > > > > > > >
> > > > > > > > > > the constructions in this paper need more positional embedding.
> > > > > > > > >
> > > > > > > > > We believe the positional encoding in [1] is actually more complex than ours (provided in their Appendix C.4.1, Eq (32-34)), which involves one-hot indicator vectors $\mathbf{e}_i$ where $i$ is the position of the token. By contrast, our positional encoding only involves an indicator for being the final (test) token (cf. our Eq(3)).

---

> > > > > > > > > > ### Author Response · Authors · 2023-08-21
> > > > > > > > > > **Summary of concerns and our contributions**
> > > > > > > > > >
> > > > > > > > > > We thank the reviewer again for the thoughtful feedback on our paper, and the many efforts in all the engagement with us. We also appreciate the many constructive suggestions.
> > > > > > > > > >
> > > > > > > > > > As our discussions went lengthy, here we would like to make a **brief summary (from our perspective) for all main concerns raised by the reviewer collectively after our rebuttal, as a reference point for the reviewer/AC discussions**. Feel free to correct us for any disagreement or add in anything, either over here or in the updated review / internal discussions (as the author-reviewer discussion period is ending soon):
> > > > > > > > > >
> > > > > > > > > > * Explanations of possible mechanisms and ReLU activation
> > > > > > > > > >
> > > > > > > > > > * ICL in MLP: About the ICL capability of MLPs and RNNs
> > > > > > > > > >
> > > > > > > > > > * Novelty of Pretraining results
> > > > > > > > > >
> > > > > > > > > > * "Main hypothesis of our paper": with disagreement on whether the reviewer's proposal is the main hypothesis of our paper
> > > > > > > > > >
> > > > > > > > > > * Further technical concerns (e.g. positional encoding)
> > > > > > > > > >
> > > > > > > > > > We provided detailed answers to all concerns above, which can be found by keyword search. We would also love to hear the reviewer's thought on our final answers to those questions.
> > > > > > > > > >
> > > > > > > > > > We also briefly restate **our main contributions from our perspective**:
> > > > > > > > > > 1. algorithm selection
> > > > > > > > > > 2. theory for ICL, with more efficient constructions, covering more tasks, and analysis of pretraining
> > > > > > > > > > 3. experimental validation for both strong ICL performance in "base" tasks, and the algorithm selection phenomenon

---

### Official Review · Reviewer_pWto · 2023-07-06

**Soundness:** 4 excellent
**Presentation:** 3 good
**Contribution:** 4 excellent
**Rating:** 7
**Confidence:** 3

**Summary:**

In context Learning, is a setting in which Transformers can learn to perform new tasks when prompted with training and test examples. This work advances the understanding of the capabilities of ICL. In particular, this paper proves that for a broad class of standard machine learning algorithms like ridge regression, lasso etc, transformers can implement these methods. Moreover ICL can perform in-context algorithm selection and select simple algorithms to learn a more complex algorithm. They also show that using their proposed method they can construct a transformer that can perform nearly Bayes-optimal ICL on noisy linear models with mixed noise levels.

**Strengths:**

* This paper expands upon our understanding of ICL. It shows that transformers can implement a plethora of standard ML tasks and require much mild bounds on the number of layers and heads.
* The paper extends the analysis to in-context algorithm selection and provides two algorithm selection mechanisms. They use the proposed mechanism to construct a transformer that can perform Bayes optimal ICL on noisy linear models.
* This works opens up new directions to explore (theoretically and empirically) optimal ICL construction on other problems
* The paper is well written and easy to follow


**Weaknesses:**

* It would be good to know how results such as those in [1] compares to the one obtained in this paper.

[1] Are transformers universal approximators of sequence-to-sequence functions? a

**Questions:**

what was the reasoning behind choosing Bayes-optimal ICL on noisy linear models with mixed noise levels as an example to show case the usefulness of in context algorithm selection? Was that the most complex method for which something like theorem 12 could be proven?

---

> ### Author Rebuttal · Authors · 2023-08-07
>
> We thank the reviewer for the valuable feedback on our paper. We respond to the specific questions as follows.
>
> > How results such as those in [1] compares to the one obtained in this paper.
>
> Our approximation results and [1] are very different. [1] shows that transformers are universal approximators for a large class of sequence-to-sequence functions. However, their result is for generic continuous seq-to-seq functions, and their number of layers is exponential in the worst-case (cf. Section 4.4 of [1]).
>
> By contrast, **our results are for specific target functions (ICL algorithms) but much more efficient**—The number of layers, heads, and weight norms only depend polynomially on relevant problem parameters and the desired approximation accuracy. This happens as our constructions utilize the special structure of the ICL algorithms (our approximation target).
>
> > The reasoning behind choosing Bayes-optimal ICL on noisy linear models with mixed noise levels.
>
> There is no fundamental reason behind our choice of noisy linear models with mixed noise levels. Rather, we picked this setting as it is solvable by transformers via algorithm selection, and the setting itself is a harder one than that of Akyurek et al. 2022 which studies a single noise level.
>
> > Was that the most complex method for which something like theorem 12 could be proven?
>
> If we understand correctly, you mean the most complex “setting” (data generating model)? We believe results like Theorem 12 hold as long as 1) the model is a **mixture model**; 2) Bayes-optimal ICL can be done for each component of the mixture; and 3) $N$ is sufficiently large so that the validation losses are accurate estimations of the true population losses. Under these conditions, algorithm selection (post-ICL validation) can be used to do nearly Bayes-optimal ICL. An example is a mixture of generalized linear models with different link functions.

---

> > ### Comment · Reviewer_pWto · 2023-08-15
> > **Response to Authors**
> >
> > Thank you for clarifying my doubts.
> >
> > I have no further questions regarding the paper and I stand by my assessment that the paper is technically solid with high impact.
> >
> > PS:- One idea/further extension could be to prove a seq-2seq result like in [1] but in context. Note that this does not impact the assessment of this paper.

---

> > > ### Author Response · Authors · 2023-08-15
> > > **Response**
> > >
> > > Thank you for the response and the positive feedback on our paper!
> > >
> > > Re extension: Yes, we agree extending our in-context learning results to a seq-to-seq setting (for predicting at every token) would be an interesting direction for future work.

---

### Decision · Program_Chairs · 2023-09-21

**Decision:**

Accept (oral)

**Comment:**

In-context learning is an intriguing emerging ability of large language models. This paper makes important and practically relevant contributions towards understanding in-context learning abilities of transformers. There are more than one solid contributions. First, the authors provide efficient constructions for transformers that can learn regression tasks in-context. Their implementation is significantly more efficient than prior art. Notably, they can use a single attention layer with ReLU activation to approximate one GD step whereas prior works required multiple layers. This is insightful for practical performance of ICL. Secondly, the paper makes good contribution towards understanding model selection ability via in-context learning. For instance, prior work provided insights into Bayes optimal linear regression. This work manages to extend such findings to a broader class of algorithms and provide rigorous insights into the algorithm selection ability of TFs. I should note that there are some shortcomings of the paper that reviewer points out such as the use of ReLU attention rather than the canonical attention. But I find these fairly acceptable. I believe it will make a great addition to NeurIPS 2023 program.